# Transformers Efficiently Perform In-Context Logistic Regression via Normalized Gradient Descent

**Chenyang Zhang** [1]   **Yuan Cao** [1]

## Abstract

Transformers have demonstrated remarkable in-context learning (ICL) capabilities. The strong ICL performance of transformers is commonly believed to arise from their ability to implicitly execute certain algorithms on the context, thereby enhancing prediction and generation. In this work, we investigate how transformers with softmax attention perform in-context learning on linear classification data. We first construct a class of multi-layer transformers that can perform in-context logistic regression, with each layer exactly performing one step of normalized gradient descent on an in-context loss. Then, we show that our constructed transformer can be obtained through (i) training a single self-attention layer supervised by one-step gradient descent, and (ii) recurrently applying the trained layer to obtain a looped model. Training convergence guarantees of the self-attention layer and out-of-distribution generalization guarantees of the looped model are provided. Our results advance the theoretical understanding of ICL mechanism by showcasing how softmax transformers can effectively act as in-context learners.

## 1. Introduction

Transformers have achieved remarkable success in a wide range of applications, including natural language processing (Vaswani et al., 2017; Devlin, 2018; Touvron et al., 2023), computer vision (Dosovitskiy et al., 2020; Rao et al., 2021; Yuan et al., 2021; Zhang et al., 2025b), and reinforcement learning (Janner et al., 2021; Reed et al., 2022; Kim et al., 2025). One widely recognized interpretation for their empirical success is their ability to perform in-context learning (ICL): pre-trained transformers are capable of performing previously unseen tasks based on demonstrations and examples in the prompt, without requiring any additional task-specific fine-tuning (Brown et al., 2020).

A line of recent works interpret the in-context learning (ICL) capability of transformers from an algorithmic perspective, viewing transformers as models that can implicitly execute certain learning algorithms on the context examples. Specifically, Garg et al. (2022) proposes a theoretical framework for ICL in terms of learning a hypothesis class, and empirically shows that transformers can in-context learn the linear function class. Motivated by this empirical finding, several recent works attempt to theoretically study how transformers perform in-context learning on linear regression tasks. Akyürek et al. (2022); Von Oswald et al. (2023) construct multi-layer transformers with linear attention that can execute gradient descent on the an "in-context loss" defined on the context data, thereby enabling in-context learning of linear regression. Ahn et al. (2023); Zhang et al. (2024b); Huang et al. (2024) further provide training guarantees for single-layer transformers with linear or softmax attention, showing that such models can be trained to solve in-context regression problems. Beyond standard linear regression, Guo et al. (2023); Bai et al. (2024) further demonstrate the in-context learning capacities of transformers by constructing multi-head ReLU transformers capable of performing a variety of learning algorithms, including ridge regression, Lasso regression, and generalized linear models. More recently, Frei & Vardi (2025); Shen et al. (2025) study how single-layer transformers with linear attention can be trained to solve in-context classification on Gaussian mixture data.

Several recent works also investigate in-context learning and other learning tasks with looped transformers, in which the same parameter matrices are shared across different layers. Gatmiry et al. (2024); Chen et al. (2025a) theoretically show that looped transformers still implement gradient descent for in-context linear regression. Yang et al. (2024) empirically demonstrates that looped transformers can achieve performance comparable to that of standard transformers in in-context learning, while using significantly fewer parameters. Geiping et al. (2025) further shows that at inference time, transformers can effectively benefit from increased depth by recurrently applying a trained block.

---

[1]School of Computing & Data Science, The University of Hong Kong. Correspondence to: Yuan Cao <yuancao@hku.hk>.

*Proceedings of the 43$^{rd}$ International Conference on Machine Learning*, Seoul, South Korea. PMLR 306, 2026. Copyright 2026 by the author(s).

In this work, we study how a multi-layer (looped) transformer with softmax attention performs in context learning on classification tasks. Following the settings in Huang et al. (2025), we consider using transformers to solve an "in-context weight prediction" task, and investigate transformer's expressive power, training guarantees, and out-of-distribution (O.O.D.) generalization performance on this task. The major contributions of this work are as follows.

- We establish expressive power guarantees and demonstrate that there exists a class of multi-layer softmax transformers that can perform in-context logistic regression (defined on exponential loss function)[1] via normalized gradient descent. Specifically, the hidden layers' outputs of an $L$-layer transformer exactly match the first $L$ iterates of normalized gradient descent on the in-context loss of logistic regression. Leveraging this exact equivalence and the implicit bias of normalized gradient descent, we further prove that the transformer's output converges in direction to the maximum-margin solution of the context dataset as the depth $L$ increases.

- We also study whether our constructed models can be obtained through training. We consider the strategy to first train a single-layer transformer, and then obtain a looped transformer by recurrently applying the trained layer. For this training problem, our results precisely characterize the existence of a unique minimizer, establish a linear convergence rate, and demonstrate that the obtained model aligns well with the ones constructed in our expressive power guarantees. Interestingly, our results show that the transformer learns normalized gradient descent even though it is supervised by a "gradient descent teacher".

- We validate the capacities of the obtained looped transformers to solve in-context weight prediction by providing an O.O.D. generalization bound. Notably, this result is a point-wise high probability guarantee that holds for any input, and is stronger than the in-expectation guarantees commonly adopted in the in-context learning literature. Under mild assumptions of the number of layers $L$, the generalization bound is given as $\widetilde{\mathcal{O}}\left(\frac{d}{n}\right)$, matching the PAC learning sample complexity lower bound (Long, 1995).

- From a technical perspective, our training analysis develops several novel proof techniques, including an analysis based on an approximate training procedure, the application of the Newton–Kantorovich theorem, and the derivation of a Polyak-Lojasiewicz inequality. We believe these

novel proof techniques may help the analysis of transformers with softmax attention in broad scenarios, and thus may be of independent interest.

## 2. Problem setups

In this section, we introduce the problem setting of in-context learning for weight prediction and the multi-layer softmax transformer models considered in this work.

**In-context learning for weight predictions.** In-context learning (ICL) refers to a learning framework in which the input consists of a collection of context data pairs $D_n = \{(\mathbf{x}_i, y_i) : \mathbf{x}_i \in \mathcal{X}, y_i \in \mathcal{Y}\}_{i=1}^n \in \mathcal{D}$, together with a query input $\mathbf{x}_{\text{query}} \in \mathcal{X}$, whose label $y_{\text{query}} \in \mathcal{Y}$ is unknown. An in-context learning model $f(\cdot, \cdot) : \mathcal{X} \times \mathcal{D} \to \mathcal{Y}$ is then expected to infer the underlying feature-label mapping from $D_n$, and produce a prediction for the unknown label $y_{\text{query}}$ in the form of $\widehat{y}_{\text{query}} = f(\mathbf{x}_{\text{query}}, D_n)$. As demonstrated in recent theoretical works (Ahn et al., 2023; Bai et al., 2024; Zhang et al., 2024b), in-context learning models like transformers typically handle this type of task by implicitly performing certain learning algorithms to fit a predictor $\widehat{g}(\cdot) : \mathcal{X} \to \mathcal{Y}$ based on the context dataset $D_n$, and then generate the final prediction via $\widehat{y}_{\text{query}} = \widehat{g}(\mathbf{x}_{\text{query}})$.

Beyond the classic in-context learning setting where the goal is to output a prediction for $y_{\text{query}}$, Huang et al. (2025) further proposes the problem of "in-context weight prediction" under the linear regression setting. In this task, the predictor $\widehat{g}(\cdot) = \langle \widehat{\boldsymbol{\theta}}, \cdot \rangle$ is a linear model, and the in-context learner is required to explicitly output this weight vector $\widehat{\boldsymbol{\theta}}$. Specifically, they assume that each in-context set $D_n = \{(\mathbf{x}_i, y_i)\}_{i=1}^n$ admits a ground truth vector $\boldsymbol{\theta}^*$ such that $y_i = \langle \boldsymbol{\theta}^*, \mathbf{x}_i \rangle$, and the objective is to estimate $\boldsymbol{\theta}^*$ by $\widehat{\boldsymbol{\theta}}$. Motivated by Huang et al. (2025), in this paper, we consider "in-context weight prediction" in classification,

**Definition 2.1.** Let $\mathcal{D}_{\boldsymbol{\theta}^*}$ be a distribution over the $d$-dimensional unit sphere $\mathbb{S}^{d-1}$, and $\mathcal{D}_{\mathbf{x}}$ be a distribution over $\mathbb{R}^d$. Then the context dataset $D_n = \{(\mathbf{x}_i, y_i)\}_{i=1}^n \subset \mathbb{R}^d \times \{\pm 1\}$ and its corresponding ground-truth vector $\boldsymbol{\theta}^*$ are generated from a joint distribution $\mathcal{D}$ as:

1. The ground truth vector $\boldsymbol{\theta}^*$ is generated from $\mathcal{D}_{\boldsymbol{\theta}^*}$.

2. Each feature vector $\mathbf{x}_i$ is generated from $\mathcal{D}_{\mathbf{x}}$, $i \in [n]$.

3. Each label is determined as $y_i = \text{sign}(\langle \mathbf{x}_i, \boldsymbol{\theta}^* \rangle)$, $i \in [n]$.

Note that the sign function is invariant to positive rescaling, rendering each label $y_i$ determined solely by the direction of $\boldsymbol{\theta}^*$. Consequently, we may assume without loss of generality that $\boldsymbol{\theta}^*$ lies on the unit sphere $\mathbb{S}^{d-1}$. For the same reason, we only require the predicted weight $\widehat{\boldsymbol{\theta}}$ to approximate $\boldsymbol{\theta}^*$ up to its direction, quantified by $\left\| \frac{\widehat{\boldsymbol{\theta}}}{\|\widehat{\boldsymbol{\theta}}\|_2} - \boldsymbol{\theta}^* \right\|_2$.

---

[1] In the implicit-bias literature, linear classification with exponential-tailed losses is often treated under the umbrella of the term "logistic regression" and analyzed under a unified framework. In this work, following this convention, we slightly abuse the term "in-context logistic regression" to refer specifically to the in-context linear classification setting with the exponential loss.

**Transformers with softmax attention.** We consider solving the in-context weight prediction tasks by transformers. The embedding matrix $\mathbf{Z}_0$ for the context dataset $D_n = \{(\mathbf{x}_i, y_i)\}_{i=1}^n$, which serves as the input to the transformer, is defined as

$$\mathbf{Z}_0 = \begin{bmatrix} \mathbf{z}_1 & \mathbf{z}_2 & \cdots & \mathbf{z}_n & \mathbf{0}_d \\ \mathbf{0}_d & \mathbf{0}_d & \cdots & \mathbf{0}_d & \boldsymbol{\theta}_0 \end{bmatrix} \in \mathbb{R}^{2d \times (n+1)}, \quad (1)$$

where $\mathbf{z}_i = y_i \cdot \mathbf{x}_i$ for all $i \in [n]$, in alignment with the common settings in linear classification. This choice does not restrict the input format: if the transformer takes the concatenated vector $[\mathbf{x}_i^\top, y_i]^\top$ as input, as commonly considered in prior works, an embedding layer can transform it into $\mathbf{z}_i = y_i \mathbf{x}_i$, and the formal derivations are provided in Appendix C. In addition, $\boldsymbol{\theta}_0$ serves as an initialization for the prediction of $\boldsymbol{\theta}^*$. With the input matrix in the form of (1), a standard self-attention layer (Vaswani et al., 2017) is defined as

$$\mathtt{SA}(\mathbf{Z}; \mathbf{V}, \mathbf{W}) = \mathbf{V}\mathbf{Z}\mathrm{softmax}(\mathbf{Z}^\top \mathbf{W} \mathbf{Z} + \mathbf{M}). \quad (2)$$

In the formulation above, $\mathbf{V}, \mathbf{W} \in \mathbb{R}^{2d \times 2d}$ denote the value and key-query parameter matrices of the self-attention layer, respectively. Following the convention of most theoretical studies (Zhang et al., 2024b;c; Huang et al., 2024; Wang et al., 2024b; Zhang et al., 2025a), we reparameterize the original key and query matrices into a single trainable parameter matrix $\mathbf{W}$. The mask matrix $\mathbf{M} = \begin{bmatrix} \mathbf{0}_{n \times (n+1)} \\ -\infty \cdot \mathbf{1}_{n+1}^\top \end{bmatrix}$ prevents attention to the last query column. To define an $L$-layer transformer, we denote $(\mathbf{V}_{0:L-1}, \mathbf{W}_{0:L-1}) = [(\mathbf{V}_0, \mathbf{W}_0), \dots, (\mathbf{V}_{L-1}, \mathbf{W}_{L-1})]$ as the collection of parameter pairs across layers. Then, building upon the single self-attention layer defined in (2), an $L$-layer transformer with residual connections and parameters $(\mathbf{V}_{0:L-1}, \mathbf{W}_{0:L-1})$ is defined recursively as

$$\mathtt{TF}(\mathbf{Z}_0; \mathbf{V}_{0:L-1}, \mathbf{W}_{0:L-1}) = \mathbf{Z}_L \in \mathbb{R}^{2d \times (n+1)},$$
$$\mathbf{Z}_{l+1} = \mathbf{Z}_l + \mathtt{SA}(\mathbf{Z}_l; \mathbf{V}_l, \mathbf{W}_l), \ l = 0, \dots, L-1. \quad (3)$$

We read the entries in $\mathbf{Z}_L$ located at the same position as $\boldsymbol{\theta}_0$ in $\mathbf{Z}_0$, i.e., $\boldsymbol{\theta}_L = [\mathbf{Z}_L]_{d+1:2d, \, n+1}$, as the predicted weight vector corresponding to the input $\mathbf{Z}_0$. This setup is consistent with the setting in Huang et al. (2025).

## 3. Main results

### 3.1. Overview of Theoretical Results

In this section, we present the theoretical results on how transformers can perform in-context logistic regression and solve in-context weight prediction tasks in classification. Before presenting the technical details, we first provide a high-level roadmap and summary of our these conclusions. We begin by introducing the in-context loss for linear classification, the empirical risk defined on the context

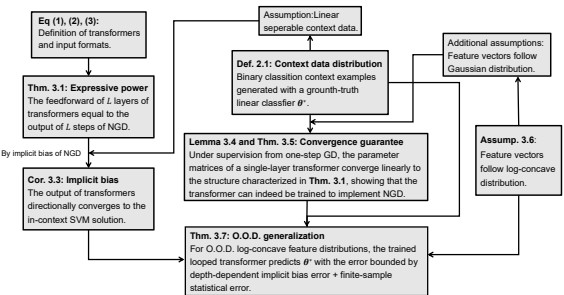

*Figure 1.* High-level roadmap of the theoretical framework, illustrating the assumptions, main results, and logical flow underlying our expressivity, implicit-bias, trainability, and O.O.D. generalization guarantees.

dataset. Theorem 3.1 then establishes the expressive-power result that under appropriate parameterizations, an $L$-layer softmax transformer can exactly implement $L$ steps of normalized gradient descent on this in-context loss. Building on this characterization, Corollary 3.3 applies the implicit bias theory of normalized gradient descent and shows that, for linearly separable context data, the transformers' output directionally converges to the in-context maximum-margin solution. We then move from expressivity to learning guarantees. In Theorem 3.5, we consider training a single-layer softmax transformer using supervision from a one-step gradient-descent teacher, and show that the trained parameters exactly converge to an NGD-implementing structures. This demonstrates that the NGD mechanism characterized in Theorem 3.1 can be achieved through training, rather than merely existing as an explicit expressivity construction. Finally, Theorem 3.7 validates the O.O.D. generalization behavior of the looped transformer obtained by recurrently applying the trained single-layer block from Theorem 3.5. Under log-concave feature distributions, Theorem 3.7 shows that its prediction error is controlled by the implicit-bias error from Corollary 3.3 and the finite-sample statistical error of the in-context maximum-margin solution. Figure 1 provides a schematic illustration of this roadmap, highlighting the assumptions, main results, and logical flow underlying our theoretical guarantees.

### 3.2. Deep transformers can perform in-context logistic regression via normalized gradient descent

To study how transformers solve in-context logistic regression, we define the empirical risk on the context dataset and refer to it as "in-context loss". Specifically, for any $\boldsymbol{\theta} \in \mathbb{R}^d$, its in-context loss on $D_n = \{(\mathbf{x}_i, y_i)\}_{i=1}^n$ is given as

$$\mathcal{L}_{\mathrm{ICL}}(\boldsymbol{\theta}) = \frac{1}{n} \sum_{i=1}^n \ell(\langle \boldsymbol{\theta}, y_i \cdot \mathbf{x}_i \rangle), \quad (4)$$

where $\ell(\cdot) : \mathbb{R} \times \mathbb{R} \to \mathbb{R}$ is a commonly chosen exponential-tailed loss function, such as the exponential or logistic loss. In this work, we adopt the exponential loss, i.e. $\ell(x) = e^{-x}$,

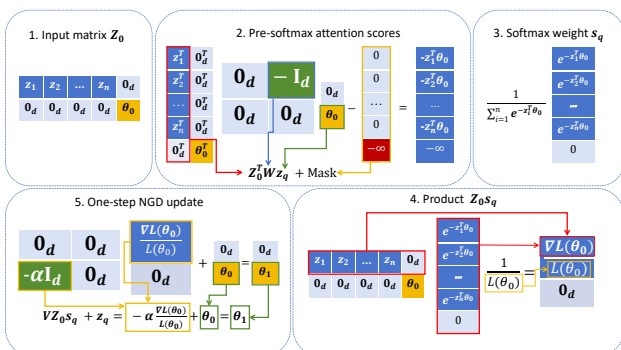

*Figure 2.* Illustration of the one-step mechanism in Theorem 3.1. Starting from the input matrix $\mathbf{Z}_0$, a single softmax self-attention layer first constructs the pre-softmax attention scores, then obtains the softmax weight vector $\mathbf{s}_q$, and finally computes the attention output that matches one step of normalized gradient descent on the in-context loss $\mathcal{L}_{\mathrm{ICL}}$.

to enable cleaner mathematical results. In the following, we show that there exists a class of $L$-layer transformers that can exactly implement $L$ steps of *normalized gradient descent* (NGD) on $\mathcal{L}_{\mathrm{ICL}}$, as formalized in Theorem 3.1.

**Theorem 3.1.** *Consider an $L$-layer transformer* $\mathrm{TF}(\cdot)$ *in* (3) *with parameter matrices* $(\mathbf{V}_{0:L-1}, \mathbf{W}_{0:L-1})$ *of the form*

$$\mathbf{V}_l = \begin{bmatrix} \mathbf{0}_{d\times d} & \mathbf{0}_{d\times d} \\ \widetilde{\alpha}_l \cdot \mathbf{I}_d & \mathbf{0}_{d\times d} \end{bmatrix}, \quad \mathbf{W}_l = \begin{bmatrix} \mathbf{0}_{d\times d} & -\mathbf{I}_d \\ \mathbf{A}_{1,l} & \mathbf{A}_{2,l} \end{bmatrix}$$

*for* $l = 0, 1, \ldots, L-1$, *where* $\mathbf{A}_{1,l}, \mathbf{A}_{2,l}$ *are arbitrary* $d \times d$ *matrices, and* $\widetilde{\alpha}_l > 0$. *Then for any input matrix* $\mathbf{Z}_0$ *of the form* (1), *the transformer gives hidden layer outputs* $\mathbf{Z}_l$, $l = 1, \ldots, L$, *such that* $\{\boldsymbol{\theta}_l = [\mathbf{Z}_l]_{d+1:2d, n+1}\}_{l=1}^{L}$ *are the iterates of normalized gradient descent on* $\mathcal{L}_{\mathrm{ICL}}(\boldsymbol{\theta})$ *with learning rates* $\{\widetilde{\alpha}_l\}_{l=0}^{L-1}$:

$$\boldsymbol{\theta}_{l+1} = \boldsymbol{\theta}_l - \widetilde{\alpha}_l \frac{\nabla \mathcal{L}_{\mathrm{ICL}}(\boldsymbol{\theta}_l)}{\mathcal{L}_{\mathrm{ICL}}(\boldsymbol{\theta}_l)}, \quad l = 0, 1, \ldots, L-1. \quad (5)$$

Theorem 3.1 demonstrates that, under appropriate parameterizations, the outputs of softmax attention layers exactly match the iterates of normalized gradient descent applied to the in-context loss $\mathcal{L}_{\mathrm{ICL}}$. Consequently, the forward pass of an $L$-layer transformer can be interpreted as performing in-context logistic regression through $L$ steps of normalized gradient descent. Specifically, Figure 2 illustrates how a single softmax self-attention layer constructs the attention weights, recovers the normalized-gradient direction, and implements one step of normalized gradient descent.

As an expressive-power result, Theorem 3.1 does not impose any assumptions on the context examples. In addition, the attention-only construction should be understood as a minimal construction that isolates the role of softmax attention: adding common architectural components, such as MLP layers, gated attention, or positional encodings, does not affect the validity of the expressive-power conclusion,

since these components can be parameterized so that the resulting model preserves the same input-output mapping as the attention-only transformers defined in (3). Moreover, we note that the update rule in (5) slightly differs from the standard definition of normalized gradient descent, as it normalizes the gradient by $\mathcal{L}_{\mathrm{ICL}}(\boldsymbol{\theta})$ instead of $\|\nabla\mathcal{L}_{\mathrm{ICL}}(\boldsymbol{\theta})\|_2$. However, this form of normalization term is commonly adopted in theoretical studies of logistic regression (Nacson et al., 2019; Ji & Telgarsky, 2021; Wang et al., 2024a), and is also referred to as normalized gradient descent. Our use of this terminology follows this convention. We further note that if an RMSNorm-style layer normalization is incorporated into the construction, the induced update can be transformed into the standard normalized-gradient-descent form, with normalization by $\|\nabla\mathcal{L}_{\mathrm{ICL}}(\boldsymbol{\theta})\|_2$.

Several recent works (Ahn et al., 2023; Bai et al., 2024) show that transformers with linear/ReLU attention can perform in-context linear regression with gradient descent. In comparison, our result in Theorem 3.1 shows that transformers with softmax attention can perform in-context logistic regression with normalized gradient descent. Notably, Bai et al. (2024) also covers results on in-context logistic regression, and shows that multi-head ReLU attention layers can approximate gradient descent updates. However, their results rely on universal approximation by multi-head ReLU attention, and require $\widetilde{\mathcal{O}}(\epsilon^{-2})$ heads per layer to achieve an approximation error $\epsilon$. In comparison, our result considers softmax attention, only requires a single head per layer, and the correspondence to normalized gradient descent is exact and does not suffer from any approximation error.

*Remark* 3.2. Theorem 3.1 also enjoys an important advantage that it accommodates arbitrary parameterizations of the blocks $\mathbf{A}_{1,l}$ and $\mathbf{A}_{2,l}$ within $\mathbf{W}_l$. In fact, this parameterization of $\mathbf{W}_l$ can be further generalized to $\mathbf{W}_l = \begin{bmatrix} \mathbf{0}_{d\times d} & -\widetilde{\beta} \cdot \mathbf{I}_d \\ \mathbf{A}_{1,l} & \mathbf{A}_{2,l} \end{bmatrix}$, with $\widetilde{\beta}$ being any positive scalar. Under this more general form, it can be shown that $L$-layer transformers still perform in-context logistic regression via normalized gradient descent, but on a rescaled context dataset $\widetilde{D}_n = \{\widetilde{\beta} \cdot \mathbf{x}_i, y_i\}_{i=1}^{n}$, and with rescaled learning rates $\{\widetilde{\alpha}_l/\widetilde{\beta}\}_{l=0}^{L-1}$. This flexibility in allowing a broad class of parameterizations for $\mathbf{W}_l$ plays a key role in our subsequent analysis in Subsection 3.3. The proof of Theorem 3.1 is also demonstrated for this generalized version in Appendix D

Notably, regardless of the distributions $\mathcal{D}_{\boldsymbol{\theta}^*}, \mathcal{D}_{\mathbf{x}}$, any context dataset $D_n$ following Definition 2.1 can always be linearly separated by its corresponding $\boldsymbol{\theta}^*$. For such linear separable datasets, a remarkable line of works (Soudry et al., 2018; Ji & Telgarsky, 2019; Nacson et al., 2019; Wang et al., 2024a) have shown that (normalized) gradient descent on logistic/exponential loss has an implicit bias towards the maximum-margin solution $\boldsymbol{\theta}_{\mathrm{SVM}}(D_n) =$

$\operatorname{argmax}_{\|\boldsymbol{\theta}\|_2 \leq 1} \min_{i \in [n]} \langle \boldsymbol{\theta}, y_i \cdot \mathbf{x}_i \rangle$. Specifically, Theorem 4.3 in Ji & Telgarsky (2021) shows that the $L$-th iterate of normalized gradient descent with constant learning rate $\widetilde{\alpha}$ converges to the maximum margin solution in direction with a convergence rate $\mathcal{O}(\log(n)/(\widetilde{\alpha}L))$. Combining this result and Theorem 3.1, we have the following corollary.

**Corollary 3.3.** *Suppose that an L-layer transformer is parameterized as in Theorem 3.1 with $\widetilde{\alpha}_l = \widetilde{\alpha} \leq \mathcal{O}(1)$ for all $l = 0, \ldots, L-1$. Then for any context dataset $D_n = \{(\mathbf{x}_i, y_i)\}_{i=1}^n$ following Definition 2.1 and any $\boldsymbol{\theta}_0$ with $\|\boldsymbol{\theta}_0\|_2 = \mathcal{O}(1)$, the predicted weight $\boldsymbol{\theta}_L$ by this transformer directionally converges to $\boldsymbol{\theta}_{\mathrm{SVM}}(D_n)$ as*

$$\left\| \frac{\boldsymbol{\theta}_L}{\|\boldsymbol{\theta}_L\|_2} - \boldsymbol{\theta}_{\mathrm{SVM}}(D_n) \right\| \leq \mathcal{O}\left( \frac{\log n}{\widetilde{\alpha}L} \right).$$

Corollary 3.3 indicates that the predicted weight $\boldsymbol{\theta}_L$ produced by $L$-layer transformers converges in direction to its maximum margin solution $\boldsymbol{\theta}_{\mathrm{SVM}}(D_n)$ at a rate inversely proportional to $L$. With this result, evaluating the quality of $\boldsymbol{\theta}_L$ as a weight predictor for $\boldsymbol{\theta}^*$ reduces to characterizing the discrepancy between $\boldsymbol{\theta}_{\mathrm{SVM}}(D_n)$ and $\boldsymbol{\theta}^*$. We elaborate this in Subsection 3.4.

### 3.3. Training of single softmax-attention layer

In the previous section, we have shown that under appropriate parameterizations, transformers can perform in-context logistic regression via normalized gradient descent. However, this result only demonstrates the expressive power of transformers. To give a more comprehensive analysis, in this section, we investigate whether such transformers can indeed be obtained via training.

An interesting observation is that, the parameterizations in Theorem 3.1 naturally admit a looped implementation, where all layers share the same weights, i.e., $\mathbf{V} = \mathbf{V}_l$ and $\mathbf{W} = \mathbf{W}_l$ for all $l \in [L]$. In addition, Geiping et al. (2025) empirically demonstrates that recurrently applying the trained block enables transformers to achieve better performance at the inference stage. Motivated by these observations, we consider an effective training setup: we first train a single-layer transformer $\mathrm{TF}(\cdot; \mathbf{V}, \mathbf{W})$, and then obtain a multi-layer looped transformer by recurrently applying this trained layer. Notably, we adopt the one-step gradient descent, rather than normalized gradient descent, as a "teacher model" to supervise the single-layer transformer. This setup is inspired by similar settings considered in Huang et al. (2025), and allows us to test whether the model can still learn normalized gradient descent even if the teacher is a different algorithm. The one-step GD update on the context data can be expressed as

$$\boldsymbol{\theta}_{\mathrm{GD}} = \boldsymbol{\theta}_0 - \alpha \nabla \mathcal{L}_{\mathrm{ICL}}(\boldsymbol{\theta}_0),$$

where $\boldsymbol{\theta}_0$ represents the initialization, and $\alpha$ denotes the learning rate for one-step GD update. We consider minimizing the discrepancy between $\boldsymbol{\theta}_{\mathrm{GD}}$ and the output of the single-layer transformer, i.e. $\boldsymbol{\theta}_1 = [\mathrm{TF}(\mathbf{Z}_0, \mathbf{V}, \mathbf{W})]_{d+1;2d,n+1}$. The training objective is defined as the population mean-squared error:

$$\mathcal{L}_{\mathrm{train}}(\mathbf{V}, \mathbf{W}) = \mathbb{E}_{D_n, \boldsymbol{\theta}_0}\left[ \|\boldsymbol{\theta}_1 - \boldsymbol{\theta}_{\mathrm{GD}}\|_2^2 \right].$$

The expectation is taken over the context dataset $D_n$ and the initialization $\boldsymbol{\theta}_0$, where $\boldsymbol{\theta}_0$ is assumed to follow $\mathcal{U}(\mathbb{S}^{d-1})$. Moreover, we assume that $D_n$ is generated following Definition 2.1, with the feature distribution $\mathcal{D}_{\mathbf{x}}$ being $\mathcal{N}(\mathbf{0}, \sigma^2 \mathbf{I}_d)$, and the true classifier distribution $\mathcal{D}_{\boldsymbol{\theta}^*}$ being $\mathcal{U}(\mathbb{S}^{d-1})$. We consider using gradient descent with zero initialization $\mathbf{V}^{(0)}, \mathbf{W}^{(0)} = \mathbf{0}_{2d \times 2d}$ to minimize the training loss $\mathcal{L}_{\mathrm{train}}$:

$$\mathbf{V}^{(t+1)} = \mathbf{V}^{(t)} - \eta \nabla_{\mathbf{V}} \mathcal{L}_{\mathrm{train}}(\mathbf{V}^{(t)}, \mathbf{W}^{(t)}); \tag{6}$$

$$\mathbf{W}^{(t+1)} = \mathbf{W}^{(t)} - \eta \nabla_{\mathbf{W}} \mathcal{L}_{\mathrm{train}}(\mathbf{V}^{(t)}, \mathbf{W}^{(t)}), \tag{7}$$

where $\eta$ denotes the learning rate. Our goal is then to theoretically study this training procedure defined above and verify whether the trained transformer learns to perform one-step normalized gradient descent.

Our first observation is that the iterates $\mathbf{V}^{(t)}, \mathbf{W}^{(t)}$ of gradient descent always preserve certain structured forms, which is summarized in the following lemma.

**Lemma 3.4.** *The iterates $\mathbf{V}^{(t)}$ and $\mathbf{W}^{(t)}$ of the training procedure* (6), (7) *always follow a structured form as*

$$\mathbf{V}^{(t)} = \begin{bmatrix} \mathbf{0}_{d \times d} & \mathbf{0}_{d \times d} \\ C_1^{(t)} \cdot \mathbf{I}_d & \mathbf{0}_{d \times d} \end{bmatrix}, \mathbf{W}^{(t)} = \begin{bmatrix} \mathbf{0}_{d \times d} & -C_2^{(t)} \cdot \mathbf{I}_d \\ \mathbf{0}_{d \times d} & \mathbf{0}_{d \times d} \end{bmatrix},$$

*where $C_1^{(t)}$ and $C_2^{(t)}$ are two scalar coefficients.*

Lemma 3.4 plays a key role in our training analysis: it reduces the original optimization problem concerning the evolutions of full $d \times d$ parameter matrices $\mathbf{V}, \mathbf{W}$ to a much simpler one involving only two scalars $C_1, C_2$. The coefficient vector $\mathbf{C}^{(t)} = [C_1^{(t)}, C_2^{(t)}]^\top$ equivalently follows gradient descent starting from zero initialization $\mathbf{C}^{(0)} = \mathbf{0}$ to minimize a proxy training loss:

$$\mathbf{C}^{(t+1)} = \mathbf{C}^{(t)} - \eta \nabla_{\mathbf{C}} \widetilde{\mathcal{L}}_{\mathrm{train}}(\mathbf{C}^{(t)}),$$

$$\widetilde{\mathcal{L}}_{\mathrm{train}}(\mathbf{C}) = \mathcal{L}_{\mathrm{train}}\left( \begin{bmatrix} \mathbf{0}_{d \times d} & \mathbf{0}_{d \times d} \\ C_1 \cdot \mathbf{I}_d & \mathbf{0}_{d \times d} \end{bmatrix}, \begin{bmatrix} \mathbf{0}_{d \times d} & -C_2 \cdot \mathbf{I}_d \\ \mathbf{0}_{d \times d} & \mathbf{0}_{d \times d} \end{bmatrix} \right).$$

The following theorem characterizes the convergence of this equivalent training procedure.

**Theorem 3.5.** *Suppose that $n = \Omega(d^2)$, $\eta \leq \mathcal{O}\left(\frac{1}{n}\right)$, and $\alpha, \sigma \leq \mathcal{O}(1)$. Then the following results hold.*

1. ***Invariant compact set $\mathcal{R}$.** For all $t \geq 0$, the iterate $\mathbf{C}^{(t)}$ always remains in a compact set $\mathcal{R}$ defined as*

$$\mathcal{R} = [0, 2\alpha e^{\sigma^2/2}] \times [0, 2].$$

2. **Unique local minimizer in $\mathcal{R}$.** *The loss $\widetilde{\mathcal{L}}_{\text{train}}(\mathbf{C})$ has a unique local minimizer $\mathbf{C}^* = [C_1^*, C_2^*]^\top$ in $\mathcal{R}$. In addition, this local minimizer satisfies that*

$$\left\| \mathbf{C}^* - [\alpha e^{\sigma^2/2}, 1]^\top \right\|_2 \leq \mathcal{O}\left(\tfrac{1}{d}\right).$$

3. **Linear convergence of the loss and iterates.** *For $t \geq 0$, the training loss enjoys a linear convergence rate:*

$$\widetilde{\mathcal{L}}_{\text{train}}(\mathbf{C}^{(t)}) - \mathcal{L}_{\text{train}}(\mathbf{C}^*)$$
$$\leq \left( 1 - \frac{\eta \mu_{1,\alpha,\sigma}}{d} \right)^t \left( \widetilde{\mathcal{L}}_{\text{train}}(\mathbf{C}^{(0)}) - \widetilde{\mathcal{L}}_{\text{train}}(\mathbf{C}^*) \right).$$

*Moreover, the iterates $\mathbf{C}^{(t)}$ converges linearly to $\mathbf{C}^*$:*

$$\left\| \mathbf{C}^{(t)} - \mathbf{C}^* \right\|_2 \leq \mu_{2,\alpha,\sigma} \left( 1 - \frac{\eta \mu_{1,\alpha,\sigma}}{d} \right)^{t/2} \|\mathbf{C}^*\|_2.$$

*Here, $\mu_{1,\alpha,\sigma}$ and $\mu_{2,\alpha,\sigma}$ are positive constants solely depending on $\alpha$ and $\sigma$.*

Theorem 3.5 establishes rigorous training convergence guarantees. The first and second conclusions describe the loss landscape of $\widetilde{\mathcal{L}}_{\text{train}}$ and proves the existence of a unique local minimizer $\mathbf{C}^*$. The third conclusion gives accurate convergence guarantees with linear rates. Importantly, by the second and third conclusions, as $t \to \infty$, one has $C_1^{(t)} \approx \alpha e^{\sigma^2/2}$ and $C_2^{(t)} \approx 1$, which implies that the trained transformer layer approximately matches the form of our constructed layers in Theorem 3.1. This demonstrates that:

> *The trained transformer can indeed perform normalized gradient descent update, even though the model is supervised by a gradient descent teacher.*

This reveals a nontrivial separation between the supervising algorithm and the learned in-context algorithm, suggesting that transformers are not merely algorithm imitators, but may also discover algorithmic mechanisms distinct from the supervising algorithm. Moreover, the fact that $C_2^{(t)}$ is not exactly one does not affect the conclusion that the trained transformer layer can exactly perform one-step normalized gradient descent. As discussed in Remark 3.2, the coefficients $C_1$ and $C_2$ admit a clear algorithmic interpretation: $C_2$ acts as the rescaling factor for feature vectors, and the ratio $C_1/C_2$ determines the learning rate. Therefore, the trained single-layer transformer in Theorem 3.5 essentially performs one step of normalized gradient descent on the slightly rescaled dataset $\{(C_2^{(t)} \cdot \mathbf{x}_i, y_i)\}_{i=1}^n$, with the learning rate $C_1^{(t)}/C_2^{(t)} \approx \alpha e^{\sigma^2/2}$.

From a technical perspective, Theorem 3.5 introduces new theoretical tools. While a line of recent works have studied the training of softmax transformers (Jelassi et al., 2022; Wang et al., 2024b; Li et al., 2024b; Zhang et al., 2025a; Shi & Cao, 2025), we note that existing analyses are mostly

under the setting where the learning tasks can be perfectly solved by having softmax attention perform certain "sparse selection". As a result, existing convergence guarantees mostly focus on showing that certain pre-softmax attention scores diverge to infinity, and that they diverge at a faster rate compared to the rest of the scores. In comparison, the learning task we consider is fundamentally different in multiple aspects. First, since the "teacher model" is gradient descent, the learning task is "misspecified" and zero training loss cannot be perfectly achieved. In addition, as is shown in Theorem 3.5, training converges to a finite minimizer $\mathbf{C}^*$ instead of giving diverging parameters in $\mathbf{W}$. More importantly, the model with parameters defined by $\mathbf{C}^*$ does not perform "sparse selection", as the softmax score from the last token to the $i$-th token is $\frac{e^{-C_2^* \langle \mathbf{z}_i, \boldsymbol{\theta}_0 \rangle}}{\sum_{i'=1}^n e^{-C_2^* \langle \mathbf{z}_{i'}, \boldsymbol{\theta}_0 \rangle}}$, which defines a dense, weighted average over all the tokens. Finally, for our learning task, the training loss and its gradient do not admit closed-form expressions, further complicating the optimization analysis. To overcome these challenges, we develop several novel proof techniques, which are summarized in the brief proof sketch as follows.

**Step 1.** We derive explicit non-asymptotic approximations of the gradients (Lemma E.2), and show that $[\alpha e^{\sigma^2/2}, 1]^\top$ is a fixed point of the approximated training process.

**Step 2.** We then apply the Newton–Kantorovich theorem to show the existence of a fixed point $\mathbf{C}^*$ of the original training process that is close to $[\alpha e^{\sigma^2/2}, 1]^\top$ (Lemma E.7).

**Step 3.** We further prove a Polyak-Lojasiewicz (PL) inequality (Lemma E.8) despite the non-convexity of the training loss, which leads to the linear convergence rate.

### 3.4. Multi-layer looped transformers efficiently solve in-context weight prediction

Theorems 3.1 and 3.5 together show that we can recurrently apply the trained transformer layer characterized in Theorem 3.5 to obtain a multi-layer looped transformer that solves in-context logistic regression via normalized gradient descent. In this section, we establish the final theoretical guarantee on the performance of such looped transformers in solving in-context weight prediction. In contrast to the assumption that $\mathcal{D}_{\mathbf{x}}$ during training follows $\mathcal{N}(\mathbf{0}, \sigma^2 \mathbf{I}_d)$, here we study out-of-distribution (O.O.D.) generalization performance on new "test" in-context datasets $D_n$ for which $\mathcal{D}_{\mathbf{x}}$ is a general log-concave distribution.

**Assumption 3.6** (Log-concave distribution). For the feature distribution $\mathcal{D}_{\mathbf{x}}$ in Definition 2.1, let $f(\cdot)$ be its probability density function. Then it holds that

1. **Log-concavity:** $\log[f(\mathbf{x})]$ is a concave function.
2. **Moment conditions:** For any $\mathbf{x} \sim \mathcal{D}_{\mathbf{x}}$, it holds that $\mathbb{E}[\mathbf{x}] = \mathbf{0}$, and $\mathbb{E}[\mathbf{x}\mathbf{x}^\top] = \boldsymbol{\Sigma} \succ 0$.

Assumption 3.6 covers a broad class of distributions, such as centered Gaussian, uniform, and Laplace distributions. Based on it, we have the following theorem.

**Theorem 3.7.** *Let $\mathbf{V}^{(t)}$ and $\mathbf{W}^{(t)}$ be the parameter matrices trained in Theorem 3.5 after $t = \widetilde{\Omega}\left(\frac{d}{\eta\mu_{1,\alpha,\sigma}}\right)$ iterations. Denote $\mathtt{TF}(\cdot, [\mathbf{V}^{(t)}]^{\otimes L}, [\mathbf{W}^{(t)}]^{\otimes L})$ the $L$-layer looped transformer, with $\mathbf{V}^{(t)}$ and $\mathbf{W}^{(t)}$ being its shared weights across layers. Suppose that the feature distribution $\mathcal{D}_{\mathbf{x}}$ in Definition 2.1 follows Assumption 3.6. Then for any input matrix $\mathbf{Z}_0$ of the form (1), with probability at least $1 - \delta$, the looped transformers' prediction $\boldsymbol{\theta}_L = [\mathtt{TF}(\mathbf{Z}_0, [\mathbf{V}^{(t)}]^{\otimes L}, [\mathbf{W}^{(t)}]^{\otimes L})]_{d+1:2d,\, n+1}$ satisfies*

$$\left\| \frac{\boldsymbol{\theta}_L}{\|\boldsymbol{\theta}_L\|_2} - \boldsymbol{\theta}^* \right\|_2 \leq \mathcal{O}\left( \frac{\log n}{\alpha L} + \frac{d\log\rho}{n} \right). \qquad (8)$$

*where $\rho = \max\{n, d, \lambda_{\min}^{-1}(\boldsymbol{\Sigma}), \delta^{-1}\}$,*

Theorem 3.7 provides an O.O.D. generalization guarantee for the looped transformer to solve in-context weight prediction. We note that in recent theoretical studies of ICL (Zhang et al., 2024b; Frei & Vardi, 2025; Huang et al., 2025), the generalization bounds are typically presented in expectation over the distribution of the test context dataset $D_n$. In comparison, Theorem 3.7 establishes a point-wise high-probability guarantee, which holds for any fixed input $\mathbf{Z}_0$. In particular, by choosing $\delta = \mathcal{O}\left(\frac{d}{n}\right)$, the conclusion of Theorem 3.7 can immediately induce an in-expectation bound $\mathbb{E}_{D_n}\left[\left\|\frac{\boldsymbol{\theta}_L}{\|\boldsymbol{\theta}_L\|_2} - \boldsymbol{\theta}^*\right\|_2\right] \leq \mathcal{O}\left(\frac{\log n}{\alpha L} + \frac{d\log\rho}{n}\right)$, validating that our result is stronger than the classic in-expectation generalization bound. Moreover, this O.O.D. generalization ability stems from the good property that the output $\boldsymbol{\theta}_L$ in direction converges to the maximum margin solution $\boldsymbol{\theta}_{\mathrm{SVM}}(D_n)$. It is natural to decompose $\left\|\frac{\boldsymbol{\theta}_L}{\|\boldsymbol{\theta}_L\|_2} - \boldsymbol{\theta}^*\right\|_2 \leq \left\|\frac{\boldsymbol{\theta}_L}{\|\boldsymbol{\theta}_L\|_2} - \boldsymbol{\theta}_{\mathrm{SVM}}(D_n)\right\|_2 + \|\boldsymbol{\theta}_{\mathrm{SVM}}(D_n) - \boldsymbol{\theta}^*\|_2$ by triangle inequality. The first term $\left\|\frac{\boldsymbol{\theta}_L}{\|\boldsymbol{\theta}_L\|_2} - \boldsymbol{\theta}_{\mathrm{SVM}}(D_n)\right\|_2$ is controlled in Corollary 3.3, directly yielding the term $\mathcal{O}\left(\frac{\log n}{\alpha L}\right)$. In addition, the second term $\|\boldsymbol{\theta}_{\mathrm{SVM}}(D_n) - \boldsymbol{\theta}^*\|_2$ quantifies how well the maximum-margin solution learned from the context dataset $D_n$ approximates the true classifier $\boldsymbol{\theta}^*$. The upper bound for this statistical error is demonstrated to be $\mathcal{O}\left(\frac{d\log\rho}{n}\right)$, exactly the second term of (8). The proof for Theorem 3.7 is deferred to Appendix F.

Several recent works Frei & Vardi (2025); Shen et al. (2025) investigate how single-layer transformers with linear attention can be trained to solve in-context classification on Gaussian-mixture data, and establish in-distribution generalization. In contrast, our Theorem 3.7 establishes O.O.D. generalization for multi-layer transformers with softmax attention. Compared with another recent work Bai et al. (2024), our work gives better bounds thanks to the fast convergence rate of normalized gradient descent. Specifically, as is discussed above, the first term in the bound of Theorem 3.7

quantifies the in-direction convergence of $\boldsymbol{\theta}_L$ towards the maximum-margin solution, and the rate is given by Corollary 3.3. Consequently, as long as $L = \widetilde{\Omega}\left(\frac{n}{\alpha d}\right)$, the generalization bound can be given as $\left\|\frac{\boldsymbol{\theta}_L}{\|\boldsymbol{\theta}_L\|_2} - \boldsymbol{\theta}^*\right\|_2 \leq \widetilde{\mathcal{O}}\left(\frac{d}{n}\right)$, which matches the sample complexity lower bound in classic PAC learning (Long, 1995). In contrast, Bai et al. (2024) constructs multi-head ReLU transformers that approximate standard gradient descent for in-context logistic regression. If similar analyses are applied to their setting, then by the implicit bias results of gradient descent (Soudry et al., 2018), their constructed model's output approaches the maximum-margin solution only at the rate $\mathcal{O}\left(\frac{\log\log L}{\log L}\right)$. As a result, unless the depth of the model $L$ is exponentially large in the problem parameters, this term always dominates the statistical error term $\widetilde{\mathcal{O}}\left(\frac{d}{n}\right)$, and therefore fundamentally limits the performance in solving the weight prediction task.

## 4. Experiments

In this section, we present the experimental results. We consider three experimental settings: (i) training a single-layer transformer; (ii) constructing a multi-layer looped transformer from the trained layer, and evaluating its O.O.D. generalization in solving weight prediction; (iii) training a multiple-layer looped transformer from scratch, and evaluating its capacity in solving weight prediction.

### 4.1. Training a single-layer transformer

We first consider training a single-layer transformer to validate our Theorem 3.5. The architecture of single-layer transformer follows the definition in (3) with $L = 1$, and the training strategy aligns with the theoretical settings in Subsection 3.3. We adopt an online gradient descent algorithm to simulate training over the population loss $\mathcal{L}_{\mathrm{train}}$. Specifically, at each iteration, we generate a new batch of $K = 400$ context datasets $\{D_{n,k}\}_{k=1}^K$, where each dataset $D_{n,k}$ is generated following Definition 2.1 with $\mathcal{D}_{\mathbf{x}}$ being $\mathcal{N}(\mathbf{0}_d, \mathbf{I}_d)$ and $\mathcal{D}_{\boldsymbol{\theta}^*}$ being $\mathcal{U}(\mathbb{S}^{d-1})$. For each $D_{n,k}$, we generate a corresponding $\boldsymbol{\theta}_{0,k}$ from $\mathcal{U}(\mathbb{S}^{d-1})$. Then we can obtain a batch of $K$ input matrices $\{\mathbf{Z}_{0,k}\}_{k=1}^K$ embedded of the form in (1), and the gradient descent update in (6), (7) is conducted on this batch of inputs, with the learning rate $\eta = 0.1$. In addition, we consider two gradient descent teachers $\boldsymbol{\theta}_{\mathrm{GD}}$ with $\alpha = 0.5$ and $\alpha = 1$, respectively. For each case, we conduct experiments under three different configurations: $(n, d) \in \{(60, 20), (100, 25), (150, 30)\}$.

Figure 3 reports the curves of training losses. In all settings, the training losses consistently converge near zero. Notably, configurations with a larger dimension $d$ exhibit slower convergence. This observation aligns with our Theorem 3.5, as the linear convergence factor $1 - \eta\mu_{1,\alpha,\sigma}/d$ grows with $d$, thereby slowing down the optimization process.

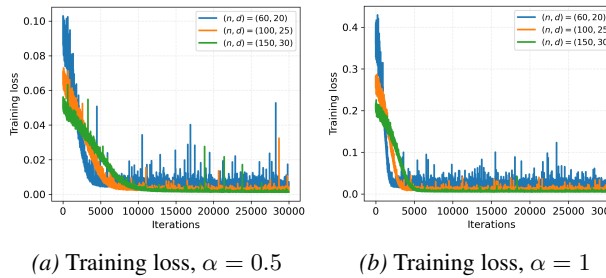

*(a)* Training loss, $\alpha = 0.5$     *(b)* Training loss, $\alpha = 1$

*Figure 3.* Training loss under two settings: $\alpha = 0.5$, and $\alpha = 1$.

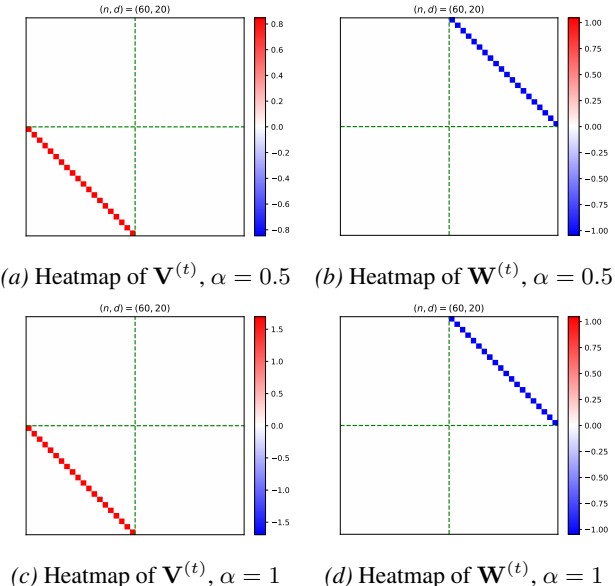

*(a)* Heatmap of $\mathbf{V}^{(t)}$, $\alpha = 0.5$    *(b)* Heatmap of $\mathbf{W}^{(t)}$, $\alpha = 0.5$

*(c)* Heatmap of $\mathbf{V}^{(t)}$, $\alpha = 1$    *(d)* Heatmap of $\mathbf{W}^{(t)}$, $\alpha = 1$

*Figure 4.* Heatmaps of the parameter matrices $\mathbf{V}^{(t)}$ and $\mathbf{W}^{(t)}$ at convergence. Results are shown for $\alpha = 0.5$ and $\alpha = 1$ respectively, with $(n, d) = (60, 20)$.

Figure 4 displays the heatmaps of the parameter matrices $\mathbf{V}^{(t)}$ and $\mathbf{W}^{(t)}$ obtained after training. These results demonstrate that the trained $\mathbf{V}^{(t)}$ and $\mathbf{W}^{(t)}$ follow the structured pattern described in Lemma 3.4: the bottom-left block of $\mathbf{V}$ and the top-right block of $\mathbf{W}$ are almost proportional to the identity matrix, with coefficients $C_1 > 0$ and $-C_2 < 0$, respectively, and all other blocks remain almost zero. Due to the limited space, here we only present the results for $(n, d) = (60, 20)$. Other results are deferred to Appendix G.

Figure 5 further presents the trajectories of $\mathbf{C}^{(t)} = [C_1^{(t)}, C_2^{(t)}]^\top$. The trajectories exhibit clear convergence behavior, as evidenced by the dense accumulations of iterates near the end of the curves. In addition, in all settings, the iterates consistently converge to points close to $[\alpha e^{\sigma^2/2}, 1]^\top$, aligning with the third conclusion in Theorem 3.5.

The experiments in Figures 3, 4, and 5 all match our theoretical conclusions regarding the training of a single-layer transformer, validating that a single-layer transformer can be trained to conduct a normalized gradient descent update.

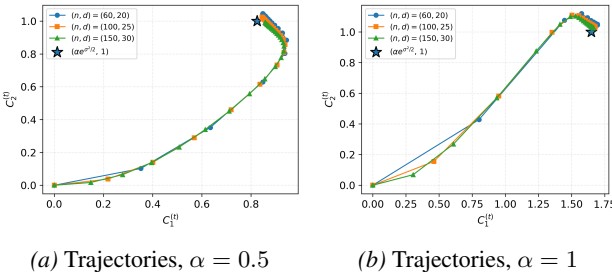

*(a)* Trajectories, $\alpha = 0.5$     *(b)* Trajectories, $\alpha = 1$

*Figure 5.* Trajectories of the coefficient $C_1^{(t)}$ and $C_1^{(2)}$ under two different settings that $\alpha = 0.5$, and $\alpha = 1$.

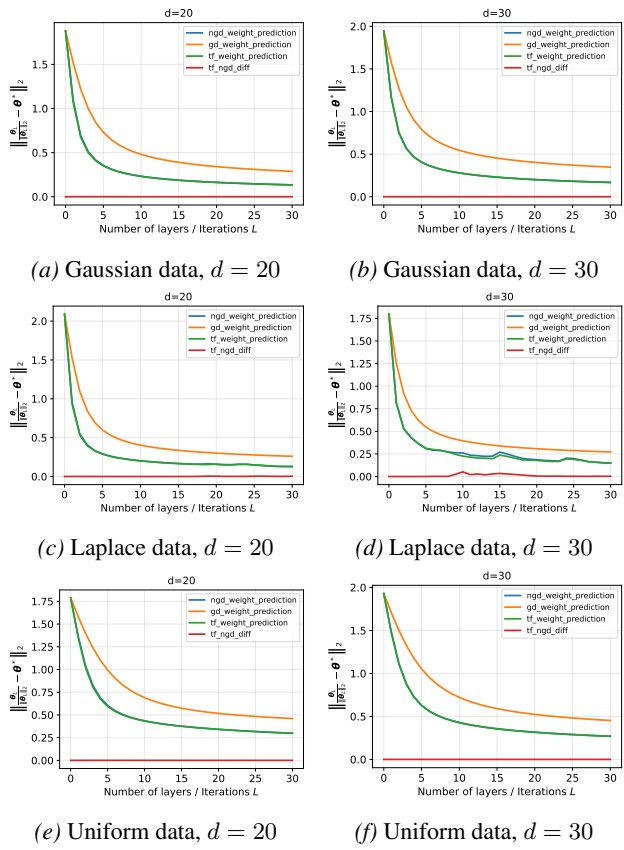

*(a)* Gaussian data, $d = 20$    *(b)* Gaussian data, $d = 30$

*(c)* Laplace data, $d = 20$    *(d)* Laplace data, $d = 30$

*(e)* Uniform data, $d = 20$    *(f)* Uniform data, $d = 30$

*Figure 6.* Evaluation of the in-context weight-prediction produced by looped transformers, NGD iterates, and standard GD iterates on different O.O.D distributed context datasets, with $d \in \{20, 30\}$.

### 4.2. O.O.D. generalization of looped transformers

Following our theoretical settings, we can obtain a multi-layer looped transformer by recurrently applying the trained single-layer transformer. In this section, we conduct experiments to validate the O.O.D. generalization of the resulting looped transformers in solving in-context weight prediction. To make sure that each O.O.D. setting covers significantly different distributions compared with the training data, we consider three different O.O.D. choices for $\mathcal{D}_\mathbf{x}$: we first marginally sample each entry of the random vector $\tilde{\mathbf{x}}$ from (i) standard Gaussian distribution $\mathcal{N}(0, 1)$; (ii) Laplace distribution $\mathrm{Laplace}(0, 1)$; (iii) uniform distribution $\mathcal{U}([0, 1])$,

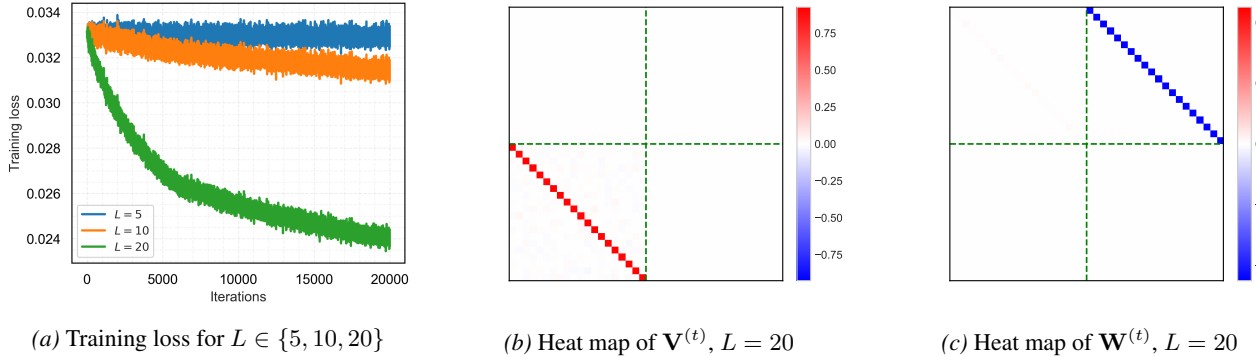

*(a)* Training loss for $L \in \{5, 10, 20\}$

*(b)* Heat map of $\mathbf{V}^{(t)}$, $L = 20$

*(c)* Heat map of $\mathbf{W}^{(t)}$, $L = 20$

*Figure 7.* Training loss curves $L \in \{5, 10, 20\}$ and the heatmaps of parameter matrices $\mathbf{V}^{(t)}$ and $\mathbf{W}^{(t)}$ of 20-layer looped transformers.

and then randomly generate a positive definite matrix $\boldsymbol{\Sigma}$ and obtain a sample $\mathbf{x}$ from $\mathcal{D}_{\mathbf{x}}$ by calculating $\mathbf{x} = \boldsymbol{\Sigma}\widetilde{\mathbf{x}}$. For each choice of $\mathcal{D}_{\mathbf{x}}$, we generate a batch of $K$ input matrices $\{\mathbf{Z}_{0,k}\}_{k=1}^{K}$ following exactly the same data generation procedure as in training, except that the feature distribution is replaced by the corresponding O.O.D. distribution. For each setting, we consider feature dimensions $d \in \{20, 30\}$, and fix the in-context sample size as $n = 500$.

We evaluate the performance of the looped transformers in in-context weight prediction. For better comparison, we also report the results from the iterates of NGD and standard GD under the same experimental settings. The results are given in Figure 6. Across all different settings, the discrepancy between the prediction produced by looped transformers and the ground-truth classifier $\boldsymbol{\theta}^*$ consistently decays to a small value as the number of layers $L$ increases, validating the capacities of deep transformers in solving in-context weight prediction. In addition, we observe that the hidden-layer outputs of the looped transformers remain very close to the iterates of NGD throughout the entire process, achieving nearly identical performance in in-context weight prediction and consistently outperforming standard GD. These observations further support our theoretical findings that multi-layer transformers can efficiently solve in-context weight prediction in in-context logistic regression via NGD.

### 4.3. Training of multi-layer looped transformers

In this section, we consider training a multi-layer looped transformer from scratch. We consider directly using the ground truth vector $\boldsymbol{\theta}^* \in \mathbb{S}^{d-1}$ to supervise the training, i.e. the training loss is defined as $\mathbb{E}\left[\left\|\frac{\boldsymbol{\theta}_L}{\|\boldsymbol{\theta}_L\|_2} - \boldsymbol{\theta}^*\right\|_2^2\right]$. Similar to the previous section, we consider using online gradient descent to minimize this training loss. In addition, each batch of inputs matrices $\{\mathbf{Z}_{0,k}\}_{k=1}^{K}$ follows the same generation process with $\mathcal{D}_{\mathbf{x}}$ being a Gaussian distribution with a randomly generated positive definite covariance matrix $\boldsymbol{\Sigma}$. We set the in-context sample size and feature dimension as $(n, d) = (60, 20)$. The experiments are conducted under three sets of the model depth $L \in \{5, 10, 20\}$.

The results are shown in Figure 7. Figure 7a shows that deeper models achieve lower loss and converge faster, suggesting that increasing depth improves in-context weight prediction. This is consistent with our theory, where the depth $L$ corresponds to the number of NGD iterations. Moreover, Figures 7b and 7c visualize the learned parameter matrices $\mathbf{V}$ and $\mathbf{W}$ of the 20-layer looped transformer. Their clear block-diagonal patterns closely match the construction in Theorem 3.1. These results indicate that, when trained from scratch, deep looped transformers can learn parameter structures that implement in-context logistic regression.

## 5. Conclusions and limitations

This work provides a comprehensive analysis of how transformers with softmax attention perform ICL on linear classification data. Specifically, we construct a class of softmax transformers capable of performing in-context logistic regression. We demonstrate that these transformers can be obtained by training a single-layer model and recurrently applying the trained layer. Furthermore, we establish an O.O.D. generalization for the trained model in in-context weight prediction, and experiments back up our theory.

Our analysis has several limitations. First, we study in-context logistic regression, a simplified setting relative to the real-world tasks. Nevertheless, this setting isolates the role of softmax attention and enables a rigorous characterization of a nontrivial mechanism: the implementation and learning of normalized gradient descent. Second, our expressive-power result is based on an attention-only construction. Although additional components such as MLPs, gated attention, and positional encodings can be parameterized to preserve the same input-output mapping, our theory does not fully characterize their interaction with the NGD mechanism during training. Finally, while experiments suggest that end-to-end trained multi-layer looped transformers can also match our construction, our current training analysis does not cover this setting. Extending the theory to richer architectures, broader tasks, and end-to-end multi-layer training remains an important direction.

## Acknowledgments

We would like to thank the anonymous reviewers and area chairs for their helpful comments. Yuan Cao is supported in part by NSFC 12301657, Hong Kong RGC ECS 27308624, and Hong Kong RGC GRF 17301825.

## Impact Statement

This paper presents theoretical study on the capability of transformers to perform in-context learning. We believe that the results do not have any direct negative societal impact.

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

# A. Related works

**In-context learning of transformers.** In-context learning capacities of transformers are first formalized in Garg et al. (2022), with experiments on linear function hypothesis classes. Given this formulation, many recent works attempt to investigate the in-context learning capabilities under different settings. Akyürek et al. (2022); Von Oswald et al. (2023) theoretically demonstrate the expressive power of transformers, showing that they can implicitly perform algorithms on context data like gradient descent. Ahn et al. (2023) explicitly constructs a multi-layer linear transformer capable of conducting preconditioned gradient descent for linear regression, and provides the characterization of the loss landscape and training convergence. Zhang et al. (2024b) investigates the training of a single-layer linear transformer on the context data embedded from linear regression samples, and provides both in-distribution and out-of-distribution generalization guarantees. Huang et al. (2024) extends this result to the single-layer softmax transformer, while requiring a strong assumption regarding the training strategy. Also, for in-context linear regression, Chen et al. (2024b;c) investigates the mechanism of the multi-heads from the training and expressive power perspectives respectively. Zhang et al. (2024c) investigates the training of a single-layer transformer connected with an MLP block, and shows it can simulate the one-step of gradient descent with learnable initialization. (Bai et al., 2024; Guo et al., 2023) further extend the scope of in-context algorithms implemented by transformers, showing that they can perform empirical risk minimization for linear regression, ridge regression, Lasso, and more general generalized linear models. Frei & Vardi (2025); Shen et al. (2025) investigates the training of single-layer linear transformer to solve in-context classification with Gaussian mixture inputs. Chen & Zou (2024) study the role of depth in transformers through a set of sequence learning tasks, showing that while a single attention layer can achieve memorization, reasoning and generalization require multiple attention layers. Chen et al. (2025b) studies how test-time computation in transformers can be understood through in-context linear regression with randomness and sampling. Cao et al. (2025) investigates the expressive power of Transformers in Bayesian network sequence modeling, showing that they can implement in-context maximum likelihood estimation and autoregressive sampling. Li et al. (2025) study the context hijacking phenomenon through the lens of an optimization procedure with heterogeneous learning rates. Anwar et al. (2024) study in-context linear regression through an adversarial lens, showing that adversarial attacks transfer poorly across transformer seeds and between transformers and classical learning algorithms.

**Optimization of transformers.** Besides the studies focusing on the ICL capacities of transformers, several recent works study the training behavior under other settings. Zhang et al. (2020); Kunstner et al. (2023); Pan & Li (2023); Li et al. (2024a); Zhang et al. (2024d) investigate how transformers behave when trained by different optimizers, covering distinct theoretical and empirical settings. Li et al. (2023); Jelassi et al. (2022) studies the training of shallow ViT-type transformers under certain specified initializations. Gao et al. (2024) addresses the global convergence of transformers given certain prerequisites. Ildiz et al. (2024); Chen et al. (2024a); Nichani et al. (2024); Shi & Cao (2025) study how transformers are trained on Markovian data and how attention mechanisms recover the underlying transition structure. Specifically, Ildiz et al. (2024) characterize self-attention as a context-conditioned Markov chain, Chen et al. (2024a) show that induction heads emerge as a copier–selector–classifier mechanism on $n$-gram data, Nichani et al. (2024) demonstrates that attention gradients recover latent causal graphs, and Shi & Cao (2025) shows that attention selects parent states while values implement Markov transitions in random walks. In addition, some works provide learning guarantees of transformers in certain statistical tasks, including the sparse-linear classifier (Zhang et al., 2025a), sparse token selection (Wang et al., 2024b), one-nearest neighbor selection (Li et al., 2024b), maximum hard-margin classifier (Tarzanagh et al., 2023a;b), the compositions of functions (Wang et al., 2025), implementation of spectral methods and EM updates on Gaussian mixture models (He et al., 2025a;b), and "teacher-student" distillation (Zhang et al., 2026).

**Implicit bias of logistic regression.** A line of theoretical works studies the implicit bias of different optimizers on logistic regression (Soudry et al., 2018; Ji & Telgarsky, 2019; Nacson et al., 2019; Qian & Qian, 2019; Ji & Telgarsky, 2021; Wang et al., 2022; Zhang et al., 2024a; Wang et al., 2024a). Soudry et al. (2018) proves that the iterates of gradient descent directionally converge to the maximum $\ell_2$-margin solution on separable data, and Ji & Telgarsky (2019) extends this result to the settings with non-separable data. While the previous results are established for full-batch gradient descent, Nacson et al. (2019) studies the implicit bias of stochastic gradient descent, and demonstrates the same directional convergence results as full-batch gradient descent. Ji & Telgarsky (2021) propose a primal-dual framework, and demonstrate a fast polynomial convergence rate for implicit bias of normalized gradient descent. Wang et al. (2024a) further proposes an exponentially adaptive learning rate for gradient descent, which achieves a linear convergence rate. Wu et al. (2023) studies the implicit bias of gradient descent under the "edge of stability" setting, where the learning rate can be set as arbitrarily large. Besides gradient descent, several works have investigated adaptive gradient-based optimization methods that incorporate momentum. Qian & Qian (2019) studies the implicit bias of AdaGrad, and shows a directional convergence toward a maximum-margin

solution under certain preconditioned norm. Wang et al. (2022) studies shows that the momentum does not change the implicit bias of gradient descent. Zhang et al. (2024a) demonstrates an implicit bias towards the maximum $\ell_\infty$-margin of Adam.

## B. Notation

In this section, we introduce the following mathematical notations that we use throughout the paper.

**Notation.** Given two sequences $\{x_n\}$ and $\{y_n\}$, we denote $x_n = \mathcal{O}(y_n)$ if there exist some absolute constant $C_1 > 0$ and $N > 0$ such that $|x_n| \leq C_1|y_n|$ for all $n \geq N$. Similarly, we denote $x_n = \Omega(y_n)$ if there exist $C_2 > 0$ and $N > 0$ such that $|x_n| \geq C_2|y_n|$ for all $n > N$. We say $x_n = \Theta(y_n)$ if $x_n = \mathcal{O}(y_n)$ and $x_n = \Omega(y_n)$ both holds. We use $\widetilde{\mathcal{O}}(\cdot)$, $\widetilde{\Omega}(\cdot)$, and $\widetilde{\Theta}(\cdot)$ to hide logarithmic factors in these notations respectively. Moreover, we denote $x_n = \mathrm{poly}(y_n)$ if $x_n = O(y_n^D)$ for some positive constant $D$, and $x_n = \mathrm{polylog}(y_n)$ if $x_n = \mathrm{poly}(\log(y_n))$. For two scalars $a$ and $b$, we denote $a \vee b = \max\{a, b\}$ and $a \wedge b = \min\{a, b\}$. For any $n \in \mathbb{N}_+$, we use $[n]$ to denote the set $\{1, 2, \cdots, n\}$. We use $\mathbf{1}_n$ to denote a $n$-dimensional vector with all 1 entries. For any $d_1, d_2 \in \mathbb{N}_+$, we denote $\mathbf{I}_{d_1}$ the $d_1 \times d_1$ identity matrix, and $\mathbf{0}_{d_1 \times d_2}$ the $d_1 \times d_2$ matrix with all its entries being zero. We denote $\mathbb{S}^{d-1}$ the $d$-dimensional unit sphere, and $\mathcal{U}(\mathbb{S}^{d-1})$ the uniform distribution over $\mathbb{S}^{d-1}$. For any $\mathbf{a} \in \mathbb{R}^d$ and a positive definite matrix $\mathbf{\Sigma} \in \mathbb{R}^{d \times d}$, $\mathcal{N}(\mathbf{a}, \mathbf{\Sigma})$ denotes the $d$-dimensional Gaussian distribution with mean $\mathbf{a}$ and covariance matrix $\mathbf{\Sigma}$. We use $\phi(\cdot)$, and $\Phi(\cdot)$ to denote the p.d.f. and c.d.f. of standard normal distribution, respectively. For any matrix $\mathbf{A}$, we use $\lambda_i(\mathbf{A})$ to denote its $i$-th eigenvalue. In addition, we write $[\mathbf{A}]_{i:j,k}$ to denote the subvector formed by entries in rows $i$ through $j$ of the $k$-th column. Similarly, $[\mathbf{A}]_{i,j:k}$ denote the subvector formed by entries in columns $j$ through $k$ of the $i$-th row.

## C. Embedding concatenated inputs into $y_i \cdot \mathbf{x}_i$

In this section, we show that the embedding vector $\mathbf{z}_i = y_i \cdot \mathbf{x}_i$ used in Eq. (1) can be exactly obtained from the concatenated input $[\mathbf{x}_i^\top, y_i]^\top$ through a standard embedding layer. Therefore, our analysis also applies to the setting where each in-context example is provided in the form $[\mathbf{x}_i^\top, y_i]^\top$.

**Lemma C.1.** *Suppose $y \in \{-1, 1\}$ and $\|\mathbf{x}\|_\infty \leq M$. Let $\mathtt{ReLU}(\cdot)$ denote the ReLU activation applied entrywise. Define*

$$f(\widetilde{\mathbf{x}}; \mathbf{W}_1, \boldsymbol{b}_1, \mathbf{W}_2) = \mathbf{W}_2 \mathtt{ReLU}(\mathbf{W}_1\widetilde{\mathbf{x}} + \boldsymbol{b}_1), \qquad \widetilde{\mathbf{x}} = \begin{bmatrix} \mathbf{x} \\ y \end{bmatrix} \in \mathbb{R}^{d+1},$$

*where*

$$\mathbf{W}_1 = \begin{bmatrix} \mathbf{I}_d & M \cdot \mathbf{1}_d \\ -\mathbf{I}_d & M \cdot \mathbf{1}_d \\ \mathbf{I}_d & -M \cdot \mathbf{1}_d \\ -\mathbf{I}_d & -M \cdot \mathbf{1}_d \end{bmatrix} \in \mathbb{R}^{4d \times (d+1)}, \qquad \boldsymbol{b}_1 = -M \cdot \mathbf{1}_{4d},$$

*and*

$$\mathbf{W}_2 = \begin{bmatrix} \mathbf{I}_d & -\mathbf{I}_d & -\mathbf{I}_d & \mathbf{I}_d \end{bmatrix} \in \mathbb{R}^{d \times 4d}.$$

*Then*

$$f(\widetilde{\mathbf{x}}; \mathbf{W}_1, \boldsymbol{b}_1, \mathbf{W}_2) = y \cdot \mathbf{x}.$$

*Proof of Lemma C.1.* We prove the claim by considering the two possible values of $y$.

First, suppose $y = 1$. Then

$$\mathbf{W}_1 \begin{bmatrix} \mathbf{x} \\ 1 \end{bmatrix} + \boldsymbol{b}_1 = \begin{bmatrix} \mathbf{x} \\ -\mathbf{x} \\ \mathbf{x} - 2M \cdot \mathbf{1}_d \\ -\mathbf{x} - 2M \cdot \mathbf{1}_d \end{bmatrix}.$$

Since $\|\mathbf{x}\|_\infty \leq M$, we have

$$\mathbf{x} - 2M \cdot \mathbf{1}_d \leq \mathbf{0}_d, \qquad -\mathbf{x} - 2M \cdot \mathbf{1}_d \leq \mathbf{0}_d$$

entrywise. Hence

$$\text{ReLU}\left(\mathbf{W}_1 \begin{bmatrix} \mathbf{x} \\ 1 \end{bmatrix} + \boldsymbol{b}_1 \right) = \begin{bmatrix} \text{ReLU}(\mathbf{x}) \\ \text{ReLU}(-\mathbf{x}) \\ \mathbf{0}_d \\ \mathbf{0}_d \end{bmatrix}.$$

Therefore,

$$f\left(\begin{bmatrix} \mathbf{x} \\ 1 \end{bmatrix}; \mathbf{W}_1, \boldsymbol{b}_1, \mathbf{W}_2 \right) = \text{ReLU}(\mathbf{x}) - \text{ReLU}(-\mathbf{x}) = \mathbf{x}.$$

Here we used the identity $\text{ReLU}(a) - \text{ReLU}(-a) = a$ entrywise.

Next, suppose $y = -1$. Then

$$\mathbf{W}_1 \begin{bmatrix} \mathbf{x} \\ -1 \end{bmatrix} + \boldsymbol{b}_1 = \begin{bmatrix} \mathbf{x} - 2M \cdot \mathbf{1}_d \\ -\mathbf{x} - 2M \cdot \mathbf{1}_d \\ \mathbf{x} \\ -\mathbf{x} \end{bmatrix}.$$

Again, since $\|\mathbf{x}\|_\infty \le M$, the first two blocks are entrywise non-positive. Thus

$$\text{ReLU}\left(\mathbf{W}_1 \begin{bmatrix} \mathbf{x} \\ -1 \end{bmatrix} + \boldsymbol{b}_1 \right) = \begin{bmatrix} \mathbf{0}_d \\ \mathbf{0}_d \\ \text{ReLU}(\mathbf{x}) \\ \text{ReLU}(-\mathbf{x}) \end{bmatrix}.$$

Therefore,

$$f\left(\begin{bmatrix} \mathbf{x} \\ -1 \end{bmatrix}; \mathbf{W}_1, \boldsymbol{b}_1, \mathbf{W}_2 \right) = -\text{ReLU}(\mathbf{x}) + \text{ReLU}(-\mathbf{x}) = -\mathbf{x}.$$

Combining the two cases gives

$$f\left(\begin{bmatrix} \mathbf{x} \\ y \end{bmatrix}; \mathbf{W}_1, \boldsymbol{b}_1, \mathbf{W}_2 \right) = y \cdot \mathbf{x},$$

for every $y \in \{-1, 1\}$ and every $\mathbf{x}$ satisfying $\|\mathbf{x}\|_\infty \le M$. $\qquad\square$

Consequently, if the input examples are given as concatenated vectors $[\mathbf{x}_i^\top, y_i]^\top$, an embedding layer of the above form maps them exactly to $\mathbf{z}_i = y_i \cdot \mathbf{x}_i$ before they are fed into the attention layers. Since embedding layers are standard components of transformer architectures, the use of $\mathbf{z}_i = y_i \cdot \mathbf{x}_i$ in Eq. (1) does not prevent the result from applying to the concatenated-input formulation.

## D. Proof of Theorem 3.1

In this section, we provide a detailed proof for Theorem 3.1. As we have mentioned in the discussions of Theorem 3.1, the parameterization of $\mathbf{W}_l$ can be further relaxed to the form as $\mathbf{W}_l = \begin{bmatrix} \mathbf{0}_{d\times d} & -\widetilde{\beta} \cdot \mathbf{I}_d \\ \mathbf{A}_{1,l} & \mathbf{A}_{2,l} \end{bmatrix}$, with $\widetilde{\beta}$ being any positive scalar. Therefore, instead of directly proving Theorem 3.1, we prove a generalized version which allows $\mathbf{W}_l$ to be parameterized as above. The presentation and detailed proof is provided in the following.

**Theorem D.1** (Generalized version of Theorem 3.1). *Consider an $L$-layer transformers $\text{TF}(\cdot)$ as defined in (3), whose parameter matrices $(\mathbf{V}_{0:L-1}, \mathbf{W}_{0:L-1})$ satisfy that for $l = 0, 1, \dots, L-1$,*

$$\mathbf{V}_l = \begin{bmatrix} \mathbf{0}_{d\times d} & \mathbf{0}_{d\times d} \\ \widetilde{\alpha}_l \cdot \mathbf{I}_d & \mathbf{0}_{d\times d} \end{bmatrix}, \quad \mathbf{W}_l = \begin{bmatrix} \mathbf{0}_{d\times d} & -\widetilde{\beta} \cdot \mathbf{I}_d \\ \mathbf{A}_{1,l} & \mathbf{A}_{2,l} \end{bmatrix},$$

*where $\mathbf{A}_{1,l}, \mathbf{A}_{2,l}$ are arbitrary $d \times d$-dimensional real matrices, and $\widetilde{\alpha}_l, \widetilde{\beta} > 0$. For each $l \in [L]$, let $\boldsymbol{\theta}_l = [\mathbf{Z}_l]_{d+1:2d,n+1}$. Then the sequence $\{\boldsymbol{\theta}_l\}_{l=1}^L$ corresponds to iterates obtained by implementing $L$-steps normalized gradient descent on $\widetilde{\mathcal{L}}_{\text{ICL}}(\boldsymbol{\theta})$: for $l = 0, 1, \dots, L-1$,*

$$\boldsymbol{\theta}_{l+1} = \boldsymbol{\theta}_l - \frac{\widetilde{\alpha}_l}{\widetilde{\beta}} \frac{\nabla \widetilde{\mathcal{L}}_{\text{ICL}}(\boldsymbol{\theta}_l)}{\widetilde{\mathcal{L}}_{\text{ICL}}(\boldsymbol{\theta}_l)}, \tag{9}$$

where $\widetilde{\alpha}_l/\widetilde{\beta}_l$ denotes the learning rate of $l$-th iterative step. Specifically, the $\widetilde{\mathcal{L}}_{\text{ICL}}$ denotes the in-context loss defined on the rescaled context dataset $\{(\widetilde{\beta} \cdot \mathbf{x}_i, y_i)\}_{i=1}^n$ as

$$\widetilde{\mathcal{L}}_{\text{ICL}}(\boldsymbol{\theta}) = \frac{1}{n} \sum_{i=1}^n \ell(\langle \boldsymbol{\theta}, \widetilde{\beta} y_i \cdot \mathbf{x}_i \rangle).$$

It is evident that Theorem D.1 cover Theorem 3.1 when $\widetilde{\beta} = 1$. We first introduce a notation that $\mathbf{Z}_{\text{context}} = [\mathbf{z}_1, \ldots, \mathbf{z}_n] \in \mathbb{R}^{d \times n}$, where $\mathbf{z}_i = y_i \cdot \mathbf{x}_i$. Now, we are ready to prove Theorem D.1.

*Proof of Theorem D.1.* We define $\widetilde{\boldsymbol{\theta}}_l$ as the iterates of normalized gradient descent with 0 initialization, namely,

$$\widetilde{\boldsymbol{\theta}}_{l+1} = \widetilde{\boldsymbol{\theta}}_l - \widetilde{\alpha} \frac{\nabla \widetilde{\mathcal{L}}_{\text{ICL}}(\widetilde{\boldsymbol{\theta}}_l)}{\widetilde{\mathcal{L}}_{\text{ICL}}(\widetilde{\boldsymbol{\theta}}_l)}, \qquad l = 0, 1, \ldots, L-1,$$

where $\widetilde{\boldsymbol{\theta}}_0 = \mathbf{0}_d$. In simple terms, the only distinction between $\widetilde{\boldsymbol{\theta}}_l$ and $\boldsymbol{\theta}_l$ is that they have different initializations. If $\boldsymbol{\theta}_0 = \mathbf{0}_d$, then $\widetilde{\boldsymbol{\theta}}_l = \boldsymbol{\theta}_l$ for all $l \in [L]$. In the next, we prove that $\mathbf{Z}_l$ can be formulated as

$$\mathbf{Z}_l = \begin{bmatrix} \mathbf{Z}_{\text{context}} & \mathbf{0}_d \\ \widetilde{\boldsymbol{\theta}}_l \mathbf{1}_n^\top & \boldsymbol{\theta}_l \end{bmatrix}. \tag{10}$$

Then it is evident that Theorem 3.1 follows directly from (10), and we proceed by induction. Notice that the initial condition $\mathbf{Z}_0$ satisfies (10). For the inductive hypothesis, we assume that $\mathbf{Z}_l$ satisfies (10) and then demonstrate that this property is preserved for $\mathbf{Z}_{l+1}$. For $\mathbf{Z}_l$ formulated in (10), we can calculate that

$$\begin{aligned}
\mathbf{Z}_l^\top \mathbf{W} \mathbf{Z}_l &= \begin{bmatrix} \mathbf{Z}_{\text{context}}^\top & \mathbf{1}_n \widetilde{\boldsymbol{\theta}}_l^\top \\ \mathbf{0}_d^\top & \boldsymbol{\theta}_l^\top \end{bmatrix} \begin{bmatrix} \mathbf{0}_{d \times d} & -\widetilde{\beta} \cdot \mathbf{I}_d \\ \mathbf{A}_1 & \mathbf{A}_2 \end{bmatrix} \begin{bmatrix} \mathbf{Z}_{\text{context}} & \mathbf{0}_d \\ \widetilde{\boldsymbol{\theta}}_l \mathbf{1}_n^\top & \boldsymbol{\theta}_l \end{bmatrix} \\
&= \begin{bmatrix} \mathbf{1}_n \widetilde{\boldsymbol{\theta}}_l^\top \mathbf{A}_1 & -\widetilde{\beta} \cdot \mathbf{Z}_{\text{context}}^\top + \mathbf{1}_n \widetilde{\boldsymbol{\theta}}_l^\top \mathbf{A}_2 \\ \boldsymbol{\theta}_l^\top \mathbf{A}_1 & \boldsymbol{\theta}_l^\top \mathbf{A}_2 \end{bmatrix} \begin{bmatrix} \mathbf{Z}_{\text{context}} & \mathbf{0}_d \\ \widetilde{\boldsymbol{\theta}}_l \mathbf{1}_n^\top & \boldsymbol{\theta}_l \end{bmatrix} \\
&= \begin{bmatrix} \mathbf{1}_n \widetilde{\boldsymbol{\theta}}_l^\top (\mathbf{A}_1 \mathbf{Z}_{\text{context}} + \mathbf{A}_2 \widetilde{\boldsymbol{\theta}}_l \mathbf{1}_n^\top) - \widetilde{\beta} \cdot \mathbf{Z}_{\text{context}}^\top \widetilde{\boldsymbol{\theta}}_l \mathbf{1}_n^\top & -\widetilde{\beta} \cdot \mathbf{Z}_{\text{context}}^\top \boldsymbol{\theta}_l + \mathbf{1}_n \widetilde{\boldsymbol{\theta}}_l^\top \mathbf{A}_2 \boldsymbol{\theta}_l \\ \boldsymbol{\theta}_l^\top \mathbf{A}_1 \mathbf{Z}_{\text{context}} + \boldsymbol{\theta}_l^\top \mathbf{A}_2 \widetilde{\boldsymbol{\theta}}_l \mathbf{1}_n^\top & \boldsymbol{\theta}_l^\top \mathbf{A}_2 \boldsymbol{\theta}_l \end{bmatrix}.
\end{aligned}$$

Notice that the softmax operation in the self-attention layer (2) is applied column-wise. Consequently, all rank-one blocks proportional to $\mathbf{1}_n$ yield identical values across coordinates, and hence can be omitted as they do not affect the softmax output. Moreover, due to the presence of the mask matrix $\mathbf{M}$, the attention weights corresponding to the last query column remain zero. Combining all these observations, we can calculate that

$$\text{softmax}(\mathbf{Z}_l^\top \mathbf{W} \mathbf{Z}_l + \mathbf{M}) = \begin{bmatrix} \frac{e^{-\langle \widetilde{\beta} \cdot \mathbf{z}_1, \widetilde{\boldsymbol{\theta}} \rangle}}{\sum_{i=1}^n e^{-\langle \widetilde{\beta} \cdot \mathbf{z}_i, \widetilde{\boldsymbol{\theta}} \rangle}} \mathbf{1}_n^\top & \frac{e^{-\langle \widetilde{\beta} \cdot \mathbf{z}_1, \boldsymbol{\theta} \rangle}}{\sum_{i=1}^n e^{-\langle \widetilde{\beta} \cdot \mathbf{z}_i, \boldsymbol{\theta} \rangle}} \\ \frac{e^{-\langle \widetilde{\beta} \cdot \mathbf{z}_2, \widetilde{\boldsymbol{\theta}} \rangle}}{\sum_{i=1}^n e^{-\langle \widetilde{\beta} \cdot \mathbf{z}_i, \widetilde{\boldsymbol{\theta}} \rangle}} \mathbf{1}_n^\top & \frac{e^{-\langle \widetilde{\beta} \cdot \mathbf{z}_2, \boldsymbol{\theta} \rangle}}{\sum_{i=1}^n e^{-\langle \widetilde{\beta} \cdot \mathbf{z}_i, \boldsymbol{\theta} \rangle}} \\ \cdots & \cdots \\ \frac{e^{-\langle \widetilde{\beta} \cdot \mathbf{z}_n, \widetilde{\boldsymbol{\theta}} \rangle}}{\sum_{i=1}^n e^{-\langle \widetilde{\beta} \cdot \mathbf{z}_i, \widetilde{\boldsymbol{\theta}} \rangle}} \mathbf{1}_n^\top & \frac{e^{-\langle \widetilde{\beta} \cdot \mathbf{z}_n, \boldsymbol{\theta} \rangle}}{\sum_{i=1}^n e^{-\langle \widetilde{\beta} \cdot \mathbf{z}_i, \boldsymbol{\theta} \rangle}} \\ \mathbf{0}_n^\top & 0 \end{bmatrix} = \begin{bmatrix} -\frac{\ell'(\langle \widetilde{\beta} \cdot \mathbf{z}_1, \widetilde{\boldsymbol{\theta}} \rangle)}{\widetilde{\mathcal{L}}_{\text{ICL}}(\widetilde{\boldsymbol{\theta}})} \mathbf{1}_n^\top & -\frac{\ell'(\langle \widetilde{\beta} \cdot \mathbf{z}_1, \boldsymbol{\theta} \rangle)}{\widetilde{\mathcal{L}}_{\text{ICL}}(\boldsymbol{\theta})} \\ -\frac{\ell'(\langle \widetilde{\beta} \cdot \mathbf{z}_2, \widetilde{\boldsymbol{\theta}} \rangle)}{\widetilde{\mathcal{L}}_{\text{ICL}}(\widetilde{\boldsymbol{\theta}})} \mathbf{1}_n^\top & -\frac{\ell'(\langle \widetilde{\beta} \cdot \mathbf{z}_2, \boldsymbol{\theta} \rangle)}{\widetilde{\mathcal{L}}_{\text{ICL}}(\boldsymbol{\theta})} \\ \cdots & \cdots \\ -\frac{\ell'(\langle \widetilde{\beta} \cdot \mathbf{z}_n, \widetilde{\boldsymbol{\theta}} \rangle)}{\widetilde{\mathcal{L}}_{\text{ICL}}(\widetilde{\boldsymbol{\theta}})} \mathbf{1}_n^\top & -\frac{\ell'(\langle \widetilde{\beta} \cdot \mathbf{z}_n, \boldsymbol{\theta} \rangle)}{\widetilde{\mathcal{L}}_{\text{ICL}}(\boldsymbol{\theta})} \\ \mathbf{0}_n^\top & 0 \end{bmatrix}.$$

Based on this result, it can be directly calculated that

$$\mathbf{Z}_{l+1} = \mathbf{Z}_l + \text{SA}(\mathbf{Z}_l; \mathbf{V}, \mathbf{W}) = \begin{bmatrix} \mathbf{Z}_{\text{context}} & \mathbf{0}_d \\ \widetilde{\boldsymbol{\theta}}_l \mathbf{1}_n^\top & \boldsymbol{\theta}_l \end{bmatrix} + \begin{bmatrix} \mathbf{0}_{d \times n} & \mathbf{0}_d \\ -\frac{\widetilde{\alpha}_l}{\widetilde{\beta}} \frac{\nabla \widetilde{\mathcal{L}}_{\text{ICL}}(\widetilde{\boldsymbol{\theta}})}{\widetilde{\mathcal{L}}_{\text{ICL}}(\widetilde{\boldsymbol{\theta}})} \mathbf{1}_n^\top & -\frac{\widetilde{\alpha}_l}{\widetilde{\beta}} \frac{\nabla \widetilde{\mathcal{L}}_{\text{ICL}}(\boldsymbol{\theta})}{\widetilde{\mathcal{L}}_{\text{ICL}}(\boldsymbol{\theta})} \end{bmatrix} = \begin{bmatrix} \mathbf{Z}_{\text{context}} & \mathbf{0}_d \\ \widetilde{\boldsymbol{\theta}}_{l+1} \mathbf{1}_n^\top & \boldsymbol{\theta}_{l+1} \end{bmatrix},$$

where the last equality holds by the iterative rules for $\boldsymbol{\theta}$ and $\widetilde{\boldsymbol{\theta}}$. We demonstrate that (10) holds for $\mathbf{Z}_{l+1}$ and hence complete the proof. $\square$

# E. Proof of Lemma 3.4 and Theorem 3.5

In this section, we provide a comprehensive proof for Lemma 3.4 and Theorem 3.5. We first introduce a notation regarding the blocks inner parameter matrix $\mathbf{V}$ and $\mathbf{W}$. To better align with the parameter patterns defined in Lemma 3.4, we express $\mathbf{V}$ and $\mathbf{W}$ into the form of four blocks as

$$\mathbf{V} = \begin{bmatrix} \mathbf{V}_{11} & \mathbf{V}_{12} \\ \mathbf{V}_{21} & \mathbf{V}_{22} \end{bmatrix}, \qquad \mathbf{W} = \begin{bmatrix} \mathbf{W}_{11} & \mathbf{W}_{12} \\ \mathbf{W}_{21} & \mathbf{W}_{22} \end{bmatrix},$$

where $\mathbf{V}_{k_1,k_2}, \mathbf{W}_{k_1,k_2} \in \mathbb{R}^{d \times d}$ for all $k_1, k_2 \in [2]$. Since Lemma 3.4 and Theorem 3.5 contain too many results, we separate these contents into several lemmas and theorems and prove them respectively. Specifically, the conclusion of Lemma 3.4 is separated into Lemma E.1 and E.2. Lemma E.1 provides the gradient calculations of (6) and (6), and demonstrates that except $\mathbf{V}_{21}$ and $\mathbf{W}_{12}$, other blocks of $\mathbf{V}$ and $\mathbf{W}$ always remain $\mathbf{0}$. Lemma E.2 demonstrates that $\mathbf{V}_{21}^{(t)} = C_1^{(t)}\mathbf{I}_d$ and $\mathbf{W}_{12}^{(t)} = -C_2^{(t)}\mathbf{I}_d$, and further provides the updating rules for $C_1^{(t)}$ and $C_2^{(t)}$. Theorem 3.5 is separated into Lemma E.5, E.7, and Theorem E.9, corresponding to the three conclusion respectively. Now, we start our proof.

**Lemma E.1** (Restatement of Lemma 3.4, part I). *For the blocks $\mathbf{V}_{11}, \mathbf{V}_{12}, \mathbf{V}_{22}, \mathbf{W}_{11}, \mathbf{W}_{21}, \mathbf{W}_{22}$, the gradient of loss with respect to them remain zero, implying that $\mathbf{V}_{11}^{(t)}, \mathbf{V}_{12}^{(t)}, \mathbf{V}_{22}^{(t)}, \mathbf{W}_{11}^{(t)}, \mathbf{W}_{21}^{(t)}, \mathbf{W}_{22}^{(t)} = \mathbf{0}_{d \times d}$ for all $t \geq 0$. For $\mathbf{V}_{21}$ and $\mathbf{W}_{12}$, their gradient can be expressed as*

$$\nabla_{\mathbf{V}_{21}}\mathcal{L}_{\mathrm{train}}(\mathbf{V}, \mathbf{W}) = -\mathbb{E}\left[\left(\frac{\alpha}{n}\sum_{i=1}^n e^{-\langle \mathbf{z}_i, \boldsymbol{\theta}_0\rangle}\mathbf{z}_i - \mathbf{V}_{21}\sum_{i=1}^n \mathbf{z}_i\mathbf{s}_i\right)\sum_{i=1}^n \mathbf{z}_i^\top \mathbf{s}_i\right];$$

$$\nabla_{\mathbf{W}_{12}}\mathcal{L}_{\mathrm{train}}(\mathbf{V}, \mathbf{W}) = -\mathbb{E}\left[\mathbf{Z}_{\mathrm{context}}\left(\mathrm{diag}(\mathbf{s}) - \mathbf{s}\mathbf{s}^\top\right)\mathbf{Z}_{\mathrm{context}}^\top \mathbf{V}_{21}^\top\left(\frac{\alpha}{n}\sum_{i=1}^n e^{-\langle \mathbf{z}_i, \boldsymbol{\theta}_0\rangle}\mathbf{z}_i - \mathbf{V}_{21}\sum_{i=1}^n \mathbf{z}_i\mathbf{s}_i\right)\boldsymbol{\theta}_0^\top\right].$$

*Proof of Lemma E.1.* Notice that the mask matrix $\mathbf{M}$ prevents attending to the last query column, resulting in that

$$\mathrm{softmax}(\mathbf{Z}^\top\mathbf{W}[\mathbf{0}_d, \boldsymbol{\theta}_0]^\top + \mathbf{M}) = \left[\mathrm{softmax}\left([\mathbf{Z}_{\mathrm{context}}^\top, \mathbf{0}_{n \times d}]\mathbf{W}[\mathbf{0}_d, \boldsymbol{\theta}_0]^\top\right), 0\right]$$
$$= \left[\mathrm{softmax}\left(\mathbf{Z}_{\mathrm{context}}^\top\mathbf{W}_{12}\boldsymbol{\theta}_0\right), 0\right] \in \mathbb{R}^{n+1},$$

where the last equation is directly simplified as the zero blocks in the quadratic form have no effect on the final results. By denoting $\mathbf{s} = \mathrm{softmax}\left(\mathbf{Z}_{\mathrm{context}}^\top\mathbf{W}_{12}\boldsymbol{\theta}_0\right) \in \mathbb{R}^d$, we can rewrite that

$$[\mathtt{SA}(\mathbf{Z}_0, \mathbf{V}, \mathbf{W})]_{d+1:2d,n+1} = \mathbf{V}_{21}\mathbf{Z}_{\mathrm{context}}\mathbf{s} + \mathbf{V}_{22}\mathbf{0}_{d \times n}\mathbf{s} = \mathbf{V}_{21}\mathbf{Z}_{\mathrm{context}}\mathbf{s}.$$

This further implies that

$$\boldsymbol{\theta}_1 = [\mathtt{TF}(\mathbf{Z}_0; \mathbf{V}, \mathbf{W}]_{d+1:2d,n+1} = \boldsymbol{\theta}_0 + [\mathtt{SA}(\mathbf{Z}_0, \mathbf{V}, \mathbf{W})]_{d+1:2d,n+1} = \boldsymbol{\theta}_0 + \mathbf{V}_{21}\mathbf{Z}_{\mathrm{context}}\mathbf{s}.$$

Following a similar calculations in Wang et al. (2024b), we can obtain that

$$\mathrm{d}\mathcal{L}_{\mathrm{train}}(\mathbf{V}, \mathbf{W}) = \mathbb{E}\left[-\left(\frac{\alpha}{n}\sum_{i=1}^n e^{-\langle \mathbf{z}_i, \boldsymbol{\theta}_0\rangle}\mathbf{z}_i - \mathbf{V}_{21}\mathbf{Z}_{\mathrm{context}}\mathbf{s}\right)^\top \mathrm{d}\mathbf{V}_{21}\mathbf{Z}_{\mathrm{context}}\mathbf{s}\right]$$
$$+ \mathbb{E}\left[-\left(\frac{\alpha}{n}\sum_{i=1}^n e^{-\langle \mathbf{z}_i, \boldsymbol{\theta}_0\rangle}\mathbf{z}_i - \mathbf{V}_{21}\mathbf{Z}_{\mathrm{context}}\mathbf{s}\right)^\top \mathbf{V}_{21}\mathbf{Z}_{\mathrm{context}}\mathrm{dsoftmax}\left(\mathbf{Z}_{\mathrm{context}}^\top\mathbf{W}_{12}\boldsymbol{\theta}_0\right)\right]$$
$$= \mathbb{E}\left[-\left(\frac{\alpha}{n}\sum_{i=1}^n e^{-\langle \mathbf{z}_i, \boldsymbol{\theta}_0\rangle}\mathbf{z}_i - \mathbf{V}_{21}\mathbf{Z}_{\mathrm{context}}\mathbf{s}\right)^\top \mathrm{d}\mathbf{V}_{21}\mathbf{Z}_{\mathrm{context}}\mathbf{s}\right]$$
$$+ \mathbb{E}\left[-\left(\frac{\alpha}{n}\sum_{i=1}^n e^{-\langle \mathbf{z}_i, \boldsymbol{\theta}_0\rangle}\mathbf{z}_i - \mathbf{V}_{21}\mathbf{Z}_{\mathrm{context}}\mathbf{s}\right)^\top \mathbf{V}_{21}\mathbf{Z}_{\mathrm{context}}\left(\mathrm{diag}(\mathbf{s}) - \mathbf{s}\mathbf{s}^\top\right)\mathbf{Z}_{\mathrm{context}}^\top\mathrm{d}\mathbf{W}_{12}\boldsymbol{\theta}_0\right].$$

From the differential calculation above, we directly conclude that only the gradients with respect to the blocks $\mathbf{V}_{21}$ and $\mathbf{W}_{12}$ are non-zero, while those of other blocks remain zero throughout the training process, as the loss is irrelevant with

them. Therefore, we conclude that for all $t \in \mathbb{N}$, $\mathbf{V}_{11}^{(t)}, \mathbf{V}_{12}^{(t)}, \mathbf{V}_{22}^{(t)}, \mathbf{W}_{11}^{(t)}, \mathbf{W}_{21}^{(t)}, \mathbf{V}_{22}^{(t)} = \mathbf{0}_{d \times d}$. For the block $\mathbf{V}_{21}$ and $\mathbf{W}_{12}$, their gradients can be expressed as

$$\nabla_{\mathbf{V}_{21}} \mathcal{L}_{\text{train}}(\mathbf{V}, \mathbf{W}) = -\mathbb{E}\left[ \left( \frac{\alpha}{n} \sum_{i=1}^{n} e^{-\langle \mathbf{z}_i, \boldsymbol{\theta}_0 \rangle} \mathbf{z}_i - \mathbf{V}_{21} \sum_{i=1}^{n} \mathbf{z}_i \mathbf{s}_i \right) \sum_{i=1}^{n} \mathbf{z}_i^\top \mathbf{s}_i \right];$$

$$\nabla_{\mathbf{W}_{12}} \mathcal{L}_{\text{train}}(\mathbf{V}, \mathbf{W}) = -\mathbb{E}\left[ \mathbf{Z}_{\text{context}} \left( \text{diag}(\mathbf{s}) - \mathbf{s}\mathbf{s}^\top \right) \mathbf{Z}_{\text{context}}^\top \mathbf{V}_{21}^\top \left( \frac{\alpha}{n} \sum_{i=1}^{n} e^{-\langle \mathbf{z}_i, \boldsymbol{\theta}_0 \rangle} \mathbf{z}_i - \mathbf{V}_{21} \sum_{i=1}^{n} \mathbf{z}_i \mathbf{s}_i \right) \boldsymbol{\theta}_0^\top \right].$$

This completes the proof. $\qquad\square$

Lemma E.1 demonstrates that except $\mathbf{V}_{21}$ and $\mathbf{W}_{12}$, other blocks of parameter matrices $\mathbf{V}$ and $\mathbf{W}$ remains zero throughout the training. To finish the proof for the specific patterns in Lemma 3.4, it suffices to show that there exist two time-dependent scalars $C_1^{(t)}$ and $C_2^{(t)}$, such that $\mathbf{V}_{21} = C_1^{(t)} \cdot \mathbf{I}_d$ and $\mathbf{W}_{12} = -C_2^{(t)} \cdot \mathbf{I}_d$ for all $t \geq 0$, which are demonstrated in the following Lemma E.2.

**Lemma E.2** (Restatement of Lemma 3.4, part II). *Under the same conditions of Theorem 3.5, there exist time dependent scalars $C_1^{(t)}$ and $C_2^{(t)}$ such that for $t \geq 0$,*

$$\mathbf{V}_{21}^{(t)} = C_1^{(t)} \cdot \mathbf{I}_d; \quad \mathbf{W}_{12}^{(t)} = -C_2^{(t)} \cdot \mathbf{I}_d.$$

*In addition, $C_1^{(t)}$ and $C_2^{(t)}$ follow the iterative rules as*

$$C_1^{(t+1)} = C_1^{(t)} + \frac{\eta \sigma^2}{d} \left[ \alpha e^{\frac{\sigma^2}{2}} \left( \frac{2}{\pi} + C_2^{(t)} \sigma^2 + \frac{d}{n} e^{C_2^{(t)} \sigma^2} + \frac{F_{1,\sigma}(C_2^{(t)})}{n} + \frac{F_{2,\sigma}(C_2^{(t)})}{d} \right) \right.$$

$$\left. - C_1^{(t)} \left( \frac{2}{\pi} + [C_2^{(t)}]^2 \sigma^2 + \frac{d}{n} e^{[C_2^{(t)}]^2 \sigma^2} + \frac{F_{3,\sigma}(C_2^{(t)})}{n} + \frac{F_{4,\sigma}(C_2^{(t)})}{d} \right) \right];$$

$$C_2^{(t+1)} = C_2^{(t)} + \frac{\eta C_1^{(t)} \sigma^4}{d} \left[ \alpha e^{\frac{\sigma^2}{2}} \left( 1 + \frac{d e^{C_2^{(t)} \sigma^2}}{n} + \frac{F_{5,\sigma}(C_2^{(t)})}{n} + \frac{F_{6,\sigma}(C_2^{(t)})}{d} \right) \right.$$

$$\left. - C_1^{(t)} C_2^{(t)} \left( 1 + \frac{d e^{[C_2^{(t)}]^2 \sigma^2}}{n} + \frac{F_{7,\sigma}(C_2^{(t)})}{n} + \frac{F_{8,\sigma}(C_2^{(t)})}{d} \right) \right]. \tag{11}$$

*Here, for each $k \in [8]$, $F_{k,\sigma}(\cdot)$ is continuously differentiable with respect to its argument, where $\sigma$ is treated as a fixed constant.*

These conclusions are proved by induction. It is straightforward to verify that they hold at $t = 0$, since the parameters are initialized as $\mathbf{V}^{(0)} = \mathbf{0}_{2d \times 2d}$ and $\mathbf{W}^{(0)} = \mathbf{0}_{2d \times 2d}$. To streamline the exposition, we reorganize the content of Lemma E.2 into two separate lemmas, namely Lemmas E.3 and E.4, which focus on the updates of $\mathbf{V}$ and $\mathbf{W}$, respectively. This decomposition allows us to present the relevant arguments more clearly and avoids an overly lengthy proof within a single lemma. Accordingly, we establish Lemmas E.3 and E.4 independently.

In the inductive step, we assume that the conclusions of both Lemma E.3 and Lemma E.4 hold at the current iteration, and then show that the conclusion of the lemma under consideration continues to hold at the next iteration. This procedure does not constitute circular reasoning. Indeed, all arguments could equivalently be organized into a single Lemma E.2. The inductive assumption simply reflects the fact that the parameter updates are coupled, and it suffices to verify that, starting from a valid initialization, the stated conclusions are preserved from one iteration to the next.

**Lemma E.3** (Restatement of Lemma E.2, part I). *Under the same conditions of Theorem 3.5, there exist a time dependent scalars $C_1^{(t)}$ such that for $t \geq 0$,*

$$\mathbf{V}_{21}^{(t)} = C_1^{(t)} \cdot \mathbf{I}_d.$$

*In addition, $C_1^{(t)}$ follows the iterative rules as*

$$C_1^{(t+1)} = C_1^{(t)} + \frac{\eta \sigma^2}{d} \left[ \alpha e^{\frac{\sigma^2}{2}} \left( \frac{2}{\pi} + C_2^{(t)} \sigma^2 + \frac{d}{n} e^{C_2^{(t)} \sigma^2} + \frac{F_{1,\sigma}(C_2^{(t)})}{n} + \frac{F_{2,\sigma}(C_2^{(t)})}{d} \right) \right.$$

$$-C_1^{(t)}\left(\frac{2}{\pi}+[C_2^{(t)}]^2\sigma^2+\frac{d}{n}e^{[C_2^{(t)}]^2\sigma^2}+\frac{F_{3,\sigma}(C_2^{(t)})}{n}+\frac{F_{4,\sigma}(C_2^{(t)})}{d}\right)\right].$$

Here, for each $k\in[8]$, $F_{k,\sigma}(\cdot)$ is continuously differentiable with respect to its argument, where $\sigma$ is treated as a fixed constant.

*Proof of Lemma E.3.* Suppose that $\mathbf{V}_{21}=C_1^{(t)}\cdot\mathbf{I}_d$ and $\mathbf{W}_{21}=-C_2^{(t)}\cdot\mathbf{I}_d$, then by Lemma E.1, we can calculate that

$$\nabla_{\mathbf{V}_{21}}\mathcal{L}_{\text{train}}(\mathbf{V}^{(t)},\mathbf{W}^{(t)})=-\mathbb{E}\left[\left(\frac{\alpha}{n}\sum_{i=1}^n e^{-\langle\mathbf{z}_i,\boldsymbol{\theta}_0\rangle}\mathbf{z}_i-\mathbf{V}_{21}^{(t)}\sum_{i=1}^n\mathbf{z}_i\mathbf{s}_i^{(t)}\right)\sum_{i=1}^n\mathbf{z}_i^\top\mathbf{s}_i^{(t)}\right]$$

$$=-\frac{\alpha}{n}\underbrace{\mathbb{E}\left[\sum_{i_1=1}^n\sum_{i_2=1}^n e^{-\langle\mathbf{z}_{i_1},\boldsymbol{\theta}_0\rangle}\mathbf{s}_{i_2}^{(t)}\mathbf{z}_{i_1}\mathbf{z}_{i_2}^\top\right]}_{I_1}+C_1^{(t)}\underbrace{\mathbb{E}\left[\sum_{i_1=1}^n\sum_{i_2=1}^n\mathbf{s}_{i_1}^{(t)}\mathbf{s}_{i_2}^{(t)}\mathbf{z}_{i_1}\mathbf{z}_{i_2}^\top\right]}_{I_2}.$$

In the next, we analyze the value of $I_1$ and $I_2$ respectively. For any given $\boldsymbol{\theta}_*$ and $\boldsymbol{\theta}_0$, let $\mathbf{A}$ be an orthogonal matrix defined as

$$\mathbf{A}=\left[\boldsymbol{\theta}_*,\frac{(\mathbf{I}_d-\boldsymbol{\theta}_*\boldsymbol{\theta}_*^\top)\boldsymbol{\theta}_0}{\|(\mathbf{I}_d-\boldsymbol{\theta}_*\boldsymbol{\theta}_*^\top)\boldsymbol{\theta}_0\|_2},\boldsymbol{\xi}_3,\ldots,\boldsymbol{\xi}_d\right]\in\mathbb{R}^d,$$

where $\boldsymbol{\xi}_3\ldots,\boldsymbol{\xi}_d$ are normalized vectors orthogonal to $\boldsymbol{\theta}_*$ and $\boldsymbol{\theta}_0$ (For notational consistency, we will use $\boldsymbol{\xi}_1$ and $\boldsymbol{\xi}_2$ to represent $\boldsymbol{\theta}_*$ and $\frac{(\mathbf{I}_d-\boldsymbol{\theta}_*\boldsymbol{\theta}_*^\top)\boldsymbol{\theta}_0}{\|(\mathbf{I}_d-\boldsymbol{\theta}_*\boldsymbol{\theta}_*^\top)\boldsymbol{\theta}_0\|_2}$, respectively, in certain summation contexts.). In addition, We define that $\widetilde{\mathbf{x}}_i=\mathbf{A}^\top\mathbf{x}_i$, which implies that $y_i=\text{sign}(\widetilde{\mathbf{x}}_{i,1})$. We further denote that $\widetilde{\mathbf{z}}_i=y_i\widetilde{\mathbf{x}}_i$. Then by Lemma H.22, we know that all coordinates of $\widetilde{\mathbf{z}}_i$ are independent with each other. The first coordinate $\widetilde{\mathbf{z}}_{i,1}$ follows a folded normal distribution, and other coordinates still follow normal distributions. Notice that we can re-write $\langle\boldsymbol{\theta}_0,\mathbf{z}_i\rangle=\langle\boldsymbol{\theta}_0,\boldsymbol{\theta}_*\rangle\widetilde{\mathbf{z}}_{i,1}+\|(\mathbf{I}_d-\boldsymbol{\theta}_*\boldsymbol{\theta}_*^\top)\boldsymbol{\theta}_0\|_2\widetilde{\mathbf{z}}_{i,2}$, which implies that both $e^{-\langle\boldsymbol{\theta}_0,\mathbf{z}_i\rangle}$ and $\mathbf{s}_i^{(t)}$ are all independent with the coordinates from $\widetilde{\mathbf{z}}_{i,3}$ to $\widetilde{\mathbf{z}}_{i,d}$. In addition, Lemma H.21 guarantees that $\widetilde{\mathbf{z}}_i$ are independent with $\boldsymbol{\theta}_*,\boldsymbol{\theta}_0$, and $\boldsymbol{\xi}_3,\ldots,\boldsymbol{\xi}_d$. Based on all these preliminaries, we calculate $I_1$ as

$$I_1=\mathbb{E}\left[\sum_{i_1=1}^n\sum_{i_2=1}^n e^{-\langle\mathbf{z}_{i_1},\boldsymbol{\theta}_0\rangle}\mathbf{s}_{i_2}^{(t)}\mathbf{A}\mathbf{A}^\top\mathbf{z}_{i_1}\mathbf{z}_{i_2}^\top\mathbf{A}\mathbf{A}^\top\right]=\mathbb{E}\left[\sum_{i_1=1}^n\sum_{i_2=1}^n e^{-\langle\mathbf{z}_{i_1},\boldsymbol{\theta}_0\rangle}\mathbf{s}_{i_2}^{(t)}\mathbf{A}\widetilde{\mathbf{z}}_{i_1}\widetilde{\mathbf{z}}_{i_2}^\top\mathbf{A}^\top\right]$$

$$=\underbrace{\mathbb{E}\left[\sum_{i_1=1}^n\sum_{i_2=1}^n e^{-\langle\mathbf{z}_{i_1},\boldsymbol{\theta}_0\rangle}\mathbf{s}_{i_2}^{(t)}\widetilde{\mathbf{z}}_{i_1,1}\widetilde{\mathbf{z}}_{i_2,1}\boldsymbol{\theta}_*\boldsymbol{\theta}_*^\top\right]}_{I_{1,1}}$$

$$+\underbrace{\mathbb{E}\left[\sum_{i_1=1}^n\sum_{i_2=1}^n e^{-\langle\mathbf{z}_{i_1},\boldsymbol{\theta}_0\rangle}\mathbf{s}_{i_2}^{(t)}\widetilde{\mathbf{z}}_{i_1,2}\widetilde{\mathbf{z}}_{i_2,2}\frac{(\mathbf{I}_d-\boldsymbol{\theta}_*\boldsymbol{\theta}_*^\top)\boldsymbol{\theta}_0(\mathbf{I}_d-\boldsymbol{\theta}_*\boldsymbol{\theta}_*^\top)\boldsymbol{\theta}_0^\top}{\|(\mathbf{I}_d-\boldsymbol{\theta}_*\boldsymbol{\theta}_*^\top)\boldsymbol{\theta}_0\|_2^2}\right]}_{I_{1,2}}$$

$$+\underbrace{\mathbb{E}\left[\sum_{i_1=1}^n\sum_{i_2=1}^n\sum_{j=3}^d e^{-\langle\mathbf{z}_{i_1},\boldsymbol{\theta}_0\rangle}\mathbf{s}_{i_2}^{(t)}\widetilde{\mathbf{z}}_{i_1,j}\widetilde{\mathbf{z}}_{i_2,j}\boldsymbol{\xi}_j\boldsymbol{\xi}_j^\top\right]}_{I_{1,3}}+\underbrace{\mathbb{E}\left[\sum_{i_1=1}^n\sum_{i_2=1}^n\sum_{j_1=1}^d\sum_{j_2\neq j_1}e^{-\langle\mathbf{z}_{i_1},\boldsymbol{\theta}_0\rangle}\mathbf{s}_{i_2}^{(t)}\widetilde{\mathbf{z}}_{i_1,j_1}\widetilde{\mathbf{z}}_{i_2,j_2}\boldsymbol{\xi}_{j_1}\boldsymbol{\xi}_{j_2}^\top\right]}_{I_{1,4}}. \quad (12)$$

Through the independence stated above, and the calculations demonstrated in Section H.1, we have

$$I_{1,1}=\mathbb{E}\left[\sum_{i_1=1}^n\sum_{i_2=1}^n e^{-\langle\mathbf{z}_{i_1},\boldsymbol{\theta}_0\rangle}\mathbf{s}_{i_2}^{(t)}\widetilde{\mathbf{z}}_{i_1,1}\widetilde{\mathbf{z}}_{i_2,1}\right]\mathbb{E}\left[\boldsymbol{\theta}_*\boldsymbol{\theta}_*^\top\right]=\frac{1}{d}\left(\frac{2n}{\pi}\sigma^2 e^{\frac{\sigma^2}{2}}+f_1(C_2^{(t)},\sigma)+\frac{nf_2(C_2^{(t)},\sigma)}{d}\right)\cdot\mathbf{I}_d,$$

where the expectation of the first term is derived in Lemma H.3, with $f_1,f_2$ being two analytic functions of $C_2^{(t)}$ and $\sigma$, and $\mathbb{E}[\boldsymbol{\theta}_*\boldsymbol{\theta}_*^\top]=\mathbf{I}_d/d$ as demonstrated in Lemma H.20. Similarly, we also have

$$I_{1,2}=\mathbb{E}\left[\sum_{i_1=1}^n\sum_{i_2=1}^n e^{-\langle\mathbf{z}_{i_1},\boldsymbol{\theta}_0\rangle}\mathbf{s}_{i_2}^{(t)}\widetilde{\mathbf{z}}_{i_1,2}\widetilde{\mathbf{z}}_{i_2,2}\right]\mathbb{E}\left[\frac{(\mathbf{I}_d-\boldsymbol{\theta}_*\boldsymbol{\theta}_*^\top)\boldsymbol{\theta}_0(\mathbf{I}_d-\boldsymbol{\theta}_*\boldsymbol{\theta}_*^\top)\boldsymbol{\theta}_0^\top}{\|(\mathbf{I}_d-\boldsymbol{\theta}_*\boldsymbol{\theta}_*^\top)\boldsymbol{\theta}_0\|_2^2}\right]$$

$$= \frac{1}{d}\left(nC_2^{(t)}\sigma^4 e^{\frac{\sigma^2}{2}} + f_3(C_2^{(t)},\sigma) + \frac{nf_4(C_2^{(t)},\sigma)}{d}\right)\cdot\mathbf{I}_d,$$

where the expectation of the first term is derived in Lemma H.1. For $I_{1,3}$, we have

$$I_{1,3} = \sum_{j=3}^{d}\sum_{i=1}^{n}\mathbb{E}\big[e^{-\langle\mathbf{z}_i,\boldsymbol{\theta}_0\rangle}\mathbf{s}_i^{(t)}\big]\mathbb{E}[\widetilde{\mathbf{z}}_{i,j}^2]\mathbb{E}\big[\boldsymbol{\xi}_j\boldsymbol{\xi}_j^\top\big] + \sum_{i_1=1}^{n}\sum_{i_2\neq i_1}\sum_{j=3}^{d}\mathbb{E}\big[e^{-\langle\mathbf{z}_{i_1},\boldsymbol{\theta}_0\rangle}\mathbf{s}_{i_2}^{(t)}\big]\mathbb{E}[\widetilde{\mathbf{z}}_{i_1,j}\widetilde{\mathbf{z}}_{i_2,j}]\mathbb{E}\big[\boldsymbol{\xi}_j\boldsymbol{\xi}_j^\top\big]$$

$$= \sum_{j=3}^{d}\sum_{i=1}^{n}\mathbb{E}\big[e^{-\langle\mathbf{z}_i,\boldsymbol{\theta}_0\rangle}\mathbf{s}_i^{(t)}\big]\mathbb{E}[\widetilde{\mathbf{z}}_{i,j}^2]\mathbb{E}\big[\boldsymbol{\xi}_j\boldsymbol{\xi}_j^\top\big] = \frac{d-2}{d}\left(\sigma^2 e^{C_2^{(t)}\sigma^2+\frac{1}{2}\sigma^2} + \frac{f_5(C_2^{(t)},\sigma)}{n} + \frac{f_6(C_2^{(t)},\sigma)}{d}\right)\cdot\mathbf{I}_d,$$

where the second equation holds as $\mathbb{E}[\widetilde{\mathbf{z}}_{i_1,j}\widetilde{\mathbf{z}}_{i_2,j}] = \mathbb{E}[\widetilde{\mathbf{z}}_{i_1,j}]\mathbb{E}[\widetilde{\mathbf{z}}_{i_2,j}] = 0$, and the expectation of the last equation is given in Lemma H.5. Lastly, for the term $I_{1,4}$, by the independence among these random variables and the fact that each $\boldsymbol{\xi}_j$ is symmetric, we have

$$I_{1,4} = \sum_{i_1=1}^{n}\sum_{i_2=1}^{n}\sum_{j_1=1}^{d}\sum_{j_2\neq j_1}\mathbb{E}\big[e^{-\langle\mathbf{z}_{i_1},\boldsymbol{\theta}_0\rangle}\mathbf{s}_{i_2}^{(t)}\widetilde{\mathbf{z}}_{i_1,j_1}\widetilde{\mathbf{z}}_{i_2,j_2}\big]\mathbb{E}[\boldsymbol{\xi}_{j_1}]\mathbb{E}[\boldsymbol{\xi}_{j_2}^\top] = \mathbf{0}_{d\times d}.$$

Combining these results into (12), we obtain that

$$I_1 = \frac{n\sigma^2 e^{\frac{\sigma^2}{2}}}{d}\left(\frac{2}{\pi} + C_2^{(t)}\sigma^2 + \frac{d}{n}e^{C_2^{(t)}\sigma^2} + \frac{F_{1,\sigma}(C_2^{(t)})}{n} + \frac{F_{2,\sigma}(C_2^{(t)})}{d}\right)\mathbf{I}_d. \tag{13}$$

Similarly, we can also separate $I_2$ as

$$I_2 = \mathbb{E}\left[\sum_{i_1=1}^{n}\sum_{i_2=1}^{n}\mathbf{s}_{i_1}^{(t)}\mathbf{s}_{i_2}^{(t)}\mathbf{A}\mathbf{A}^\top\mathbf{z}_{i_1}\mathbf{z}_{i_2}^\top\mathbf{A}\mathbf{A}^\top\right] = \mathbb{E}\left[\sum_{i_1=1}^{n}\sum_{i_2=1}^{n}\mathbf{s}_{i_1}^{(t)}\mathbf{s}_{i_2}^{(t)}\mathbf{A}\widetilde{\mathbf{z}}_{i_1}\widetilde{\mathbf{z}}_{i_2}^\top\mathbf{A}^\top\right]$$

$$= \underbrace{\mathbb{E}\left[\sum_{i_1=1}^{n}\sum_{i_2=1}^{n}\mathbf{s}_{i_1}^{(t)}\mathbf{s}_{i_2}^{(t)}\widetilde{\mathbf{z}}_{i_1,1}\widetilde{\mathbf{z}}_{i_2,1}\boldsymbol{\theta}_*\boldsymbol{\theta}_*^\top\right]}_{I_{2,1}} + \underbrace{\mathbb{E}\left[\sum_{i_1=1}^{n}\sum_{i_2=1}^{n}\mathbf{s}_{i_1}^{(t)}\mathbf{s}_{i_2}^{(t)}\widetilde{\mathbf{z}}_{i_1,2}\widetilde{\mathbf{z}}_{i_2,2}\frac{(\mathbf{I}_d - \boldsymbol{\theta}_*\boldsymbol{\theta}_*^\top)\boldsymbol{\theta}_0(\mathbf{I}_d - \boldsymbol{\theta}_*\boldsymbol{\theta}_*^\top)\boldsymbol{\theta}_0^\top}{\|(\mathbf{I}_d - \boldsymbol{\theta}_*\boldsymbol{\theta}_*^\top)\boldsymbol{\theta}_0\|_2^2}\right]}_{I_{2,2}}$$

$$+ \underbrace{\mathbb{E}\left[\sum_{i_1=1}^{n}\sum_{i_2=1}^{n}\sum_{j=3}^{d}\mathbf{s}_{i_1}^{(t)}\mathbf{s}_{i_2}^{(t)}\widetilde{\mathbf{z}}_{i_1,j}\widetilde{\mathbf{z}}_{i_2,j}\boldsymbol{\xi}_j\boldsymbol{\xi}_j^\top\right]}_{I_{2,3}} + \underbrace{\mathbb{E}\left[\sum_{i_1=1}^{n}\sum_{i_2=1}^{n}\sum_{j_1=1}^{d}\sum_{j_2\neq j_1}\mathbf{s}_{i_1}^{(t)}\mathbf{s}_{i_2}^{(t)}\widetilde{\mathbf{z}}_{i_1,j_1}\widetilde{\mathbf{z}}_{i_2,j_2}\boldsymbol{\xi}_{j_1}\boldsymbol{\xi}_{j_2}^\top\right]}_{I_{2,4}}. \tag{14}$$

We calculate each term following a similar procedure. For $I_{2,1}$, we have

$$I_{2,1} = \mathbb{E}\left[\sum_{i_1=1}^{n}\sum_{i_2=1}^{n}\mathbf{s}_{i_1}^{(t)}\mathbf{s}_{i_2}^{(t)}\widetilde{\mathbf{z}}_{i_1,1}\widetilde{\mathbf{z}}_{i_2,1}\right]\mathbb{E}\big[\boldsymbol{\theta}_*\boldsymbol{\theta}_*^\top\big] = \frac{\sigma^2}{d}\left(\frac{2}{\pi} + \frac{f_7(C_2^{(t)},\sigma)}{n} + \frac{f_8(C_2^{(t)},\sigma)}{d}\right)\cdot\mathbf{I}_d,$$

where the expectation is provided in Lemma H.4. For $I_{2,2}$, we have

$$I_{2,2} = \mathbb{E}\left[\sum_{i_1=1}^{n}\sum_{i_2=1}^{n}\mathbf{s}_{i_1}^{(t)}\mathbf{s}_{i_2}^{(t)}\widetilde{\mathbf{z}}_{i_1,2}\widetilde{\mathbf{z}}_{i_2,2}\right]\mathbb{E}\left[\frac{(\mathbf{I}_d - \boldsymbol{\theta}_*\boldsymbol{\theta}_*^\top)\boldsymbol{\theta}_0(\mathbf{I}_d - \boldsymbol{\theta}_*\boldsymbol{\theta}_*^\top)\boldsymbol{\theta}_0^\top}{\|(\mathbf{I}_d - \boldsymbol{\theta}_*\boldsymbol{\theta}_*^\top)\boldsymbol{\theta}_0\|_2^2}\right]$$

$$= \frac{\sigma^2}{d}\left([C_2^{(t)}]^2\sigma^2 + f_9(C_2^{(t)},\sigma) + \frac{nf_{10}(C_2^{(t)},\sigma)}{d}\right)\cdot\mathbf{I}_d,$$

where the expectation is provided in Lemma H.2. For $I_{2,3}$, we have

$$I_{2,3} = \sum_{j=3}^{d}\sum_{i=1}^{n}\mathbb{E}\big[(\mathbf{s}_i^{(t)})^2\big]\mathbb{E}[\widetilde{\mathbf{z}}_{i,j}^2]\mathbb{E}\big[\boldsymbol{\xi}_j\boldsymbol{\xi}_j^\top\big] + \sum_{i_1=1}^{n}\sum_{i_2\neq i_1}\sum_{j=3}^{d}\mathbb{E}\big[\mathbf{s}_{i_1}^{(t)}\mathbf{s}_{i_2}^{(t)}\big]\mathbb{E}[\widetilde{\mathbf{z}}_{i_1,j}\widetilde{\mathbf{z}}_{i_2,j}]\mathbb{E}\big[\boldsymbol{\xi}_j\boldsymbol{\xi}_j^\top\big]$$

$$= \sum_{j=3}^{d} \sum_{i=1}^{n} \mathbb{E}\big[(\mathbf{s}_i^{(t)})^2\big] \mathbb{E}[\widetilde{z}_{i,j}^2] \mathbb{E}\big[\boldsymbol{\xi}_j \boldsymbol{\xi}_j^\top\big] = \frac{(d-2)\sigma^2}{d} \left( e^{[C_2^{(t)}]^2 \sigma^2} + \frac{f_{11}(C_2^{(t)}, \sigma)}{n} + \frac{f_{12}(C_2^{(t)}, \sigma)}{d} \right) \cdot \mathbf{I}_d,$$

where the expectation is provided in Lemma H.6. In addition, $I_{2,4} = \mathbf{0}_{d \times d}$ by the symmetry of $\boldsymbol{\xi}_j$. Substituting these results into (14), we obtain that

$$I_2 = \frac{\sigma^2}{d} \left( \frac{2}{\pi} + [C_2^{(t)}]^2 \sigma^2 + \frac{d}{n} e^{[C_2^{(t)}]^2 \sigma^2} + \frac{F_{3,\sigma}(C_2^{(t)})}{n} + \frac{F_{4,\sigma}(C_2^{(t)})}{d} \right) \mathbf{I}_d. \tag{15}$$

Hence, we prove the induction that by assuming at $t$-th iteration, $\mathbf{V}_{21}^{(t)} = C_1^{(t)} \cdot \mathbf{I}_d$ and $\mathbf{W}_{12}^{(t)} = -C_2^{(t)} \cdot \mathbf{I}_d$, the gradient $\nabla_{\mathbf{V}_{21}} \mathcal{L}_{\text{train}}(\mathbf{V}^{(t)}, \mathbf{W}^{(t)})$ is also proportional to the identity matrix $\mathbf{I}_d$. In addition, (13) and (15) establish the iterative rule for the coefficient $C_1^{(t)}$ as

$$\begin{aligned}
C_1^{(t+1)} =& C_1^{(t)} + \frac{\eta \sigma^2}{d} \left[ \alpha e^{\frac{\sigma^2}{2}} \left( \frac{2}{\pi} + C_2^{(t)} \sigma^2 + \frac{d}{n} e^{C_2^{(t)} \sigma^2} + \frac{F_{1,\sigma}(C_2^{(t)})}{n} + \frac{F_{2,\sigma}(C_2^{(t)})}{d} \right) \right. \\
& \left. - C_1^{(t)} \left( \frac{2}{\pi} + [C_2^{(t)}]^2 \sigma^2 + \frac{d}{n} e^{[C_2^{(t)}]^2 \sigma^2} + \frac{F_{3,\sigma}(C_2^{(t)})}{n} + \frac{F_{4,\sigma}(C_2^{(t)})}{d} \right) \right].
\end{aligned}$$

This completes the proof. $\qquad \square$

**Lemma E.4** (Restatement of Lemma E.2, part II). *Under the same conditions of Theorem 3.5, there exist a time dependent scalar $C_2^{(t)}$ such that for $t \geq 0$,*

$$\mathbf{W}_{12}^{(t)} = -C_2^{(t)} \cdot \mathbf{I}_d.$$

*In addition, $C_2^{(t)}$ follows the iterative rules as*

$$\begin{aligned}
C_2^{(t+1)} =& C_2^{(t)} + \frac{\eta C_1^{(t)} \sigma^4}{d} \left[ \alpha e^{\frac{\sigma^2}{2}} \left( 1 + \frac{d e^{C_2^{(t)} \sigma^2}}{n} + \frac{F_{5,\sigma}(C_2^{(t)})}{n} + \frac{F_{6,\sigma}(C_2^{(t)})}{d} \right) \right. \\
& \left. - C_1^{(t)} C_2^{(t)} \left( 1 + \frac{d e^{[C_2^{(t)}]^2 \sigma^2}}{n} + \frac{F_{7,\sigma}(C_2^{(t)})}{n} + \frac{F_{8,\sigma}(C_2^{(t)})}{d} \right) \right].
\end{aligned}$$

*Here, for each $k \in [8]$, $F_{k,\sigma}(\cdot)$ is continuously differentiable with respect to its argument, where $\sigma$ is treated as a fixed constant.*

*Proof of Lemma E.4.* Suppose that $\mathbf{V}_{21} = C_1^{(t)} \cdot \mathbf{I}_d$ and $\mathbf{W}_{21} = -C_2^{(t)} \cdot \mathbf{I}_d$, then by Lemma E.1, we can calculate that

$$\begin{aligned}
& \nabla_{\mathbf{W}_{12}} \mathcal{L}_{\text{train}}(\mathbf{V}^{(t)}, \mathbf{W}^{(t)}) \\
=& -\mathbb{E}\left[ \mathbf{Z}_{\text{context}} \left( \text{diag}(\mathbf{s}^{(t)}) - \mathbf{s}^{(t)}(\mathbf{s}^{(t)})^\top \right) \mathbf{Z}_{\text{context}}^\top (\mathbf{V}_{21}^{(t)})^\top \left( \frac{\alpha}{n} \sum_{i=1}^{n} e^{-\langle \mathbf{z}_i, \boldsymbol{\theta}_0 \rangle} \mathbf{z}_i - \mathbf{V}_{21}^{(t)} \sum_{i=1}^{n} \mathbf{z}_i \mathbf{s}_i^{(t)} \right) \boldsymbol{\theta}_0 \right] \\
=& -\frac{\alpha C_1^{(t)}}{n} \underbrace{\mathbb{E}\left[ \sum_{i=1}^{n} \sum_{i_1=1}^{n} \sum_{i_2=1}^{n} e^{-\langle \mathbf{z}_i, \boldsymbol{\theta}_0 \rangle} \mathbf{s}_{i_1}^{(t)} \mathbf{s}_{i_2}^{(t)} (\mathbf{z}_{i_1} - \mathbf{z}_{i_2}) \mathbf{z}_{i_1}^\top \mathbf{z}_i \boldsymbol{\theta}_0^\top \right]}_{I_3} \\
& + C_1^2(t) \underbrace{\mathbb{E}\left[ \sum_{i=1}^{n} \sum_{i_1=1}^{n} \sum_{i_2=1}^{n} \mathbf{s}_i^{(t)} \mathbf{s}_{i_1}^{(t)} \mathbf{s}_{i_2}^{(t)} (\mathbf{z}_{i_1} - \mathbf{z}_{i_2}) \mathbf{z}_{i_1}^\top \mathbf{z}_i \boldsymbol{\theta}_0^\top \right]}_{I_4}
\end{aligned}$$

Utilizing the same definition of the orthogonal matrix $\mathbf{A}$, $\widetilde{\mathbf{z}}_i$ for all $i \in [n]$, and $\boldsymbol{\xi}_j$ for all $j \in [d]$ as in the proof of Lemma E.3, we have

$$I_3 = \mathbb{E}\left[ \sum_{i=1}^{n} \sum_{i_1=1}^{n} \sum_{i_2=1}^{n} e^{-\langle \mathbf{z}_i, \boldsymbol{\theta}_0 \rangle} \mathbf{s}_{i_1}^{(t)} \mathbf{s}_{i_2}^{(t)} \mathbf{A} \mathbf{A}^\top (\mathbf{z}_{i_1} - \mathbf{z}_{i_2}) \mathbf{z}_{i_1}^\top \mathbf{A} \mathbf{A}^\top \mathbf{z}_i \boldsymbol{\theta}_0^\top \right]$$

$$=\mathbb{E}\left[\sum_{i=1}^{n}\sum_{i_1=1}^{n}\sum_{i_2=1}^{n}e^{-\langle\mathbf{z}_i,\boldsymbol{\theta}_0\rangle}\mathbf{s}_{i_1}^{(t)}\mathbf{s}_{i_2}^{(t)}\mathbf{A}(\widetilde{\mathbf{z}}_{i_1}-\widetilde{\mathbf{z}}_{i_2})\widetilde{\mathbf{z}}_{i_1}^{\top}\widetilde{\mathbf{z}}_i\boldsymbol{\theta}_0^{\top}\right]$$

$$=\mathbb{E}\left[\sum_{i=1}^{n}\sum_{i_1=1}^{n}\sum_{i_2=1}^{n}\sum_{j=1}^{d}e^{-\langle\mathbf{z}_i,\boldsymbol{\theta}_0\rangle}\mathbf{s}_{i_1}^{(t)}\mathbf{s}_{i_2}^{(t)}(\widetilde{\mathbf{z}}_{i_1,j}-\widetilde{\mathbf{z}}_{i_2,j})\widetilde{\mathbf{z}}_{i_1}^{\top}\widetilde{\mathbf{z}}_i\boldsymbol{\xi}_j\boldsymbol{\theta}_0^{\top}\right]$$

$$=\mathbb{E}\left[\|(\mathbf{I}_d-\boldsymbol{\theta}_*\boldsymbol{\theta}_*^{\top})\boldsymbol{\theta}_0\|_2\sum_{i=1}^{n}\sum_{i_1=1}^{n}\sum_{i_2=1}^{n}e^{-\langle\mathbf{z}_i,\boldsymbol{\theta}_0\rangle}\mathbf{s}_{i_1}^{(t)}\mathbf{s}_{i_2}^{(t)}(\widetilde{\mathbf{z}}_{i_1,2}-\widetilde{\mathbf{z}}_{i_2,2})\widetilde{\mathbf{z}}_{i_1}^{\top}\widetilde{\mathbf{z}}_i\boldsymbol{\xi}_2\boldsymbol{\xi}_2^{\top}\right]$$

$$+\mathbb{E}\left[\sum_{i=1}^{n}\sum_{i_1=1}^{n}\sum_{i_2=1}^{n}\sum_{j\neq 2}e^{-\langle\mathbf{z}_i,\boldsymbol{\theta}_0\rangle}\mathbf{s}_{i_1}^{(t)}\mathbf{s}_{i_2}^{(t)}(\widetilde{\mathbf{z}}_{i_1,j}-\widetilde{\mathbf{z}}_{i_2,j})\widetilde{\mathbf{z}}_{i_1}^{\top}\widetilde{\mathbf{z}}_i\boldsymbol{\xi}_j\boldsymbol{\theta}_0^{\top}\right]$$

$$=\frac{1}{d}\mathbb{E}\left[\|(\mathbf{I}_d-\boldsymbol{\theta}_*\boldsymbol{\theta}_*^{\top})\boldsymbol{\theta}_0\|_2\sum_{i=1}^{n}\sum_{i_1=1}^{n}\sum_{i_2=1}^{n}e^{-\langle\mathbf{z}_i,\boldsymbol{\theta}_0\rangle}\mathbf{s}_{i_1}^{(t)}\mathbf{s}_{i_2}^{(t)}(\widetilde{\mathbf{z}}_{i_1,2}-\widetilde{\mathbf{z}}_{i_2,2})\widetilde{\mathbf{z}}_{i_1}^{\top}\widetilde{\mathbf{z}}_i\right]\cdot\mathbf{I}_d. \tag{16}$$

Here the second term $\mathbb{E}\left[\sum_{i=1}^{n}\sum_{i_1=1}^{n}\sum_{i_2=1}^{n}\sum_{j\neq 2}e^{-\langle\mathbf{z}_i,\boldsymbol{\theta}_0\rangle}\mathbf{s}_{i_1}^{(t)}\mathbf{s}_{i_2}^{(t)}(\widetilde{\mathbf{z}}_{i_1,j}-\widetilde{\mathbf{z}}_{i_2,j})\widetilde{\mathbf{z}}_{i_1}^{\top}\widetilde{\mathbf{z}}_i\boldsymbol{\xi}_j\boldsymbol{\theta}_0^{\top}\right]$ because for any $j\neq 2$, we have $\mathbb{E}\left[e^{-\langle\mathbf{z}_i,\boldsymbol{\theta}_0\rangle}\mathbf{s}_{i_1}^{(t)}\mathbf{s}_{i_2}^{(t)}(\widetilde{\mathbf{z}}_{i_1,j}-\widetilde{\mathbf{z}}_{i_2,j})\widetilde{\mathbf{z}}_{i_1}^{\top}\widetilde{\mathbf{z}}_i\boldsymbol{\xi}_j\boldsymbol{\theta}_0^{\top}\right]=\mathbb{E}\left[e^{-\langle\mathbf{z}_i,\boldsymbol{\theta}_0\rangle}\mathbf{s}_{i_1}^{(t)}\mathbf{s}_{i_2}^{(t)}(\widetilde{\mathbf{z}}_{i_1,j}-\widetilde{\mathbf{z}}_{i_2,j})\widetilde{\mathbf{z}}_{i_1}^{\top}\widetilde{\mathbf{z}}_i\boldsymbol{\xi}_j\right]\mathbb{E}[\boldsymbol{\theta}_0^{\top}]=\mathbf{0}$ through a similar argument in Lemma E.3. Then we get (16) by plugging the definition that $\boldsymbol{\xi}_2=\frac{(\mathbf{I}_d-\boldsymbol{\theta}_*\boldsymbol{\theta}_*^{\top})\boldsymbol{\theta}_0}{\|(\mathbf{I}_d-\boldsymbol{\theta}_*\boldsymbol{\theta}_*^{\top})\boldsymbol{\theta}_0\|_2}$, and the fact that $\mathbb{E}[\boldsymbol{\xi}_j\boldsymbol{\xi}_j^{\top}]=\frac{1}{d}\mathbf{I}_d$. This demonstrates that $I_3$ is always proportional to $\mathbf{I}_d$. In the next, we calculate the coefficient in (16). We first separate the term $\mathbb{E}\left[\|(\mathbf{I}_d-\boldsymbol{\theta}_*\boldsymbol{\theta}_*^{\top})\boldsymbol{\theta}_0\|_2\sum_{i=1}^{n}\sum_{i_1=1}^{n}\sum_{i_2=1}^{n}e^{-\langle\mathbf{z}_i,\boldsymbol{\theta}_0\rangle}\mathbf{s}_{i_1}^{(t)}\mathbf{s}_{i_2}^{(t)}(\widetilde{\mathbf{z}}_{i_1,2}-\widetilde{\mathbf{z}}_{i_2,2})\widetilde{\mathbf{z}}_{i_1}^{\top}\widetilde{\mathbf{z}}_i\right]$ as

$$\mathbb{E}\left[\|(\mathbf{I}_d-\boldsymbol{\theta}_*\boldsymbol{\theta}_*^{\top})\boldsymbol{\theta}_0\|_2\sum_{i=1}^{n}\sum_{i_1=1}^{n}\sum_{i_2=1}^{n}e^{-\langle\mathbf{z}_i,\boldsymbol{\theta}_0\rangle}\mathbf{s}_{i_1}^{(t)}\mathbf{s}_{i_2}^{(t)}(\widetilde{\mathbf{z}}_{i_1,2}-\widetilde{\mathbf{z}}_{i_2,2})\widetilde{\mathbf{z}}_{i_1}^{\top}\widetilde{\mathbf{z}}_i\right]$$

$$=\underbrace{\mathbb{E}\left[\|(\mathbf{I}_d-\boldsymbol{\theta}_*\boldsymbol{\theta}_*^{\top})\boldsymbol{\theta}_0\|_2\sum_{i=1}^{n}\sum_{i_1=1}^{n}e^{-\langle\mathbf{z}_i,\boldsymbol{\theta}_0\rangle}\mathbf{s}_{i_1}^{(t)}\widetilde{\mathbf{z}}_{i_1,2}\widetilde{\mathbf{z}}_{i_1,1}\widetilde{\mathbf{z}}_{i,1}\right]}_{I_{3,1}}$$

$$+\underbrace{\mathbb{E}\left[\|(\mathbf{I}_d-\boldsymbol{\theta}_*\boldsymbol{\theta}_*^{\top})\boldsymbol{\theta}_0\|_2\sum_{i=1}^{n}\sum_{i_1=1}^{n}e^{-\langle\mathbf{z}_i,\boldsymbol{\theta}_0\rangle}\mathbf{s}_{i_1}^{(t)}\widetilde{\mathbf{z}}_{i_1,2}\widetilde{\mathbf{z}}_{i_1,2}\widetilde{\mathbf{z}}_{i,2}\right]}_{I_{3,2}}$$

$$+\underbrace{\mathbb{E}\left[\|(\mathbf{I}_d-\boldsymbol{\theta}_*\boldsymbol{\theta}_*^{\top})\boldsymbol{\theta}_0\|_2\sum_{i=1}^{n}\sum_{i_1=1}^{n}\sum_{j=3}^{d}e^{-\langle\mathbf{z}_i,\boldsymbol{\theta}_0\rangle}\mathbf{s}_{i_1}^{(t)}\widetilde{\mathbf{z}}_{i_1,2}\widetilde{\mathbf{z}}_{i_1,j}\widetilde{\mathbf{z}}_{i,j}\right]}_{I_{3,3}}$$

$$-\underbrace{\mathbb{E}\left[\|(\mathbf{I}_d-\boldsymbol{\theta}_*\boldsymbol{\theta}_*^{\top})\boldsymbol{\theta}_0\|_2\sum_{i=1}^{n}\sum_{i_1=1}^{n}\sum_{i_2=1}^{n}e^{-\langle\mathbf{z}_i,\boldsymbol{\theta}_0\rangle}\mathbf{s}_{i_1}^{(t)}\mathbf{s}_{i_2}^{(t)}\widetilde{\mathbf{z}}_{i_2,2}\widetilde{\mathbf{z}}_{i_1,1}\widetilde{\mathbf{z}}_{i,1}\right]}_{I_{3,4}}$$

$$-\underbrace{\mathbb{E}\left[\|(\mathbf{I}_d-\boldsymbol{\theta}_*\boldsymbol{\theta}_*^{\top})\boldsymbol{\theta}_0\|_2\sum_{i=1}^{n}\sum_{i_1=1}^{n}\sum_{i_2=1}^{n}e^{-\langle\mathbf{z}_i,\boldsymbol{\theta}_0\rangle}\mathbf{s}_{i_1}^{(t)}\mathbf{s}_{i_2}^{(t)}\widetilde{\mathbf{z}}_{i_2,2}\widetilde{\mathbf{z}}_{i_1,2}\widetilde{\mathbf{z}}_{i,2}\right]}_{I_{3,5}}$$

$$-\underbrace{\mathbb{E}\left[\|(\mathbf{I}_d-\boldsymbol{\theta}_*\boldsymbol{\theta}_*^{\top})\boldsymbol{\theta}_0\|_2\sum_{i=1}^{n}\sum_{i_1=1}^{n}\sum_{i_2=1}^{n}\sum_{j=3}^{d}e^{-\langle\mathbf{z}_i,\boldsymbol{\theta}_0\rangle}\mathbf{s}_{i_1}^{(t)}\mathbf{s}_{i_2}^{(t)}\widetilde{\mathbf{z}}_{i_2,2}\widetilde{\mathbf{z}}_{i_1,j}\widetilde{\mathbf{z}}_{i,j}\right]}_{I_{3,6}}. \tag{17}$$

We carefully calculate these terms trough the lemmas provided in Appendix H.1, respectively. For the term $I_{3,1}$, by Lemma H.8, we have

$$I_{3,1}=\frac{2}{\pi}\sigma^4 e^{\frac{\sigma^2}{2}}nC_2^{(t)}+f_1(C_2^{(t)},\sigma)+\frac{nf_2(C_2^{(t)},\sigma)}{d},$$

where $f_1$, $f_2$ are two analytic functions of $C_2^{(t)}$ and $\sigma$. For the term $I_{3,2}$, by Lemma H.7, we have

$$I_{3,2} = \sigma^4 e^{\frac{\sigma^2}{2}} n\big(\sigma^2 [C_2^{(t)}]^2 + 1\big) + f_3(C_2^{(t)}, \sigma) + \frac{n f_4(C_2^{(t)}, \sigma)}{d}.$$

For the term $I_{3,3}$, we have

$$I_{3,3} = \sum_{i=1}^{n} \sum_{j=3}^{d} \mathbb{E}\big[\|(\mathbf{I}_d - \boldsymbol{\theta}_* \boldsymbol{\theta}_*^\top)\boldsymbol{\theta}_0\|_2 e^{-\langle \mathbf{z}_i, \boldsymbol{\theta}_0 \rangle} \mathbf{s}_i^{(t)} \widetilde{\mathbf{z}}_{i,2}\big] \mathbb{E}[\widetilde{\mathbf{z}}_{i,j}^2]$$

$$+ \sum_{i=1}^{n} \sum_{i_1 \neq i} \sum_{j=3}^{d} \mathbb{E}\big[\|(\mathbf{I}_d - \boldsymbol{\theta}_* \boldsymbol{\theta}_*^\top)\boldsymbol{\theta}_0\|_2 e^{-\langle \mathbf{z}_i, \boldsymbol{\theta}_0 \rangle} \mathbf{s}_i^{(t)} \widetilde{\mathbf{z}}_{i,2}\big] \mathbb{E}[\widetilde{\mathbf{z}}_{i,j}] \mathbb{E}[\widetilde{\mathbf{z}}_{i_1,j}]$$

$$= \sum_{i=1}^{n} \sum_{j=3}^{d} \mathbb{E}\big[\|(\mathbf{I}_d - \boldsymbol{\theta}_* \boldsymbol{\theta}_*^\top)\boldsymbol{\theta}_0\|_2 e^{-\langle \mathbf{z}_i, \boldsymbol{\theta}_0 \rangle} \mathbf{s}_i^{(t)} \widetilde{\mathbf{z}}_{i,2}\big] \mathbb{E}[\widetilde{\mathbf{z}}_{i,j}^2]$$

$$= d\big(1 + C_2^{(t)}\big)\sigma^4 e^{(2C_2^{(t)}+1)\sigma^2/2} + f_5(C_2^{(t)}, \sigma) + \frac{n f_6(C_2^{(t)}, \sigma)}{d}.$$

The second equation holds as by the independence between $\widetilde{\mathbf{z}}_{i_1,j}$ and $\widetilde{\mathbf{z}}_{i,j}$ when $i_1 \neq i$, and the fact that $\mathbb{E}[\widetilde{\mathbf{z}}_{i_1,j}] = \mathbb{E}[\widetilde{\mathbf{z}}_{i,j}] = 0$. The last equation is calculated based on Lemma H.9. For the term $I_{3,4}$, by Lemma H.11, we have

$$I_{3,4} = \frac{2}{\pi} \sigma^4 e^{\frac{\sigma^2}{2}} n C_2^{(t)} + f_7(C_2^{(t)}, \sigma) + \frac{n f_8(C_2^{(t)}, \sigma)}{d}.$$

For the term $I_{3,5}$, by Lemma H.10, we have

$$I_{3,5} = \sigma^6 e^{\frac{\sigma^2}{2}} n [C_2^{(t)}]^2 + f_9(C_2^{(t)}, \sigma) + \frac{n f_{10}(C_2^{(t)}, \sigma)}{d}.$$

For the term $I_{3,6}$, similar to the procedure of calculation for $I_{3,3}$ and utilizing the result of Lemma H.12, we have

$$I_{3,6} = \sum_{i=1}^{n} \sum_{i_2=1}^{n} \sum_{j=3}^{d} \mathbb{E}\big[\|(\mathbf{I}_d - \boldsymbol{\theta}_* \boldsymbol{\theta}_*^\top)\boldsymbol{\theta}_0\|_2 e^{-\langle \mathbf{z}_i, \boldsymbol{\theta}_0 \rangle} \mathbf{s}_i^{(t)} \mathbf{s}_{i_2}^{(t)} \widetilde{\mathbf{z}}_{i,2}\big] \mathbb{E}[\widetilde{\mathbf{z}}_{i,j}^2]$$

$$= d\sigma^4 e^{(2C_2^{(t)}+1)\sigma^2/2} C_2^{(t)} + f_{11}(C_2^{(t)}, \sigma) + \frac{n f_{12}(C_2^{(t)}, \sigma)}{d}.$$

Plugging all these results into (17), we obtain that

$$I_3 = \frac{n\sigma^4 e^{\frac{\sigma^2}{2}}}{d}\left(1 + \frac{d e^{C_2^{(t)} \sigma^2}}{n} + \frac{F_{5,\sigma}(C_2^{(t)})}{n} + \frac{F_{6,\sigma}(C_2^{(t)})}{d}\right)\mathbf{I}_d. \tag{18}$$

In the next, we consider calculating $I_4$ through a similar procedure as

$$I_4 = \mathbb{E}\left[\sum_{i=1}^{n} \sum_{i_1=1}^{n} \sum_{i_2=1}^{n} \mathbf{s}_i^{(t)} \mathbf{s}_{i_1}^{(t)} \mathbf{s}_{i_2}^{(t)} \mathbf{A}\mathbf{A}^\top (\mathbf{z}_{i_1} - \mathbf{z}_{i_2}) \mathbf{z}_{i_1}^\top \mathbf{A}\mathbf{A}^\top \mathbf{z}_i \boldsymbol{\theta}_0^\top\right]$$

$$= \mathbb{E}\left[\sum_{i=1}^{n} \sum_{i_1=1}^{n} \sum_{i_2=1}^{n} \mathbf{s}_i^{(t)} \mathbf{s}_{i_1}^{(t)} \mathbf{s}_{i_2}^{(t)} \mathbf{A}(\widetilde{\mathbf{z}}_{i_1} - \widetilde{\mathbf{z}}_{i_2}) \widetilde{\mathbf{z}}_{i_1}^\top \widetilde{\mathbf{z}}_i \boldsymbol{\theta}_0^\top\right]$$

$$= \mathbb{E}\left[\sum_{i=1}^{n} \sum_{i_1=1}^{n} \sum_{i_2=1}^{n} \sum_{j=1}^{d} \mathbf{s}_i^{(t)} \mathbf{s}_{i_1}^{(t)} \mathbf{s}_{i_2}^{(t)} (\widetilde{\mathbf{z}}_{i_1,j} - \widetilde{\mathbf{z}}_{i_2,j}) \widetilde{\mathbf{z}}_{i_1}^\top \widetilde{\mathbf{z}}_i \boldsymbol{\xi}_j \boldsymbol{\theta}_0^\top\right]$$

$$= \frac{1}{d} \mathbb{E}\left[\|(\mathbf{I}_d - \boldsymbol{\theta}_* \boldsymbol{\theta}_*^\top)\boldsymbol{\theta}_0\|_2 \sum_{i=1}^{n} \sum_{i_1=1}^{n} \sum_{i_2=1}^{n} \mathbf{s}_i^{(t)} \mathbf{s}_{i_1}^{(t)} \mathbf{s}_{i_2}^{(t)} (\widetilde{\mathbf{z}}_{i_1,2} - \widetilde{\mathbf{z}}_{i_2,2}) \widetilde{\mathbf{z}}_{i_1}^\top \widetilde{\mathbf{z}}_i\right] \cdot \mathbf{I}_d.$$

This result demonstrates that $I_4$ is also proportional to $\mathbf{I}_d$. For the coefficient
$\mathbb{E}\big[\|(\mathbf{I}_d - \boldsymbol{\theta}_*\boldsymbol{\theta}_*^\top)\boldsymbol{\theta}_0\|_2 \sum_{i=1}^n \sum_{i_1=1}^n \sum_{i_2=1}^n \mathbf{s}_i^{(t)}\mathbf{s}_{i_1}^{(t)}\mathbf{s}_{i_2}^{(t)}(\widetilde{\mathbf{z}}_{i_1,2} - \widetilde{\mathbf{z}}_{i_2,2})\widetilde{\mathbf{z}}_{i_1}^\top\widetilde{\mathbf{z}}_i\big]$, we separate it as

$$\mathbb{E}\Big[\|(\mathbf{I}_d - \boldsymbol{\theta}_*\boldsymbol{\theta}_*^\top)\boldsymbol{\theta}_0\|_2 \sum_{i=1}^n \sum_{i_1=1}^n \sum_{i_2=1}^n \mathbf{s}_i^{(t)}\mathbf{s}_{i_1}^{(t)}\mathbf{s}_{i_2}^{(t)}(\widetilde{\mathbf{z}}_{i_1,2} - \widetilde{\mathbf{z}}_{i_2,2})\widetilde{\mathbf{z}}_{i_1}^\top\widetilde{\mathbf{z}}_i\Big]$$

$$= \underbrace{\mathbb{E}\Big[\|(\mathbf{I}_d - \boldsymbol{\theta}_*\boldsymbol{\theta}_*^\top)\boldsymbol{\theta}_0\|_2 \sum_{i=1}^n \sum_{i_1=1}^n \mathbf{s}_i^{(t)}\mathbf{s}_{i_1}^{(t)}\widetilde{\mathbf{z}}_{i_1,2}\widetilde{\mathbf{z}}_{i_1,1}\widetilde{\mathbf{z}}_{i,1}\Big]}_{I_{4,1}} + \underbrace{\mathbb{E}\Big[\|(\mathbf{I}_d - \boldsymbol{\theta}_*\boldsymbol{\theta}_*^\top)\boldsymbol{\theta}_0\|_2 \sum_{i=1}^n \sum_{i_1=1}^n \mathbf{s}_i^{(t)}\mathbf{s}_{i_1}^{(t)}\widetilde{\mathbf{z}}_{i_1,2}\widetilde{\mathbf{z}}_{i_1,2}\widetilde{\mathbf{z}}_{i,2}\Big]}_{I_{4,2}}$$

$$+ \underbrace{\mathbb{E}\Big[\|(\mathbf{I}_d - \boldsymbol{\theta}_*\boldsymbol{\theta}_*^\top)\boldsymbol{\theta}_0\|_2 \sum_{i=1}^n \sum_{i_1=1}^n \sum_{j=3}^d \mathbf{s}_i^{(t)}\mathbf{s}_{i_1}^{(t)}\widetilde{\mathbf{z}}_{i_1,2}\widetilde{\mathbf{z}}_{i_1,j}\widetilde{\mathbf{z}}_{i,j}\Big]}_{I_{4,3}}$$

$$- \underbrace{\mathbb{E}\Big[\|(\mathbf{I}_d - \boldsymbol{\theta}_*\boldsymbol{\theta}_*^\top)\boldsymbol{\theta}_0\|_2 \sum_{i=1}^n \sum_{i_1=1}^n \sum_{i_2=1}^n \mathbf{s}_i^{(t)}\mathbf{s}_{i_1}^{(t)}\mathbf{s}_{i_2}^{(t)}\widetilde{\mathbf{z}}_{i_2,2}\widetilde{\mathbf{z}}_{i_1,1}\widetilde{\mathbf{z}}_{i,1}\Big]}_{I_{4,4}}$$

$$- \underbrace{\mathbb{E}\Big[\|(\mathbf{I}_d - \boldsymbol{\theta}_*\boldsymbol{\theta}_*^\top)\boldsymbol{\theta}_0\|_2 \sum_{i=1}^n \sum_{i_1=1}^n \sum_{i_2=1}^n \mathbf{s}_i^{(t)}\mathbf{s}_{i_1}^{(t)}\mathbf{s}_{i_2}^{(t)}\widetilde{\mathbf{z}}_{i_2,2}\widetilde{\mathbf{z}}_{i_1,2}\widetilde{\mathbf{z}}_{i,2}\Big]}_{I_{4,5}}$$

$$- \underbrace{\mathbb{E}\Big[\|(\mathbf{I}_d - \boldsymbol{\theta}_*\boldsymbol{\theta}_*^\top)\boldsymbol{\theta}_0\|_2 \sum_{i=1}^n \sum_{i_1=1}^n \sum_{i_2=1}^n \sum_{j=3}^d \mathbf{s}_i^{(t)}\mathbf{s}_{i_1}^{(t)}\mathbf{s}_{i_2}^{(t)}\widetilde{\mathbf{z}}_{i_2,2}\widetilde{\mathbf{z}}_{i_1,j}\widetilde{\mathbf{z}}_{i,j}\Big]}_{I_{4,6}}. \tag{19}$$

We carefully calculate these terms trough the lemmas provided in Appendix H.1, respectively. For the term $I_{4,1}$, by Lemma H.14, we have

$$I_{4,1} = \frac{2}{\pi}\sigma^4 C_2^{(t)} + \frac{f_{13}(C_2^{(t)}, \sigma)}{n} + \frac{f_{14}(C_2^{(t)}, \sigma)}{d}.$$

For the term $I_{4,2}$, by Lemma H.13, we have

$$I_{4,2} = C_2^{(t)}\sigma^4\big(\sigma^2[C_2^{(t)}]^2 + 1\big) + \frac{f_{15}(C_2^{(t)}, \sigma)}{n} + \frac{f_{16}(C_2^{(t)}, \sigma)}{d}.$$

For the term $I_{4,3}$, by Lemma H.15, we have

$$I_{4,3} = \sum_{i=1}^n \sum_{j=3}^d \mathbb{E}\big[\|(\mathbf{I}_d - \boldsymbol{\theta}_*\boldsymbol{\theta}_*^\top)\boldsymbol{\theta}_0\|_2 \big(\mathbf{s}_i^{(t)}\big)^2\widetilde{\mathbf{z}}_{i,2}\big]\mathbb{E}[\widetilde{\mathbf{z}}_{i,j}^2]$$

$$= \frac{2dC_2^{(t)}\sigma^4 e^{\sigma^2[C_2^{(t)}]^2}}{n} + \frac{f_{17}(C_2^{(t)}, \sigma)}{n} + \frac{f_{18}(C_2^{(t)}, \sigma)}{d}.$$

For the term $I_{4,4}$, by Lemma H.17, we have

$$I_{4,4} = \frac{2}{\pi}\sigma^4 C_2^{(t)} + \frac{f_{19}(C_2^{(t)}, \sigma)}{n} + \frac{f_{20}(C_2^{(t)}, \sigma)}{d}.$$

For the term $I_{4,5}$, by Lemma H.16, we have

$$I_{4,5} = C_2^3(t)\sigma^6 + \frac{f_{21}(C_2^{(t)}, \sigma)}{n} + \frac{f_{22}(C_2^{(t)}, \sigma)}{d}.$$

For the term $I_{4,6}$, similar to the procedure of calculation for $I_{4,3}$ and utilizing the result of Lemma H.18, we have

$$
\begin{aligned}
I_{4,6} &= \sum_{i=1}^{n} \sum_{i_2=1}^{n} \sum_{j=3}^{d} \mathbb{E}\big[\|(\mathbf{I}_d - \boldsymbol{\theta}_* \boldsymbol{\theta}_*^\top)\boldsymbol{\theta}_0\|_2 \big(\mathbf{s}_i^{(t)}\big)^2 \mathbf{s}_{i_2}^{(t)} \widetilde{\mathbf{z}}_{i,2}\big] \mathbb{E}[\widetilde{\mathbf{z}}_{i,j}^2] \\
&= \frac{d C_2^{(t)} \sigma^4 e^{\sigma^2 [C_2^{(t)}]^2}}{n} + \frac{f_{23}(C_2^{(t)}, \sigma)}{n} + \frac{f_{24}(C_2^{(t)}, \sigma)}{d}
\end{aligned}
$$

Plugging all these results into (19), we obtain that

$$
I_4 = \frac{C_2^{(t)} \sigma^4}{d}\left(1 + \frac{d e^{[C_2^{(t)}]^2 \sigma^2}}{n} + \frac{F_{7,\sigma}(C_2^{(t)})}{n} + \frac{F_{8,\sigma}(C_2^{(t)})}{d}\right)\mathbf{I}_d. \tag{20}
$$

Therefore, we prove the induction that by assuming at $t$-th iteration, $\mathbf{V}_{21}^{(t)} = C_1^{(t)} \cdot \mathbf{I}_d$ and $\mathbf{W}_{12}^{(t)} = -C_2^{(t)} \cdot \mathbf{I}_d$, the gradient $\nabla_{\mathbf{W}_{12}} \mathcal{L}_{\text{train}}(\mathbf{V}^{(t)}, \mathbf{W}^{(t)})$ is also proportional to the identity matrix $\mathbf{I}_d$. In addition, (18) and (20) establish the iterative rule for the coefficient $C_1^{(t)}$ as

$$
\begin{aligned}
C_2^{(t+1)} =\, & C_2^{(t)} + \frac{\eta C_1^{(t)} \sigma^4}{d}\left[\alpha e^{\frac{\sigma^2}{2}}\left(1 + \frac{d e^{C_2^{(t)} \sigma^2}}{n} + \frac{F_{5,\sigma}(C_2^{(t)})}{n} + \frac{F_{6,\sigma}(C_2^{(t)})}{d}\right) \right. \\
& \left. - C_1^{(t)} C_2^{(t)}\left(1 + \frac{d e^{[C_2^{(t)}]^2 \sigma^2}}{n} + \frac{F_{7,\sigma}(C_2^{(t)})}{n} + \frac{F_{8,\sigma}(C_2^{(t)})}{d}\right)\right].
\end{aligned}
$$

This completes the proof. $\qquad\square$

Therefore, we have shown that throughout training, the parameter matrices $\mathbf{V}^{(t)}$ and $\mathbf{W}^{(t)}$ always remain in the structured parameterization defined in Lemma 3.4. Consequently, for our learning task, analyzing the training loss $\mathcal{L}_{\text{train}}(\mathbf{V}, \mathbf{W})$ along the iterates $\mathbf{V}^{(t)}, \mathbf{W}^{(t)}$ is equivalent to studying a reduced two-dimensional proxy loss $\widetilde{\mathcal{L}}_{\text{train}}(C_1, C_2)$ with respect to the coefficients $C_1^{(t)}$ and $C_2^{(t)}$. Specifically, the proxy loss $\widetilde{\mathcal{L}}_{\text{train}}(C_1, C_2)$ is defined as

$$
\widetilde{\mathcal{L}}_{\text{train}}(C_1, C_2) := \mathcal{L}_{\text{train}}\left(\begin{bmatrix} \mathbf{0}_{d\times d} & \mathbf{0}_{d\times d} \\ C_1 \cdot \mathbf{I}_d & \mathbf{0}_{d\times d} \end{bmatrix}, \begin{bmatrix} \mathbf{0}_{d\times d} & -C_2 \cdot \mathbf{I}_d \\ \mathbf{0}_{d\times d} & \mathbf{0}_{d\times d} \end{bmatrix}\right).
$$

Under this parameterization, the update rules for $C_1^{(t)}$ and $C_2^{(t)}$ given in (11) correspond exactly to gradient descent applied to $\widetilde{\mathcal{L}}_{\text{train}}(C_1, C_2)$ with zero initialization. From (11), we further derive that $\widetilde{\mathcal{L}}_{\text{train}}(C_1, C_2)$ admits the following explicit form:

$$
\begin{aligned}
\widetilde{\mathcal{L}}_{\text{train}}(C_1, C_2) = \frac{\sigma^2}{d}\bigg[ & \frac{C_1^2}{2}\left(\frac{2}{\pi} + C_2^2 \sigma^2 + \frac{d}{n}e^{C_2^2 \sigma^2} + \frac{F_{3,\sigma}(C_2)}{n} + \frac{F_{4,\sigma}(C_2)}{d}\right) \\
& - \alpha e^{\sigma^2/2} C_1\left(\frac{2}{\pi} + C_2 \sigma^2 + \frac{d}{n}e^{C_2 \sigma^2} + \frac{F_{1,\sigma}(C_2)}{n} + \frac{F_{2,\sigma}(C_2)}{d}\right)\bigg] + c,
\end{aligned}
$$

where $c$ is a constant independent of $C_1$ and $C_2$. And we can immediately conclude that $F'_{1,\sigma}(C_2) = F_{5,\sigma}(C_2)$, $F'_{2,\sigma}(C_2) = F_{6,\sigma}(C_2)$, $F'_{3,\sigma}(C_2) = F_{7,\sigma}(C_2)$, $F'_{4,\sigma}(C_2) = F_{8,\sigma}(C_2)$. In addition, we define that

$$
\begin{aligned}
G_1(C_2) =\, & \frac{2}{\pi} + C_2 \sigma^2 + \frac{d}{n}e^{C_2 \sigma^2} + \frac{F_{1,\sigma}(C_2)}{n} + \frac{F_{2,\sigma}(C_2)}{d}; \\
G_2(C_2) =\, & \frac{2}{\pi} + C_2^2 \sigma^2 + \frac{d}{n}e^{C_2^2 \sigma^2} + \frac{F_{3,\sigma}(C_2)}{n} + \frac{F_{4,\sigma}(C_2)}{d}; \\
G_3(C_2) =\, & 1 + \frac{d e^{C_2 \sigma^2}}{n} + \frac{F_{5,\sigma}(C_2)}{n} + \frac{F_{6,\sigma}(C_2)}{d}; \\
G_4(C_2) =\, & 1 + \frac{d e^{C_2^2 \sigma^2}}{n} + \frac{F_{7,\sigma}(C_2)}{n} + \frac{F_{8,\sigma}(C_2)}{d}.
\end{aligned} \tag{21}
$$

With these notations, we prove the first conclusion in Theorem 3.5, which states that there exists an invariant compact set $\mathcal{R} = [0, 2\alpha e^{\sigma^2/2}] \times [0, 2]$ for $\mathbf{C}^{(t)} = [C_1^{(t)}, C_2^{(t)}]^\top$ during the training.

**Lemma E.5** (First conclusion in Theorem 3.5). *Consider the iterative rules for* $\mathbf{C}^{(t)} = [C_1^{(t)}, C_2^{(t)}]^\top$ *given in* (11) *with initializations* $C_1^{(0)}, C_2^{(0)} = 0$. *Suppose that* $\eta \leq \mathcal{O}(\frac{1}{n})$, $\alpha \leq \mathcal{O}(1)$ *and* $n \geq \Omega(d^2)$, *then it holds that:*

$$0 \leq C_1^{(t)} \leq 2\alpha e^{\sigma^2/2}; \ 0 \leq C_2^{(t)} \leq 2, \tag{22}$$

*for all* $t \in \mathbb{N}$.

*Proof of Lemma E.5.* We prove this lemma by induction. It's evident that (22) holds at $t = 0$. In the next, we assume it holds at $t$-th iteration and attempt to prove it still holds at $t + 1$-th iteration. Since (22) holds, there exists $K \leq O(1)$ such that $|F_{i,\sigma}(C_2^{(t)})|, e^{4\sigma^2} \leq \frac{K}{8}$ for all $i \in [8]$. Consequently, combined with the condition that $n \geq \Omega(d^2)$, we can directly obtain that

$$\frac{2}{\pi} - \frac{K}{d} \leq G_1(C_2^{(t)}) \leq \frac{3}{\pi} + \frac{K}{d}; \quad \frac{2}{\pi} - \frac{K}{d} \leq G_2(C_2^{(t)}) \leq \frac{4}{\pi} + \frac{K}{d};$$
$$1 - \frac{K}{d} \leq G_3(C_2^{(t)}) \leq 1 + \frac{K}{d}; \quad 1 - \frac{K}{d} \leq G_4(C_2^{(t)}) \leq 1 + \frac{K}{d};$$
$$\frac{3}{5} \leq \frac{\frac{3}{\pi} - \frac{K}{d}}{\frac{4}{\pi} + \frac{K}{d}} \leq \frac{G_1(C_2^{(t)})}{G_2(C_2^{(t)})} \leq \frac{\frac{9}{4\pi} + \frac{K}{d}}{\frac{17}{8\pi} - \frac{K}{d}} \leq \frac{3}{2}; \quad 1 - \frac{K}{d} \leq \frac{G_3(C_2^{(t)})}{G_4(C_2^{(t)})} \leq 1 + \frac{K}{d}; \quad \frac{G_3(C_2^{(t)})}{G_1(C_2^{(t)})} \leq 2.$$

We first prove that $C_1^{(t+1)} \geq 0$ from two cases: (1).$C_1^{(t)} \leq \frac{3\alpha e^{\sigma^2/2}}{5}$; and (2).$C_1^{(t)} > \frac{3\alpha e^{\sigma^2/2}}{5}$. If $C_1^{(t)} \leq \frac{3\alpha e^{\sigma^2/2}}{5}$, we have

$$\alpha e^{\sigma^2/2} G_1(C_2^{(t)}) - C_1^{(t)} G_2(C_2^{(t)}) \geq \alpha e^{\sigma^2/2} G_1(C_2^{(t)}) - \frac{3\alpha e^{\sigma^2/2}}{5} \frac{5}{3} G_1(C_2^{(t)}) \geq 0,$$

which implies that $C_1^{(t+1)} \geq C_1^{(t)} \geq 0$. If $C_1^{(t)} > \frac{3\alpha e^{\sigma^2/2}}{5}$, then we have

$$\alpha e^{\sigma^2/2} G_1(C_2^{(t)}) - C_1^{(t)} G_2(C_2^{(t)}) \geq -C_1^{(t)} G_2(C_2^{(t)}) \geq -2\alpha e^{\sigma^2/2}\left(\frac{4}{\pi} + \frac{K}{d}\right),$$

where we replace $C_1^{(t)}$ and $G_2(C_2^{(t)})$ with their upper bounds respectively. Since $\eta \leq \mathcal{O}(\frac{1}{n})$, we can obtain that

$$C_1^{(t+1)} \geq C_1^{(t)} + \frac{\eta\sigma^2}{d}\left(\alpha e^{\sigma^2/2} G_1(C_2^{(t)}) - C_1^{(t)} G_2(C_2^{(t)})\right)$$
$$\geq \frac{3\alpha e^{\sigma^2/2}}{5} - \frac{2\eta\sigma^2 \alpha e^{\sigma^2/2}}{d}\left(\frac{4}{\pi} + \frac{K}{d}\right) \geq 0.$$

The last inequality holds as $\eta \leq \mathcal{O}(\frac{1}{n})$ is sufficiently small. This proves that $C_1^{(t+1)} \geq 0$. Similarly, to prove $C_1^{(t+1)} \leq 2\alpha e^{\sigma^2/2}$, we also consider two cases: (1). $C_1^{(t)} \geq \frac{3\alpha e^{\sigma^2/2}}{2}$; and (2). $C_1^{(t)} < \frac{3\alpha e^{\sigma^2/2}}{2}$. For the first case where $C_1^{(t)} \geq \frac{3\alpha e^{\sigma^2/2}}{2}$, we have

$$\alpha e^{\sigma^2/2} G_1(C_2^{(t)}) - C_1^{(t)} G_2(C_2^{(t)}) \leq \frac{3\alpha e^{\sigma^2/2}}{2} G_2(C_2^{(t)}) - C_1^{(t)} G_2(C_2^{(t)}) \leq 0,$$

which implies that $C_1^{(t+1)} \leq C_1^{(t)} \leq 2\alpha e^{\sigma^2/2}$. On the other hand where $C_1^{(t)} < \frac{3\alpha e^{\sigma^2/2}}{2}$, we can calculate that

$$\alpha e^{\sigma^2/2} G_1(C_2^{(t)}) - C_1^{(t)} G_2(C_2^{(t)}) \leq \alpha e^{\sigma^2/2} G_1(C_2^{(t)}) \leq \alpha e^{\sigma^2/2}\left(\frac{3}{\pi} + \frac{K}{d}\right).$$

This can further implies that

$$C_1^{(t+1)} = C_1^{(t)} + \eta\left(\alpha e^{1/2} G_1(C_2^{(t)}) - C_1^{(t)} G_2(C_2^{(t)})\right) \leq \frac{3\alpha e^{\sigma^2/2}}{2} + \frac{\eta\sigma^2 \alpha e^{\sigma^2/2}}{d}\left(\frac{3}{\pi} + \frac{K}{d}\right) \leq 2\alpha e^{\sigma^2/2}.$$

This completes the proof that $C_1^{(t+1)} \leq 2\alpha e^{\sigma^2/2}$. In the following, we proceed with $C_2^{(t)}$ with the similar techniques. We first prove that $C_2^{(t+1)} \geq 0$ under (1). $C_2^{(t)} \leq \frac{1}{4}$; and (2). $C_2^{(t)} > \frac{1}{4}$. For the first case where $C_2^{(t)} \leq \frac{1}{4}$, we have

$$\alpha e^{\sigma^2/2} C_1^{(t)} G_3(C_2^{(t)}) - C_1^2(t) C_2^{(t)} G_4(C_2^{(t)}) \geq C_1^{(t)} \left( \alpha e^{\sigma^2/2} G_3(C_2^{(t)}) - 4\alpha e^{\sigma^2/2} \frac{1}{4} G_3(C_2^{(t)}) \right) \geq 0,$$

which implies that $C_2^{(t+1)} \geq C_2^{(t)} \geq 0$. On the other hand when $C_2^{(t)} > \frac{1}{4}$, we can obtain that

$$\alpha e^{\sigma^2/2} C_1^{(t)} G_3(C_2^{(t)}) - C_1^2(t) C_2^{(t)} G_4(C_2^{(t)}) \geq -C_1^2(t) C_2^{(t)} G_4(C_2^{(t)}) \geq -8\alpha^2 e^{\sigma^2},$$

where the last inequality holds as we substitute the upper bounds that $C_1^{(t)} \leq 2\alpha e^{\sigma^2/2}$ and $C_2^{(t)} \leq 2$. This result helps us further derive that

$$C_2^{(t+1)} = C_2^{(t)} + \frac{\sigma^4 \eta}{d} \left( \alpha e^{\sigma^2/2} C_1^{(t)} G_3(C_2^{(t)}) - C_1^2(t) C_2^{(t)} G_4(C_2^{(t)}) \right) \geq \frac{1}{4} - \frac{8\eta \sigma^4 \alpha^2 e^{\sigma^2}}{d} \geq 0.$$

This completes the proof that $C_2^{(t+1)} \geq 0$. In the next, we prove that $C_2^{(t)}$ is always smaller than 2. This would be a little tricky, and we consider two different phases. We first prove that there exists an iteration $T_1$, which serves the first time such that $C_1$ reaches $\frac{\alpha e^{\sigma^2/2}(3d-\pi K)}{4d+\pi K}$. We prove that for all $t \leq T_1$, it holds that $C_2^{(t)} \leq 1$ by induction. Notice that for any $t \leq T_1$, we have

$$\Delta C_1^{(t)} = C_1^{(t+1)} - C_1^{(t)} = \frac{\eta \sigma^2}{d} \left( \alpha e^{\frac{\sigma^2}{2}} G_1(C_2^{(t)}) - C_1^{(t)} G_2(C_2^{(t)}) \right)$$

$$\geq \frac{\eta \sigma^2}{d} \left( \alpha e^{\frac{\sigma^2}{2}} G_1(C_2^{(t)}) - \frac{\alpha e^{\sigma^2/2}(3d - \pi K)}{4d + \pi K} \frac{3d + 3\pi K/4}{3d - \pi K} G_1(C_2^{(t)}) \right) \geq \frac{\eta \sigma^2}{4d} \alpha e^{\frac{\sigma^2}{2}} G_1(C_2^{(t)}).$$

On the other hand, we can upper bound the increments of $C_2^{(t)}$ as

$$\Delta C_2^{(t)} = C_2^{(t+1)} - C_2^{(t)} = \frac{\eta \sigma^4}{d} \left( \alpha e^{\sigma^2/2} C_1^{(t)} G_3(C_2^{(t)}) - C_1^2(t) C_2^{(t)} G_4(C_2^{(t)}) \right)$$

$$\leq \frac{\eta \sigma^4}{d} \alpha e^{\sigma^2/2} C_1^{(t)} G_3(C_2^{(t)}) \leq \frac{2\eta \sigma^4}{d} \alpha e^{\sigma^2/2} G_1(C_2^{(t)}) \frac{\alpha e^{\sigma^2/2}(3d - \pi K)}{4d + \pi K}.$$

This implies that for all $t \leq T_1$, $\frac{\Delta C_2^{(t)}}{\Delta C_1^{(t)}} \leq \frac{8\alpha \sigma^2 e^{\sigma^2/2}(3d-\pi K)}{4d+\pi K}$. Consequently, we obtain that for all $t \leq T_1$, $C_2^{(t)} \leq$ $\frac{8\alpha^2 \sigma^2 e^{\sigma^2}(3d-\pi K)^2}{(4d+\pi K)^2} \leq 1$, which holds as $\sigma^2 \leq \frac{1}{2\pi}$ and $\alpha \leq 1$. Next, we prove for the case when $t \geq T_1$. We first prove that $C_1^{(t)} \geq \frac{\alpha e^{\sigma^2/2}(3d-\pi K)}{4d+\pi K} - \eta$ for any $t \geq T_1$ by induction. When $\frac{\alpha e^{\sigma^2/2}(3d-\pi K)}{4d+\pi K} - \eta \leq C_1^{(t)} \leq \frac{\alpha e^{\sigma^2/2}(3d-\pi K)}{4d+\pi K}$, we have

$$\alpha e^{\sigma^2/2} G_1(C_2^{(t)}) - C_1^{(t)} G_2(C_2^{(t)})$$

$$\geq \alpha e^{\sigma^2/2} G_1(C_2^{(t)}) - \frac{\alpha e^{\sigma^2/2}(3d - \pi K)}{4d + \pi K} \frac{4d + \pi K}{3d - \pi K} G_1(C_2^{(t)}) \geq 0.$$

This implies that $C_1^{(t+1)} \geq C_1^{(t)} \geq \frac{\alpha e^{\sigma^2/2}(3d-\pi K)}{4d+\pi K} - \eta$. When $C_1^{(t)} > \frac{\alpha e^{\sigma^2/2}(3d-\pi K)}{4d+\pi K}$, we have

$$\alpha e^{\sigma^2/2} G_1(C_2^{(t)}) - C_1^{(t)} G_2(C_2^{(t)}) \geq -C_1^{(t)} G_2(C_2^{(t)}) \geq -2\alpha e^{\sigma^2/2} \left( \frac{4}{\pi} + \frac{K}{d} \right).$$

Consequently, we have

$$C_1^{(t+1)} \geq C_1^{(t)} + \frac{\eta \sigma^2}{d} \left( \alpha e^{\sigma^2/2} G_1(C_2^{(t)}) - C_1^{(t)} G_2(C_2^{(t)}) \right)$$

$$\geq \frac{\alpha e^{\sigma^2/2}(3d - \pi K)}{4d + \pi K} - \frac{2\eta \sigma^2 \alpha e^{\sigma^2/2}}{d} \left( \frac{4}{\pi} + \frac{K}{d} \right) \geq \frac{\alpha e^{\sigma^2/2}(3d - \pi K)}{4d + \pi K} - \eta.$$

Now, by establishing the fact $C_1^{(t)} \geq \frac{\alpha e^{\sigma^2/2}(3d-\pi K)}{4d+\pi K} - \eta$, we can follow the previous proof techniques by induction. To prove that $C_2^{(t+1)} \leq 2$, we also consider two cases: (1). $C_2^{(t)} \geq \frac{19}{10}$; and (2). $C_2^{(t)} < \frac{19}{10}$. For the first case where $C_2^{(t)} \geq \frac{19}{10}$, we can obtain that

$$\alpha e^{\sigma^2/2} G_3(C_2^{(t)}) - C_1^{(t)} C_2^{(t)} G_4(C_2^{(t)})$$

$$\leq \alpha e^{\sigma^2/2} G_3(C_2^{(t)}) - \left( \frac{\alpha e^{\sigma^2/2}(3d-\pi K)}{4d+\pi K} - \eta \right) \frac{19}{10} \frac{G_3(C_2^{(t)})}{1+\frac{K}{d}}$$

$$\leq \alpha e^{\sigma^2/2} G_3(C_2^{(t)}) - \alpha e^{\sigma^2/2} G_3(C_2^{(t)}) + 2\eta G_3(C_2^{(t)}) - \alpha e^{\sigma^2/2} G_3(C_2^{(t)}) \frac{\frac{17d}{10} - (4+2\pi)K}{(4d+\pi K)(1+\frac{K}{d})}$$

$$\leq 2\eta G_3(C_2^{(t)}) - \frac{3\alpha e^{\sigma^2/2}}{8} G_3(C_2^{(t)}) \leq 0,$$

where the last inequality holds as $\eta \leq \mathcal{O}\left(\frac{1}{n}\right)$ is sufficiently small. This implies that $C_2^{(t+1)} \leq C_2^{(t)} \leq 2$. In addition, when $C_2^{(t)} \leq \frac{19}{10}$, we have

$$C_1^{(t)} \left( \alpha e^{\sigma^2/2} G_3(C_2^{(t)}) - C_1^{(t)} C_2^{(t)} G_4(C_2^{(t)}) \right) \leq C_1^{(t)} \alpha e^{\sigma^2/2} G_3(C_2^{(t)}) \leq 4\alpha^2 e^{\sigma^2}.$$

Consequently, we can obtain that

$$C_2^{(t+1)} = C_2^{(t)} + \frac{\eta \sigma^4}{d} C_1^{(t)} \left( \alpha e^{\sigma^2/2} G_3(C_2^{(t)}) - C_1^{(t)} C_2^{(t)} G_4(C_2^{(t)}) \right) \leq \frac{19}{10} + \frac{4\eta \sigma^4 \alpha^2 e^{\sigma^2}}{d} \leq 2,$$

where the last inequality holds as $\eta \leq \mathcal{O}\left(\frac{1}{n}\right)$ is sufficiently small. This completes the proof that $C_2^{(t)} \leq 2$ for all $t \in \mathbb{N}$. $\square$

We have demonstrated in Lemma E.5 that the iterates of $C_1^{(t)}$ and $C_2^{(t)}$ always stay inner the region $\mathcal{R} = [0, 2\alpha e^{\sigma^2/2}] \times [0, 2]$. We next prove that within this region, there exists a unique local minimum $(C_1^*, C_2^*)$ of the loss function $\widetilde{\mathcal{L}}_{\text{train}}(C_1, C_2)$. We first introduce Newton–Kantorovich Theorem, which helps demonstrate our results.

**Theorem E.6** (Newton–Kantorovich Theorem, cf. Theorem 5.5.1 in Kelley (1995)). *Let $F : \mathbb{R}^d \to \mathbb{R}^d$ be a continuously differentiable function. Suppose there exists a point $\mathbf{x}_0 \in \mathbb{R}^d$ and constants $\beta, \gamma, L$ such that:*

1. *$F(\cdot)$ is differentiable at $\mathbf{x}_0$, and $F'(\mathbf{x}_0)$ is invertible, satisfying that*

$$\|F'(\mathbf{x}_0)^{-1}\| \leq \beta; \quad \|F'(\mathbf{x}_0)^{-1} F(\mathbf{x}_0)\| \leq \gamma.$$

2. *$F'(\cdot)$ is Lipschitz continuous with constant $\iota$ in a neighborhood of $\mathbf{x}_0$ radius $\bar{r}$ satisfying that*

$$\bar{r} \geq r_- = \frac{1 - \sqrt{1 - 2\beta\gamma\iota}}{\beta\iota}$$

3. *The constants $\beta, \gamma, \iota$ satisfy that $\beta\gamma\iota \leq \frac{1}{2}$.*

*Then there exists a unique fixed point $\mathbf{x}^*$ of $F(\cdot)$ in the neighborhood of $\mathbf{x}_0$ with radius equal to $\max\{\bar{r}, \frac{1+\sqrt{1-2\beta\gamma\iota}}{\beta\iota}\}$. In addition, $\mathbf{x}^*$ satisfies that $\|\mathbf{x}^* - \mathbf{x}_0\| \leq r_-$.*

Then the following Lemma E.7 demonstrates that there exists a unique local minimum of the proxy loss $\widetilde{\mathcal{L}}_{\text{train}}(C_1, C_2)$ by utilizing the Newton-Kantorovich Theorem E.6.

**Lemma E.7** (Second conclusion in Theorem 3.5). *Let $\mathcal{R} := [0, 2\alpha e^{\sigma^2/2}] \times [0, 2]$. Then there exists a unique local minimum $\mathbf{C}^* = [C_1^*, C_2^*]^\top$ of the loss function $\widetilde{\mathcal{L}}_{\text{train}}(C_1, C_2)$ inner $\mathcal{R}$, and this local minimum satisfies that*

$$\left| C_1^* - \alpha e^{\sigma^2/2} \right|, \left| C_2^* - 1 \right| \leq \mathcal{O}\left(\frac{1}{d}\right).$$

*Proof of Lemma E.7.* We prove this lemma by demonstrating that $\nabla\widetilde{\mathcal{L}}_{\text{train}}(C_1, C_2)$ only has one unique fixed point by Newton–Kantorovich Theorem E.6. Since Newton–Kantorovich Theorem E.6 does not require a particular norm, we specify the $\ell_\infty$ norm $\|\cdot\|_\infty$ in our proof. We first calculate the Hessian matrix of $\widetilde{\mathcal{L}}_{\text{train}}(C_1, C_2)$ as

$$\frac{\partial^2\widetilde{\mathcal{L}}_{\text{train}}(C_1, C_2)}{\partial C_1^2} = \frac{\sigma^2}{d}\left(\frac{2}{\pi} + C_2^2\sigma^2 + \frac{d}{n}e^{C_2^2\sigma^2} + \frac{F_{3,\sigma}(C_2)}{n} + \frac{F_{4,\sigma}(C_2)}{d}\right)$$

$$\frac{\partial^2\widetilde{\mathcal{L}}_{\text{train}}(C_1, C_2)}{\partial C_1\partial C_2} = -\frac{\sigma^4}{d}\left[\alpha e^{\frac{\sigma^2}{2}}\left(1 + \frac{d}{n}e^{C_2\sigma^2} + \frac{F'_{1,\sigma}(C_2)}{n} + \frac{F'_{2,\sigma}(C_2)}{d}\right)\right.$$
$$\left. - 2C_1C_2\left(1 + \frac{d}{n}e^{C_2^2\sigma^2} + \frac{F'_{3,\sigma}(C_2)}{n} + \frac{F'_{4,\sigma}(C_2)}{d}\right)\right]$$

$$\frac{\partial^2\widetilde{\mathcal{L}}_{\text{train}}(C_1, C_2)}{\partial C_2\partial C_1} = -\frac{\sigma^4}{d}\left[\alpha e^{\frac{\sigma^2}{2}}\left(1 + \frac{d}{n}e^{C_2\sigma^2} + \frac{F_{5,\sigma}(C_2)}{n} + \frac{F_{6,\sigma}(C_2)}{d}\right)\right.$$
$$\left. - 2C_1C_2\left(1 + \frac{d}{n}e^{C_2^2\sigma^2} + \frac{F_{7,\sigma}(C_2)}{n} + \frac{F_{8,\sigma}(C_2)}{d}\right)\right]$$

$$\frac{\partial^2\widetilde{\mathcal{L}}_{\text{train}}(C_1, C_2)}{\partial C_2^2} = -\frac{C_1\sigma^4}{d}\left[\alpha e^{\frac{\sigma^2}{2}}\left(\frac{\sigma^2de^{C_2\sigma^2}}{n} + \frac{F'_{5,\sigma}(C_2)}{n} + \frac{F'_{6,\sigma}(C_2)}{d}\right)\right.$$
$$\left. - C_1\left(1 + \frac{(1 + 2C_2^2)de^{C_2^2\sigma^2}}{n} + \frac{F'_{7,\sigma}(C_2)}{n} + \frac{F'_{8,\sigma}(C_2)}{d}\right)\right].$$

In addition, since we have $(C_1, C_2) \in \mathcal{R} = [0, 2\alpha e^{\sigma^2/2}] \times [0, 2]$, which is a compact set. Similar to the proof of Lemma E.5, we can find $K \leq \mathcal{O}(1)$, such that $\max\{|F_{k,\sigma}(C_2)|, |F'_{k,\sigma}(C_2)|, 9e^{4\sigma^2}\} \leq \frac{K}{8}$ for all $k \in [8]$. To check the conditions of Newton-Kantorovich Theorem, we let $(\widetilde{C}_1, \widetilde{C}_2) = (\alpha e^{\sigma^2/2}, 1)$. Then we can calculate that

$$\left|\frac{\partial^2\widetilde{\mathcal{L}}_{\text{train}}(C_1, C_2)}{\partial C_1^2}\right|_{C_1=\widetilde{C}_1, C_2=\widetilde{C}_2} - \frac{\sigma^2}{d}\left(\frac{2}{\pi} + \sigma^2\right)\right| \leq \frac{K}{d^2}$$

$$\left|\frac{\partial^2\widetilde{\mathcal{L}}_{\text{train}}(C_1, C_2)}{\partial C_1\partial C_2}\right|_{C_1=\widetilde{C}_1, C_2=\widetilde{C}_2} - \frac{\alpha e^{\sigma^2/2}\sigma^4}{d}\right| \leq \frac{K}{d^2}$$

$$\left|\frac{\partial^2\widetilde{\mathcal{L}}_{\text{train}}(C_1, C_2)}{\partial C_2\partial C_1}\right|_{C_1=\widetilde{C}_1, C_2=\widetilde{C}_2} - \frac{\alpha e^{\sigma^2/2}\sigma^4}{d}\right| \leq \frac{K}{d^2}$$

$$\left|\frac{\partial^2\widetilde{\mathcal{L}}_{\text{train}}(C_1, C_2)}{\partial C_2^2}\right|_{C_1=\widetilde{C}_1, C_2=\widetilde{C}_2} - \frac{\alpha^2 e^{\sigma^2}\sigma^4}{d}\right| \leq \frac{K}{d^2}.$$

From which we can calculate that $\det\left[\nabla^2\widetilde{\mathcal{L}}_{\text{train}}(\widetilde{C}_1, \widetilde{C}_2)\right] \geq \frac{2\alpha^2 e^{\sigma^2}\sigma^6}{\pi d^2} - \frac{5K}{d^3}$. Consequently, we can further calculate that

$$\left\|[\nabla^2\widetilde{\mathcal{L}}_{\text{train}}(\widetilde{C}_1, \widetilde{C}_2)]^{-1}\right\|_\infty \leq \frac{\max\left\{\left|\frac{\partial^2\widetilde{\mathcal{L}}_{\text{train}}(\widetilde{C}_1, \widetilde{C}_2)}{\partial C_2^2}\right|, \left|\frac{\partial^2\widetilde{\mathcal{L}}_{\text{train}}(\widetilde{C}_1, \widetilde{C}_2)}{\partial C_1^2}\right|\right\} + \left|\frac{\partial^2\widetilde{\mathcal{L}}_{\text{train}}(\widetilde{C}_1, \widetilde{C}_2)}{\partial C_1\partial C_2}\right|}{\det\left[\nabla^2\widetilde{\mathcal{L}}_{\text{train}}(\widetilde{C}_1, \widetilde{C}_2)\right]}$$

$$\leq \frac{(\pi\alpha^2 e^{\sigma^2}\sigma^2 + \pi\alpha e^{\sigma^2/2}\sigma^2 + 2 + \pi\sigma^2)d}{\alpha^2 e^{\sigma^2}\sigma^4} = \beta,$$

which completes the first condition of Newton-Kantorovich Theorem E.6. In the next, we can further calculate that

$$\left|\frac{\partial\widetilde{\mathcal{L}}_{\text{train}}(C_1, C_2)}{\partial C_1}\right|_{C_1=\widetilde{C}_1, C_2=\widetilde{C}_2}\right| \leq \frac{K}{d^2}; \quad \left|\frac{\partial\widetilde{\mathcal{L}}_{\text{train}}(C_1, C_2)}{\partial C_2}\right|_{C_1=\widetilde{C}_1, C_2=\widetilde{C}_2}\right| \leq \frac{K}{d^2}.$$

Consequently, we have that

$$\left\|[\nabla^2\widetilde{\mathcal{L}}_{\text{train}}(\widetilde{C}_1, \widetilde{C}_2)]^{-1}\nabla\widetilde{\mathcal{L}}_{\text{train}}(\widetilde{C}_1, \widetilde{C}_2)\right\|_\infty \leq \left\|[\nabla^2\widetilde{\mathcal{L}}_{\text{train}}(\widetilde{C}_1, \widetilde{C}_2)]^{-1}\right\|_\infty\left\|\nabla\widetilde{\mathcal{L}}_{\text{train}}(\widetilde{C}_1, \widetilde{C}_2)\right\|_\infty$$

$$\leq \frac{(\pi\alpha^2 e^{\sigma^2}\sigma^2 + \pi\alpha e^{\sigma^2/2}\sigma^2 + 2 + \pi\sigma^2)K}{\alpha^2 e^{\sigma^2}\sigma^4 d} = \gamma.$$

In addition, for all $(C_1, C_2) \in \mathcal{R}$, we can calculate that

$$\left\|\nabla^2 \widetilde{\mathcal{L}}_{\text{train}}(C_1, C_2)\right\|_\infty \leq \frac{16\alpha^2\sigma^4 e^{\sigma^2} + 9\alpha e^{\sigma^2/2}\sigma^4 + 4\sigma^4}{d} = \iota,$$

implying that $\nabla^2 \widetilde{\mathcal{L}}_{\text{train}}(C_1, C_2)$ is Lipschitz continuous with the constant $\iota$. And we can check that $\beta\gamma\iota = \mathcal{O}(\frac{1}{d}) \leq \frac{1}{2}$. For now, we have verified that all conditions of Theorem E.6 hold. Therefore, there exists a unique fixed point $(C_1^*, C_2^*)$ of $\nabla\widetilde{\mathcal{L}}_{\text{train}}(C_1, C_2)$ inner $\mathcal{R}$, and satisfying that

$$\left|C_1^* - \alpha e^{\sigma^2/2}\right|, \left|C_2^* - 1\right| \leq r_- \leq \gamma \leq \mathcal{O}\left(\frac{1}{d}\right).$$

In the next, we prove that this unique fixed point $(C_1^*, C_2^*)$ is a local minimum. Since $|C_1^* - \alpha e^{\sigma^2/2}|, |C_2^* - 1| \leq \mathcal{O}(\frac{1}{d})$, we can easily obtain that $\left\|\nabla^2\widetilde{\mathcal{L}}_{\text{train}}(C_1^*, C_2^*) - \nabla^2\widetilde{\mathcal{L}}_{\text{train}}(\widetilde{C}_1, \widetilde{C}_2)\right\|_\infty \leq \mathcal{O}(\frac{1}{d^2})$. Consequently, we can obtain that

$$\det\left[\nabla^2\widetilde{\mathcal{L}}_{\text{train}}(C_1^*, C_2^*)\right] \geq \det\left[\nabla^2\widetilde{\mathcal{L}}_{\text{train}}(\widetilde{C}_1, \widetilde{C}_2)\right] - 2\|\nabla^2\widetilde{\mathcal{L}}_{\text{train}}(C_1^*, C_2^*) - \nabla^2\widetilde{\mathcal{L}}_{\text{train}}(\widetilde{C}_1, \widetilde{C}_2)\|_\infty^2$$
$$- 2\|\nabla^2\widetilde{\mathcal{L}}_{\text{train}}(C_1^*, C_2^*) - \nabla^2\widetilde{\mathcal{L}}_{\text{train}}(\widetilde{C}_1, \widetilde{C}_2)\|_\infty\|\nabla^2\widetilde{\mathcal{L}}_{\text{train}}(\widetilde{C}_1, \widetilde{C}_2)\|_\infty$$
$$\geq \frac{2\alpha^2 e^{\sigma^2}\sigma^6}{\pi d^2} - \mathcal{O}\left(\frac{1}{d^3}\right) > 0.$$

In addition, we can also check that

$$\text{tr}\left[\nabla^2\widetilde{\mathcal{L}}_{\text{train}}(C_1^*, C_2^*)\right] \geq \text{tr}\left[\nabla^2\widetilde{\mathcal{L}}_{\text{train}}(\widetilde{C}_1, \widetilde{C}_2)\right] - 2\|\nabla^2\widetilde{\mathcal{L}}_{\text{train}}(C_1^*, C_2^*) - \nabla^2\widetilde{\mathcal{L}}_{\text{train}}(\widetilde{C}_1, \widetilde{C}_2)\|_\infty$$
$$\geq \frac{\alpha^2 e^{\sigma^2}\sigma^4 + 2\sigma^2/\pi + \sigma^4}{d} - \mathcal{O}\left(\frac{1}{d^2}\right) > 0.$$

These two results demonstrate that $\nabla^2\widetilde{\mathcal{L}}_{\text{train}}(C_1^*, C_2^*)$ is strictly positive definite. Therefore, the fixed point $\mathbf{C}^* = [C_1^*, C_2^*]^\top$ of the gradient $\nabla\widetilde{\mathcal{L}}_{\text{train}}(C_1, C_2)$ is local minimum of the loss function $\widetilde{\mathcal{L}}_{\text{train}}(C_1, C_2)$, which completes the proof. $\square$

For now, we have completed the first and second conclusions in Theorem 3.5. In the next lemma, we demonstrate that inside the region $\mathcal{R}$, the loss function $\widetilde{\mathcal{L}}_{\text{train}}(C_1, C_2)$ follows a Polyak-Lojasiewicz (PL) condition, which serves as a critical step for final linear convergence rate.

**Lemma E.8** (PL condition inside $\mathcal{R}$). *There exists a strictly positive constant $\mu_{1,\alpha,\sigma}$ solely depending on $\alpha$ and $\sigma$, such that for all $[C_1, C_2]^\top$ inner $\mathcal{R}$, the loss function $\widetilde{\mathcal{L}}_{\text{train}}(C_1, C_2)$ satisfies the following Polyak-Lojasiewicz (PL) condition*

$$\left\|\nabla\widetilde{\mathcal{L}}_{\text{train}}(C_1, C_2)\right\|_2^2 \geq \frac{2\mu_{\alpha,\sigma}}{d}\left(\widetilde{\mathcal{L}}_{\text{train}}(C_1, C_2) - \widetilde{\mathcal{L}}_{\text{train}}(C_1^*, C_2^*)\right).$$

*Proof of Lemma E.8.* To prove this lemma, we first introduce the nullcline of $C_1$ in the gradient regarding $C_1$. By setting $[\nabla\widetilde{\mathcal{L}}_{\text{train}}(C_1, C_2)]_1 = 0$, we can derive the nullcline of $C_1$ as $G_*(C_2) = \frac{\alpha e^{\frac{\sigma^2}{2}}G_1(C_2)}{G_2(C_2)}$, and define that $\Delta(C_1, C_2) = C_1 - G_*(C_2)$. Then we can decompose the loss $\widetilde{\mathcal{L}}_{\text{train}}(C_1, C_2)$ as

$$\widetilde{\mathcal{L}}_{\text{train}}(C_1, C_2) = \frac{\sigma^2}{d}\left[\frac{\Delta^2(C_1, C_2)G_2(C_2)}{2} - \frac{\alpha^2 e^{\sigma^2}G_1^2(C_2)}{2G_2(C_2)}\right] + c,$$

where $c$ is a constants irrelevant with $C_1$ and $C_2$. Since $(C_1^*, C_2^*)$ is the fixed point of $\nabla\widetilde{\mathcal{L}}_{\text{train}}(C_1, C_2)$, we can obtain that $\Delta(C_1^*, C_2^*) = 0$, which helps us to further derive that

$$\widetilde{\mathcal{L}}_{\text{train}}(C_1, C_2) - \widetilde{\mathcal{L}}_{\text{train}}(C_1^*, C_2^*) = \frac{\sigma^2\Delta^2(C_1, C_2)G_2(C_2)}{2d} + \psi(C_2) - \psi(C_2^*), \tag{23}$$

where $\psi(C_2) = -\frac{\alpha^2 e^{\sigma^2} G_1^2(C_2)}{2dG_2(C_2)}$. In addition, since

$$\frac{\partial \widetilde{\mathcal{L}}_{\text{train}}(C_1^*, C_2^*)}{\partial C_2} = \frac{\sigma^2 \Delta(C_1^*, C_2^*)}{2d}\left(2\frac{\partial \Delta(C_1^*, C_2^*)}{\partial C_2}G_2(C_2^*) + \Delta(C_1^*, C_2^*)G_2'(C_2^*)\right) + \psi'(C_2^*) = 0,$$

we can immediately conclude that $\psi'(C_2^*) = 0$. In addition, we can calculate that

$$\psi''(C_2) = \frac{\alpha^2 e^{\sigma^2}}{2dG_2^3(C_2)}\left[4G_1(C_2)G_1'(C_2)G_2(C_2)G_2'(C_2) + G_1^2(C_2)G_2(C_2)G_2''(C_2)\right.$$

$$\left. - 2G_2^2(C_2)\left(G_1(C_2)G_1''(C_2) + \left[G_1'(C_2)\right]^2\right) - 2G_1^2(C_2)\left[G_2'(C_2)\right]^2\right]$$

$$= \frac{\alpha^2 e^{\sigma^2}}{2dG_2^3(C_2)}\widetilde{G}(C_2). \tag{24}$$

Similar to the previous proof of Lemma E.5 and Lemma E.7, we can find $K \leq \mathcal{O}(1)$ such that $\max\{|F_{k,\sigma}(C_2)|, |F_{k,\sigma}'(C_2)|, 9e^{4\sigma^2}\} \leq \frac{K}{8}$. For the terms $G_1(C_2)$ and $G_2(C_2)$ and their derivatives, we can obtain their uniform upper and lower bounds for $C_2$ in $[0, 2]$ as

$$\frac{2}{\pi} - \frac{K}{d} \leq G_1(C_2) \leq \frac{3}{\pi} + \frac{K}{d}; \quad \frac{2}{\pi} - \frac{K}{d} \leq G_2(C_2) \leq \frac{4}{\pi} + \frac{K}{d}$$

$$|G_1'(C_2) - \sigma^2| \leq \frac{K}{d}; \quad |G_1''(C_2)| \leq \frac{K}{d}; \quad |G_2'(C_2) - 2\sigma^2 C_2| \leq \frac{K}{d}; \quad |G_2''(C_2) - 2\sigma^2| \leq \frac{K}{d}. \tag{25}$$

This further implies that $\frac{1}{2}\alpha e^{\sigma^2/2} \leq \Delta(C_1, C_2) \leq 2\alpha e^{\sigma^2/2}$. Substituting these terms into the $\widetilde{G}(C_2)$ defined in (24), we can obtain that

$$\left|\widetilde{G}(C_2) - \left(\frac{16\sigma^2}{\pi^3} - \frac{8\sigma^4}{\pi^2} + \frac{48\sigma^4}{\pi^2}C_2 - \frac{24\sigma^4}{\pi^2}C_2^2 + \frac{12\sigma^6}{\pi}C_2^2 - \frac{8\sigma^6}{\pi}C_2^3\right)\right| \leq \frac{90K}{\pi^3 d}. \tag{26}$$

By substituting the facts that $0 \leq C_2 \leq 2$ inner $\mathcal{R}$, and $\sigma^2 \leq \frac{1}{2\pi}$ into (26), we can derive the lower and upper bounds for $\widetilde{G}(C_2)$ as

$$\frac{8\sigma^2}{\pi^3} \leq \widetilde{G}(C_2) \leq \frac{29\sigma^2}{\pi^3}.$$

For now, we substitute the previously derived upper and lower bounds on $\widetilde{G}(C_2)$, together with the corresponding bounds on $G_2(C_2)$ that $\frac{2}{\pi} - \frac{K}{d} \leq \frac{4}{\pi} + \frac{K}{d}$ as established in the proof of Lemma E.5, into (24). This yields the following bounds of $\psi''(C_2)$ uniformly for all $C_2$ in $\mathcal{R}$ as

$$\frac{4\alpha^2\sigma^2 e^{\sigma^2}}{65d} \leq \psi''(C_2) \leq \frac{3\alpha^2\sigma^2 e^{\sigma^2}}{d}. \tag{27}$$

Consequently, by utilizing Taylor's expansion, the fact $\psi'(C_2^*) = 0$, and the uniform upper and lower bounds for $\psi''(C_2)$, we can derive that

$$\psi(C_2) - \psi(C_2^*) = \psi'(C_2^*)(C_2 - C_2^*) + \frac{\psi''(\bar{C}_2)}{2}(C_2 - C_2^*)^2 \leq \frac{3\alpha^2\sigma^2 e^{\sigma^2}}{2d}(C_2 - C_2^*)^2, \tag{28}$$

where $\bar{C}_2$ is an intermediate value between $C_2$ and $C_2^*$. On the other hand, we can also have

$$\left|\psi'(C_2)\right| = \left|\psi'(C_2^*) + \psi''(\bar{C}_2)(C_2 - C_2^*)\right| \geq \frac{4\alpha^2\sigma^2 e^{\sigma^2}}{65d}\left|C_2 - C_2^*\right|. \tag{29}$$

By square on both sides of (29) and compare it with (28), we obtain that

$$\left[\psi'(C_2)\right]^2 \geq \frac{\alpha^2\sigma^2 e^{\sigma^2}}{600d}\left(\psi(C_2) - \psi(C_2^*)\right), \tag{30}$$

which establishes a PL condition for $\psi(C_2)$. In the next, we prove that this can imply a PL condition for $\widetilde{\mathcal{L}}_{\text{train}}(C_1, C_2)$. We first consider two components $\frac{\partial \widetilde{\mathcal{L}}_{\text{train}}(C_1, C_2)}{\partial C_1}$ and $\frac{\partial \widetilde{\mathcal{L}}_{\text{train}}(C_1, C_2)}{\partial C_2}$ of $\|\nabla \widetilde{\mathcal{L}}_{\text{train}}(C_1, C_2)\|_2$ as following:

$$\frac{\partial \widetilde{\mathcal{L}}_{\text{train}}(C_1, C_2)}{\partial C_1} = \frac{\sigma^2}{d} G_2(C_2) \Delta(C_1, C_2);$$

$$\frac{\partial \widetilde{\mathcal{L}}_{\text{train}}(C_1, C_2)}{\partial C_2} = \psi'(C_2) + \frac{\sigma^2}{d} \frac{\alpha e^{\sigma^2/2} \big[ G_1'(C_2)G_2(C_2) - G_1(C_2)G_2'(C_2) \big]}{G_2(C_1, C_2)} \Delta(C_1, C_2) + \frac{\sigma^2}{2d} G_2'(C_2) \Delta^2(C_1, C_2).$$

By utilizing the results in (25), we can obtain that

$$\left| \frac{\partial \widetilde{\mathcal{L}}_{\text{train}}(C_1, C_2)}{\partial C_1} \right|^2 \geq \frac{17\sigma^4}{\pi^2 d^2} \Delta^2(C_1, C_2), \tag{31}$$

and

$$\left| \frac{\partial \widetilde{\mathcal{L}}_{\text{train}}(C_1, C_2)}{\partial C_2} \right|^2$$

$$= \left( \psi'(C_2) + \frac{\sigma^2}{d} \frac{\alpha e^{\sigma^2/2} \big[ G_1'(C_2)G_2(C_2) - G_1(C_2)G_2'(C_2) \big]}{G_2(C_2)} \Delta(C_1, C_2) + \frac{\sigma^2}{2d} G_2'(C_2) \Delta^2(C_1, C_2) \right)^2$$

$$\geq \frac{1}{2} \big[ \psi'(C_2) \big]^2 - \left( \frac{\sigma^2}{d} \frac{\alpha e^{\sigma^2/2} \big[ G_1'(C_2)G_2(C_2) - G_1(C_2)G_2'(C_2) \big]}{G_2(C_2)} \Delta(C_1, C_2) + \frac{\sigma^2}{2d} G_2'(C_2) \Delta^2(C_1, C_2) \right)^2$$

$$\geq \frac{1}{2} \big[ \psi'(C_2) \big]^2 - \frac{5\alpha \sigma^6 e^{\frac{\sigma^2}{2}}}{d^2} \Delta^2(C_1, C_2), \tag{32}$$

where the first inequality holds as $a^2 + b^2 \geq \frac{a^2}{2} - b^2$ for all $a, b \in \mathbb{R}$, and the second inequality is derived by utilizing the upper and lower bounds in (25). We let $\kappa = \min\{1, \frac{17}{10\pi^2 \alpha \sigma^2 e^{\sigma^2/2}}\}$, and then can calculate that

$$\|\nabla \widetilde{\mathcal{L}}_{\text{train}}(C_1, C_2)\|_2^2 = \left| \frac{\partial \widetilde{\mathcal{L}}_{\text{train}}(C_1, C_2)}{\partial C_1} \right|^2 + \left| \frac{\partial \widetilde{\mathcal{L}}_{\text{train}}(C_1, C_2)}{\partial C_2} \right|^2$$

$$\geq \left| \frac{\partial \widetilde{\mathcal{L}}_{\text{train}}(C_1, C_2)}{\partial C_1} \right|^2 + \kappa \left| \frac{\partial \widetilde{\mathcal{L}}_{\text{train}}(C_1, C_2)}{\partial C_2} \right|^2$$

$$\geq \frac{17\sigma^4}{\pi^2 d^2} \Delta^2(C_1, C_2) + \frac{\kappa}{2} \big[ \psi'(C_2) \big]^2 - \kappa \frac{5\alpha \sigma^6 e^{\frac{\sigma^2}{2}}}{d^2} \Delta^2(C_1, C_2)$$

$$\geq \frac{17\sigma^4}{2\pi^2 d^2} \Delta^2(C_1, C_2) + \frac{\kappa}{2} \big[ \psi'(C_2) \big]^2$$

$$\geq \frac{2\sigma^2}{5\pi d} \frac{\sigma^2}{2d} G_2(C_2) \Delta^2(C_1, C_2) + \frac{\alpha^2 \sigma^2 e^{\sigma^2} \kappa}{1200} \big( \psi(C_2) - \psi(C_2^*) \big)$$

$$\geq \frac{2}{d} \min\left\{ \frac{\sigma^2}{5\pi d}, \frac{\alpha^2 \sigma^2 e^{\sigma^2}}{600d}, \frac{\alpha e^{\sigma^2/2}}{375\pi^2} \right\} \left( \frac{\sigma^2}{2d} G_2(C_2) \Delta^2(C_1, C_2) + \psi(C_2) - \psi(C_2^*) \right)$$

$$\geq \frac{2\mu_{1,\alpha,\sigma}}{d} \left( \widetilde{\mathcal{L}}_{\text{train}}(C_1, C_2) - \widetilde{\mathcal{L}}_{\text{train}}(C_1^*, C_2^*) \right),$$

where $\mu_{1,\alpha,\sigma} = \min\{ \frac{\sigma^2}{5\pi d}, \frac{\alpha^2 \sigma^2 e^{\sigma^2}}{600d}, \frac{\alpha e^{\sigma^2/2}}{375\pi^2} \}$ is a constant solely depending on $\alpha$ and $\sigma$. Here, the first inequality holds as $\kappa \leq 1$. The second inequality is derived by applying the results of (31) and (32). The third inequality holds as $\kappa \frac{5\alpha \sigma^6 e^{\frac{\sigma^2}{2}}}{d^2} \leq \frac{17\sigma^4}{2\pi^2 d^2}$ by definition of $\kappa$. The forth inequality is established by substituting the upper bound for $G_2(C_2)$ and applying the result of (30). Finally, we obtain the last inequality by applying the result of (23). This completes the proof. $\square$

Now, we are ready to prove the last conclusion of Theorem 3.5, i.e. the linear convergence rate for both training loss and parameters, which are presented in the following Theorem E.9.

**Theorem E.9** (Third conclusion in Theorem 3.5). *Under the same conditions as Theorem 3.5, the iterates $C_1^{(t)}, C_2^{(t)}$ of gradient descent updates with respect to the training loss $\widetilde{\mathcal{L}}_{\text{train}}(C_1, C_2)$ given in Lemma E.2, converge linearly to the unique local minimizer $\mathbf{C}^* = [C_1^*, C_2^*]^\top$ of the training loss $\widetilde{\mathcal{L}}_{\text{train}}(C_1, C_2)$. In particular, for all $t \geq 0$, the training loss decays as*

$$\widetilde{\mathcal{L}}_{\text{train}}(C_1^{(t)}, C_2^{(t)}) - \widetilde{\mathcal{L}}_{\text{train}}(C_1^*, C_2^*) \leq \left(1 - \frac{\eta \mu_{1,\alpha,\sigma}}{d}\right)^t \left(\widetilde{\mathcal{L}}_{\text{train}}(C_1^{(0)}, C_2^{(0)}) - \widetilde{\mathcal{L}}_{\text{train}}(C_1^*, C_2^*)\right).$$

*Moreover, the coefficient vector $\mathbf{C}^{(t)} = \left[C_1^{(t)}, C_2^{(t)}\right]^\top$ converges linearly to $\mathbf{C}^*$ as*

$$\left\|\mathbf{C}^{(t)} - \mathbf{C}^*\right\|_2 \leq \mu_{2,\alpha,\sigma} \left(1 - \frac{\eta \mu_{1,\alpha,\sigma}}{d}\right)^{t/2} \|\mathbf{C}^*\|_2.$$

*Here, $\mu_{1,\alpha,\sigma}$ and $\mu_{2,\alpha,\sigma}$ are both positive constants solely depending on $\alpha$ and $\sigma$.*

*Proof of Theorem E.9.* Notice that we have demonstrated in the proof of Lemma E.7 that $\|\nabla^2 \widetilde{\mathcal{L}}_{\text{train}}(C_1, C_2)\|_\infty \leq \frac{16\alpha^2 \sigma^4 e^{\sigma^2} + 9\alpha e^{\sigma^2/2} \sigma^4 + 4\sigma^4}{d} = \iota$ for all $[C_1, C_2]^\top \in \mathcal{R}$, which can implies that $\|\nabla^2 \widetilde{\mathcal{L}}_{\text{train}}(C_1, C_2)\|_2 \leq \sqrt{2}\iota$. Then we can obtain that

$$\widetilde{\mathcal{L}}_{\text{train}}(C_1^{(t+1)}, C_2^{(t+1)}) \leq \widetilde{\mathcal{L}}_{\text{train}}(C_1^{(t)}, C_2^{(t)}) - \eta\left\|\nabla\widetilde{\mathcal{L}}_{\text{train}}(C_1^{(t)}, C_2^{(t)})\right\|_2^2 + \frac{\sqrt{2}\eta^2\iota}{2}\left\|\nabla\widetilde{\mathcal{L}}_{\text{train}}(C_1^{(t)}, C_2^{(t)})\right\|_2^2$$

$$\leq \widetilde{\mathcal{L}}_{\text{train}}(C_1^{(t)}, C_2^{(t)}) - \eta\left(1 - \frac{\sqrt{2}\eta\iota}{2}\right)\left\|\nabla\widetilde{\mathcal{L}}_{\text{train}}(C_1^{(t)}, C_2^{(t)})\right\|_2^2$$

$$\leq \widetilde{\mathcal{L}}_{\text{train}}(C_1^{(t)}, C_2^{(t)}) - \frac{\eta\mu_{1,\alpha,\sigma}}{d}\left(\widetilde{\mathcal{L}}_{\text{train}}(C_1^{(t)}, C_2^{(t)}) - \widetilde{\mathcal{L}}_{\text{train}}(C_1^*, C_2^*)\right),$$

where the first inequality is established by second-order Taylor's expansion and the fact that $[C_1^{(t+1)}, C_2^{(t+1)}]^\top - [C_1^{(t)}, C_2^{(t)}]^\top = \eta\nabla\widetilde{\mathcal{L}}_{\text{train}}(C_1^{(t)}, C_2^{(t)})$. The last inequality is obtained by PL condition established in Lemma E.8, and $\sqrt{2}\eta\iota \leq 1$. Then by minus $\widetilde{\mathcal{L}}_{\text{train}}(C_1^*, C_2^*)$ on both sides of the inequality above, we can obtain that

$$\widetilde{\mathcal{L}}_{\text{train}}(C_1^{(t+1)}, C_2^{(t+1)}) - \widetilde{\mathcal{L}}_{\text{train}}(C_1^*, C_2^*) \leq \left(1 - \frac{\eta\mu_{1,\alpha,\sigma}}{d}\right)\left(\widetilde{\mathcal{L}}_{\text{train}}(C_1^{(t)}, C_2^{(t)}) - \widetilde{\mathcal{L}}_{\text{train}}(C_1^*, C_2^*)\right)$$

$$\leq \cdots$$

$$\leq \left(1 - \frac{\eta\mu_{1,\alpha,\sigma}}{d}\right)^{t+1}\left(\widetilde{\mathcal{L}}_{\text{train}}(C_1^{(0)}, C_2^{(0)}) - \widetilde{\mathcal{L}}_{\text{train}}(C_1^*, C_2^*)\right). \tag{33}$$

This completes the proof for the linear convergence rate of loss decay. In the next, we prove the parameter convergence. Notice that for any $[C_1, C_2]^\top \in \mathcal{R}$, utilizing the definitions of $G_*(C_2)$ and $\Delta(C_1, C_2)$ in Lemma E.8, we have

$$\left|C_1 - C_1^*\right| = \left|\Delta(C_1, C_2) + G_*(C_2) - G_*(C_2^*)\right| \leq \left|\Delta(C_1, C_2)\right| + 2\alpha e^{\sigma^2/2}\left|C_2 - C_2^*\right|$$

where the equality holds by the definition of $\Delta(C_1, C_2)$, and the fact that $G_1^* = G_*(C_2^*)$. The second inequality is derived by triangle inequality and $|G_*'(C_2)| \leq 2\alpha e^{\sigma^2/2}$. Therefore, by $(a+b)^2 \leq 2a^2 + 2b^2$, and the inequality established above, we can further obtain that

$$\left|C_1 - C_1^*\right|^2 + \left|C_2 - C_2^*\right|^2 \leq 2\Delta^2(C_1, C_2) + \left(1 + 4\alpha e^{\sigma^2}\right)\left|C_2 - C_2^*\right|^2. \tag{34}$$

On the other hand, we can calculate that

$$\widetilde{\mathcal{L}}_{\text{train}}(C_1, C_2) - \mathcal{L}_{\text{train}}(C_1^*, C_2^*) = \frac{\sigma^2 \Delta^2(C_1, C_2) G_2(C_2)}{2d} + \psi(C_2) - \psi(C_2^*)$$

$$\geq \frac{\sigma^2}{2\pi d}\Delta^2(C_1, C_2) + \frac{2\alpha^2\sigma^2 e^{\sigma^2}}{65d}\left|C_2 - C_2^*\right|^2, \tag{35}$$

where the second inequality holds as $G_2(C_2) \geq \frac{1}{\pi}$ as demonstrated in (25), and $\psi''(C_2) \geq \frac{4\alpha^2\sigma^2 e^{\sigma^2}}{65d}$ as demonstrated in (27). Compare (34) and (35), we can obtain that for all $[C_1, C_2]^\top \in \mathcal{R}$, it holds

$$\left| C_1 - C_1^* \right|^2 + \left| C_2 - C_2^* \right|^2 \leq \frac{130\pi d \max\{2, 1 + 4\alpha^2\sigma^2 e^{\sigma^2}\}}{\min\{65\sigma^2, 4\pi\alpha^2\sigma^2 e^{\sigma^2}\}} \left( \widetilde{\mathcal{L}}_{\text{train}}(C_1, C_2) - \widetilde{\mathcal{L}}_{\text{train}}(C_1^*, C_2^*) \right). \tag{36}$$

On the other hand, for the initialization $C_1^{(0)} = C_2^{(0)} = 0$, we have $\left| G_*(C_2^{(0)}) - G_*(C_2^*) \right| \leq \mathcal{O}\left(\frac{1}{d}\right)$ and $\left| \Delta(C_1^{(0)}, C_2^{(0)}) + \alpha e^{\sigma^2/2} \right| \leq \mathcal{O}\left(\frac{1}{d}\right)$. This implies that

$$\left| C_1^{(0)} - C_1^* \right| = \left| \Delta(C_1^{(0)}, C_2^{(0)}) + G_*(C_2^{(0)}) - G_*(C_2^*) \right| \geq \frac{1}{\sqrt{2}} \left| \Delta(C_1^{(0)}, C_2^{(0)}) \right|.$$

Consequently,

$$\left\| \mathbf{C}^* \right\|_2^2 = \left\| \mathbf{C}^{(0)} - \mathbf{C}^* \right\|_2^2 = \left| C_1^{(0)} - C_1^* \right|^2 + \left| C_2^{(0)} - C_2^* \right|^2 \geq \frac{1}{2}\Delta^2(C_1^{(0)}, C_2^{(0)}) + \left| C_2^{(0)} - C_2^* \right|^2. \tag{37}$$

Similar to the calculations in (35), we can obtain an upper bound for $\widetilde{\mathcal{L}}_{\text{train}}(C_1, C_2) - \widetilde{\mathcal{L}}_{\text{train}}(C_1^*, C_2^*)$ as

$$\begin{aligned}
\widetilde{\mathcal{L}}_{\text{train}}(C_1, C_2) - \widetilde{\mathcal{L}}_{\text{train}}(C_1^*, C_2^*) &= \frac{\sigma^2 \Delta^2(C_1, C_2) G_2(C_2)}{2d} + \psi(C_2) - \psi(C_2^*) \\
&\leq \frac{5\sigma^2}{2\pi d}\Delta^2(C_1, C_2) + \frac{3\alpha^2\sigma^2 e^{\sigma^2}}{2d}\left| C_2 - C_2^* \right|^2,
\end{aligned} \tag{38}$$

where the second inequality holds as $G_2(C_2) \leq \frac{5}{\pi}$ as demonstrated in (25), and $\psi''(C_2) \geq 3\alpha^2\sigma^2 e^{\sigma^2}$ as demonstrated in (27). Compare (37) and (38), we can conclude that

$$\widetilde{\mathcal{L}}_{\text{train}}(C_1^{(0)}, C_2^{(0)}) - \widetilde{\mathcal{L}}_{\text{train}}(C_1^*, C_2^*) \leq \frac{\max\{2\sigma^2, 3\alpha^2\sigma^2 e^{\sigma^2}\}}{d} \left\| \mathbf{C}^* \right\|_2^2. \tag{39}$$

For now, we have finished all the required inequalities, and we can finally derive that

$$\begin{aligned}
\left\| \mathbf{C}^{(t)} - \mathbf{C}^* \right\|_2^2 &= \left| C_1^{(t)} - C_1^* \right|^2 + \left| C_2^{(t)} - C_2^* \right|^2 \\
&\leq \frac{130\pi d \max\{2, 1 + 4\alpha^2\sigma^2 e^{\sigma^2}\}}{\min\{65\sigma^2, 4\pi\alpha^2\sigma^2 e^{\sigma^2}\}} \left( \widetilde{\mathcal{L}}_{\text{train}}(C_1^{(t)}, C_2^{(t)}) - \mathcal{L}_{\text{train}}(C_1^*, C_2^*) \right) \\
&\leq \frac{130\pi d \max\{2, 1 + 4\alpha^2\sigma^2 e^{\sigma^2}\}}{\min\{65\sigma^2, 4\pi\alpha^2\sigma^2 e^{\sigma^2}\}} \left( 1 - \frac{\eta\mu_{1,\alpha,\sigma}}{d} \right)^t \left( \widetilde{\mathcal{L}}_{\text{train}}(C_1^{(0)}, C_2^{(0)}) - \mathcal{L}_{\text{train}}(C_1^*, C_2^*) \right) \\
&\leq \frac{130\pi \max\{2, 1 + 4\alpha^2\sigma^2 e^{\sigma^2}\} \cdot \max\{2\sigma^2, 3\alpha^2\sigma^2 e^{\sigma^2}\}}{\min\{65\sigma^2, 4\pi\alpha^2\sigma^2 e^{\sigma^2}\}} \left( 1 - \frac{\eta\mu_{1,\alpha,\sigma}}{d} \right)^t \left\| \mathbf{C}^* \right\|_2^2 \\
&= \mu_{2,\alpha,\sigma}^2 \left( 1 - \frac{\eta\mu_{1,\alpha,\sigma}}{d} \right)^t \left\| \mathbf{C}^* \right\|_2^2.
\end{aligned}$$

Here, the inequality is established by (36). The second inequality is derived by (33). And the last inequality is obtained by implying the results of (39). This completes the proof.

$\square$

## F. Proof of Theorem 3.7

In this section, we present the proof of Theorem 3.7. We begin by outlining the key insights underlying the argument. Corollary 3.3 establishes that the weight prediction produced by an $L$-layer transformer, when parameterized as in Theorem 3.1, converges to the maximum-margin solution on the context data. Moreover, Theorem 3.5 shows that, under supervision from a one-step GD, the parameter matrices $\mathbf{V}^{(t)}$ and $\mathbf{W}^{(t)}$ of the self-attention layer converge to the specific parameterization described in Theorem 3.1. Taken together, these results imply that it suffices to derive an upper bound on the Euclidean distance between the maximum-margin solution $\boldsymbol{\theta}_{\text{SVM}}(\mathbf{Z}_0)$ and the ground truth $\boldsymbol{\theta}^*$. We first introduce several lemmas which will be utilized in proof.

**Lemma F.1.** *Suppose that $\mathbf{x}_i$ are i.i.d. samples from a $d$-dimensional log-concave distribution with a positive definite covariance matrix $\boldsymbol{\Sigma}$ for all $i \in [n]$. Then for any $\delta > 0$ and any $\boldsymbol{\theta} \in \mathbb{S}^{d-1}$, the following conclusion holds with probability at least $1 - \delta$:*

$$|\langle \boldsymbol{\theta}, \mathbf{x}_i \rangle| \geq \frac{\sqrt{\lambda_{\min}(\boldsymbol{\Sigma})}\delta}{2n}, \quad \text{for all } i \in [n].$$

*Proof of Lemma F.1.* For all $i \in [n]$, we know that $\langle \boldsymbol{\theta}, \mathbf{x}_i \rangle$ follows a one-dimensional log-concave distribution with mean $\mu_l = \langle \boldsymbol{\theta}, \mathbb{E}[\mathbf{x}_i] \rangle$ and variance $\sigma_l^2 = \boldsymbol{\theta}^\top \boldsymbol{\Sigma} \boldsymbol{\theta}$. This implies that $\frac{\langle \boldsymbol{\theta}, \mathbf{x}_i \rangle - \mu_l}{\sigma_l}$ follows one-dimensional isotropic log-concave distribution. Consequently, for any $s > 0$, we have

$$\mathbb{P}(|\langle \boldsymbol{\theta}, \mathbf{x}_i \rangle| < s) = \mathbb{P}\left( \left| \frac{\langle \boldsymbol{\theta}, \mathbf{x}_i \rangle - \mu_l}{\sigma_l} + \frac{\mu_l}{\sigma_l} \right| < \frac{s}{\sigma_l} \right)$$

$$= \mathbb{P}\left( -\frac{s}{\sigma_l} - \frac{\mu_l}{\sigma_l} \leq \frac{\langle \boldsymbol{\theta}, \mathbf{x}_i \rangle - \mu_l}{\sigma_l} < \frac{s}{\sigma_l} - \frac{\mu_l}{\sigma_l} \right) \leq \frac{2s}{\sigma_l},$$

where the last inequality holds as the probability density function of isotropic log concave distribution is always smaller than 1 as demonstrated in Lemma H.23. Taking a union bound for all $i \in [n]$, we have

$$\mathbb{P}\left( \min_{i \in [n]} |\langle \boldsymbol{\theta}, \mathbf{x}_i \rangle| < s \right) \leq \sum_{i=1}^{n} \mathbb{P}(|\langle \boldsymbol{\theta}, \mathbf{x}_i \rangle| < s) \leq \frac{2ns}{\sigma_l}.$$

Let the right hand side of the above inequality smaller than $\delta$, we derive that $s \geq \frac{\sigma_l \delta}{2n} \geq \frac{\sqrt{\lambda_{\min}(\boldsymbol{\Sigma})}\delta}{2n}$, which completes the proof. □

**Lemma F.2.** *Suppose that $\mathbf{x}_i$ are i.i.d. samples from a $d$-dimensional log-concave distribution with mean vector $\boldsymbol{\mu}$ and a positive definite covariance matrix $\boldsymbol{\Sigma}$ for all $i \in [n]$. Then for any $\delta > 0$, the following conclusion holds with probability at least $1 - \delta$:*

$$\|\mathbf{x}_i\|_2 \leq \mathcal{O}\left( \sqrt{d} + \log\left( \frac{n}{\delta} \right) \right), \quad \text{for all } i \in [n].$$

*Proof.* Notice that $\widetilde{\mathbf{x}}_i = \boldsymbol{\Sigma}^{-\frac{1}{2}}(\mathbf{x}_i - \boldsymbol{\mu})$ follows a $d$-dimensional isotropic log-concave distribution. Then by Lemma H.25, for any $s > 1$ and $i \in [n]$, it holds

$$\mathbb{P}(\|\widetilde{\mathbf{x}}_i\|_2 \geq s\sqrt{d}) \leq e^{-\frac{s\sqrt{d}}{c}},$$

where $c$ is an absolute positive constant. In addition, we have

$$\|\mathbf{x}_i\|_2 \leq \left\| \boldsymbol{\Sigma}^{\frac{1}{2}} \widetilde{\mathbf{x}}_i \right\|_2 + \|\boldsymbol{\mu}\|_2 \leq \sqrt{\lambda_{\max}(\boldsymbol{\Sigma})} \|\widetilde{\mathbf{x}}_i\|_2 + \|\boldsymbol{\mu}\|_2.$$

Combine these two results, we can obtain that for any $s > 1$,

$$\mathbb{P}(\|\mathbf{x}_i\|_2 \geq \sqrt{\lambda_{\max}(\boldsymbol{\Sigma})}s\sqrt{d} + \|\boldsymbol{\mu}\|_2) \leq \mathbb{P}(\|\widetilde{\mathbf{x}}_i\|_2 \geq s\sqrt{d}) \leq e^{-\frac{s\sqrt{d}}{c}}.$$

Taking a union bound for all $i \in [n]$, we have

$$\mathbb{P}\left( \max_{i \in [n]} \|\mathbf{x}_i\|_2 \geq \sqrt{\lambda_{\max}(\boldsymbol{\Sigma})}s\sqrt{d} + \|\boldsymbol{\mu}\|_2 \right) \leq \sum_{i=1}^{n} \mathbb{P}(\|\mathbf{x}_i\|_2 \geq \sqrt{\lambda_{\max}(\boldsymbol{\Sigma})}s\sqrt{d} + \|\boldsymbol{\mu}\|_2) \leq n e^{-\frac{s\sqrt{d}}{c}}.$$

Let the right hand side of the above inequality smaller than $\delta$, we derive that $s \geq \frac{c}{\sqrt{d}} \log\left( \frac{n}{\delta} \right)$. Therefore, by setting $s = 1 + \frac{c}{\sqrt{d}} \log\left( \frac{n}{\delta} \right)$, we complete the proof. □

Now, we are ready to prove Theorem 3.7

*Proof of Theorem 3.7.* Combining the results of Theorem D.1, and Theorem 3.5, we know that the weight prediction $\boldsymbol{\theta}_L = [\mathrm{TF}(\mathbf{Z}_0; [\mathbf{V}^{(t)}]^{\otimes L}, [\mathbf{W}^{(t)}]^{\otimes L})]_{d+1:2d,n+1}$ of the looped transformers equals to that obtained by applying normalized gradient descent on rescaled dataset $\{(C_2^{(t)}\mathbf{x}_i, y_i)\}_{i=1}^n$ with a learning rate $C_1^{(t)}/C_2^{(t)}$. In addition, the rescaled dataset $\{(C_2^{(t)}\mathbf{x}_i, y_i)\}_{i=1}^n$ shares the same maximum margin solution with the original context data, i.e. $\boldsymbol{\theta}_{\mathrm{SVM}}(\mathbf{Z}_0)$. When $t \geq \Omega\left(\frac{d[\log(3\|\mathbf{C}^*\|_2) - \log(1 \wedge \alpha e^{\sigma^2/2})]}{\eta\mu_{1,\alpha,\sigma}}\right)$, the convergence results in Theorem 3.5 implies that

$$\frac{\alpha e^{\sigma^2/2}}{2} \leq C_1^{(t)} \leq 2\alpha e^{\sigma^2/2}; \quad \frac{1}{2} \leq C_2^{(t)} \leq 2.$$

Consequently, we have $\frac{C_1^{(t)}}{C_2^{(t)}} = \Theta(\alpha) \leq \mathcal{O}(1)$, and Theorem 3.3 guarantees that

$$\left\|\frac{\boldsymbol{\theta}_L}{\|\boldsymbol{\theta}_L\|_2} - \boldsymbol{\theta}_{\mathrm{SVM}}(\mathbf{Z}_0)\right\|_2 \leq \mathcal{O}\left(\frac{\log n}{\alpha L}\right). \tag{40}$$

This result demonstrates that it suffices to provide an upper bound for $\|\boldsymbol{\theta}_{\mathrm{SVM}}(\mathbf{Z}_0) - \boldsymbol{\theta}^*\|_2$, which can be converted to the test error of $\boldsymbol{\theta}_{\mathrm{SVM}}(\mathbf{Z}_0)$ as

$$\left\|\boldsymbol{\theta}_{\mathrm{SVM}}(\mathbf{Z}_0) - \boldsymbol{\theta}^*\right\|_2 = 2\sin\left(\frac{\angle(\boldsymbol{\theta}_{\mathrm{SVM}}(\mathbf{Z}_0), \boldsymbol{\theta}^*)}{2}\right) \leq \angle(\boldsymbol{\theta}_{\mathrm{SVM}}(\mathbf{Z}_0), \boldsymbol{\theta}^*) \tag{41}$$

$$\leq c_-^{-1}\mathbb{P}_{\mathbf{x} \sim \mathcal{D}_{\mathbf{x}}}\left(\mathrm{sign}(\langle\boldsymbol{\theta}_{\mathrm{SVM}}(\mathbf{Z}_0), \mathbf{x}\rangle) \neq \mathrm{sign}(\langle\boldsymbol{\theta}^*, \mathbf{x}\rangle)\right) = c_-^{-1}p\left(\boldsymbol{\theta}_{\mathrm{SVM}}(\mathbf{Z}_0)\right), \tag{42}$$

where the last inequality holds by Lemma H.24, and $c_-$ is an absolute positive constant. In addition, $p(\boldsymbol{\theta})$ is defined as:

$$p(\boldsymbol{\theta}) = \mathbb{P}_{\mathbf{x} \sim \mathcal{D}_{\mathbf{x}}}\left(\mathrm{sign}(\langle\boldsymbol{\theta}, \mathbf{x}\rangle) \neq \mathrm{sign}(\langle\boldsymbol{\theta}^*, \mathbf{x}\rangle)\right)$$

for all $\boldsymbol{\theta} \in \mathbb{S}^{d-1}$, exactly representing the test error of $\boldsymbol{\theta}$. In the following, we focus on providing the upper bound for this term. By Lemma F.1 and Lemma F.2, with probability at least $1 - \frac{2}{3}\delta$, it holds that

$$\min_{i \in [n]} y_i\langle\boldsymbol{\theta}^*, \mathbf{x}_i\rangle \geq \frac{\sqrt{\lambda_{\min}(\boldsymbol{\Sigma})}\delta}{6n}; \quad \max_{i \in [n]}\|\mathbf{x}_i\|_2 \leq c_1\left(\sqrt{d} + \log\left(\frac{n}{\delta}\right)\right), \tag{43}$$

where $c_1$ is an absolute positive constant. We set $\epsilon = \frac{\sqrt{\lambda_{\min}(\boldsymbol{\Sigma})}\delta}{12c_1 n\left(\sqrt{d} + \log(n\delta^{-1})\right)}$, and denote $N(\mathbb{S}^{d-1}, \epsilon)$ as the $\epsilon$-net on the $d$-dimensional unit sphere $\mathbb{S}^{d-1}$. Then by the definition of $\epsilon$-net, there exist $\widetilde{\boldsymbol{\theta}} \in N(\mathbb{S}^{d-1}, \epsilon)$ such that $\|\boldsymbol{\theta}_{\mathrm{SVM}}(\mathbf{Z}_0) - \widetilde{\boldsymbol{\theta}}\|_2 \leq \epsilon$. Then for $\widetilde{\boldsymbol{\theta}}$, it can still achieve zero classification error on context dataset $\{(\mathbf{x}_i, y_i)\}_{i=1}^n$, as for any $i \in [n]$,

$$y_i\langle\mathbf{x}_i, \widetilde{\boldsymbol{\theta}}\rangle \geq \min_{i \in [n]} y_i\langle\mathbf{x}_i, \boldsymbol{\theta}_{\mathrm{SVM}}(\mathbf{Z}_0)\rangle + y_i\langle\mathbf{x}_i, \widetilde{\boldsymbol{\theta}} - \boldsymbol{\theta}_{\mathrm{SVM}}(\mathbf{Z}_0)\rangle$$

$$\geq \min_{i \in [n]} y_i\langle\mathbf{x}_i, \boldsymbol{\theta}^*\rangle - \|\mathbf{x}_i\|_2\|\boldsymbol{\theta} - \boldsymbol{\theta}_{\mathrm{SVM}}(\mathbf{Z}_0)\|_2$$

$$\geq \frac{\sqrt{\lambda_{\min}(\boldsymbol{\Sigma})}\delta}{6n} - c_1\left(\sqrt{d} + \log\left(\frac{n}{\delta}\right)\right)\epsilon \geq \frac{\sqrt{\lambda_{\min}(\boldsymbol{\Sigma})}\delta}{12n}.$$

Here, the second inequality holds as $\min_{i \in [n]} y_i\langle\mathbf{x}_i, \boldsymbol{\theta}_{\mathrm{SVM}}(\mathbf{Z}_0)\rangle \geq \min_{i \in [n]} y_i\langle\mathbf{x}_i, \boldsymbol{\theta}^*\rangle$ by the definition of SVM solution, and $y_i\langle\mathbf{x}_i, \widetilde{\boldsymbol{\theta}} - \boldsymbol{\theta}_{\mathrm{SVM}}(\mathbf{Z}_0)\rangle \geq -\|\mathbf{x}_i\|_2\|\boldsymbol{\theta} - \boldsymbol{\theta}_{\mathrm{SVM}}(\mathbf{Z}_0)\|_2$ by Cauchy-Schwarz inequality. The third inequality is derived by (43), and the last inequality holds by our definition of $\epsilon$. Then for any $\varepsilon > 0$, we can derive that

$$\mathbb{P}\left(p(\widetilde{\boldsymbol{\theta}}) \geq \varepsilon\right) \leq \mathbb{P}\left(\{\exists\,\boldsymbol{\theta} \in N(\mathbb{S}^{d-1}, \epsilon) : p(\boldsymbol{\theta}) \geq \varepsilon, \text{ and } \min_{i \in [n]} y_i\langle\boldsymbol{\theta}, \mathbf{x}_i\rangle > 0\}\right)$$

$$\leq \left|N(\mathbb{S}^{d-1}, \epsilon)\right|e^{-n\varepsilon} \leq \left(\frac{3}{\epsilon}\right)^d e^{-n\varepsilon}.$$

Here, the second inequality holds as for any $\boldsymbol{\theta} \in \mathbb{S}^{d-1}$, $\mathbb{P}(\min_{i \in [n]} y_i\langle\boldsymbol{\theta}, \mathbf{x}_i\rangle > 0) \leq (1 - p(\boldsymbol{\theta}))^n \leq e^{-np(\boldsymbol{\theta})}$, and we take an union bound for all $\boldsymbol{\theta} \in N(\mathbb{S}^{d-1}, \epsilon)$. The last inequality holds as Lemma H.26 guarantees that $\left|N(\mathbb{S}^{d-1}, \epsilon)\right| \leq \left(\frac{3}{\epsilon}\right)^d$.

By setting $\left(\frac{3}{\epsilon}\right)^d e^{-n\varepsilon} \leq \frac{\delta}{3}$ and replacing the definition of $\epsilon$, we can derive that $\varepsilon = \Theta\left(\frac{d\log\left(\frac{n\sqrt{d}}{\sqrt{\lambda_{\min}(\boldsymbol{\Sigma})\delta^2}}\right)}{n}\right)$. Therefore, combined with fact that (43) holds with probability at least $1 - \frac{2\delta}{3}$, we can conclude that

$$p(\widetilde{\boldsymbol{\theta}}) \leq \mathcal{O}\left(\frac{d\left[\log(n\sqrt{d}) - \log(\sqrt{\lambda_{\min}(\boldsymbol{\Sigma})\delta^2})\right]}{n}\right)$$

holds with probability at least $1 - \delta$. On the other hand, we can further calculate that

$$\begin{aligned}
p(\widetilde{\boldsymbol{\theta}}_{\mathrm{SVM}}(\mathbf{Z}_0)) &\leq p(\widetilde{\boldsymbol{\theta}}) + \mathbb{P}_{\mathbf{x}\sim\mathcal{D}_{\mathbf{x}}}\left(\mathrm{sign}(\langle\boldsymbol{\theta}_{\mathrm{SVM}}(\mathbf{Z}_0),\mathbf{x}\rangle) \neq \mathrm{sign}(\langle\widetilde{\boldsymbol{\theta}},\mathbf{x}\rangle)\right) \\
&\leq p(\widetilde{\boldsymbol{\theta}}) + c_+\pi\left\|\boldsymbol{\theta}_{\mathrm{SVM}}(\mathbf{Z}_0) - \widetilde{\boldsymbol{\theta}}\right\|_2 \leq p(\widetilde{\boldsymbol{\theta}}) + c_+\pi\epsilon \\
&\leq \mathcal{O}\left(\frac{d\left[\log(n\sqrt{d}) - \log(\sqrt{\lambda_{\min}(\boldsymbol{\Sigma})\delta^2})\right]}{n}\right).
\end{aligned}$$

Here, the second inequality holds by Lemma H.24, where $c_+$ is a positive absolute constant. The third inequality is derived as $\left\|\boldsymbol{\theta}_{\mathrm{SVM}}(\mathbf{Z}_0) - \widetilde{\boldsymbol{\theta}}\right\|_2 \leq \epsilon$ by the choice of $\widetilde{\boldsymbol{\theta}}$ in $N(\mathbb{S}^{d-1}, \epsilon)$. And the last inequality holds as $\epsilon = \frac{\sqrt{\lambda_{\min}(\boldsymbol{\Sigma})\delta}}{12c_1 n\left(\sqrt{d} + \log(n\delta^{-1})\right)} \leq \mathcal{O}\left(\frac{d}{n}\right)$. Combined this result with (40) and (41), we finally conclude that

$$\begin{aligned}
\left\|\frac{\boldsymbol{\theta}_L}{\|\boldsymbol{\theta}_L\|_2} - \boldsymbol{\theta}^*\right\|_2 &\leq \left\|\frac{\boldsymbol{\theta}_L}{\|\boldsymbol{\theta}_L\|_2} - \boldsymbol{\theta}_{\mathrm{SVM}}(\mathbf{Z}_0)\right\|_2 + \left\|\boldsymbol{\theta}_{\mathrm{SVM}}(\mathbf{Z}_0) - \boldsymbol{\theta}^*\right\|_2 \\
&\leq \mathcal{O}\left(\frac{\log n}{\alpha L} + \frac{d\log\left(\max\{n, d, \lambda_{\min}(\boldsymbol{\Sigma})^{-1}, \delta^{-1}\}\right)}{n}\right).
\end{aligned}$$

This completes the proof. $\qquad\square$

## G. Additional experimental results

In this section, we provide further experimental results. We first plot the heatmaps of trained parameter matrices $\mathbf{V}^{(t)}$ and $\mathbf{W}^{(t)}$ for $(n, d) \in \{(100, 25), (150, 30)\}$.

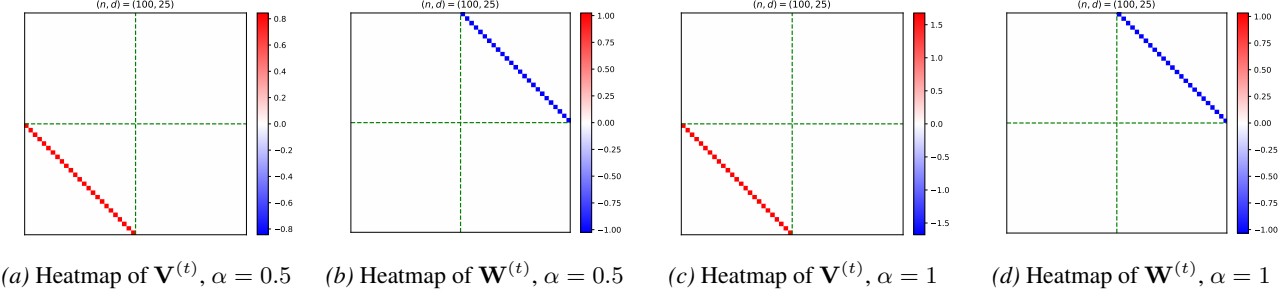

*(a)* Heatmap of $\mathbf{V}^{(t)}$, $\alpha = 0.5$    *(b)* Heatmap of $\mathbf{W}^{(t)}$, $\alpha = 0.5$    *(c)* Heatmap of $\mathbf{V}^{(t)}$, $\alpha = 1$    *(d)* Heatmap of $\mathbf{W}^{(t)}$, $\alpha = 1$

*Figure 8.* Heatmaps of the parameters matrices $\mathbf{V}^{(t)}$ and $\mathbf{W}^{(t)}$ when training loss converge. The results are presented under two different settings that $\alpha = 0.5$ and $\alpha = 1$, with $(n, d) = (100, 25)$.

Figure 8 and 9 show that under the setting $(n, d) = (100, 25)$ and $(n, d) = (150, 30)$, the trained matrices still follow the structured pattern described in Lemma 3.4.

In addition, we further validate the equivalence between softmax transformers and normalized gradient descent in Theorem 3.1 on real-world datasets. We consider two datasets from different modalities: MNIST for image classification and SST-2 for sentiment classification. For MNIST, we use the binary classification task between digits 1 and 2, and convert each image into a $d = 20$ dimensional vector representation using a trained CNN encoder. For SST-2, we use a pretrained BERT encoder followed by a projection layer to obtain $d = 20$ dimensional sentence representations. In both cases, given the encoded feature vector $\mathbf{x}_i$ and binary label $y_i \in \{-1, 1\}$, we form the signed feature vector $\mathbf{z}_i = y_i \cdot \mathbf{x}_i$ and construct the input matrix $\mathbf{Z}_0$ in the same form as (1). We then compare the hidden-layer outputs of the manually

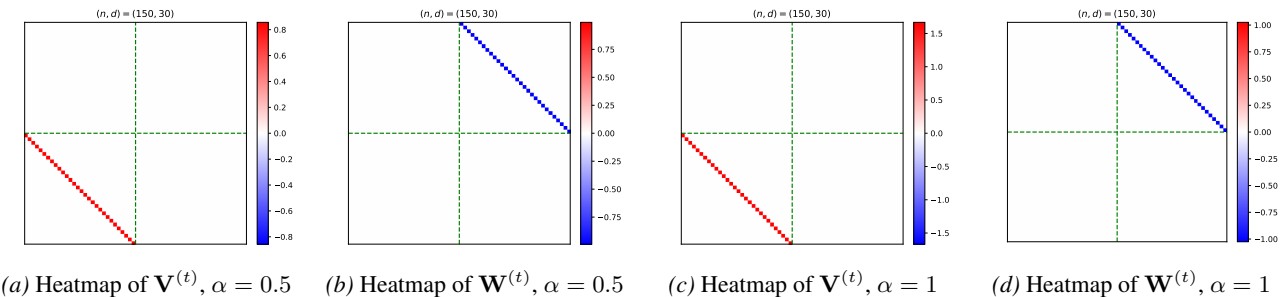

*(a)* Heatmap of $\mathbf{V}^{(t)}$, $\alpha = 0.5$     *(b)* Heatmap of $\mathbf{W}^{(t)}$, $\alpha = 0.5$     *(c)* Heatmap of $\mathbf{V}^{(t)}$, $\alpha = 1$     *(d)* Heatmap of $\mathbf{W}^{(t)}$, $\alpha = 1$

*Figure 9.* Heatmaps of the parameters matrices $\mathbf{V}^{(t)}$ and $\mathbf{W}^{(t)}$ when training loss converge. The results are presented under two different settings that $\alpha = 0.5$ and $\alpha = 1$, with $(n, d) = (150, 30)$.

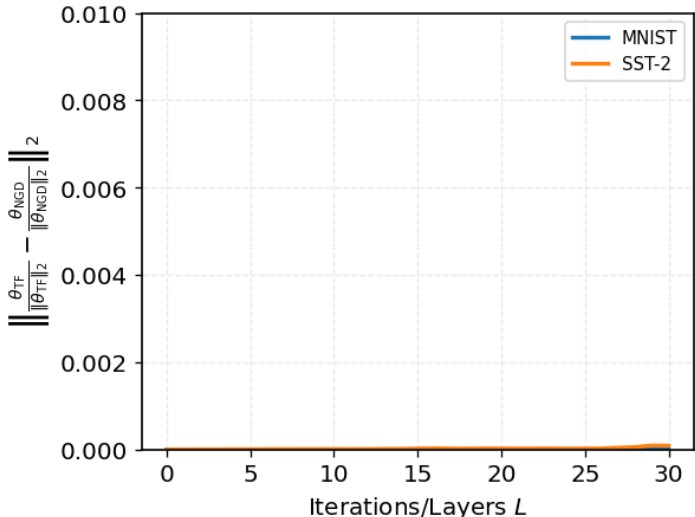

*Figure 10.* Difference between the normalized transformer output and the normalized NGD iterate on MNIST and SST-2.

constructed looped softmax transformer in Theorem 3.1 with the corresponding normalized-gradient-descent iterates over $L = 30$ layers/iterations.

Figure 10 reports the discrepancy $\left\| \frac{\boldsymbol{\theta}_{\mathrm{TF}}}{\|\boldsymbol{\theta}_{\mathrm{TF}}\|_2} - \frac{\boldsymbol{\theta}_{\mathrm{NGD}}}{\|\boldsymbol{\theta}_{\mathrm{NGD}}\|_2} \right\|_2$ between the transformer output and the NGD iterate across layers. The discrepancy remains negligible on both MNIST and SST-2 throughout the entire trajectory. This shows that, after raw inputs are converted into vector representations by standard encoders, the constructed softmax transformer continues to closely match the NGD dynamics. These results provide empirical evidence that the equivalence in Theorem 3.1 is robust across different input modalities, including image and text data.

## H. Technical lemmas

### H.1. Expectation calculations

**Lemma H.1.** *Let $x_1, x_2, \ldots, x_n \sim \mathcal{N}(0, \sigma^2)$ be $n$ i.i.d Gaussian random variables, and $\widetilde{x}_1, \widetilde{x}_2, \ldots, \widetilde{x}_n \sim \mathcal{N}(0, \sigma^2)$ be another $n$ i.i.d Gaussian random variables. In addition, $a$ is a positive scalar, and $\boldsymbol{\theta}_*, \boldsymbol{\theta}_0 \in \mathbb{R}^d$ are two independent random vectors following uniform $d$-dimensional sphere distribution, then we have*

$$\left| \mathbb{E}\left[ \frac{\sum_{i_1=1}^{n} \sum_{i_2=1}^{n} e^{-\langle \boldsymbol{\theta}_0, \boldsymbol{\theta}_* \rangle (|\widetilde{x}_{i_1}| + a|\widetilde{x}_{i_2}|) - \|(\mathbf{I}_d - \boldsymbol{\theta}_* \boldsymbol{\theta}_*^\top)\boldsymbol{\theta}_0\|_2 (x_{i_1} + a x_{i_2}) } x_{i_1} x_{i_2}}{\sum_{i=1}^{n} e^{-a\langle \boldsymbol{\theta}_0, \boldsymbol{\theta}_* \rangle |\widetilde{x}_i| - a\|(\mathbf{I}_d - \boldsymbol{\theta}_* \boldsymbol{\theta}_*^\top)\boldsymbol{\theta}_0\|_2 x_i} } \right] - 2na\sigma^4 e^{\frac{\sigma^2}{2}} B \right| \leq f_1(a, \sigma).$$

*Here, $B = \mathbb{E}[\|(\mathbf{I}_d - \boldsymbol{\theta}_* \boldsymbol{\theta}_*^\top)\boldsymbol{\theta}_0\|_2^2 \Phi(-\langle \boldsymbol{\theta}_0, \boldsymbol{\theta}_* \rangle \sigma)]$ is an absolute constant independent of $a, n$ satisfying that $\left| B - \frac{1}{2} \right| \leq \frac{c_1(a, \sigma)}{d}$, and $f_1(a, \sigma)$ is an analytic function of $a$ and $\sigma$, and irrelevant with $n, d$.*

*Proof of Lemma H.1.* We denote $K_1 = \langle \boldsymbol{\theta}_0, \boldsymbol{\theta}_* \rangle$, and $K_2 = \|(\mathbf{I}_d - \boldsymbol{\theta}_* \boldsymbol{\theta}_*^\top) \boldsymbol{\theta}_0\|_2$, then we have $K_1^2 + K_2^2 = 1$, and the p.d.f. of $K_1$ is given in Lemma H.20. In the next, by utilizing the Laplace transform identity $\frac{1}{S} = \int_0^\infty e^{-sS} ds$, we have

$$\frac{1}{\sum_{i=1}^n e^{-aK_1|\widetilde{x}_i| - aK_2 x_i}} = \int_0^\infty \exp\left( -s \sum_{i=1}^n e^{-aK_1|\widetilde{x}_i| - aK_2 x_i} \right) ds.$$

Substituting this into the expectation and utilizing Fubini's Theorem to exchange the order of integral calculations, we obtain

$$\mathbb{E}\left[ \frac{\sum_{i_1=1}^n \sum_{i_2=1}^n e^{-K_1(|\widetilde{x}_{i_1}| + a|\widetilde{x}_{i_2}|) - K_2(x_{i_1} + a x_{i_2})} x_{i_1} x_{i_2}}{\sum_{i=1}^n e^{-aK_1|\widetilde{x}_i| - aK_2 x_i}} \right]$$

$$= \int_0^\infty \mathbb{E}\left[ \sum_{i_1, i_2 = 1}^n e^{-K_1(|\widetilde{x}_{i_1}| + a|\widetilde{x}_{i_2}|) - K_2(x_{i_1} + a x_{i_2})} x_{i_1} x_{i_2} \exp\left( -s' \sum_{i=1}^n e^{-aK_1|\widetilde{x}_i| - aK_2 x_i} \right) \right] ds'$$

$$= n \mathbb{E}\left[ \int_0^\infty \mathbb{E}\left[ e^{-K_1(1+a)|\widetilde{x}_1| - K_2(1+a)x_1} x_1^2 e^{-s' e^{-a(K_1|\widetilde{x}_1| + K_2 x_1)}} | K_1 \right] \left( \mathbb{E}\left[ e^{-s' e^{-a(K_1|\widetilde{x}_1| + K_2 x_1)}} | K_1 \right] \right)^{n-1} ds' \right]$$

$$+ n(n-1) \mathbb{E}\left[ \int_0^\infty \mathbb{E}\left[ e^{-K_1(|\widetilde{x}_1| + a|\widetilde{x}_2|) - K_2(x_1 + a x_2)} x_1 x_2 e^{-s' e^{-a(K_1|\widetilde{x}_1| + K_2 x_1)} - s' e^{-a(K_1|\widetilde{x}_2| + K_2 x_2)}} | K_1 \right] \right.$$

$$\left. \cdot \left( \mathbb{E}\left[ e^{-s' e^{-a(K_1|\widetilde{x}_1| + K_2 x_1)}} | K_1 \right] \right)^{n-2} ds' \right]$$

$$= \mathbb{E}\left[ \int_0^\infty \mathbb{E}\left[ e^{-K_1(1+a)|\widetilde{x}_1| - K_2(1+a)x_1} x_1^2 e^{-\frac{s}{n} e^{-a(K_1|\widetilde{x}_1| + K_2 x_1)}} | K_1 \right] \left( \mathbb{E}\left[ e^{-\frac{s}{n} e^{-a(K_1|\widetilde{x}_1| + K_2 x_1)}} | K_1 \right] \right)^{n-1} ds \right]$$

$$+ (n-1) \mathbb{E}\left[ \int_0^\infty \mathbb{E}\left[ e^{-K_1(|\widetilde{x}_1| + a|\widetilde{x}_2|) - K_2(x_1 + a x_2)} x_1 x_2 e^{-\frac{s}{n} e^{-a(K_1|\widetilde{x}_1| + K_2 x_1)} - \frac{s}{n} e^{-a(K_1|\widetilde{x}_2| + K_2 x_2)}} | K_1 \right] \right.$$

$$\left. \cdot \left( \mathbb{E}\left[ e^{-\frac{s}{n} e^{-a(K_1|\widetilde{x}_1| + K_2 x_1)}} | K_1 \right] \right)^{n-2} ds \right]$$

$$= \mathbb{E}\left[ \underbrace{\int_0^\infty A_1(s/n, 1+a, a, K_1) [M(s/n, a, K_1)]^{n-1} ds}_{(I)} \right]$$

$$+ (n-1) \mathbb{E}\left[ \underbrace{\int_0^\infty A_2(s/n, 1, a, K_1) A_2(s/n, a, a, K_1) [M(s/n, a, K_1)]^{n-2} ds}_{(II)} \right], \tag{44}$$

where the second equation holds by the symmetries among $x_1, \ldots, x_n$ and $\widetilde{x}_1, \ldots, \widetilde{x}_n$, and the third equation holds by replacing $s'$ with $s/n$ in the integral. In addition, the terms $A_1(\lambda, \alpha, a, K_1)$, $A_2(\lambda, \alpha, a, K_1)$ and $M(\lambda, a, K_1)$ are defined as:

$$A_1(\lambda, \alpha, a, K_1) = \mathbb{E}\left[ e^{-\alpha K_1 |\widetilde{x}_1| - \alpha K_2 x_1} x_1^2 e^{-\lambda e^{-a(K_1|\widetilde{x}_1| + K_2 x_1)}} | K_1 \right];$$

$$A_2(\lambda, \alpha, a, K_1) = \mathbb{E}\left[ e^{-\alpha K_1 |\widetilde{x}_1| - \alpha K_2 x_1} x_1 e^{-\lambda e^{-a(K_1|\widetilde{x}_1| + K_2 x_1)}} | K_1 \right];$$

$$M(\lambda, a, K_1) = \mathbb{E}\left[ e^{-\lambda e^{-a(K_1|\widetilde{x}_1| + K_2 x_1)}} | K_1 \right].$$

For the term $A_1(\lambda, \alpha, a, K_1)$, by the fact that $1 - z \le e^{-z} \le 1$ for all $z > 0$, we have that

$$A_1(\lambda, \alpha, a, K_1) \ge \mathbb{E}\left[ e^{-\alpha K_1 |\widetilde{x}_1| - \alpha K_2 x_1} x_1^2 | K_1 \right] - \lambda \mathbb{E}\left[ e^{-(a+\alpha) K_1 |\widetilde{x}_1| - (a+\alpha) K_2 x_1} x_1^2 | K_1 \right];$$

$$A_1(\lambda, \alpha, a, K_1) \le \mathbb{E}\left[ e^{-\alpha K_1 |\widetilde{x}_1| - \alpha K_2 x_1} x_1^2 | K_1 \right]. \tag{45}$$

We can establish the upper and lower bounds for the term $(I)$ based on the inequalities (45) above. We first derive the lower bound as

$$(I) \ge \int_0^\infty \mathbb{E}\left[ e^{-K_1(1+a)|\widetilde{x}_1| - K_2(1+a)x_1} x_1^2 | K_1 \right] [M(s/n, a, K_1)]^{n-1} ds$$

$$- \frac{1}{n} \int_0^\infty s \mathbb{E}\left[ e^{-K_1(1+2a)|\widetilde{x}_1| - K_2(1+2a)x_1} x_1^2 | K_1 \right] [M(s/n, a, K_1)]^{n-1} ds$$

$$=2\sigma^2\big(K_2^2(1+a)^2\sigma^2+1\big)e^{\frac{(a+1)^2\sigma^2}{2}}\Phi(-K_1(a+1)\sigma)\int_0^\infty\Big(\mathbb{E}\big[e^{-\frac{s}{n}e^{-a(K_1|\widetilde{x}_1|+K_2x_1)}}|K_1\big]\Big)^{n-1}\mathrm{d}s$$

$$-\frac{K_2^2(1+2a)^2\sigma^2+1}{n}e^{\frac{(2a+1)^2\sigma^2}{2}}2\Phi(-K_1(2a+1)\sigma)\int_0^\infty s\Big(\mathbb{E}\big[e^{-\frac{s}{n}e^{-a(K_1|\widetilde{x}_1|+K_2x_1)}}|K_1\big]\Big)^{n-1}\mathrm{d}s$$

$$=2\sigma^2\big(K_2^2(1+a)^2\sigma^2+1\big)e^{\frac{(a+1)^2\sigma^2}{2}}\Phi(-K_1(a+1)\sigma)\mathbb{E}\bigg[\int_0^\infty e^{-\frac{s}{n}\sum_{i=2}^n e^{-a(K_1|\widetilde{x}_i|+K_2x_i)}}\mathrm{d}s\Big|K_1\bigg]$$

$$-\frac{K_2^2(1+2a)^2\sigma^2+1}{n}e^{\frac{(2a+1)^2\sigma^2}{2}}2\Phi(-K_1(2a+1)\sigma)\mathbb{E}\bigg[\int_0^\infty se^{-\frac{s}{n}\sum_{i=2}^n e^{-a(K_1|\widetilde{x}_i|+K_2x_i)}}\mathrm{d}s\Big|K_1\bigg]$$

$$=2\sigma^2\big(K_2^2(1+a)^2\sigma^2+1\big)e^{\frac{(a+1)^2\sigma^2}{2}}\Phi(-K_1(a+1)\sigma)\mathbb{E}\bigg[\frac{n}{\sum_{i=2}^n e^{-a(K_1|\widetilde{x}_i|+K_2x_i)}}\Big|K_1\bigg]$$

$$-\frac{K_2^2(1+2a)^2\sigma^2+1}{n}e^{\frac{(2a+1)^2\sigma^2}{2}}2\Phi(-K_1(2a+1)\sigma)\mathbb{E}\bigg[\frac{n^2}{\big(\sum_{i=2}^n e^{-a(K_1|\widetilde{x}_i|+K_2x_i)}\big)^2}\Big|K_1\bigg]$$

$$\geq\sigma^2\big(K_2^2(1+a)^2\sigma^2+1\big)e^{\frac{(2a+1)\sigma^2}{2}}\frac{\Phi(-K_1(a+1)\sigma)}{\Phi(-K_1a\sigma)}-\frac{c_1(a\sigma)}{n}\geq\sigma^2 e^{\frac{(2a+1)\sigma^2}{2}}\Phi(-(a+1)\sigma)-1 \qquad (46)$$

The first equality is true because for a normal variable $z\sim\mathcal{N}(0,\sigma^2)$ and any scalar $c$, we have $\mathbb{E}[z^2e^{-cz}]=\sigma^2(c^2\sigma^2+1)e^{c^2\sigma^2/2}$ and $\mathbb{E}[e^{-c|z|}]=2e^{c^2\sigma^2/2}\Phi(-c\sigma)$. The second equality is obtained by applying Fubini's theorem to exchange the order of integration, and the third follows from direct calculation. The penultimate inequality is derived by applying Lemma H.19, where $c_1(a)$ is a constant solely depending on $a$. Following a similar (but simpler) calculation, we can also get the upper bound for $(I)$ as

$$(I)\leq\int_0^\infty\mathbb{E}\big[e^{-K_1(1+a)|\widetilde{x}_1|-K_2(1+a)x_1}x_1^2|K_1\big]\big[M(s/n,a,K_1)\big]^{n-1}\mathrm{d}s$$

$$=2\sigma^2\big(K_2^2(1+a)^2\sigma^2+1\big)e^{\frac{(a+1)^2\sigma^2}{2}}\Phi(-K_1(a+1)\sigma)\mathbb{E}\bigg[\frac{n}{\sum_{i=2}^n e^{-a(K_1|\widetilde{x}_i|+K_2x_i)}}\Big|K_1\bigg]$$

$$\leq\sigma^2\big(K_2^2(1+a)^2\sigma^2+1\big)e^{\frac{(2a+1)\sigma^2}{2}}\frac{\Phi(-K_1(a+1)\sigma)}{\Phi(-K_1a\sigma)}-\frac{c_1(a\sigma)}{n}\leq\frac{\sigma^2((1+a)^2\sigma^2+1)}{\Phi(-a\sigma)}e^{\frac{(2a+1)\sigma^2}{2}}+1. \qquad (47)$$

To calculate the upper and lower bounds for the term $(II)$ is a little tricky, and we first calculate the derivatives of $A_2(\lambda,\alpha,a,K_1)$ w.r.t. $\lambda$ as

$$\left|\frac{\mathrm{d}A_2(\lambda,\alpha,a,K_1)}{\mathrm{d}\lambda}\right|=\left|-\mathbb{E}\big[e^{-(\alpha+a)(K_1|\widetilde{x}_1|+K_2x_1)}x_1 e^{-\lambda e^{-a(K_1|\widetilde{x}_1|+K_2x_1)}}|K_1\big]\right|$$

$$\leq\mathbb{E}\big[|x_1|e^{-(\alpha+a)(K_1|\widetilde{x}_1|+K_2x_1)}\big]\leq2\sigma e^{(a+\alpha)^2\sigma^2},$$

which implies that both $A_2(\lambda,\alpha,a,K_1)$ is Lipschitz continuous w.r.t. $\lambda$. Therefore, we can further derive that

$$|A_2(\lambda,1,a,K_1)A_2(\lambda,a,a,K_1)-A_2(0,1,a,K_1)A_2(0,a,a,K_1)|$$
$$=|(A_2(\lambda,1,a,K_1)-A_2(0,1,a,K_1))A_2(\lambda,a,a,K_1)+A_2(0,1,a,K_1)(A_2(\lambda,a,a,K_1)-A_2(0,a,a,K_1))|$$
$$\leq|A_2(\lambda,1,a,K_1)-A_2(0,1,a,K_1)||A_2(\lambda,a,a,K_1)|+|A_2(0,1,a,K_1)||A_2(\lambda,a,a,K_1)-A_2(0,a,a,K_1)|$$
$$\leq|A_2(\lambda,1,a,K_1)-A_2(0,1,a,K_1)|(|A_2(\lambda,a,a,K_1)-A_2(0,a,a,K_1)|+|A_2(0,a,a,K_1)|)$$
$$+|A_2(0,1,a,K_1)||A_2(\lambda,a,a,K_1)-A_2(0,a,a,K_1)|\leq8\lambda\sigma^3 ae^{5(a\vee1)^2\sigma^2}+4\lambda^2\sigma^2 e^{8(a\vee1)^2\sigma^2}$$

where the last inequality holds by using the Lipschitz continuous properties of $A_2(\lambda,\alpha,a,K_1)$, and the facts that $A_2(0,1,a,K_1)=-2\sigma^2 K_2 e^{\sigma^2/2}\Phi(-K_1\sigma)$, $A_2(0,a,a,K_1)=-2a\sigma^2 K_2 e^{a^2\sigma^2/2}\Phi(-aK_1\sigma)$ and $|K_1|,|K_2|\leq1$. With the inequality established above and the triangle inequality, we have

$$A_2(\lambda,1,a,K_1)A_2(\lambda,a,a,K_1)\leq4a\sigma^4 e^{\frac{(a^2+1)\sigma^2}{2}}K_2^2\Phi(-K_1\sigma)\Phi(-aK_1\sigma)+8\lambda\sigma^3 ae^{5(a\vee1)^2\sigma^2}+4\lambda^2\sigma^4 e^{8(a\vee1)^2\sigma^2};$$

$$A_2(\lambda,1,a,K_1)A_2(\lambda,a,a,K_1)\geq4a\sigma^4 e^{\frac{(a^2+1)\sigma^2}{2}}K_2^2\Phi(-K_1\sigma)\Phi(-aK_1\sigma)-8\lambda\sigma^3 ae^{5(a\vee1)^2\sigma^2}-4\lambda^2\sigma^2 e^{8(a\vee1)^2\sigma^2}. \qquad (48)$$

Now, we are ready to derive the lower and upper bounds for the term $(II)$ based on (48). We first derive the lower bound as

$$
\begin{aligned}
(II) =& \int_0^\infty A_2(s/n, 1, a, K_1) A_2(s/n, a, a, K_1)[M(s/n, a, K_1)]^{n-2} \mathrm{d}s \\
\geq& 4a\sigma^4 e^{\frac{(a^2+1)\sigma^2}{2}} K_2^2 \Phi(-K_1\sigma)\Phi(-aK_1\sigma) \int_0^\infty [M(s/n, a, K_1)]^{n-2} \mathrm{d}s \\
& - \frac{8\sigma^3 a e^{5(a\vee 1)^2\sigma^2}}{n} \int_0^\infty s[M(s/n, a, K_1)]^{n-2} \mathrm{d}s - \frac{4\sigma^2 e^{8(a\vee 1)^2\sigma^2}}{n^2} \int_0^\infty s^2[M(s/n, a, K_1)]^{n-2} \mathrm{d}s \\
=& 4a\sigma^4 e^{\frac{(a^2+1)\sigma^2}{2}} K_2^2 \Phi(-K_1\sigma)\Phi(-aK_1\sigma) \mathbb{E}\left[\frac{n}{\sum_{i=3}^n e^{-a(K_1|\widetilde{x}_i|+K_2 x_i)}}\Big| K_1\right] \\
& - \frac{8\sigma^3 a e^{5(a\vee 1)^2\sigma^2}}{n}\mathbb{E}\left[\frac{n^2}{\left(\sum_{i=3}^n e^{-a(K_1|\widetilde{x}_i|+K_2 x_i)}\right)^2}\right] - \frac{8\sigma^2 e^{8(a\vee 1)^2\sigma^2}}{n^2}\mathbb{E}\left[\frac{n^3}{\left(\sum_{i=3}^n e^{-a(K_1|\widetilde{x}_i|+K_2 x_i)}\right)^3}\right] \\
\geq& 2a\sigma^4 e^{\frac{\sigma^2}{2}} K_2^2 \Phi(-K_1\sigma) - \frac{c_3(a,\sigma)}{n},
\end{aligned}
\tag{49}
$$

where $c_3(a,\sigma) = \frac{2a\sigma^3 e^{4(a\vee 1)^2 4\sigma^2}}{\Phi^2(-a\sigma)}$, a continuous function of $a$ and $\sigma$. Here the last equation is derived by Fubini's theorem to exchange the order of the integral, and the last inequality is established by Lemma H.19. Similarly, we can also obtain that

$$
\begin{aligned}
(II) \leq& 4a\sigma^4 e^{\frac{(a^2+1)\sigma^2}{2}} K_2^2 \Phi(-K_1\sigma)\Phi(-aK_1\sigma) \int_0^\infty [M(s/n, a, K_1)]^{n-2} \mathrm{d}s \\
& + \frac{8a\sigma^3 e^{5(a\vee 1)^2\sigma^2}}{n} \int_0^\infty s[M(s/n, a, K_1)]^{n-2} \mathrm{d}s + \frac{4\sigma^2 e^{8(a\vee 1)^2\sigma^2}}{n^2} \int_0^\infty s^2[M(s/n, a, K_1)]^{n-2} \mathrm{d}s \\
\leq& 2a\sigma^4 e^{\frac{\sigma^2}{2}} K_2^2 \Phi(-K_1\sigma) + \frac{c_3(a,\sigma)}{n}.
\end{aligned}
\tag{50}
$$

Substituting the results of (46), (47), (49), and (50) into (44), we complete the proof. □

**Lemma H.2.** *Let $x_1, x_2, \ldots, x_n \sim \mathcal{N}(0, \sigma^2)$ be $n$ i.i.d Gaussian random variables, and $\widetilde{x}_1, \widetilde{x}_2, \ldots, \widetilde{x}_n \sim \mathcal{N}(0, \sigma^2)$ be another $n$ i.i.d standard Gaussian random variables. In addition, $a$ is a positive scalar, and $\boldsymbol{\theta}_*, \boldsymbol{\theta}_0 \in \mathbb{R}^d$ are two independent random vectors following uniform $d-$dimensional sphere distribution, then we have*

$$
\left|\mathbb{E}\left[\frac{\sum_{i_1=1}^n \sum_{i_2=1}^n e^{-a\langle\boldsymbol{\theta}_0,\boldsymbol{\theta}_*\rangle(|\widetilde{x}_{i_1}|+|\widetilde{x}_{i_2}|)-a\|(\mathbf{I}_d-\boldsymbol{\theta}_*\boldsymbol{\theta}_*^\top)\boldsymbol{\theta}_0\|_2(x_{i_1}+x_{i_2})} x_{i_1} x_{i_2}}{\left(\sum_{i=1}^n e^{-a\langle\boldsymbol{\theta}_0,\boldsymbol{\theta}_*\rangle|\widetilde{x}_i|-a\|(\mathbf{I}_d-\boldsymbol{\theta}_*\boldsymbol{\theta}_*^\top)\boldsymbol{\theta}_0\|_2 x_i}\right)^2}\right] - a^2\sigma^4\left(1-\frac{1}{d}\right)\right| \leq \frac{f_2(a,\sigma)}{n},
$$

*where $f_2(a,\sigma)$ is an analytic function of $a$ and $\sigma$, and irrelevant with $n, d$.*

*Proof of Lemma H.2.* The proof of this lemma is quite similar to that of Lemma H.1. We repeatedly use the previous notations that $K_1 = \langle\boldsymbol{\theta}_0,\boldsymbol{\theta}_*\rangle$, and $K_2 = \|(\mathbf{I}_d - \boldsymbol{\theta}_*\boldsymbol{\theta}_*^\top)\boldsymbol{\theta}_0\|_2$. For this lemma, we leverage another identity $\frac{1}{S^2} = \int_0^\infty s e^{-sS} \mathrm{d}s$ to obtain that

$$
\frac{1}{\left(\sum_{i=1}^n e^{-aK_1|\widetilde{x}_i|-aK_2 x_i}\right)^2} = \int_0^\infty s\exp\left(-s\sum_{i=1}^n e^{-aK_1|\widetilde{x}_i|-aK_2 x_i}\right)\mathrm{d}s.
$$

Following a similar procedure in the proof of Lemma H.1, we substitute the identity above into the expectation and utilize Fubini's Theorem to exchange the order of integral calculations to obtain,

$$
\begin{aligned}
&\mathbb{E}\left[\frac{\sum_{i_1=1}^n \sum_{i_2=1}^n e^{-aK_1(|\widetilde{x}_{i_1}|+|\widetilde{x}_{i_2}|)-aK_2(x_{i_1}+x_{i_2})} x_{i_1} x_{i_2}}{\left(\sum_{i=1}^n e^{-aK_1|\widetilde{x}_i|-aK_2 x_i}\right)^2}\right] \\
&= \int_0^\infty s'\mathbb{E}\left[\sum_{i_1,i_2=1}^n e^{-aK_1(|\widetilde{x}_{i_1}|+|\widetilde{x}_{i_2}|)-aK_2(x_{i_1}+x_{i_2})} x_{i_1} x_{i_2} \exp\left(-s'\sum_{i=1}^n e^{-aK_1|\widetilde{x}_i|-aK_2 x_i}\right)\right]\mathrm{d}s' \\
&= n\mathbb{E}\left[\int_0^\infty s'\mathbb{E}\left[e^{-2aK_1|\widetilde{x}_1|-2aK_2 x_1} x_1^2 e^{-s'e^{-a(K_1|\widetilde{x}_1|+K_2 x_1)}}\Big| K_1\right]\left(\mathbb{E}\left[e^{-s'e^{-a(K_1|\widetilde{x}_1|+K_2 x_1)}}\Big| K_1\right]\right)^{n-1}\mathrm{d}s'\right]
\end{aligned}
$$

$$+ n(n-1)\mathbb{E}\left[\int_0^\infty s'\mathbb{E}\left[e^{-aK_1(|\tilde{x}_1|+|\tilde{x}_2|)-aK_2(x_1+x_2)}x_1x_2e^{-s'e^{-a(K_1|\tilde{x}_1|+K_2x_1)}-s'e^{-a(K_1|\tilde{x}_2|+K_2x_2)}}\Big|K_1\right]\right.$$
$$\left.\cdot\left(\mathbb{E}\left[e^{-s'e^{-a(K_1|\tilde{x}_1|+K_2x_1)}}\Big|K_1\right]\right)^{n-2}\mathrm{d}s'\right]$$

$$=\frac{1}{n}\mathbb{E}\left[\int_0^\infty s\mathbb{E}\left[e^{-2aK_1|\tilde{x}_1|-2aK_2x_1}x_1^2e^{-\frac{s}{n}e^{-a(K_1|\tilde{x}_1|+K_2x_1)}}\Big|K_1\right]\left(\mathbb{E}\left[e^{-\frac{s}{n}e^{-a(K_1|\tilde{x}_1|+K_2x_1)}}\Big|K_1\right]\right)^{n-1}\mathrm{d}s\right]$$
$$+\frac{n-1}{n}\mathbb{E}\left[\int_0^\infty s\mathbb{E}\left[e^{-aK_1(|\tilde{x}_1|+|\tilde{x}_2|)-aK_2(x_1+x_2)}x_1x_2e^{-\frac{s}{n}e^{-a(K_1|\tilde{x}_1|+K_2x_1)}-\frac{s}{n}e^{-a(K_1|\tilde{x}_2|+K_2x_2)}}\Big|K_1\right]\right.$$
$$\left.\cdot\left(\mathbb{E}\left[e^{-\frac{s}{n}e^{-a(K_1|\tilde{x}_1|+K_2x_1)}}\Big|K_1\right]\right)^{n-2}\mathrm{d}s\right]$$

$$=\frac{1}{n}\mathbb{E}\left[\underbrace{\int_0^\infty sA_1(s/n,2a,a,K_1)[M(s/n,a,K_1)]^{n-1}\mathrm{d}s}_{(I)}\right]$$
$$+\frac{n-1}{n}\mathbb{E}\left[\underbrace{\int_0^\infty s[A_2(s/n,a,a,K_1)]^2[M(s/n,a,K_1)]^{n-2}\mathrm{d}s}_{(II)}\right]. \tag{51}$$

Here, $A_1(\lambda,\alpha,a,K_1)$, $A_2(\lambda,\alpha,a,K_1)$ and $M(\lambda,a,K_1)$ share the same definitions as in the proof in Lemma H.1. We can use similar procedures in (46) and (47) from the proof of Lemma H.1 to calculate the upper and lower bounds for the term $(I)$ as

$$(I)\geq\int_0^\infty s\mathbb{E}\left[e^{-2aK_1|\tilde{x}_1|-2aK_2x_1}x_1^2|K_1\right][M(s/n,a,K_1)]^{n-1}\mathrm{d}s$$
$$-\frac{1}{n}\int_0^\infty s^2\mathbb{E}\left[e^{-3aK_1|\tilde{x}_1|-3aK_2x_1}x_1^2|K_1\right][M(s/n,a,K_1)]^{n-1}\mathrm{d}s$$
$$=2\sigma^2\left(4a^2K_2^2\sigma^2+1\right)e^{2a^2\sigma^2}\Phi(-2a\sigma K_1)\mathbb{E}\left[\int_0^\infty se^{-\frac{s}{n}\sum_{i=2}^n e^{-a(K_1|\tilde{x}_i|+K_2x_i)}}\mathrm{d}s\Big|K_1\right]$$
$$-\frac{9a^2\sigma^2K_2^2+1}{n}e^{\frac{9a^2\sigma^2}{2}}2\Phi(-3a\sigma K_1)\mathbb{E}\left[\int_0^\infty s^2e^{-\frac{s}{n}\sum_{i=2}^n e^{-a(K_1|\tilde{x}_i|+K_2x_i)}}\mathrm{d}s\Big|K_1\right]$$
$$=2\sigma^2\left(4a^2\sigma^2K_2^2+1\right)e^{2a^2\sigma^2}\Phi(-2a\sigma K_1)\mathbb{E}\left[\frac{n^2}{\left(\sum_{i=2}^n e^{-a(K_1|\tilde{x}_i|+K_2x_i)}\right)^2}\Big|K_1\right]$$
$$-\frac{9a^2\sigma^2K_2^2+1}{n}e^{\frac{9a^2\sigma^2}{2}}4\Phi(-3a\sigma K_1)\mathbb{E}\left[\frac{n^3}{\left(\sum_{i=2}^n e^{-a(K_1|\tilde{x}_i|+K_2x_i)}\right)^3}\Big|K_1\right]$$
$$\geq\sigma^2\left(4a^2\sigma^2K_2^2+1\right)e^{\frac{3a^2\sigma^2}{2}}\frac{\Phi(-2aK_1\sigma)}{2[\Phi(-K_1a\sigma)]^2}-\frac{c_1(a,\sigma)}{n}\geq\frac{\sigma^2e^{3a^2\sigma^2/2}\Phi(-2a\sigma)}{2}-1 \tag{52}$$

The first equality is true because for a normal random variable $z\sim\mathcal{N}(0,\sigma^2)$ and any scalar $c$, we have $\mathbb{E}[z^2e^{-cz}]=\sigma^2(c^2\sigma^2+1)e^{c^2\sigma^2/2}$ and $\mathbb{E}[e^{-c|z|}]=2e^{c^2\sigma^2/2}\Phi(-c\sigma)$. The second equality is obtained by applying Fubini's theorem to exchange the order of integration, and the third follows from direct calculation. The penultimate inequality is derived by applying Lemma H.19, where $c_1(a)$ is a constant solely depending on $a$. Following a similar (but simpler) calculation, we can also get the upper bound for $(I)$ as

$$(I)\leq\int_0^\infty s\mathbb{E}\left[e^{-2aK_1|\tilde{x}_1|-2aK_2x_1}x_1^2|K_1\right][M(s/n,a,K_1)]^{n-1}\mathrm{d}s$$
$$=2\sigma^2\left(4a^2\sigma^2K_2^2+1\right)e^{2a^2\sigma^2}\Phi(-2a\sigma K_1)\mathbb{E}\left[\frac{n^2}{\left(\sum_{i=2}^n e^{-a(K_1|\tilde{x}_i|+K_2x_i)}\right)^2}\Big|K_1\right]$$
$$\leq\sigma^2\left(4a^2\sigma^2K_2^2+1\right)e^{\frac{3a^2\sigma^2}{2}}\frac{\Phi(-2a\sigma K_1)}{2[\Phi(-K_1a\sigma)]^2}+\frac{c_1(a,\sigma)}{n}\leq\frac{\sigma^2(4a^2\sigma^2+1)}{2\Phi^2(a\sigma)}e^{\frac{3a^2\sigma^2}{2}}+1, \tag{53}$$

With the Lipschitz continuity of $A_2(\lambda, \alpha, a, K_1)$ derived in the proof of Lemma H.1, we can also demonstrate that

$$
\begin{aligned}
&\left|[A_2(\lambda, a, a, K_1)]^2 - [A_2(0, a, a, K_1)]^2\right| \\
=& |A_2(\lambda, a, a, K_1) - A_2(0, a, a, K_1)| |A_2(\lambda, a, a, K_1) + A_2(0, a, a, K_1)| \\
\leq & 2|A_2(\lambda, a, a, K_1) - A_2(0, a, a, K_1)| |A_2(0, a, a, K_1)| + |A_2(\lambda, a, a, K_1) - A_2(0, a, a, K_1)|^2 \\
\leq & 8\lambda a\sigma^3 e^{9a^2\sigma^2/2} + 4\lambda^2\sigma^2 e^{8a^2\sigma^2}.
\end{aligned}
$$

Now, by using the triangle inequality to establish upper and lower bounds for $[A_2(\lambda, a, a, K_1)]^2$ from the inequalities above, we are ready to derive the lower and upper bounds for the term $(II)$. We first derive the lower bound as

$$
\begin{aligned}
(II) =& \int_0^\infty s[A_2(s/n, a, a, K_1)]^2 [M(s/n, a, K_1)]^{n-2} ds \\
\geq & 4a^2\sigma^4 e^{a^2\sigma^2} K_2^2 [\Phi(-a\sigma K_1)]^2 \int_0^\infty s[M(s/n, a, K_1)]^{n-2} ds - \frac{8a\sigma^3 e^{9a^2\sigma^2/2}}{n} \int_0^\infty s^2 [M(s/n, a, K_1)]^{n-2} ds \\
& - \frac{4\sigma^2 e^{8a^2\sigma^2}}{n^2} \int_0^\infty s^3 [M(s/n, a, K_1)]^{n-2} ds \\
=& 4a^2\sigma^4 e^{a^2\sigma^2} K_2^2 [\Phi(-a\sigma K_1)]^2 \mathbb{E}\left[\frac{n^2}{\left(\sum_{i=3}^n e^{-a(K_1|\widetilde{x}_i|+K_2 x_i)}\right)^2}\bigg| K_1\right] \\
& - \frac{16a\sigma^3 e^{9a^2\sigma^2/2}}{n} \mathbb{E}\left[\frac{n^3}{\left(\sum_{i=3}^n e^{-a(K_1|\widetilde{x}_i|+K_2 x_i)}\right)^3}\right] - \frac{24e^{8a^2\sigma^2}}{n^2} \mathbb{E}\left[\frac{n^4}{\left(\sum_{i=3}^n e^{-a(K_1|\widetilde{x}_i|+K_2 x_i)}\right)^4}\right] \\
\geq & a^2\sigma^4 K_2^2 - \frac{c_3(a, \sigma)}{n}, \tag{54}
\end{aligned}
$$

where $c_3(a, \sigma) = \frac{3e^{3a^2\sigma^3}}{\Phi^3(-a\sigma)}$ is a positive continuous function of $a$ and $\sigma$. Here the last equation is derived by Fubini's theorem to exchange the order of the integral, and the last inequality is established by Lemma H.19. Similarly, we can also obtain that

$$
\begin{aligned}
(II) \leq & 4a^2\sigma^4 e^{a^2\sigma^2} K_2^2 [\Phi(-a\sigma K_1)]^2 \int_0^\infty s[M(s/n, a, K_1)]^{n-2} ds + \frac{8a\sigma^3 e^{9a^2,\sigma^2/2}}{n} \int_0^\infty s^2 [M(s/n, a, K_1)]^{n-2} ds \\
& + \frac{4\sigma^2 e^{8a^2}}{n^2} \int_0^\infty s^3 [M(s/n, a, K_1)]^{n-2} ds \\
\leq & a^2\sigma^4 K_2^2 + \frac{c_3(a, \sigma)}{n}. \tag{55}
\end{aligned}
$$

Substituting the results of (52), (53), (54), and (55) into (51), and utilizing the fact that $\mathbb{E}[K_2^2] = 1 - \frac{1}{d}$, we complete the proof. $\qquad\square$

**Lemma H.3.** *Let $x_1, x_2, \ldots, x_n \sim \mathcal{N}(0, \sigma^2)$ be $n$ i.i.d Gaussian random variables, and $\widetilde{x}_1, \widetilde{x}_2, \ldots, \widetilde{x}_n \sim \mathcal{N}(0, \sigma^2)$ be another $n$ i.i.d Gaussian random variables. In addition, $a$ is a positive scalar, and $\boldsymbol{\theta}_*, \boldsymbol{\theta}_0 \in \mathbb{R}^d$ are two independent random vectors following uniform $d$-dimensional sphere distribution, then we have*

$$
\left|\mathbb{E}\left[\frac{\sum_{i_1=1}^n \sum_{i_2=1}^n e^{-\langle\boldsymbol{\theta}_0, \boldsymbol{\theta}_*\rangle(|\widetilde{x}_{i_1}|+a|\widetilde{x}_{i_2}|) - \|(\mathbf{I}_d - \boldsymbol{\theta}_*\boldsymbol{\theta}_*^\top)\boldsymbol{\theta}_0\|_2(x_{i_1}+ax_{i_2})} |\widetilde{x}_{i_1}||\widetilde{x}_{i_2}|}{\sum_{i=1}^n e^{-a\langle\boldsymbol{\theta}_0, \boldsymbol{\theta}_*\rangle|\widetilde{x}_i| - a\|(\mathbf{I}_d - \boldsymbol{\theta}_*\boldsymbol{\theta}_*^\top)\boldsymbol{\theta}_0\|_2 x_i}}\right] - nB_{a,\sigma,1}\right| \leq f_3(a, \sigma) + \frac{nf_4(a, \sigma)}{d},
$$

*where $B_{a,\sigma,1} = \frac{2\sigma^2 e^{\frac{\sigma^2}{2}}}{\pi}$, and $f_3(a, \sigma), f_4(a, \sigma)$ are both smooth functions of $a$ and $\sigma$, and irrelevant with $n, d$.*

*Proof of Lemma H.3.* We repeatedly use the previous notations that $K_1 = \langle\boldsymbol{\theta}_0, \boldsymbol{\theta}_*\rangle$, $K_2 = \|(\mathbf{I}_d - \boldsymbol{\theta}_*\boldsymbol{\theta}_*^\top)\boldsymbol{\theta}_0\|_2$, and leverage $\frac{1}{S} = \int_0^\infty e^{-sS} ds$ to obtain that

$$
\frac{1}{\sum_{i=1}^n e^{-aK_1|\widetilde{x}_i| - aK_2 x_i}} = \int_0^\infty \exp\left(-s\sum_{i=1}^n e^{-aK_1|\widetilde{x}_i| - aK_2 x_i}\right) ds.
$$

Following a similar procedure in the proof of Lemma H.1, we substitute the identity above into the expectation and utilize Fubini's Theorem to exchange the order of integral calculations to obtain,

$$
\mathbb{E}\left[\frac{\sum_{i_1=1}^n \sum_{i_2=1}^n e^{-K_1(|\widetilde{x}_{i_1}|+a|\widetilde{x}_{i_2}|)-K_2(x_{i_1}+ax_{i_2})}|\widetilde{x}_{i_1}||\widetilde{x}_{i_2}|}{\sum_{i=1}^n e^{-aK_1|\widetilde{x}_i|-aK_2x_i}}\right]
$$

$$
=\int_0^\infty \mathbb{E}\left[\sum_{i_1,i_2=1}^n e^{-K_1(|\widetilde{x}_{i_1}|+a|\widetilde{x}_{i_2}|)-K_2(x_{i_1}+ax_{i_2})}|\widetilde{x}_{i_1}||\widetilde{x}_{i_2}|\exp\left(-s'\sum_{i=1}^n e^{-aK_1|\widetilde{x}_i|-aK_2x_i}\right)\right]ds'
$$

$$
=n\mathbb{E}\left[\int_0^\infty \mathbb{E}\left[e^{-K_1(1+a)|\widetilde{x}_1|-K_2(1+a)x_1}|\widetilde{x}_1|^2 e^{-s'e^{-a(K_1|\widetilde{x}_1|+K_2x_1)}}|K_1\right]\left(\mathbb{E}\left[e^{-s'e^{-a(K_1|\widetilde{x}_1|+K_2x_1)}}|K_1\right]\right)^{n-1}ds'\right]
$$

$$
+n(n-1)\mathbb{E}\left[\int_0^\infty \mathbb{E}\left[e^{-K_1(|\widetilde{x}_1|+a|\widetilde{x}_2|)-K_2(x_1+ax_2)}|\widetilde{x}_1||\widetilde{x}_2|e^{-s'e^{-a(K_1|\widetilde{x}_1|+K_2x_1)}-s'e^{-a(K_1|\widetilde{x}_2|+K_2x_2)}}|K_1\right]\right.
$$

$$
\left.\cdot\left(\mathbb{E}\left[e^{-s'e^{-a(K_1|\widetilde{x}_1|+K_2x_1)}}|K_1\right]\right)^{n-2}ds'\right]
$$

$$
=\mathbb{E}\left[\int_0^\infty \mathbb{E}\left[e^{-K_1(1+a)|\widetilde{x}_1|-K_2(1+a)x_1}|\widetilde{x}_1|^2 e^{-\frac{s}{n}e^{-a(K_1|\widetilde{x}_1|+K_2x_1)}}|K_1\right]\left(\mathbb{E}\left[e^{-\frac{s}{n}e^{-a(K_1|\widetilde{x}_1|+K_2x_1)}}|K_1\right]\right)^{n-1}ds\right]
$$

$$
+(n-1)\mathbb{E}\left[\int_0^\infty \mathbb{E}\left[e^{-K_1(|\widetilde{x}_1|+a|\widetilde{x}_2|)-K_2(x_1+ax_2)}|\widetilde{x}_1||\widetilde{x}_2|e^{-\frac{s}{n}e^{-a(K_1|\widetilde{x}_1|+K_2x_1)}-\frac{s}{n}e^{-a(K_1|\widetilde{x}_2|+K_2x_2)}}|K_1\right]\right.
$$

$$
\left.\cdot\left(\mathbb{E}\left[e^{-\frac{s}{n}e^{-a(K_1|\widetilde{x}_1|+K_2x_1)}}|K_1\right]\right)^{n-2}ds\right]
$$

$$
=\mathbb{E}\left[\underbrace{\int_0^\infty A_3(s/n,1+a,a,K_1)[M(s/n,a,K_1)]^{n-1}ds}_{(I)}\right]
$$

$$
+(n-1)\mathbb{E}\left[\underbrace{\int_0^\infty A_4(s/n,1,a,K_1)A_4(s/n,a,a,K_1)[M(s/n,a,K_1)]^{n-2}ds}_{(II)}\right], \tag{56}
$$

Here, $M(\lambda,a,K_1)$ share the same definitions as in the proof in Lemma H.1, and the terms $A_3(\lambda,\alpha,a,K_1)$, $A_4(\lambda,\alpha,a,K_1)$ are defined as:

$$
A_3(\lambda,\alpha,a,K_1)=\mathbb{E}\left[e^{-\alpha K_1|\widetilde{x}_1|-\alpha K_2x_1}|\widetilde{x}_1|^2 e^{-\lambda e^{-a(K_1|\widetilde{x}_1|+K_2x_1)}}|K_1\right];
$$

$$
A_4(\lambda,\alpha,a,K_1)=\mathbb{E}\left[e^{-\alpha K_1|\widetilde{x}_1|-\alpha K_2x_1}|\widetilde{x}_1|e^{-\lambda e^{-a(K_1|\widetilde{x}_1|+K_2x_1)}}|K_1\right].
$$

We use the fact $1-z\le e^{-z}\le 1$ to derive the upper and lower bounds for $A_3(\lambda,\alpha,a,K_1)$ as

$$
A_3(\lambda,\alpha,a,K_1)\ge \mathbb{E}\left[e^{-\alpha K_1|\widetilde{x}_1|-\alpha K_2x_1}|\widetilde{x}_1|^2|K_1\right]-\lambda\mathbb{E}\left[e^{-K_1(\alpha+a)|\widetilde{x}_1|-K_2(\alpha+a)x_1}|\widetilde{x}_1|^2|K_1\right];
$$

$$
A_3(\lambda,\alpha,a,K_1)\le \mathbb{E}\left[e^{-\alpha K_1|\widetilde{x}_1|-\alpha K_2x_1}|\widetilde{x}_1|^2|K_1\right].
$$

We can establish the upper and lower bounds for the term $(I)$ based on the inequalities above as,

$$
(I)\ge \int_0^\infty \mathbb{E}\left[e^{-K_1(1+a)|\widetilde{x}_1|-K_2(1+a)x_1}|\widetilde{x}_1|^2|K_1\right][M(s/n,a,K_1)]^{n-1}ds
$$

$$
-\frac{1}{n}\int_0^\infty s\mathbb{E}\left[e^{-K_1(1+2a)|\widetilde{x}_1|-K_2(1+2a)x_1}|\widetilde{x}_1|^2|K_1\right][M(s/n,a,K_1)]^{n-1}ds
$$

$$
=\sigma^2\left(2(1+K_1^2(1+a)^2\sigma^2)e^{\frac{K_1^2(a+1)^2\sigma^2}{2}}\Phi(-K_1(a+1)\sigma)-\sqrt{\frac{2}{\pi}}K_1(1+a)\sigma\right)
$$

$$
\cdot e^{\frac{K_2^2(a+1)^2\sigma^2}{2}}\int_0^\infty [M(s/n,a,K_1)]^{n-1}ds
$$

$$
-\frac{\sigma^2\left(2(1+K_1^2(1+2a)^2\sigma^2)e^{\frac{K_1^2(2a+1)^2\sigma^2}{2}}\Phi(-K_1(2a+1)\sigma)-\sqrt{\frac{2}{\pi}}K_1(1+2a)\sigma\right)}{n}
$$

$$\cdot e^{\frac{K_2^2(2a+1)^2\sigma^2}{2}}\int_0^\infty s[M(s/n,a,K_1)]^{n-1}\mathrm{d}s$$

$$\geq -\sigma(a+1)e^{\frac{(a+1)^2\sigma^2}{2}}\mathbb{E}\left[\frac{n}{\sum_{i=2}^n e^{-a(K_1|\widetilde{x}_i|+K_2x_i)}}\bigg|K_1\right]$$

$$-\frac{\sigma^2(8a^2+10a+5)}{n}e^{\frac{(2a+1)^2\sigma^2}{2}}\mathbb{E}\left[\frac{n^2}{\left(\sum_{i=2}^n e^{-a(K_1|\widetilde{x}_i|+K_2x_i)}\right)^2}\bigg|K_1\right]$$

$$\geq -\frac{\sigma(a+1)}{\Phi(-a\sigma)}e^{\frac{(2a+1)\sigma^2}{2}}-1 \tag{57}$$

The first equality is true because for a standard normal variable $z\sim\mathcal{N}(0,\sigma^2)$ and any scalar $c$, we have $\mathbb{E}[|z|^2e^{-c|z|}]=\sigma^2\left(2(1+c^2\sigma^2)e^{c^2\sigma^2/2}\Phi(-c\sigma)-c\sigma\sqrt{\frac{2}{\pi}}\right)$. The second equality is obtained by applying Fubini's theorem to exchange the order of integration, and the fact that $|K_1|,|K_2|\leq 1$. The last inequality is derived by applying Lemma H.19, where $c_1(a)$ is a constant solely depending on $a$. Following a similar (but simpler) calculation, we can also get the upper bound for $(I)$ as

$$(I)\leq\int_0^\infty\mathbb{E}\left[e^{-K_1(1+a)|\widetilde{x}_1|-K_2(1+a)x_1}|\widetilde{x}_1|^2|K_1\right][M(s/n,a,K_1)]^{n-1}\mathrm{d}s$$

$$\leq\sigma^2(2a^2+5a+5)e^{\frac{(a+1)^2\sigma^2}{2}}\mathbb{E}\left[\frac{n}{\sum_{i=2}^n e^{-a(K_1|\widetilde{x}_i|+K_2x_i)}}\bigg|K_1\right]$$

$$\leq\frac{\sigma^2(2a^2+5a+5)}{\Phi(-a\sigma)}e^{\frac{(2a+1)\sigma^2}{2}}+1 \tag{58}$$

Following a similar procedure in the proof of Lemma H.1, we first calculate the derivative of $A_4(\lambda,a,K_1)$ w.r.t. $\lambda$ as

$$\left|\frac{\mathrm{d}A_4(\lambda,\alpha,a,K_1)}{\mathrm{d}\lambda}\right|=\left|-\mathbb{E}\left[e^{-(a+\alpha)(K_1|\widetilde{x}_1|+K_2x_1)}|\widetilde{x}_1|e^{-\lambda e^{-a(K_1|\widetilde{x}_1|+K_2x_1)}}|K_1\right]\right|$$

$$\leq\mathbb{E}\left[|\widetilde{x}_1|e^{-(a+\alpha)(K_1|\widetilde{x}_1|+K_2x_1)}|K_1\right]\leq\sigma(2|a+\alpha|\sigma+1)e^{(a+\alpha)^2\sigma^2},$$

which implies that both $A_4(\lambda,\alpha,a,K_1)$ and is Lipschitz continuous w.r.t. $\lambda$. Therefore, we can further derive that

$$|A_4(\lambda,1,a,K_1)A_4(\lambda,a,a,K_1)-A_4(0,1,a,K_1)A_4(0,a,a,K_1)|$$
$$=|(A_4(\lambda,1,a,K_1)-A_4(0,1,a,K_1))A_4(\lambda,a,a,K_1)+A_4(0,1,a,K_1)(A_4(\lambda,a,a,K_1)-A_4(0,a,a,K_1))|$$
$$\leq|A_4(\lambda,1,a,K_1)-A_4(0,1,a,K_1)||A_4(\lambda,a,a,K_1)|+|A_4(0,1,a,K_1)||A_4(\lambda,a,a,K_1)-A_4(0,a,a,K_1)|$$
$$\leq|A_4(\lambda,1,a,K_1)-A_4(0,1,a,K_1)|(|A_4(\lambda,a,a,K_1)-A_4(0,a,a,K_1))|+|A_4(0,a,a,K_1)|)$$
$$+|A_4(0,1,a,K_1)||A_4(\lambda,a,a,K_1)-A_4(0,a,a,K_1)|$$
$$\leq 30\lambda\max\{a,1,\sigma\}^3e^{9(|a|\vee 1)^2\sigma^2/2}+25\lambda^2\max\{a,1,\sigma\}^4e^{8(|a|\vee 1)^2\sigma^2},$$

where the last inequality holds by using the Lipschitz continuous properties of $A_4(\lambda,\alpha,a,K_1)$, the facts that $A_4(0,1,a,K_1)=\sqrt{\frac{2}{\pi}}\sigma e^{K_2^2\sigma^2/2}-2\sigma^2K_1\Phi(-K_1\sigma)e^{\sigma^2/2}$, $A_4(0,a,a,K_1)=\sqrt{\frac{2}{\pi}}\sigma e^{a^2\sigma^2K_2^2/2}-2aK_1\sigma^2\Phi(-aK_1\sigma)e^{a^2\sigma^2/2}$ and $|K_1|,|K_2|\leq 1$. With the inequality established above and the triangle inequality, we calculate the lower and upper bounds for the term $(II)$ as

$$(II)=\int_0^\infty A_4(s/n,1,a,K_1)A_4(s/n,a,a,K_1)[M(s/n,a,K_1)]^{n-2}\mathrm{d}s$$

$$\geq A_4(0,1,a,K_1)A_4(0,a,a,K_1)\int_0^\infty[M(s/n,a,K_1)]^{n-2}\mathrm{d}s-$$

$$-\frac{30\max\{a,1,\sigma\}^3e^{5(|a|\vee 1)^2\sigma^2}}{n}\int_0^\infty s[M(s/n,a,K_1)]^{n-2}\mathrm{d}s$$

$$-\frac{25\max\{a,1,\sigma\}^4e^{8(|a|\vee 1)^2}}{n^2}\int_0^\infty s^2[M(s/n,a,K_1)]^{n-2}\mathrm{d}s$$

$$
\begin{aligned}
=&A_4(0,1,a,K_1)A_4(0,a,a,K_1)\mathbb{E}\left[\frac{n}{\sum_{i=3}^n e^{-a(K_1|\widetilde{x}_i|+K_2 x_i)}}\Bigg|K_1\right] \\
&-\frac{30\max\{a,1,\sigma\}^3 e^{5(|a|\vee 1)^2\sigma^2}}{n}\mathbb{E}\left[\frac{n^2}{\left(\sum_{i=3}^n e^{-a(K_1|\widetilde{x}_i|+K_2 x_i)}\right)^2}\right] \\
&-\frac{50\max\{a,1,\sigma\}^4 e^{8(|a|\vee 1)^2}}{n^2}\mathbb{E}\left[\frac{n^3}{\left(\sum_{i=3}^n e^{-a(K_1|\widetilde{x}_i|+K_2 x_i)}\right)^3}\right] \\
\geq&\frac{A_4(0,1,a,K_1)A_4(0,a,a,K_1)}{2e^{a^2\sigma^2/2}\Phi(-aK_1\sigma)}-\frac{c_3(a,\sigma)}{n},
\end{aligned}
\tag{59}
$$

where $c_3(a,\sigma)=\frac{8\max\{a,1,\sigma\}^3 e^{7(|a|\vee 1)^2\sigma^2}}{\Phi^2(-a\sigma)}$, a positive continuous function of $a$ and $\sigma$. Here the last equation is derived by Fubini's theorem to exchange the order of the integral, and the last inequality is established by Lemma H.19. Similarly, we can also obtain that

$$
\begin{aligned}
(II)\leq& A_4(0,1,a,K_1)A_4(0,a,a,K_1)\int_0^\infty [M(s/n,a,K_1)]^{n-2}\mathrm{d}s \\
&+\frac{30\max\{a,1,\sigma\}^3 e^{5(|a|\vee 1)^2\sigma^2}}{n}\int_0^\infty s[M(s/n,a,K_1)]^{n-2}\mathrm{d}s \\
&+\frac{25\max\{a,1,\sigma\}^4 e^{8(|a|\vee 1)^2}}{n^2}\int_0^\infty s^2[M(s/n,a,K_1)]^{n-2}\mathrm{d}s \\
\leq&\frac{A_4(0,1,a,K_1)A_4(0,a,a,K_1)}{2e^{a^2\sigma^2/2}\Phi(-aK_1\sigma)}+\frac{c_3(a,\sigma)}{n}.
\end{aligned}
\tag{60}
$$

To further derive a concrete bounds for the leading term $F(a,K_1)=\frac{A_4(0,1,a,K_1)A_4(0,a,a,K_1)}{2e^{a^2\sigma^2/2}\Phi(-aK_1)}$, we consider the second order Taylor's expansion of $F(a,K_1)$ at $K_1=0$. By utilizing the conclusion that $K_1$ has a symmetric distribution, we have

$$
\mathbb{E}[F(a,K_1)]=F(a,0)+\mathbb{E}\left[\frac{F''(a,\xi)K_1^2}{2}\right],
$$

where $\xi$ is a random variable between $0$ and $K_1$. Since the function $F(a,k)$ is analytic w.r.t. $k$ on $[-1,1]$, and $|K_1|\leq 1$ (hence $|\xi|\leq 1$), its second-order derivative $F''(a,k)$ is continuous and bounded on this compact interval. Therefore, there exists a constant $c_4(a,\sigma)$ such that $|F''(a,\xi)|\leq c_4(a,\sigma)$. By $\mathbb{E}[K_1^2]=1/d$ and $F(a,0)=\frac{2\sigma^2 e^{\sigma^2/2}}{\pi}$, we can further derive that

$$
\left|\mathbb{E}[F(a,K_1)]-\frac{2\sigma^2 e^{\frac{\sigma^2}{2}}}{\pi}\right|\leq\mathbb{E}\left[\frac{c_4(a,\sigma)}{d}\right].
\tag{61}
$$

Substituting these results of (57), (58), (59), (60), and (61) into (56), we complete the proof. $\qquad\square$

**Lemma H.4.** *Let $x_1,x_2,\ldots,x_n\sim\mathcal{N}(0,\sigma^2)$ be $n$ i.i.d Gaussian random variables, and $\widetilde{x}_1,\widetilde{x}_2,\ldots,\widetilde{x}_n\sim\mathcal{N}(0,\sigma^2)$ be another $n$ i.i.d standard Gaussian random variables. In addition, $a$ is a positive scalar, and $\boldsymbol{\theta}_*,\boldsymbol{\theta}_0\in\mathbb{R}^d$ are two independent random vectors following uniform $d-$dimensional sphere distribution, then we have*

$$
\left|\mathbb{E}\left[\frac{\sum_{i_1=1}^n\sum_{i_2=1}^n e^{-a\langle\boldsymbol{\theta}_0,\boldsymbol{\theta}_*\rangle(|\widetilde{x}_{i_1}|+|\widetilde{x}_{i_2}|)-a\|(\mathbf{I}_d-\boldsymbol{\theta}_*\boldsymbol{\theta}_*^\top)\boldsymbol{\theta}_0\|_2(x_{i_1}+x_{i_2})}|\widetilde{x}_{i_1}||\widetilde{x}_{i_2}|}{\left(\sum_{i=1}^n e^{-a\langle\boldsymbol{\theta}_0,\boldsymbol{\theta}_*\rangle|\widetilde{x}_i|-a\|(\mathbf{I}_d-\boldsymbol{\theta}_*\boldsymbol{\theta}_*^\top)\boldsymbol{\theta}_0\|_2 x_i}\right)^2}\right]-\frac{2\sigma^2}{\pi}\right|\leq\frac{f_5(a,\sigma)}{n}+\frac{f_6(a,\sigma)}{d},
$$

*where $f_5(a,\sigma)$ and $f_6(a,\sigma)$ are both analytic functions of $a$ and $\sigma$, and irrelevant with $n,d$.*

*Proof of Lemma H.4.* We repeatedly use the previous notations that $K_1=\langle\boldsymbol{\theta}_0,\boldsymbol{\theta}_*\rangle$, and $K_2=\|(\mathbf{I}_d-\boldsymbol{\theta}_*\boldsymbol{\theta}_*^\top)\boldsymbol{\theta}_0\|_2$. Following a similar procedure in the proof of Lemma H.2, we can obtain that

$$
\mathbb{E}\left[\frac{\sum_{i_1=1}^n\sum_{i_2=1}^n e^{-aK_1(|\widetilde{x}_{i_1}|+|\widetilde{x}_{i_2}|)-aK_2(x_{i_1}+x_{i_2})}|\widetilde{x}_{i_1}||\widetilde{x}_{i_2}|}{\left(\sum_{i=1}^n e^{-aK_1|\widetilde{x}_i|-aK_2 x_i}\right)^2}\right]
$$

$$= \int_0^\infty s' \mathbb{E}\left[\sum_{i_1,i_2=1}^n e^{-aK_1(|\widetilde{x}_{i_1}|+|\widetilde{x}_{i_2}|)-aK_2(x_{i_1}+x_{i_2})}|\widetilde{x}_{i_1}||\widetilde{x}_{i_2}|\exp\left(-s'\sum_{i=1}^n e^{-aK_1|\widetilde{x}_i|-aK_2x_i}\right)\right]ds'$$

$$= n\mathbb{E}\left[\int_0^\infty s'\mathbb{E}\left[e^{-2aK_1|\widetilde{x}_1|-2aK_2x_1}|\widetilde{x}_1|^2 e^{-s'e^{-a(K_1|\widetilde{x}_1|+K_2x_1)}}\Big|K_1\right]\left(\mathbb{E}\left[e^{-s'e^{-a(K_1|\widetilde{x}_1|+K_2x_1)}}\Big|K_1\right]\right)^{n-1}ds'\right]$$

$$+ n(n-1)\mathbb{E}\left[\int_0^\infty s'\mathbb{E}\left[e^{-aK_1(|\widetilde{x}_1|+|\widetilde{x}_2|)-aK_2(x_1+x_2)}|\widetilde{x}_1||\widetilde{x}_2|e^{-s'e^{-a(K_1|\widetilde{x}_1|+K_2x_1)}-s'e^{-a(K_1|\widetilde{x}_2|+K_2x_2)}}\Big|K_1\right]\right.$$
$$\left.\cdot\left(\mathbb{E}\left[e^{-s'e^{-a(K_1|\widetilde{x}_1|+K_2x_1)}}\Big|K_1\right]\right)^{n-2}ds'\right]$$

$$= \frac{1}{n}\mathbb{E}\left[\int_0^\infty s\mathbb{E}\left[e^{-2aK_1|\widetilde{x}_1|-2aK_2x_1}|\widetilde{x}_1|^2 e^{-\frac{s}{n}e^{-a(K_1|\widetilde{x}_1|+K_2x_1)}}\Big|K_1\right]\left(\mathbb{E}\left[e^{-\frac{s}{n}e^{-a(K_1|\widetilde{x}_1|+K_2x_1)}}\Big|K_1\right]\right)^{n-1}ds\right]$$

$$+ \frac{n-1}{n}\mathbb{E}\left[\int_0^\infty s\mathbb{E}\left[e^{-aK_1(|\widetilde{x}_1|+|\widetilde{x}_2|)-aK_2(x_1+x_2)}|\widetilde{x}_1||\widetilde{x}_2|e^{-\frac{s}{n}e^{-a(K_1|\widetilde{x}_1|+K_2x_1)}-\frac{s}{n}e^{-a(K_1|\widetilde{x}_2|+K_2x_2)}}\Big|K_1\right]\right.$$
$$\left.\cdot\left(\mathbb{E}\left[e^{-\frac{s}{n}e^{-a(K_1|\widetilde{x}_1|+K_2x_1)}}\Big|K_1\right]\right)^{n-2}ds\right]$$

$$= \frac{1}{n}\mathbb{E}\left[\underbrace{\int_0^\infty s A_3(s/n, 2a, a, K_1)[M(s/n, a, K_1)]^{n-1}ds}_{(I)}\right]$$

$$+ \frac{n-1}{n}\mathbb{E}\left[\underbrace{\int_0^\infty s[A_4(s/n, a, a, K_1)]^2[M(s/n, a, K_1)]^{n-2}ds}_{(II)}\right]. \tag{62}$$

Here, $A_3(\lambda, \alpha, a, K_1)$, $A_4(\lambda, \alpha, a, K_1)$ and $M(\lambda, a, K_1)$ share the same definitions as in the proof in Lemma H.3. Through similar calculation procedures in (57) and (58) from the proof of Lemma H.3, we can calculate the upper and lower bounds for the term $(I)$ as

$$(I) \geq \int_0^\infty s\mathbb{E}\left[e^{-2aK_1|\widetilde{x}_1|-2aK_2x_1}|\widetilde{x}_1|^2\Big|K_1\right][M(s/n, a, K_1)]^{n-1}ds$$

$$- \frac{1}{n}\int_0^\infty s^2\mathbb{E}\left[e^{-3aK_1|\widetilde{x}_1|-3aK_2x_1}|\widetilde{x}_1|^2\Big|K_1\right][M(s/n, a, K_1)]^{n-1}ds$$

$$= \sigma^2\left(2(1+4K_1^2a^2\sigma^2)e^{2K_1^2a^2\sigma^2}\Phi(-2K_1a\sigma) - 2\sqrt{\frac{2}{\pi}}K_1a\sigma\right)e^{2K_2^2a^2\sigma^2}\int_0^\infty s[M(s/n, a, K_1)]^{n-1}ds$$

$$- \frac{2(1+9K_1^2a^2\sigma^2)e^{\frac{9K_1^2a^2\sigma^2}{2}}\Phi(-3K_1a\sigma) - \sqrt{\frac{2}{\pi}}3K_1a\sigma}{n}\sigma^2 e^{\frac{9K_2^2a^2\sigma^2}{2}}\int_0^\infty s^2[M(s/n, a, K_1)]^{n-1}ds$$

$$\geq -2a\sigma e^{2a^2\sigma^2}\mathbb{E}\left[\frac{n^2}{\left(\sum_{i=2}^n e^{-a(K_1|\widetilde{x}_i|+K_2x_i)}\right)^2}\Big|K_1\right]$$

$$- \frac{2\sigma^2(18a^2+3a+2)}{n}e^{\frac{9a^2\sigma^2}{2}}\mathbb{E}\left[\frac{n^3}{\left(\sum_{i=2}^n e^{-a(K_1|\widetilde{x}_i|+K_2x_i)}\right)^3}\Big|K_1\right]$$

$$\geq -\frac{a\sigma}{2\Phi^2(-a\sigma)}e^{a^2\sigma^2} - 1. \tag{63}$$

The first equality is true because for a normal random variable $z \sim \mathcal{N}(0, \sigma^2)$ and any scalar $c$, we have $\mathbb{E}[|z|^2 e^{-c|z|}] = \sigma^2\left(2(1+c^2\sigma^2)e^{c^2\sigma^2/2}\Phi(-c\sigma) - c\sigma\sqrt{\frac{2}{\pi}}\right)$ and $\mathbb{E}[e^{-c|z|}] = 2e^{c^2\sigma^2/2}\Phi(-c\sigma)$. The second equality is obtained by applying Fubini's theorem to exchange the order of integration, and the third follows from direct calculation. The penultimate inequality is derived by applying Lemma H.19 Following a similar (but simpler) calculation, we can also get the upper bound for $(I)$ as

$$(I) \leq \int_0^\infty s\mathbb{E}\left[e^{-2aK_1|\widetilde{x}_1|-2aK_2x_1}|\widetilde{x}_1|^2\Big|K_1\right][M(s/n, a, K_1)]^{n-1}ds$$

$$\leq 2\sigma^2 (4a^2\sigma^2 + 1)e^{9a^2\sigma^2/2}\mathbb{E}\left[\frac{n^2}{\left(\sum_{i=2}^n e^{-a(K_1|\widetilde{x}_i|+K_2 x_i)}\right)^2}\Big| K_1\right]$$

$$\leq \frac{(4a^2\sigma^2 + 1)e^{9a^2\sigma^2/2}}{2\Phi^2(-a\sigma)} + 1. \tag{64}$$

With the Lipschitz continuity of $A_4(\lambda, \alpha, a, K_1)$ derived in the proof of Lemma H.3, we can also demonstrate that

$$|[A_4(\lambda, a, a, K_1)]^2 - [A_4(0, a, a, K_1)]^2|$$
$$=|A_4(\lambda, a, a, K_1) - A_4(0, a, a, K_1)||A_4(\lambda, a, a, K_1) + A_4(0, a, a, K_1)|$$
$$\leq 2|A_4(\lambda, a, a, K_1) - A_4(0, a, a, K_1)||A_4(0, a, a, K_1)| + |A_4(\lambda, a, a, K_1) - A_4(0, a, a, K_1)|^2$$
$$\leq 10\lambda \max\{a, 1, \sigma\}^3 e^{9a^2\sigma^2/2} + 25\lambda^2 \max\{a, 1, \sigma\}^4 e^{8a^2\sigma^2},$$

Now, by using the triangle inequality to establish upper and lower bounds for $[A_4(\lambda, a, a, K_1)]^2$ from the inequalities above, we are ready to derive the lower and upper bounds for the term $(II)$. We first derive the lower bound as

$$(II) = \int_0^\infty s[A_4(s/n, a, a, K_1)]^2 [M(s/n, a, K_1)]^{n-2}\mathrm{d}s$$
$$\geq [A_4(0, a, a, K_1)]^2 \int_0^\infty s[M(s/n, a, K_1)]^{n-2}\mathrm{d}s - \frac{10\max\{a, 1, \sigma\}^3 e^{9a^2\sigma^2/2}}{n}\int_0^\infty s^2[M(s/n, a, K_1)]^{n-2}\mathrm{d}s$$
$$- \frac{25\max\{a, 1, \sigma\}^4 e^{8a^2\sigma^2}}{n^2}\int_0^\infty s^3[M(s/n, a, K_1)]^{n-2}\mathrm{d}s$$
$$= [A_4(0, a, a, K_1)]^2\mathbb{E}\left[\frac{n^2}{\left(\sum_{i=3}^n e^{-a(K_1|\widetilde{x}_i|+K_2 x_i)}\right)^2}\Big| K_1\right]$$
$$- \frac{20\max\{a, 1, \sigma\}^3 e^{9a^2\sigma^2/2}}{n}\mathbb{E}\left[\frac{n^3}{\left(\sum_{i=3}^n e^{-a(K_1|\widetilde{x}_i|+K_2 x_i)}\right)^3}\right]$$
$$- \frac{150\max\{a, 1, \sigma\}^4 e^{8a^2\sigma^2}}{n^2}\mathbb{E}\left[\frac{n^4}{\left(\sum_{i=3}^n e^{-a(K_1|\widetilde{x}_i|+K_2 x_i)}\right)^4}\right]$$
$$\geq \frac{[A_4(0, a, a, K_1)]^2}{4e^{a^2\sigma^2}\Phi^2(-aK_1\sigma)} - \frac{c_3(a, \sigma)}{n}, \tag{65}$$

where $c_3(a, \sigma) = \frac{2\max\{a, 1, \sigma\}^4 e^{3a^2\sigma^2}}{\Phi^3(-a\sigma)}$, a positive continuous function of $a$ and $\sigma$. Here the last equation is derived by Fubini's theorem to exchange the order of the integral, and the last inequality is established by Lemma H.19. Similarly, we can also obtain that

$$(II) \leq [A_4(0, a, a, K_1)]^2 \int_0^\infty s[M(s/n, a, K_1)]^{n-2}\mathrm{d}s + \frac{10\max\{a, 1, \sigma\}^3 e^{9a^2\sigma^2/2}}{n}\int_0^\infty s^2[M(s/n, a, K_1)]^{n-2}\mathrm{d}s$$
$$+ \frac{25\max\{a, 1, \sigma\}^4 e^{8a^2\sigma^2}}{n^2}\int_0^\infty s^3[M(s/n, a, K_1)]^{n-2}\mathrm{d}s$$
$$\leq \frac{[A_4(0, a, a, K_1)]^2}{4e^{a^2\sigma^2}\Phi^2(-aK_1\sigma)} + \frac{c_3(a, \sigma)}{n}. \tag{66}$$

To further derive a concrete bounds for the leading term $F(a, K_1) = \frac{[A_4(0, a, a, K_1)]^2}{4e^{a^2\sigma^2}\Phi^2(-aK_1\sigma)}$, we consider the forth order Taylor's expansion of $F(a, K_1)$ at $K_1 = 0$. By utilizing the conclusion that $K_1$ has a symmetric distribution, we have

$$\mathbb{E}[F(a, K_1)] = F(a, 0) + \mathbb{E}\left[\frac{F''(a, \xi)K_1^2}{2}\right],$$

where $\xi$ is a random variable between $0$ and $K_1$. Since the function $F(a, k)$ is analytic w.r.t. $k$ on $[-1, 1]$, and $|K_1| \leq 1$ (hence $|\xi| \leq 1$), its fourth derivative $F^{(2)}(a, k)$ is continuous and bounded on this compact interval. Therefore, there exists

a constant $c_4(a, \sigma)$ such that $|F^{(4)}(a, \xi)| \leq c_4(a, \sigma)$. By the fact that $F(a, 0) = \frac{2}{\pi}\sigma^2$ and $\mathbb{E}[K_1^2] = 1/d$, we finally derive that

$$\left| \mathbb{E}[F(a, K_1)] - \frac{2\sigma^2}{\pi} \right| \leq \frac{c_4(a, \sigma)}{d}. \tag{67}$$

Substituting these results of (63), (64), (65), (66), and (67) into (62), we complete the proof. $\qquad \square$

**Lemma H.5.** *Let $x_1, x_2, \ldots, x_n \sim \mathcal{N}(0, \sigma^2)$ be $n$ i.i.d Gaussian random variables, and $\widetilde{x}_1, \widetilde{x}_2, \ldots, \widetilde{x}_n \sim \mathcal{N}(0, \sigma^2)$ be another $n$ i.i.d Gaussian random variables. In addition, $a$ is a positive scalar, and $\boldsymbol{\theta}_*, \boldsymbol{\theta}_0 \in \mathbb{R}^d$ are two independent random vectors following uniform $d-$dimensional sphere distribution, and we denote $K_1 = \langle \boldsymbol{\theta}_0, \boldsymbol{\theta}_* \rangle$, $K_2 = \|(\mathbf{I}_d - \boldsymbol{\theta}_*\boldsymbol{\theta}_*^\top)\boldsymbol{\theta}_0\|_2$. Then we have*

$$\left| \mathbb{E}\left[ \frac{\sum_{i=1}^n e^{-\langle \boldsymbol{\theta}_0, \boldsymbol{\theta}_* \rangle(1+a)|\widetilde{x}_i| - \|(\mathbf{I}_d - \boldsymbol{\theta}_*\boldsymbol{\theta}_*^\top)\boldsymbol{\theta}_0\|_2(x_i + ax_i)}}{\sum_{i=1}^n e^{-a\langle \boldsymbol{\theta}_0, \boldsymbol{\theta}_* \rangle|\widetilde{x}_i| - a\|(\mathbf{I}_d - \boldsymbol{\theta}_*\boldsymbol{\theta}_*^\top)\boldsymbol{\theta}_0\|_2 x_i}} \right] - B_{a,2}e^{(a + \frac{1}{2})\sigma^2} \right| \leq \frac{f_7(a, \sigma)}{n},$$

*where $B_{a,2} = \mathbb{E}\left[ \frac{\Phi(-K_1(a+1)\sigma)}{\Phi(-K_1 a\sigma)} \right]$. It satisfies that $\left| \mathbb{E}\left[ \frac{\Phi(-K_1(a+1)\sigma)}{\Phi(-K_1 a\sigma)} \right] - 1 \right| \leq \frac{f_8(a,\sigma)}{d}$, where $f_7(a, \sigma), f_8(a, \sigma)$ are both continuous functions of $a$ and $\sigma$, and irrelevant with $n, d$.*

*Proof of Lemma H.5.* Following a similar procedure in the proof of Lemma H.1 to obtain that,

$$\mathbb{E}\left[ \frac{\sum_{i=1}^n e^{-K_1(1+a)|\widetilde{x}_i| - K_2(1+a)x_i}}{\sum_{i=1}^n e^{-aK_1|\widetilde{x}_i| - aK_2 x_i}} \right] = \int_0^\infty \mathbb{E}\left[ \sum_{i=1}^n e^{-K_1(1+a)|\widetilde{x}_i| - K_2(1+a)x_i} \exp\left( -s' \sum_{i=1}^n e^{-aK_1|\widetilde{x}_i| - aK_2 x_i} \right) \right] ds'$$

$$= n\mathbb{E}\left[ \int_0^\infty \mathbb{E}\left[ e^{-K_1(1+a)|\widetilde{x}_1| - K_2(1+a)x_1} e^{-s' e^{-a(K_1|\widetilde{x}_1| + K_2 x_1)}} | K_1 \right] \left( \mathbb{E}\left[ e^{-s' e^{-a(K_1|\widetilde{x}_1| + K_2 x_1)}} | K_1 \right] \right)^{n-1} ds' \right]$$

$$= \mathbb{E}\left[ \int_0^\infty \mathbb{E}\left[ e^{-K_1(1+a)|\widetilde{x}_1| - K_2(1+a)x_1} e^{-\frac{s}{n} e^{-a(K_1|\widetilde{x}_1| + K_2 x_1)}} | K_1 \right] \left( \mathbb{E}\left[ e^{-\frac{s}{n} e^{-a(K_1|\widetilde{x}_1| + K_2 x_1)}} | K_1 \right] \right)^{n-1} ds \right]$$

$$= \mathbb{E}\left[ \underbrace{\int_0^\infty A_5(s/n, 1 + a, a, K_1)[M(s/n, a, K_1)]^{n-1} ds}_{(I)} \right], \tag{68}$$

Here, $M(\lambda, a, K_1)$ share the same definitions as in the proof in Lemma H.1, and the terms $A_5(\lambda, \alpha, a, K_1)$ is defined as:

$$A_5(\lambda, \alpha, a, K_1) = \mathbb{E}\left[ e^{-\alpha K_1|\widetilde{x}_1| - \alpha K_2 x_1} e^{-\lambda e^{-a(K_1|\widetilde{x}_1| + K_2 x_1)}} | K_1 \right].$$

We use the fact $1 - z \leq e^{-z} \leq 1$ to derive the upper and lower bounds for $A_5(\lambda, \alpha, a, K_1)$ as

$$A_5(\lambda, \alpha, a, K_1) \geq \mathbb{E}\left[ e^{-\alpha K_1|\widetilde{x}_1| - \alpha K_2 x_1} | K_1 \right] - \lambda \mathbb{E}\left[ e^{-K_1(\alpha+a)|\widetilde{x}_1| - K_2(\alpha+a)x_1} | K_1 \right];$$

$$A_5(\lambda, \alpha, a, K_1) \leq \mathbb{E}\left[ e^{-\alpha K_1|\widetilde{x}_1| - \alpha K_2 x_1} | K_1 \right].$$

We can establish the upper and lower bounds for the term $(I)$ based on the inequalities above as,

$$(I) \geq \int_0^\infty \mathbb{E}\left[ e^{-K_1(1+a)|\widetilde{x}_1| - K_2(1+a)x_1} | K_1 \right] [M(s/n, a, K_1)]^{n-1} ds$$

$$\qquad - \frac{1}{n} \int_0^\infty s\mathbb{E}\left[ e^{-K_1(1+2a)|\widetilde{x}_1| - K_2(1+2a)x_1} | K_1 \right] [M(s/n, a, K_1)]^{n-1} ds$$

$$= 2\Phi(-K_1(a+1)\sigma)e^{\frac{(a+1)^2\sigma^2}{2}} \int_0^\infty [M(s/n, a, K_1)]^{n-1} ds$$

$$\qquad - \frac{2\Phi(-K_1(2a+1)\sigma)}{n} e^{\frac{(2a+1)^2\sigma^2}{2}} \int_0^\infty s[M(s/n, a, K_1)]^{n-1} ds$$

$$\geq 2\Phi(-(a+1)\sigma)e^{\frac{(a+1)^2\sigma^2}{2}} \mathbb{E}\left[ \frac{n}{\sum_{i=2}^n e^{-a(K_1|\widetilde{x}_i| + K_2 x_i)}} \Big| K_1 \right] - \frac{2e^{\frac{(2a+1)^2\sigma^2}{2}}}{n} \mathbb{E}\left[ \frac{n^2}{\left( \sum_{i=2}^n e^{-a(K_1|\widetilde{x}_i| + K_2 x_i)} \right)^2} \Big| K_1 \right]$$

$$\geq \frac{\Phi(-K_1(a+1)\sigma)}{\Phi(-K_1 a\sigma)}e^{\frac{2a+1}{2}\sigma^2} - \frac{c_1(a,\sigma)}{n}. \tag{69}$$

The first equality is true because for a normal random variable $z \sim \mathcal{N}(0,\sigma^2)$ and any scalar $c$, we have $\mathbb{E}[e^{-c|z|}] = 2e^{c^2\sigma^2/2}\Phi(-c\sigma)$. The second equality is obtained by applying Fubini's theorem to exchange the order of integration, and the fact that $|K_1|, |K_2| \leq 1$. The last inequality is derived by applying Lemma H.19. Following a similar (but simpler) calculation, we can also get the upper bound for $(I)$ as

$$(I) \leq \int_0^\infty \mathbb{E}\big[e^{-K_1(1+a)|\widetilde{x}_1|-K_2(1+a)x_1}|K_1\big][M(s/n,a,K_1)]^{n-1}\mathrm{d}s$$

$$=2\Phi(-K_1(a+1)\sigma)e^{\frac{(a+1)^2\sigma^2}{2}}\mathbb{E}\left[\frac{n}{\sum_{i=2}^n e^{-a(K_1|\widetilde{x}_i|+K_2 x_i)}}\Big|K_1\right] \leq \frac{\Phi(-K_1(a+1)\sigma)}{\Phi(-K_1 a\sigma)}e^{\frac{2a+1}{2}\sigma^2} + \frac{c_1(a,\sigma)}{n}. \tag{70}$$

In addition, by utilizing the Taylor's expansion regarding the function $\mathbb{E}\big[\frac{\Phi(-K_1(a+1)\sigma)}{\Phi(-K_1 a\sigma)}\big]$ w.r.t. $K_1$ at $K_1 = 0$, we can derive that $\big|\mathbb{E}\big[\frac{\Phi(-K_1(a+1)\sigma)}{\Phi(-K_1 a\sigma)}\big] - 1\big| \leq \frac{c_2(a,\sigma)}{d}$. $\qquad\square$

**Lemma H.6.** *Let $x_1, x_2, \ldots, x_n \sim \mathcal{N}(0,\sigma^2)$ be $n$ i.i.d Gaussian random variables, and $\widetilde{x}_1, \widetilde{x}_2, \ldots, \widetilde{x}_n \sim \mathcal{N}(0,\sigma^2)$ be another $n$ i.i.d Gaussian random variables. In addition, $a$ is a positive scalar, and $\boldsymbol{\theta}_*, \boldsymbol{\theta}_0 \in \mathbb{R}^d$ are two independent random vectors following uniform $d-$dimensional sphere distribution, and we denote $K_1 = \langle \boldsymbol{\theta}_0, \boldsymbol{\theta}_* \rangle$, $K_2 = \|(\mathbf{I}_d - \boldsymbol{\theta}_*\boldsymbol{\theta}_*^\top)\boldsymbol{\theta}_0\|_2$. Then we have*

$$\left|\mathbb{E}\left[\frac{\sum_{i=1}^n e^{-2a\langle\boldsymbol{\theta}_0,\boldsymbol{\theta}_*\rangle|\widetilde{x}_i|-2a\|(\mathbf{I}_d-\boldsymbol{\theta}_*\boldsymbol{\theta}_*^\top)\boldsymbol{\theta}_0\|_2 x_i}}{\left(\sum_{i=1}^n e^{-a\langle\boldsymbol{\theta}_0,\boldsymbol{\theta}_*\rangle|\widetilde{x}_i|-a\|(\mathbf{I}_d-\boldsymbol{\theta}_*\boldsymbol{\theta}_*^\top)\boldsymbol{\theta}_0\|_2 x_i}\right)^2}\right] - \frac{B_{a,3}e^{a^2\sigma^2}}{n}\right| \leq \frac{f_9(a,\sigma)}{n^2}.$$

*where $B_{a,3} = \mathbb{E}\big[\frac{\Phi(-2aK_1\sigma)}{2\Phi^2(-aK_1\sigma)}\big]$. It satisfies that $\big|\mathbb{E}\big[\frac{\Phi(-2aK_1\sigma)}{2\Phi^2(-aK_1\sigma)}\big] - 1\big| \leq \frac{f_{10}(a,\sigma)}{d}$, where $f_9(a,\sigma), f_{10}(a,\sigma)$ are both continuous functions of $a$ and $\sigma$, and irrelevant with $n, d$.*

*Proof of Lemma H.6.* Following a similar procedure in the proof of Lemma H.2, we can obtain that

$$\mathbb{E}\left[\frac{\sum_{i=1}^n e^{-2aK_1|\widetilde{x}_i|-2aK_2 x_i}}{\left(\sum_{i=1}^n e^{-aK_1|\widetilde{x}_i|-aK_2 x_i}\right)^2}\right] = \int_0^\infty s'\mathbb{E}\left[\sum_{i=1}^n e^{-2aK_1|\widetilde{x}_i|-2aK_2 x_i}\exp\Big(-s'\sum_{i=1}^n e^{-aK_1|\widetilde{x}_i|-aK_2 x_i}\Big)\right]\mathrm{d}s'$$

$$=n\mathbb{E}\left[\int_0^\infty s'\mathbb{E}\big[e^{-2aK_1|\widetilde{x}_1|-2aK_2 x_1}e^{-s'e^{-a(K_1|\widetilde{x}_1|+K_2 x_1)}}|K_1\big]\Big(\mathbb{E}\big[e^{-s'e^{-a(K_1|\widetilde{x}_1|+K_2 x_1)}}|K_1\big]\Big)^{n-1}\mathrm{d}s'\right]$$

$$=\frac{1}{n}\mathbb{E}\left[\int_0^\infty s\mathbb{E}\big[e^{-2aK_1|\widetilde{x}_1|-2aK_2 x_1}e^{-\frac{s}{n}e^{-a(K_1|\widetilde{x}_1|+K_2 x_1)}}|K_1\big]\Big(\mathbb{E}\big[e^{-\frac{s}{n}e^{-a(K_1|\widetilde{x}_1|+K_2 x_1)}}|K_1\big]\Big)^{n-1}\mathrm{d}s\right]$$

$$=\frac{1}{n}\mathbb{E}\left[\underbrace{\int_0^\infty s A_5(s/n,2a,a,K_1)[M(s/n,a,K_1)]^{n-1}\mathrm{d}s}_{(I)}\right]. \tag{71}$$

Here, $A_5(\lambda,\alpha,a,K_1)$ share the same definitions as in the proof in Lemma H.5. Through similar calculation procedures in the proof of Lemma H.5, we can calculate the upper and lower bounds for the term $(I)$ as

$$(I) \geq \int_0^\infty s\mathbb{E}\big[e^{-2aK_1|\widetilde{x}_1|-2aK_2 x_1}|K_1\big][M(s/n,a,K_1)]^{n-1}\mathrm{d}s$$

$$\quad - \frac{1}{n}\int_0^\infty s^2\mathbb{E}\big[e^{-3aK_1|\widetilde{x}_1|-3aK_2 x_1}|K_1\big][M(s/n,a,K_1)]^{n-1}\mathrm{d}s$$

$$=2\Phi(-2aK_1\sigma)e^{2a^2\sigma^2}\int_0^\infty s[M(s/n,a,K_1)]^{n-1}\mathrm{d}s - \frac{2\Phi(-3aK_1\sigma)}{n}e^{2a^2\sigma^2}\int_0^\infty s^2[M(s/n,a,K_1)]^{n-1}\mathrm{d}s$$

$$\geq 2\Phi(-2aK_1\sigma)e^{2a^2\sigma^2}\mathbb{E}\left[\frac{n^2}{\sum_{i=2}^n\big(e^{-a(K_1|\widetilde{x}_i|+K_2 x_i)}\big)^2}\Big|K_1\right] - \frac{4e^{2a^2\sigma^2}}{n}\mathbb{E}\left[\frac{n^3}{\big(\sum_{i=2}^n e^{-a(K_1|\widetilde{x}_i|+K_2 x_i)}\big)^3}\Big|K_1\right]$$

$$\geq \frac{\Phi(-2aK_1\sigma)}{2\Phi^2(-aK_1\sigma)}e^{a^2\sigma^2} - \frac{c_1(a,\sigma)}{n}. \tag{72}$$

The first equality is true because for a normal random variable $z \sim \mathcal{N}(0, \sigma^2)$ and any scalar $c$, we have $\mathbb{E}[e^{-c|z|}] = 2e^{c^2\sigma^2/2}\Phi(-c\sigma)$. The second equality is obtained by applying Fubini's theorem to exchange the order of integration, and the fact that $|K_1|, |K_2| \leq 1$. The last inequality is derived by applying Lemma H.19. Following a similar (but simpler) calculation, we can also get the upper bound for $(I)$ as

$$
(I) \leq \int_0^\infty s\mathbb{E}\big[e^{-2aK_1|\widetilde{x}_1|-2aK_2x_1}|K_1\big][M(s/n, a, K_1)]^{n-1}\mathrm{d}s
$$

$$
=2\Phi(-2aK_1\sigma)e^{2a^2\sigma^2}\mathbb{E}\left[\frac{n^2}{\sum_{i=2}^n\left(e^{-a(K_1|\widetilde{x}_i|+K_2x_i)}\right)^2}\Big|K_1\right] \leq \frac{\Phi(-2aK_1\sigma)}{2\Phi^2(-aK_1\sigma)}e^{a^2\sigma^2} + \frac{c_1(a,\sigma)}{n}. \tag{73}
$$

In addition, by utilizing the Taylor's expansion regarding the function $\mathbb{E}\big[\frac{\Phi(-2aK_1\sigma)}{2\Phi^2(-aK_1\sigma)}\big]$ w.r.t. $K_1$ at $K_1 = 0$, we can derive that $\big|\mathbb{E}\big[\frac{\Phi(-2aK_1\sigma)}{2\Phi^2(-aK_1\sigma)}\big] - 1\big| \leq \frac{c_2(a,\sigma)}{d}$. $\qquad\square$

**Lemma H.7.** *Let $x_1, x_2, \ldots, x_n \sim \mathcal{N}(0, \sigma^2)$ be $n$ i.i.d Gaussian random variables, and $\widetilde{x}_1, \widetilde{x}_2, \ldots, \widetilde{x}_n \sim \mathcal{N}(0, \sigma^2)$ be another $n$ i.i.d Gaussian random variables. In addition, $a$ is a positive scalar, and $\boldsymbol{\theta}_*, \boldsymbol{\theta}_0 \in \mathbb{R}^d$ are two independent random vectors following uniform $d-$dimensional sphere distribution, and we denote $K_1 = \langle\boldsymbol{\theta}_0, \boldsymbol{\theta}_*\rangle$, $K_2 = \|(\mathbf{I}_d - \boldsymbol{\theta}_*\boldsymbol{\theta}_*^\top)\boldsymbol{\theta}_0\|_2$. Then we have*

$$
\left|\mathbb{E}\left[\|(\mathbf{I}_d - \boldsymbol{\theta}_*\boldsymbol{\theta}_*^\top)\boldsymbol{\theta}_0\|_2\frac{\sum_{i_1=1}^n\sum_{i_2=1}^n e^{-\langle\boldsymbol{\theta}_0,\boldsymbol{\theta}_*\rangle(|\widetilde{x}_{i_1}|+a|\widetilde{x}_{i_2}|)-\|(\mathbf{I}_d-\boldsymbol{\theta}_*\boldsymbol{\theta}_*^\top)\boldsymbol{\theta}_0\|_2(x_{i_1}+ax_{i_2})}x_{i_1}x_{i_2}^2}{\sum_{i=1}^n e^{-a\langle\boldsymbol{\theta}_0,\boldsymbol{\theta}_*\rangle|\widetilde{x}_i|-a\|(\mathbf{I}_d-\boldsymbol{\theta}_*\boldsymbol{\theta}_*^\top)\boldsymbol{\theta}_0\|_2 x_i}}\right] - nB_{a,4}\right| \leq f_{11}(a,\sigma).
$$

*Here, $B_{a,4} = \mathbb{E}[-2\sigma^4 e^{\frac{\sigma^2}{2}}K_2^2(a^2\sigma^2K_2^2+1)\Phi(-K_1\sigma)]$, satisfying that $|B_{a,4} + \sigma^4 e^{\frac{\sigma^2}{2}}(a^2\sigma^2+1)| \leq \frac{f_{12}(a,\sigma)}{d}$, and $f_{11}(a,\sigma)$, $f_{12}(a,\sigma)$ are both analytic functions of $a$ and $\sigma$ while irrelevant with $n$ and $d$.*

*Proof of Lemma H.7.* We repeatedly use the previous notations that $K_1 = \langle\boldsymbol{\theta}_0, \boldsymbol{\theta}_*\rangle$, and $K_2 = \|(\mathbf{I}_d - \boldsymbol{\theta}_*\boldsymbol{\theta}_*^\top)\boldsymbol{\theta}_0\|_2$. Following a similar procedure in the proof of Lemma H.1, we can obtain that

$$
\mathbb{E}\left[K_2\frac{\sum_{i_1=1}^n\sum_{i_2=1}^n e^{-K_1(|\widetilde{x}_{i_1}|+a|\widetilde{x}_{i_2}|)-K_2(x_{i_1}+ax_{i_2})}x_{i_1}x_{i_2}^2}{\sum_{i=1}^n e^{-aK_1|\widetilde{x}_i|-aK_2x_i}}\right]
$$

$$
=\int_0^\infty \mathbb{E}\left[K_2\sum_{i_1,i_2=1}^n e^{-K_1(|\widetilde{x}_{i_1}|+a|\widetilde{x}_{i_2}|)-K_2(x_{i_1}+ax_{i_2})}x_{i_1}x_{i_2}^2\exp\Big(-s'\sum_{i=1}^n e^{-aK_1|\widetilde{x}_i|-aK_2x_i}\Big)\right]\mathrm{d}s'
$$

$$
=n\mathbb{E}\left[K_2\int_0^\infty \mathbb{E}\big[e^{-K_1(1+a)|\widetilde{x}_1|-K_2(1+a)x_1}x_1^3 e^{-s'e^{-a(K_1|\widetilde{x}_1|+K_2x_1)}}|K_1\big]\Big(\mathbb{E}\big[e^{-s'e^{-a(K_1|\widetilde{x}_1|+K_2x_1)}}|K_1\big]\Big)^{n-1}\mathrm{d}s'\right]
$$

$$
+ n(n-1)\mathbb{E}\left[K_2\int_0^\infty \mathbb{E}\big[e^{-K_1(|\widetilde{x}_1|+a|\widetilde{x}_2|)-K_2(x_1+ax_2)}x_1x_2^2 e^{-s'e^{-a(K_1|\widetilde{x}_1|+K_2x_1)}-s'e^{-a(K_1|\widetilde{x}_2|+K_2x_2)}}|K_1\big]\right.
$$

$$
\left.\cdot\Big(\mathbb{E}\big[e^{-s'e^{-a(K_1|\widetilde{x}_1|+K_2x_1)}}|K_1\big]\Big)^{n-2}\mathrm{d}s'\right]
$$

$$
=\mathbb{E}\left[K_2\int_0^\infty \mathbb{E}\big[e^{-K_1(1+a)|\widetilde{x}_1|-K_2(1+a)x_1}x_1^3 e^{-\frac{s}{n}e^{-a(K_1|\widetilde{x}_1|+K_2x_1)}}|K_1\big]\Big(\mathbb{E}\big[e^{-\frac{s}{n}e^{-a(K_1|\widetilde{x}_1|+K_2x_1)}}|K_1\big]\Big)^{n-1}\mathrm{d}s\right]
$$

$$
+ (n-1)\mathbb{E}\left[K_2\int_0^\infty \mathbb{E}\big[e^{-K_1(|\widetilde{x}_1|+a|\widetilde{x}_2|)-K_2(x_1+ax_2)}x_1x_2^2 e^{-\frac{s}{n}e^{-a(K_1|\widetilde{x}_1|+K_2x_1)}-\frac{s}{n}e^{-a(K_1|\widetilde{x}_2|+K_2x_2)}}|K_1\big]\right.
$$

$$
\left.\cdot\Big(\mathbb{E}\big[e^{-\frac{s}{n}e^{-a(K_1|\widetilde{x}_1|+K_2x_1)}}|K_1\big]\Big)^{n-2}\mathrm{d}s\right]
$$

$$
=\mathbb{E}\left[K_2\underbrace{\int_0^\infty A_6(s/n, 1+a, a, K_1)[M(s/n, a, K_1)]^{n-1}\mathrm{d}s}_{(I)}\right]
$$

$$
+ (n-1)\mathbb{E}\left[K_2\underbrace{\int_0^\infty A_2(s/n, 1, a, K_1)A_1(s/n, a, a, K_1)[M(s/n, a, K_1)]^{n-2}\mathrm{d}s}_{(II)}\right]. \tag{74}
$$

Here, $A_1(\lambda, \alpha, a, K_1)$, $A_2(\lambda, \alpha, a, K_1)$ and $M(\lambda, a, K_1)$ share the same definitions as in the proof in Lemma H.1. In addition, the term $A_6(\lambda, \alpha, a, K_1)$ is defined as:

$$A_6(\lambda, \alpha, a, K_1) = \mathbb{E}\big[e^{-\alpha K_1|\widetilde{x}_1| - \alpha K_2 x_1} x_1^3 e^{-\lambda e^{-a(K_1|\widetilde{x}_1| + K_2 x_1)}} \big| K_1\big].$$

And we can further calculate the derivative of $A_6(\lambda, \alpha, a, K_1)$ w.r.t. $\lambda$ as

$$\left|\frac{\mathrm{d}A_6(\lambda, \alpha, a, K_1)}{\mathrm{d}\lambda}\right| = \left| - \mathbb{E}\big[e^{-(\alpha+a)(K_1|\widetilde{x}_1| + K_2 x_1)} x_1^3 e^{-\lambda e^{-a(K_1|\widetilde{x}_1| + K_2 x_1)}} \big| K_1\big]\right|$$

$$\leq \mathbb{E}\big[|x_1|^3 e^{-(\alpha+a)(K_1|\widetilde{x}_1| + K_2 x_1)}\big] \leq 2\sqrt{15}\sigma^3 e^{(a+\alpha)^2\sigma^2},$$

which implies that $A_6(\lambda, \alpha, a, K_1)$ is Lipschitz continuous w.r.t. $\lambda$, and $|A_6(s/n, 1 + a, a, K_1) - A_6(0, 1 + a, a, K_1)| \leq 2\sqrt{15}\sigma^3 e^{(a+\alpha)^2} s/n$. Consequently, we can establish the upper and lower bounds for the term $(I)$ as

$$(I) \geq A_6(0, 1 + a, a, K_1) \int_0^\infty [M(s/n, a, K_1)]^{n-1}\mathrm{d}s - \frac{2\sqrt{15}\sigma^3 e^{(2a+1)^2\sigma^2}}{n} \int_0^\infty s[M(s/n, a, K_1)]^{n-1}\mathrm{d}s$$

$$= - 2K_2(1 + a)\sigma^4\big(K_2^2(1 + a)^2\sigma^2 + 3\big)e^{\frac{(a+1)^2\sigma^2}{2}}\Phi(-K_1(a+1)\sigma)\mathbb{E}\left[\frac{n}{\sum_{i=2}^n e^{-a(K_1|\widetilde{x}_i| + K_2 x_i)}}\bigg|K_1\right]$$

$$- \frac{2\sqrt{15}\sigma^3 e^{(2a+1)^2\sigma^2}}{n}\mathbb{E}\left[\frac{n^2}{\big(\sum_{i=2}^n e^{-a(K_1|\widetilde{x}_i| + K_2 x_i)}\big)^2}\bigg|K_1\right]$$

$$\geq - K_2(1 + a)\sigma^4\big(K_2^2(1 + a)^2\sigma^2 + 3\big)e^{\frac{2a+1}{2}\sigma^2}\frac{\Phi(-K_1(a+1)\sigma)}{\Phi(-K_1 a\sigma)} - 1. \tag{75}$$

The first equality is true because for a normal random variable $z \sim \mathcal{N}(0, \sigma^2)$ and any scalar $c$, we have $\mathbb{E}[z^3 e^{-cz}] = \sigma^3(c^3\sigma^3 + 3c\sigma)e^{c^2\sigma^2/2}$, and $\mathbb{E}[e^{-c|z|}] = 2e^{c^2\sigma^2/2}\Phi(-c\sigma)$, and we apply Fubini's theorem to exchange the order of integration. The penultimate inequality is derived by applying Lemma H.19. Following a similar calculation, we can also get the upper bound for $(I)$ as

$$(I) \leq A_6(0, 1 + a, a, K_1) \int_0^\infty [M(s/n, a, K_1)]^{n-1}\mathrm{d}s + \frac{2\sqrt{15}\sigma^3 e^{(2a+1)^2\sigma^2}}{n} \int_0^\infty s[M(s/n, a, K_1)]^{n-1}\mathrm{d}s$$

$$\leq - K_2(1 + a)\sigma^4\big(K_2^2(1 + a)^2\sigma^2 + 3\big)e^{\frac{2a+1}{2}\sigma^2}\frac{\Phi(-K_1(a+1)\sigma)}{\Phi(-K_1 a\sigma)} + 1. \tag{76}$$

To calculate the upper and lower bounds for the term $(II)$, we also first calculate the derivatives of $A_1(\lambda, \alpha, a, K_1)$ w.r.t. $\lambda$ as

$$\left|\frac{\mathrm{d}A_1(\lambda, \alpha, a, K_1)}{\mathrm{d}\lambda}\right| = \left| - \mathbb{E}\big[e^{-(\alpha+a)(K_1|\widetilde{x}_1| + K_2 x_1)} x_1^2 e^{-\lambda e^{-a(K_1|\widetilde{x}_1| + K_2 x_1)}} \big| K_1\big]\right|$$

$$\leq \mathbb{E}\big[|x_1|^2 e^{-(\alpha+a)(K_1|\widetilde{x}_1| + K_2 x_1)}\big] \leq 2\sqrt{3}\sigma^2 e^{(a+\alpha)^2\sigma^2},$$

which implies that $A_1(\lambda, \alpha, a, K_1)$ is Lipschitz continuous w.r.t. $\lambda$. Combined with the Lipschitz continuity of $A_2(\lambda, \alpha, a, K_1)$ established in the proof of Lemma H.1, we have

$$|A_2(\lambda, 1, a, K_1)A_1(\lambda, a, a, K_1) - A_2(0, 1, a, K_1)A_1(0, a, a, K_1)|$$

$$\leq |A_2(\lambda, 1, a, K_1) - A_2(0, 1, a, K_1)|(|A_1(\lambda, a, a, K_1) - A_1(0, a, a, K_1)| + |A_1(0, a, a, K_1)|)$$

$$\quad + |A_2(0, 1, a, K_1)||A_1(\lambda, a, a, K_1) - A_1(0, a, a, K_1)|$$

$$\leq 8\lambda(a^2\sigma^2 + 1)e^{9(a\vee 1)^2/2} + 12\lambda^2\sigma^4 e^{9(a\vee 1)^2/2},$$

where the last inequality holds by using the Lipschitz continuous properties of $A_2(\lambda, \alpha, a, K_1)$, and the facts that $A_2(0, 1, a, K_1) = -2\sigma^2 K_2 e^{\sigma^2/2}\Phi(-K_1\sigma)$, $A_1(0, a, a, K_1) = 2\sigma^2(a^2 K_2^2\sigma^2 + 1)e^{a^2\sigma^2/2}\Phi(-a\sigma K_1)$ and $|K_1|, |K_2| \leq 1$. With the inequality established above and the triangle inequality, we have

$$A_2(\lambda, 1, a, K_1)A_1(\lambda, a, a, K_1) \leq -4\sigma^4 K_2(a^2 K_2^2\sigma^2 + 1)e^{\frac{a^2+1}{2}\sigma^2}\Phi(-K_1\sigma)\Phi(-aK_1\sigma) + \lambda c_2(a, \sigma) + \lambda^2 c_3(a, \sigma);$$

$$A_2(\lambda, 1, a, K_1)A_2(\lambda, a, a, K_1) \geq -4\sigma^4 K_2(a^2 K_2^2 \sigma^2 + 1)e^{\frac{a^2+1}{2}\sigma^2}\Phi(-K_1\sigma)\Phi(-aK_1\sigma) - \lambda c_2(a, \sigma) - \lambda^2 c_3(a, \sigma). \quad (77)$$

Now, we are ready to derive the lower and upper bounds for the term $(II)$ based on (77). We first derive the lower bound as

$$\begin{aligned}
(II) &= \int_0^\infty A_2(s/n, 1, a, K_1)A_1(s/n, a, a, K_1)[M(s/n, a, K_1)]^{n-2}\mathrm{d}s \\
&\geq -4\sigma^4 K_2(a^2 K_2^2 \sigma^2 + 1)e^{\frac{a^2+1}{2}\sigma^2}\Phi(-K_1\sigma)\Phi(-aK_1\sigma)\int_0^\infty [M(s/n, a, K_1)]^{n-2}\mathrm{d}s \\
&\quad - \frac{c_2(a, \sigma)}{n}\int_0^\infty s[M(s/n, a, K_1)]^{n-2}\mathrm{d}s - \frac{c_3(a, \sigma)}{n^2}\int_0^\infty s^2[M(s/n, a, K_1)]^{n-2}\mathrm{d}s \\
&= -4\sigma^4 K_2(a^2 K_2^2 \sigma^2 + 1)e^{\frac{a^2+1}{2}\sigma^2}\Phi(-K_1\sigma)\Phi(-aK_1\sigma)\mathbb{E}\left[\frac{n}{\sum_{i=3}^n e^{-a(K_1|\widetilde{x}_i|+K_2 x_i)}}\Big| K_1\right] \\
&\quad - \frac{c_2(a, \sigma)}{n}\mathbb{E}\left[\frac{n^2}{\left(\sum_{i=3}^n e^{-a(K_1|\widetilde{x}_i|+K_2 x_i)}\right)^2}\right] - \frac{2c_3(a, \sigma)}{n^2}\mathbb{E}\left[\frac{n^3}{\left(\sum_{i=3}^n e^{-a(K_1|\widetilde{x}_i|+K_2 x_i)}\right)^3}\right] \\
&\geq -2\sigma^4 e^{\frac{\sigma^2}{2}}K_2(a^2\sigma^2 K_2^2 + 1)\Phi(-K_1\sigma) - \frac{c_4(a, \sigma)}{n}, \quad (78)
\end{aligned}$$

where $c_4(a, \sigma)$ is a positive constant solely depending on $a$ and $\sigma$. Here the last equation is derived by Fubini's theorem to exchange the order of the integral, and the last inequality is established by Lemma H.19. Similarly, we can also obtain that

$$\begin{aligned}
(II) &\leq -4\sigma^4 K_2(a^2 K_2^2 \sigma^2 + 1)e^{\frac{a^2+1}{2}\sigma^2}\Phi(-K_1\sigma)\Phi(-aK_1\sigma)\int_0^\infty [M(s/n, a, K_1)]^{n-2}\mathrm{d}s \\
&\quad + \frac{c_2(a, \sigma)}{n}\int_0^\infty s[M(s/n, a, K_1)]^{n-2}\mathrm{d}s + \frac{c_3(a, \sigma)}{n^2}\int_0^\infty s^2[M(s/n, a, K_1)]^{n-2}\mathrm{d}s \\
&\leq -2\sigma^4 e^{\frac{\sigma^2}{2}}K_2(a^2\sigma^2 K_2^2 + 1)\Phi(-K_1\sigma) + \frac{c_4(a, \sigma)}{n}. \quad (79)
\end{aligned}$$

Lastly, by utilizing Taylor's expansion regarding the function $\mathbb{E}[K_2(II)] = \mathbb{E}\left[-2\sigma^4 e^{\frac{\sigma^2}{2}}K_2^2(a^2\sigma^2 K_2^2 + 1)\Phi(-K_1\sigma)\right]$ w.r.t. $K_1$ at $K_1 = 0$, and the facts that $\mathbb{E}[K_1] = 0$ and $\mathbb{E}[K_1^2] = 1/d$, we can derive that

$$\left|\mathbb{E}\left[K_2(II)\right] + \sigma^4 e^{\frac{\sigma^2}{2}}(a^2\sigma^2 + 1)\right| \leq \frac{c_5(a, \sigma)}{d}. \quad (80)$$

Substituting the results of (75), (76), (78), (79), and (80) into (74), we complete the proof. $\qquad\square$

**Lemma H.8.** *Let* $x_1, x_2, \ldots, x_n \sim \mathcal{N}(0, \sigma^2)$ *be* $n$ *i.i.d Gaussian random variables, and* $\widetilde{x}_1, \widetilde{x}_2, \ldots, \widetilde{x}_n \sim \mathcal{N}(0, \sigma^2)$ *be another* $n$ *i.i.d Gaussian random variables. In addition,* $a$ *is a positive scalar, and* $\boldsymbol{\theta}_*, \boldsymbol{\theta}_0 \in \mathbb{R}^d$ *are two independent random vectors following uniform* $d-$*dimensional sphere distribution, and we denote* $K_1 = \langle \boldsymbol{\theta}_0, \boldsymbol{\theta}_* \rangle$, $K_2 = \|(\mathbf{I}_d - \boldsymbol{\theta}_*\boldsymbol{\theta}_*^\top)\boldsymbol{\theta}_0\|_2$. *Then we have*

$$\left|\mathbb{E}\left[\frac{K_2 \sum_{i_1=1}^n \sum_{i_2=1}^n e^{-K_1(|\widetilde{x}_{i_1}|+a|\widetilde{x}_{i_2}|)-K_2(x_{i_1}+ax_{i_2})}x_{i_2}|\widetilde{x}_{i_1}||\widetilde{x}_{i_2}|}{\sum_{i=1}^n e^{-aK_1|\widetilde{x}_i|-aK_2 x_i}}\right] + \frac{2}{\pi}an\sigma^4 e^{\frac{\sigma^2}{2}}\right| \leq f_{13}(a, \sigma) + \frac{nf_{14}(a, \sigma)}{d}.$$

*Here,* $f_{13}(a, \sigma)$ *and* $f_{14}(a, \sigma)$ *are both analytic functions of* $a$ *and* $\sigma$ *while irrelevant with* $n$ *and* $d$.

*Proof of Lemma H.8.* Following a similar procedure in the proof of Lemma H.3, we can obtain that

$$\begin{aligned}
&\mathbb{E}\left[\frac{K_2 \sum_{i_1=1}^n \sum_{i_2=1}^n e^{-K_1(|\widetilde{x}_{i_1}|+a|\widetilde{x}_{i_2}|)-K_2(x_{i_1}+ax_{i_2})}x_{i_1}|\widetilde{x}_{i_1}||\widetilde{x}_{i_2}|}{\sum_{i=1}^n e^{-aK_1|\widetilde{x}_i|-aK_2 x_i}}\right] \\
&= \int_0^\infty \mathbb{E}\left[K_2 \sum_{i_1,i_2=1}^n e^{-K_1(|\widetilde{x}_{i_1}|+a|\widetilde{x}_{i_2}|)-K_2(x_{i_1}+ax_{i_2})}x_{i_1}|\widetilde{x}_{i_1}||\widetilde{x}_{i_2}|\exp\left(-s'\sum_{i=1}^n e^{-aK_1|\widetilde{x}_i|-aK_2 x_i}\right)\right]\mathrm{d}s' \\
&= n\mathbb{E}\left[K_2 \int_0^\infty \mathbb{E}\left[e^{-K_1(1+a)|\widetilde{x}_1|-K_2(1+a)x_1}x_1\widetilde{x}_1^2 e^{-s'e^{-a(K_1|\widetilde{x}_1|+K_2 x_1)}}\Big| K_1\right]\left(\mathbb{E}\left[e^{-s'e^{-a(K_1|\widetilde{x}_1|+K_2 x_1)}}\Big| K_1\right]\right)^{n-1}\mathrm{d}s'\right]
\end{aligned}$$

$$+ n(n-1)\mathbb{E}\bigg[K_2 \int_0^\infty \mathbb{E}\big[e^{-K_1(|\widetilde{x}_1|+a|\widetilde{x}_2|)-K_2(x_1+ax_2)}x_1|\widetilde{x}_1||\widetilde{x}_2|e^{-s'e^{-a(K_1|\widetilde{x}_1|+K_2x_1)}-s'e^{-a(K_1|\widetilde{x}_2|+K_2x_2)}}\big|K_1\big]$$

$$\cdot \Big(\mathbb{E}\big[e^{-s'e^{-a(K_1|\widetilde{x}_1|+K_2x_1)}}\big|K_1\big]\Big)^{n-2}\mathrm{d}s'\bigg]$$

$$=\mathbb{E}\bigg[K_2 \int_0^\infty \mathbb{E}\big[e^{-K_1(1+a)|\widetilde{x}_1|-K_2(1+a)x_1}x_1\widetilde{x}_1^2 e^{-\frac{s}{n}e^{-a(K_1|\widetilde{x}_1|+K_2x_1)}}\big|K_1\big]\Big(\mathbb{E}\big[e^{-\frac{s}{n}e^{-a(K_1|\widetilde{x}_1|+K_2x_1)}}\big|K_1\big]\Big)^{n-1}\mathrm{d}s\bigg]$$

$$+ (n-1)\mathbb{E}\bigg[K_2 \int_0^\infty \mathbb{E}\big[e^{-K_1(|\widetilde{x}_1|+a|\widetilde{x}_2|)-K_2(x_1+ax_2)}x_1|\widetilde{x}_1||\widetilde{x}_2|e^{-\frac{s}{n}e^{-a(K_1|\widetilde{x}_1|+K_2x_1)}-\frac{s}{n}e^{-a(K_1|\widetilde{x}_2|+K_2x_2)}}\big|K_1\big]$$

$$\cdot \Big(\mathbb{E}\big[e^{-\frac{s}{n}e^{-a(K_1|\widetilde{x}_1|+K_2x_1)}}\big|K_1\big]\Big)^{n-2}\mathrm{d}s\bigg]$$

$$=\mathbb{E}\bigg[K_2 \underbrace{\int_0^\infty A_8(s/n, 1+a, a, K_1)[M(s/n, a, K_1)]^{n-1}\mathrm{d}s}_{(I)}\bigg]$$

$$+ (n-1)\mathbb{E}\bigg[K_2 \underbrace{\int_0^\infty A_7(s/n, a, a, K_1)A_4(s/n, 1, a, K_1)[M(s/n, a, K_1)]^{n-2}\mathrm{d}s}_{(II)}\bigg]. \tag{81}$$

Here, $A_2(\lambda, \alpha, a, K_1)$ and $M(\lambda, a, K_1)$ share the same definitions as in the proof in Lemma H.1, $A_3(\lambda, \alpha, a, K_1)$ and $A_4(\lambda, \alpha, a, K_1)$ shares the same definitions as in the proof in Lemma H.3. In addition, the terms $A_7(\lambda, \alpha, a, K_1)$ and $A_8(\lambda, \alpha, a, K_1)$ are defined as:

$$A_7(\lambda, \alpha, a, K_1) = \mathbb{E}\big[e^{-\alpha K_1|\widetilde{x}_1|-\alpha K_2x_1}|\widetilde{x}_1|x_1 e^{-\lambda e^{-a(K_1|\widetilde{x}_1|+K_2x_1)}}\big|K_1\big];$$

$$A_8(\lambda, \alpha, a, K_1) = \mathbb{E}\big[e^{-K_1\alpha|\widetilde{x}_1|-K_2\alpha x_1}x_1|\widetilde{x}_1|^2 e^{-\lambda e^{-a(K_1|\widetilde{x}_1|+K_2x_1)}}\big|K_1\big].$$

To calculate the upper and lower bounds for the terms $(I)$ and $(II)$, we first calculate the derivatives of $A_7(\lambda, \alpha, a, K_1)$ and $A_8(\lambda, \alpha, a, K_1)$ w.r.t. $\lambda$ as

$$\left|\frac{\mathrm{d}A_7(\lambda, \alpha, a, K_1)}{\mathrm{d}\lambda}\right| = \left|-\mathbb{E}\big[e^{-(\alpha+a)K_1|\widetilde{x}_1|-(\alpha+a)K_2x_1}|\widetilde{x}_1|x_1 e^{-\lambda e^{-a(K_1|\widetilde{x}_1|+K_2x_1)}}\big|K_1\big]\right|$$

$$\leq \mathbb{E}\big[|x_1||\widetilde{x}_1|e^{-(\alpha+a)K_1|\widetilde{x}_1|-(\alpha+a)K_2x_1}\big|K_1\big] \leq 2\sigma^2 e^{(a+\alpha)^2\sigma^2};$$

$$\left|\frac{\mathrm{d}A_8(\lambda, \alpha, a, K_1)}{\mathrm{d}\lambda}\right| = \left|-\mathbb{E}\big[e^{-(\alpha+a)K_1|\widetilde{x}_1|-(\alpha+a)K_2x_1}|\widetilde{x}_1|^2 x_1 e^{-\lambda e^{-a(K_1|\widetilde{x}_1|+K_2x_1)}}\big|K_1\big]\right|$$

$$\leq \mathbb{E}\big[|x_1||\widetilde{x}_1|^2 e^{-(\alpha+a)K_1|\widetilde{x}_1|-(\alpha+a)K_2x_1}\big|K_1\big] \leq 6\sigma^2 e^{(a+\alpha)^2\sigma^2},$$

which implies that $A_7(\lambda, \alpha, a, K_1)$ and $A_8(\lambda, \alpha, a, K_1)$ is Lipschitz continuous w.r.t. $\lambda$, and $|A_7(s/n, a, a, K_1) - A_7(0, a, a, K_1)| \leq 2\sigma^2 e^{4a^2\sigma^2} s/n$, $|A_8(s/n, 1+a, a, K_1) - A_8(0, 1+a, a, K_1)| \leq 6\sigma^2 e^{(2a+1)^2\sigma^2} s/n$. Consequently, we can establish the upper and lower bounds for the term $(I)$ as

$$(I) \geq A_8(0, 1+a, a, K_1)\int_0^\infty [M(s/n, a, K_1)]^{n-1}\mathrm{d}s - \frac{c_1(a, \sigma)}{n}\int_0^\infty s[M(s/n, a, K_1)]^{n-1}\mathrm{d}s$$

$$= A_8(0, 1+a, a, K_1)\mathbb{E}\bigg[\frac{n}{\sum_{i=2}^n e^{-a(K_1|\widetilde{x}_i|+K_2x_i)}}\bigg|K_1\bigg] - \frac{c_1(a, \sigma)}{n}\mathbb{E}\bigg[\frac{n^2}{\big(\sum_{i=2}^n e^{-a(K_1|\widetilde{x}_i|+K_2x_i)}\big)^2}\bigg|K_1\bigg]$$

$$\geq -\frac{(1+a)\sigma^4 K_2 e^{\frac{(2a+1)\sigma^2}{2}}\big((1+(1+a)^2 K_1^2\sigma^2)\Phi(-(1+a)K_1\sigma) - (1+a)K_1\sigma\phi((1+a)K_1\sigma)\big)}{2\Phi^2(-a\sigma K_1)} - 1. \tag{82}$$

The first equality is true by applying Fubini's theorem to exchange the order of integration. The penultimate inequality is derived by applying Lemma H.19, and the fact that $A_8(0, 1+a, a, K_1) = -2(1+a)\sigma^4 K_2 e^{(1+a)^2\sigma^2/2}\big[(1+(1+a)^2 K_1^2\sigma^2)\Phi(-(1+a)K_1\sigma) - (1+a)K_1\sigma\,\phi((1+a)K_1\sigma)\big]$. Following a similar calculation, we can also get the upper bound for $(I)$ as

$$(I) \leq A_8(0, 1+a, a, K_1)\int_0^\infty [M(s/n, a, K_1)]^{n-1}\mathrm{d}s + \frac{c_1(a, \sigma)}{n}\int_0^\infty s[M(s/n, a, K_1)]^{n-1}\mathrm{d}s$$

$$\leq -\frac{(1+a)\sigma^4 K_2 e^{\frac{(2a+1)\sigma^2}{2}}\left((1+(1+a)^2 K_1^2\sigma^2)\Phi(-(1+a)K_1\sigma) - (1+a)K_1\sigma\phi((1+a)K_1\sigma)\right)}{2\Phi^2(-a\sigma K_1)} + 1. \quad (83)$$

Combined with the Lipschitz continuity of $A_4(\lambda, \alpha, a, K_1)$ established in the proof of Lemma H.3, we have

$$|A_7(s/n, a, a, K_1)A_4(s/n, 1, a, K_1) - A_7(0, a, a, K_1)A_4(0, 1, a, K_1)| \leq \frac{c_3(a, \sigma)s}{n} + \frac{c_4(a, \sigma)s^2}{n^2}.$$

Now, we are ready to derive the lower and upper bounds for the term $(II)$ as

$$(II) = \int_0^\infty A_7(s/n, a, a, K_1)A_4(s/n, 1, a, K_1)[M(s/n, a, K_1)]^{n-2}\mathrm{d}s$$

$$\geq A_7(0, a, a, K_1)A_4(0, 1, a, K_1)\int_0^\infty [M(s/n, a, K_1)]^{n-2}\mathrm{d}s$$

$$- \frac{c_3(a, \sigma)}{n}\int_0^\infty s[M(s/n, a, K_1)]^{n-2}\mathrm{d}s - \frac{c_4(a, \sigma)}{n^2}\int_0^\infty s^2[M(s/n, a, K_1)]^{n-2}\mathrm{d}s$$

$$= A_7(0, a, a, K_1)A_4(0, 1, a, K_1)\mathbb{E}\left[\frac{n}{\sum_{i=3}^n e^{-a(K_1|\widetilde{x}_i|+K_2 x_i)}}\Big| K_1\right]$$

$$- \frac{c_3(a, \sigma)}{n}\mathbb{E}\left[\frac{n^2}{\left(\sum_{i=3}^n e^{-a(K_1|\widetilde{x}_i|+K_2 x_i)}\right)^2}\right] - \frac{2c_4(a, \sigma)}{n^2}\mathbb{E}\left[\frac{n^3}{\left(\sum_{i=3}^n e^{-a(K_1|\widetilde{x}_i|+K_2 x_i)}\right)^3}\right]$$

$$\geq \frac{A_7(0, a, a, K_1)A_4(0, 1, a, K_1)}{2e^{a^2\sigma^2/2}\Phi(-a\sigma K_1)} - \frac{c_5(a, \sigma)}{n}, \quad (84)$$

where $c_5(a, \sigma)$ is a positive constant solely depending on $a$ and $\sigma$. Here the last equation is derived by Fubini's theorem to exchange the order of the integral, and the last inequality is established by Lemma H.19. Similarly, we can also obtain that

$$(II) \leq A_7(0, a, a, K_1)A_4(0, 1, a, K_1)\int_0^\infty [M(s/n, a, K_1)]^{n-2}\mathrm{d}s$$

$$+ \frac{c_3(a, \sigma)}{n}\int_0^\infty s[M(s/n, a, K_1)]^{n-2}\mathrm{d}s + \frac{c_4(a, \sigma)}{n^2}\int_0^\infty s^2[M(s/n, a, K_1)]^{n-2}\mathrm{d}s$$

$$\leq \frac{A_7(0, a, a, K_1)A_4(0, 1, a, K_1)}{2e^{a^2\sigma^2/2}\Phi(-a\sigma K_1)} + \frac{c_5(a, \sigma)}{n}. \quad (85)$$

To provide an exact lower and upper bound for $\mathbb{E}\left[\frac{K_2 A_7(0, a, a, K_1)A_4(0, 1, a, K_1)}{2e^{a^2\sigma^2/2}\Phi(-a\sigma K_1)}\right]$, we consider its Taylor's expansion at $K_1 = 0$, then we have

$$\left|\mathbb{E}\left[\frac{K_2 A_7(0, a, a, K_1)A_4(0, 1, a, K_1)}{2e^{a^2\sigma^2/2}\Phi(-a\sigma K_1)}\right] + \frac{2}{\pi}a\sigma^4 e^{\frac{\sigma^2}{2}}\right| \leq \frac{c_6(a, \sigma)}{d} \quad (86)$$

Substituting the results of (82), (83), (84), (85), and (86) into (81), we complete the proof. $\qquad\square$

**Lemma H.9.** *Let $x_1, x_2, \ldots, x_n \sim \mathcal{N}(0, \sigma^2)$ be $n$ i.i.d Gaussian random variables, and $\widetilde{x}_1, \widetilde{x}_2, \ldots, \widetilde{x}_n \sim \mathcal{N}(0, \sigma^2)$ be another $n$ i.i.d Gaussian random variables. In addition, $a$ is a positive scalar, and $\boldsymbol{\theta}_*, \boldsymbol{\theta}_0 \in \mathbb{R}^d$ are two independent random vectors following uniform $d-$dimensional sphere distribution, and we denote $K_1 = \langle\boldsymbol{\theta}_0, \boldsymbol{\theta}_*\rangle$, $K_2 = \|(\mathbf{I}_d - \boldsymbol{\theta}_*\boldsymbol{\theta}_*^\top)\boldsymbol{\theta}_0\|_2$. Then we have*

$$\left|\mathbb{E}\left[\frac{K_2\sum_{i=1}^n e^{-(1+a)K_1|\widetilde{x}_i|-(1+a)K_2 x_i}x_i}{\sum_{i=1}^n e^{-aK_1|\widetilde{x}_i|-aK_2 x_i}}\right] + (1+a)\sigma^2 e^{(2a+1)\sigma^2/2}\right| \leq \frac{f_{15}(a, \sigma)}{n} + \frac{f_{16}(a, \sigma)}{d}.$$

*Here, $f_{15}(a, \sigma)$ and $f_{16}(a, \sigma)$ are both analytic functions of $a$ and $\sigma$ while irrelevant with $n$ and $d$.*

*Proof of Lemma H.9.* Following a similar procedure in the proof of Lemma H.5, we can obtain that

$$\mathbb{E}\left[K_2\frac{\sum_{i=1}^n e^{-K_1(1+a)|\widetilde{x}_i|-K_2(1+a)x_i}x_i}{\sum_{i=1}^n e^{-aK_1|\widetilde{x}_i|-aK_2 x_i}}\right]$$

$$= \int_0^\infty \mathbb{E}\left[K_2 \sum_{i=1}^n e^{-K_1(1+a)|\widetilde{x}_i|-K_2(1+a)x_i} x_i \exp\left(-s' \sum_{i=1}^n e^{-aK_1|\widetilde{x}_i|-aK_2x_i}\right)\right] \mathrm{d}s'$$

$$= n\mathbb{E}\left[K_2 \int_0^\infty \mathbb{E}\left[e^{-K_1(1+a)|\widetilde{x}_1|-K_2(1+a)x_1} e^{-s'e^{-a(K_1|\widetilde{x}_1|+K_2x_1)}} x_1|K_1\right] \left(\mathbb{E}\left[e^{-s'e^{-a(K_1|\widetilde{x}_1|+K_2x_1)}}|K_1\right]\right)^{n-1} \mathrm{d}s'\right]$$

$$= \mathbb{E}\left[K_2 \int_0^\infty \mathbb{E}\left[e^{-K_1(1+a)|\widetilde{x}_1|-K_2(1+a)x_1} e^{-\frac{s}{n}e^{-a(K_1|\widetilde{x}_1|+K_2x_1)}} x_1|K_1\right] \left(\mathbb{E}\left[e^{-\frac{s}{n}e^{-a(K_1|\widetilde{x}_1|+K_2x_1)}}|K_1\right]\right)^{n-1} \mathrm{d}s\right]$$

$$= \mathbb{E}\left[K_2 \underbrace{\int_0^\infty A_2(s/n, 1+a, a, K_1)[M(s/n, a, K_1)]^{n-1}\mathrm{d}s}_{(I)}\right], \tag{87}$$

Here, $A_2(\lambda, \alpha, a, K_1)$ and $M(\lambda, a, K_1)$ share the same definitions as in the proof in Lemma H.1. By using the Lipschitz continuity of $A_2(\lambda, \alpha, a, K_1)$ established in the proof of Lemma H.1, we can establish the upper and lower bounds for the term $(I)$ as

$$(I) \geq A_2(0, 1+a, a, K_1) \int_0^\infty [M(s/n, a, K_1)]^{n-1}\mathrm{d}s - \frac{c_1(a, \sigma)}{n} \int_0^\infty s[M(s/n, a, K_1)]^{n-1}\mathrm{d}s$$

$$= A_2(0, 1+a, a, K_1)\mathbb{E}\left[\frac{n}{\sum_{i=2}^n e^{-a(K_1|\widetilde{x}_i|+K_2x_i)}}\Big|K_1\right] - \frac{c_1(a, \sigma)}{n}\mathbb{E}\left[\frac{n^2}{\left(\sum_{i=2}^n e^{-a(K_1|\widetilde{x}_i|+K_2x_i)}\right)^2}\Big|K_1\right]$$

$$\geq -\frac{(1+a)\sigma^2 K_2 e^{(2a+1)\sigma^2/2}\Phi(-(a+1)K_1\sigma)}{\Phi(-aK_1\sigma)} - \frac{c_2(a, \sigma)}{n}. \tag{88}$$

The first equality is true by applying Fubini's theorem to exchange the order of integration. The penultimate inequality is derived by applying Lemma H.19, and the fact that $A_2(0, 1+a, a, K_1) = -2(1+a)\sigma^2 K_2 e^{\frac{(1+a)^2\sigma^2}{2}}\Phi(-(1+a)K_1\sigma)$. Similarly, we also have

$$(I) \leq A_2(0, 1+a, a, K_1)\mathbb{E}\left[\frac{n}{\sum_{i=2}^n e^{-a(K_1|\widetilde{x}_i|+K_2x_i)}}\Big|K_1\right] + \frac{c_1(a, \sigma)}{n}\mathbb{E}\left[\frac{n^2}{\left(\sum_{i=2}^n e^{-a(K_1|\widetilde{x}_i|+K_2x_i)}\right)^2}\Big|K_1\right]$$

$$\leq -\frac{(1+a)\sigma^2 K_2 e^{(2a+1)\sigma^2/2}\Phi(-(a+1)K_1\sigma)}{\Phi(-aK_1\sigma)} + \frac{c_2(a, \sigma)}{n}. \tag{89}$$

To provide an exact lower and upper bound for $\mathbb{E}\left[-\frac{(1+a)\sigma^2 K_2^2 e^{(2a+1)\sigma^2/2}\Phi(-(a+1)K_1\sigma)}{\Phi(-aK_1\sigma)}\right]$, we consider its Taylor's expansion at $K_1 = 0$, then we have

$$\left|\mathbb{E}\left[-\frac{(1+a)\sigma^2 K_2^2 e^{\sigma^2/2}\Phi(-(a+1)K_1\sigma)}{\Phi(-aK_1\sigma)}\right] + (1+a)\sigma^2 e^{(2a+1)\sigma^2/2}\right| \leq \frac{c_3(a, \sigma)}{d} \tag{90}$$

Substituting the results of (88), (89), and (90) into (87), we complete the proof. $\square$

**Lemma H.10.** *Let $x_1, x_2, \ldots, x_n \sim \mathcal{N}(0, \sigma^2)$ be $n$ i.i.d Gaussian random variables, and $\widetilde{x}_1, \widetilde{x}_2, \ldots, \widetilde{x}_n \sim \mathcal{N}(0, \sigma^2)$ be another $n$ i.i.d Gaussian random variables. In addition, $a$ is a positive scalar, and $\boldsymbol{\theta}_*, \boldsymbol{\theta}_0 \in \mathbb{R}^d$ are two independent random vectors following uniform $d-$dimensional sphere distribution, and we denote $K_1 = \langle\boldsymbol{\theta}_0, \boldsymbol{\theta}_*\rangle$, $K_2 = \|(\mathbf{I}_d - \boldsymbol{\theta}_*\boldsymbol{\theta}_*^\top)\boldsymbol{\theta}_0\|_2$. Then we have*

$$\left|\mathbb{E}\left[\frac{K_2 \sum_{i_1=1}^n \sum_{i_2=1}^n \sum_{i_3=1}^n e^{-K_1(|\widetilde{x}_{i_1}|+a(|\widetilde{x}_{i_2}|+|\widetilde{x}_{i_3}|))-K_2(x_{i_1}+ax_{i_2}+ax_{i_3})}x_{i_1}x_{i_2}x_{i_3}}{\left(\sum_{i=1}^n e^{-aK_1|\widetilde{x}_i|-aK_2x_i}\right)^2}\right] + na^2\sigma^6 e^{\frac{\sigma^2}{2}}\right|$$

$$\leq f_{17}(a, \sigma) + \frac{nf_{18}(a, \sigma)}{d}.$$

*Here, $f_{17}(a, \sigma)$, $f_{18}(a, \sigma)$ are both analytic functions of $a$ and $\sigma$ while irrelevant with $n$ and $d$.*

*Proof of Lemma H.10.* Following a similar procedure in the proof of Lemma H.2, we can obtain that

$$\mathbb{E}\left[\frac{K_2 \sum_{i_1=1}^n \sum_{i_2=1}^n \sum_{i_3=1}^n e^{-K_1(|\widetilde{x}_{i_1}|+a(|\widetilde{x}_{i_2}|+|\widetilde{x}_{i_3}|))-K_2(x_{i_1}+ax_{i_2}+ax_{i_3})}x_{i_1}x_{i_2}x_{i_3}}{\left(\sum_{i=1}^n e^{-aK_1|\widetilde{x}_i|-aK_2x_i}\right)^2}\right]$$

$$
=\int_0^\infty s'\mathbb{E}\Bigg[K_2\sum_{i_1,i_2,i_3=1}^n e^{-K_1(|\widetilde{x}_{i_1}|+a(|\widetilde{x}_{i_2}|+|\widetilde{x}_{i_3}|))-K_2(x_{i_1}+ax_{i_2}+ax_{i_3})}x_{i_1}x_{i_2}x_{i_3}\exp\Big(-s'\sum_{i=1}^n e^{-aK_1|\widetilde{x}_i|-aK_2x_i}\Big)\Bigg]\mathrm{d}s'
$$

$$
=n\mathbb{E}\Bigg[K_2\int_0^\infty s'\mathbb{E}\big[e^{-K_1(1+2a)|\widetilde{x}_1|-K_2(1+2a)x_1}x_1^3 e^{-s'e^{-a(K_1|\widetilde{x}_1|+K_2x_1)}}\big|K_1\big]\Big(\mathbb{E}\big[e^{-s'e^{-a(K_1|\widetilde{x}_1|+K_2x_1)}}\big|K_1\big]\Big)^{n-1}\mathrm{d}s'\Bigg]
$$

$$
+n(n-1)\mathbb{E}\Bigg[K_2\int_0^\infty s'\mathbb{E}\big[e^{-K_1(|\widetilde{x}_1|+2a|\widetilde{x}_2|)-K_2(x_1+2ax_2)}x_1x_2^2 e^{-s'e^{-a(K_1|\widetilde{x}_1|+K_2x_1)}-s'e^{-a(K_1|\widetilde{x}_2|+K_2x_2)}}\big|K_1\big]
$$

$$
\cdot\Big(\mathbb{E}\big[e^{-s'e^{-a(K_1|\widetilde{x}_1|+K_2x_1)}}\big|K_1\big]\Big)^{n-2}\mathrm{d}s'\Bigg]
$$

$$
+2n(n-1)\mathbb{E}\Bigg[K_2\int_0^\infty s'\mathbb{E}\big[e^{-K_1(a|\widetilde{x}_1|+(a+1)|\widetilde{x}_2|)-K_2(ax_1+(a+1)x_2)}x_1x_2^2 e^{-s'e^{-a(K_1|\widetilde{x}_1|+K_2x_1)}-s'e^{-a(K_1|\widetilde{x}_2|+K_2x_2)}}\big|K_1\big]
$$

$$
\cdot\Big(\mathbb{E}\big[e^{-s'e^{-a(K_1|\widetilde{x}_1|+K_2x_1)}}\big|K_1\big]\Big)^{n-2}\mathrm{d}s'\Bigg]
$$

$$
+n(n-1)(n-2)\mathbb{E}\Bigg[K_2\int_0^\infty s'\mathbb{E}\big[e^{-K_1(|\widetilde{x}_1|+a|\widetilde{x}_2|+a|\widetilde{x}_3|)-K_2(x_1+ax_2+ax_3)}x_1x_2x_3
$$

$$
\cdot e^{-s'e^{-a(K_1|\widetilde{x}_1|+K_2x_1)}-s'e^{-a(K_1|\widetilde{x}_2|+K_2x_2)}-s'e^{-a(K_1|\widetilde{x}_3|+K_2x_3)}}\big|K_1\big]\Big(\mathbb{E}\big[e^{-s'e^{-a(K_1|\widetilde{x}_1|+K_2x_1)}}\big|K_1\big]\Big)^{n-3}\mathrm{d}s'\Bigg]
$$

$$
=\frac{1}{n}\mathbb{E}\Bigg[K_2\int_0^\infty s\mathbb{E}\big[e^{-K_1(1+2a)|\widetilde{x}_1|-K_2(1+2a)x_1}x_1^3 e^{-\frac{s}{n}e^{-a(K_1|\widetilde{x}_1|+K_2x_1)}}\big|K_1\big]\Big(\mathbb{E}\big[e^{-\frac{s}{n}e^{-a(K_1|\widetilde{x}_1|+K_2x_1)}}\big|K_1\big]\Big)^{n-1}\mathrm{d}s\Bigg]
$$

$$
+\frac{n-1}{n}\mathbb{E}\Bigg[K_2\int_0^\infty s\mathbb{E}\big[e^{-K_1(|\widetilde{x}_1|+2a|\widetilde{x}_2|)-K_2(x_1+2ax_2)}x_1x_2^2 e^{-\frac{s}{n}e^{-a(K_1|\widetilde{x}_1|+K_2x_1)}-\frac{s}{n}e^{-a(K_1|\widetilde{x}_2|+K_2x_2)}}\big|K_1\big]
$$

$$
\cdot\Big(\mathbb{E}\big[e^{-\frac{s}{n}e^{-a(K_1|\widetilde{x}_1|+K_2x_1)}}\big|K_1\big]\Big)^{n-2}\mathrm{d}s\Bigg]
$$

$$
+\frac{2(n-1)}{n}\mathbb{E}\Bigg[K_2\int_0^\infty s\mathbb{E}\big[e^{-K_1(a|\widetilde{x}_1|+(a+1)|\widetilde{x}_2|)-K_2(ax_1+(a+1)x_2)}x_1x_2^2 e^{-\frac{s}{n}e^{-a(K_1|\widetilde{x}_1|+K_2x_1)}-\frac{s}{n}e^{-a(K_1|\widetilde{x}_2|+K_2x_2)}}\big|K_1\big]
$$

$$
\cdot\Big(\mathbb{E}\big[e^{-\frac{s}{n}e^{-a(K_1|\widetilde{x}_1|+K_2x_1)}}\big|K_1\big]\Big)^{n-2}\mathrm{d}s\Bigg]
$$

$$
+\frac{(n-1)(n-2)}{n}\mathbb{E}\Bigg[K_2\int_0^\infty s\mathbb{E}\big[e^{-K_1(|\widetilde{x}_1|+a|\widetilde{x}_2|+a|\widetilde{x}_3|)-K_2(x_1+ax_2+ax_3)}x_1x_2x_3
$$

$$
\cdot e^{-\frac{s}{n}e^{-a(K_1|\widetilde{x}_1|+K_2x_1)}-\frac{s}{n}e^{-a(K_1|\widetilde{x}_2|+K_2x_2)}-\frac{s}{n}e^{-a(K_1|\widetilde{x}_3|+K_2x_3)}}\big|K_1\big]\Big(\mathbb{E}\big[e^{-\frac{s}{n}e^{-a(K_1|\widetilde{x}_1|+K_2x_1)}}\big|K_1\big]\Big)^{n-3}\mathrm{d}s\Bigg]
$$

$$
=\frac{1}{n}\mathbb{E}\Bigg[K_2\underbrace{\int_0^\infty sA_6(s/n,1+2a,a,K_1)[M(s/n,a,K_1)]^{n-1}\mathrm{d}s}_{(I)}\Bigg]
$$

$$
+\frac{n-1}{n}\mathbb{E}\Bigg[K_2\underbrace{\int_0^\infty sA_2(s/n,1,a,K_1)A_1(s/n,2a,a,K_1)[M(s/n,a,K_1)]^{n-2}\mathrm{d}s}_{(II)}\Bigg]
$$

$$
+\frac{2(n-1)}{n}\mathbb{E}\Bigg[K_2\underbrace{\int_0^\infty sA_2(s/n,a,a,K_1)A_1(s/n,a+1,a,K_1)[M(s/n,a,K_1)]^{n-2}\mathrm{d}s}_{(III)}\Bigg]
$$

$$
+\frac{(n-1)(n-2)}{n}\mathbb{E}\Bigg[K_2\underbrace{\int_0^\infty sA_2^2(s/n,a,a,K_1)A_2(s/n,1,a,K_1)[M(s/n,a,K_1)]^{n-3}\mathrm{d}s}_{(IV)}\Bigg]. \tag{91}
$$

Here, $A_1(\lambda,\alpha,a,K_1)$, $A_2(\lambda,\alpha,a,K_1)$ and $M(\lambda,a,K_1)$ share the same definitions as in the proof in Lemma H.1 and $A_6(\lambda,\alpha,a,K_1)$ shares the same definition as in the proof in Lemma H.7. Through similar procedures in the proof of

Lemma H.7 (by utilizing the Lipschitz continuities of these functions), we can derive that:

$$\left| (I) + \frac{A_6(0, 1+2a, a, K_1)}{4e^{a^2\sigma^2}\Phi^2(-a\sigma K_1)} \right|$$

$$= \left| (I) + K_2(1+2a)\sigma^4\big(K_2^2(1+2a)^2\sigma^2 + 3\big)e^{\frac{2a^2+4a+1}{2}\sigma^2}\frac{\Phi(-K_1(2a+1)\sigma)}{\Phi(-K_1 a\sigma)} \right| \leq \frac{c_1(a,\sigma)}{n};$$

$$\left| (II) + \frac{A_1(0, 2a, a, K_1)A_2(0, 1, a, K_1)}{4e^{a^2\sigma^2}\Phi^2(-a\sigma K_1)} \right|$$

$$= \left| (II) + \sigma^4 K_2\big(1 + 4a^2 K_2^2\sigma^2\big)e^{\frac{(2a^2+1)\sigma^2}{2}}\frac{\Phi(-K_1\sigma)\Phi(-2aK_1\sigma)}{\Phi^2(-aK_1\sigma)} \right| \leq \frac{c_2(a,\sigma)}{n};$$

$$\left| (III) + \frac{A_1(0, a+1, a, K_1)A_2(0, a, a, K_1)}{4e^{a^2\sigma^2}\Phi^2(-a\sigma K_1)} \right|$$

$$= \left| (III) + a\sigma^4 K_2\big(1 + (a+1)^2 K_2^2\sigma^2\big)e^{\frac{2a+1}{2}\sigma^2}\frac{\Phi(-(a+1)K_1\sigma)}{\Phi(-aK_1\sigma)} \right| \leq \frac{c_3(a,\sigma)}{n}. \tag{92}$$

Specifically, for the term $(IV)$, we have

$$\left| (IV) + \frac{A_2(0, 1, a, K_1)A_2^2(0, a, a, K_1)}{4e^{a^2\sigma^2}\Phi^2(-a\sigma K_1)} \right| = \left| (IV) + a^2\sigma^6 K_2^3 e^{\sigma^2/2}\frac{\Phi(-K_1\sigma)}{\Phi(-aK_1\sigma)} \right| \leq \frac{c_4(a,\sigma)}{n}. \tag{93}$$

Lastly, by utilizing Taylor's expansion regarding the function $\mathbb{E}\big[a^2\sigma^6 K_2^4 e^{\sigma^2/2}\frac{\Phi(-K_1\sigma)}{\Phi(-aK_1\sigma)}\big]$ w.r.t. $K_1$ at $K_1 = 0$, and the facts that $\mathbb{E}[K_1] = 0$ and $\mathbb{E}[K_1^2] = 1/d$, we can derive that

$$\left| \mathbb{E}\big[K_2(IV)\big] + a^2\sigma^6 e^{\frac{\sigma^2}{2}} \right| \leq \frac{c_4(a,\sigma)}{n} + \frac{c_5(a,\sigma)}{d}. \tag{94}$$

Substituting the results of (92), (93), and (94) into (91), we complete the proof. $\qquad\square$

**Lemma H.11.** *Let $x_1, x_2, \ldots, x_n \sim \mathcal{N}(0, \sigma^2)$ be $n$ i.i.d Gaussian random variables, and $\widetilde{x}_1, \widetilde{x}_2, \ldots, \widetilde{x}_n \sim \mathcal{N}(0, \sigma^2)$ be another $n$ i.i.d Gaussian random variables. In addition, $a$ is a positive scalar, and $\boldsymbol{\theta}_*, \boldsymbol{\theta}_0 \in \mathbb{R}^d$ are two independent random vectors following uniform $d-$dimensional sphere distribution, and we denote $K_1 = \langle \boldsymbol{\theta}_0, \boldsymbol{\theta}_* \rangle$, $K_2 = \|(\mathbf{I}_d - \boldsymbol{\theta}_*\boldsymbol{\theta}_*^\top)\boldsymbol{\theta}_0\|_2$. Then we have*

$$\left| \mathbb{E}\left[ \frac{K_2 \sum_{i_1=1}^n \sum_{i_2=1}^n \sum_{i_3=1}^n e^{-K_1(|\widetilde{x}_{i_1}|+a(|\widetilde{x}_{i_2}|+|\widetilde{x}_{i_3}|))-K_2(x_{i_1}+ax_{i_2}+ax_{i_3})}|\widetilde{x}_{i_1}||\widetilde{x}_{i_2}|x_{i_3}}{(\sum_{i=1}^n e^{-aK_1|\widetilde{x}_i|-aK_2 x_i})^2} \right] + \frac{2}{\pi}na\sigma^4 e^{\frac{\sigma^2}{2}} \right|$$

$$\leq f_{19}(a,\sigma) + \frac{nf_{20}(a,\sigma)}{d}.$$

*Here, $f_{19}(a,\sigma)$, $f_{20}(a,\sigma)$ are both analytic functions of $a$ and $\sigma$ while irrelevant with $n$ and $d$.*

*Proof of Lemma H.11.* Following a similar procedure in the proof of Lemma H.10, we can obtain that

$$\mathbb{E}\left[ \frac{K_2 \sum_{i_1=1}^n \sum_{i_2=1}^n \sum_{i_3=1}^n e^{-K_1(|\widetilde{x}_{i_1}|+a(|\widetilde{x}_{i_2}|+|\widetilde{x}_{i_3}|))-K_2(x_{i_1}+ax_{i_2}+ax_{i_3})}|\widetilde{x}_{i_1}||\widetilde{x}_{i_2}|x_{i_3}}{(\sum_{i=1}^n e^{-aK_1|\widetilde{x}_i|-aK_2 x_i})^2} \right]$$

$$= \int_0^\infty s'\mathbb{E}\left[ K_2 \sum_{i_1,i_2,i_3=1}^n e^{-K_1(|\widetilde{x}_{i_1}|+a(|\widetilde{x}_{i_2}|+|\widetilde{x}_{i_3}|))-K_2(x_{i_1}+ax_{i_2}+ax_{i_3})}|\widetilde{x}_{i_1}||\widetilde{x}_{i_2}|x_{i_3} \exp\left(-s'\sum_{i=1}^n e^{-aK_1|\widetilde{x}_i|-aK_2 x_i}\right) \right]\mathrm{d}s'$$

$$= n\mathbb{E}\left[ K_2 \int_0^\infty s'\mathbb{E}\big[e^{-K_1(1+2a)|\widetilde{x}_1|-K_2(1+2a)x_1}x_1|\widetilde{x}_1|^2 e^{-s'e^{-a(K_1|\widetilde{x}_1|+K_2 x_1)}}\big|K_1\big]\Big(\mathbb{E}\big[e^{-s'e^{-a(K_1|\widetilde{x}_1|+K_2 x_1)}}\big|K_1\big]\Big)^{n-1}\mathrm{d}s' \right]$$

$$+ n(n-1)\mathbb{E}\bigg[ K_2 \int_0^\infty s'\mathbb{E}\big[e^{-K_1(|\widetilde{x}_1|+2a|\widetilde{x}_2|)-K_2(x_1+2ax_2)}|\widetilde{x}_1||\widetilde{x}_2|x_2$$

$$\cdot e^{-s'e^{-a(K_1|\widetilde{x}_1|+K_2 x_1)}-s'e^{-a(K_1|\widetilde{x}_2|+K_2 x_1)}}\big|K_1\big]\Big(\mathbb{E}\big[e^{-s'e^{-a(K_1|\widetilde{x}_1|+K_2 x_1)}}\big|K_1\big]\Big)^{n-2}\mathrm{d}s' \bigg]$$

$$+ n(n-1)\mathbb{E}\bigg[K_2\int_0^\infty s'\mathbb{E}\big[e^{-K_1(a|\widetilde{x}_1|+(a+1)|\widetilde{x}_2|)-K_2(ax_1+(a+1)x_2)}x_1|\widetilde{x}_2|^2$$

$$\cdot e^{-s'e^{-a(K_1|\widetilde{x}_1|+K_2x_1)}-s'e^{-a(K_1|\widetilde{x}_2|+K_2x_2)}}|K_1\big]\big(\mathbb{E}\big[e^{-s'e^{-a(K_1|\widetilde{x}_1|+K_2x_1)}}|K_1\big]\big)^{n-2}\mathrm{d}s'\bigg]$$

$$+ n(n-1)\mathbb{E}\bigg[K_2\int_0^\infty s'\mathbb{E}\big[e^{-K_1(a|\widetilde{x}_1|+(a+1)|\widetilde{x}_2|)-K_2(ax_1+(a+1)x_2)}|\widetilde{x}_1||\widetilde{x}_2|x_2$$

$$\cdot e^{-s'e^{-a(K_1|\widetilde{x}_1|+K_2x_1)}-s'e^{-a(K_1|\widetilde{x}_2|+K_2x_2)}}|K_1\big]\big(\mathbb{E}\big[e^{-s'e^{-a(K_1|\widetilde{x}_1|+K_2x_1)}}|K_1\big]\big)^{n-2}\mathrm{d}s'\bigg]$$

$$+ n(n-1)(n-2)\mathbb{E}\bigg[K_2\int_0^\infty s'\mathbb{E}\big[e^{-K_1(|\widetilde{x}_1|+a|\widetilde{x}_2|+a|\widetilde{x}_3|)-K_2(x_1+ax_2+ax_3)}|\widetilde{x}_1||\widetilde{x}_2|x_3$$

$$\cdot e^{-s'e^{-a(K_1|\widetilde{x}_1|+K_2x_1)}-s'e^{-a(K_1|\widetilde{x}_2|+K_2x_2)}-s'e^{-a(K_1|\widetilde{x}_3|+K_2x_3)}}|K_1\big]\big(\mathbb{E}\big[e^{-s'e^{-a(K_1|\widetilde{x}_1|+K_2x_1)}}|K_1\big]\big)^{n-3}\mathrm{d}s'\bigg]$$

$$=\frac{1}{n}\mathbb{E}\bigg[K_2\int_0^\infty s\mathbb{E}\big[e^{-K_1(1+2a)|\widetilde{x}_1|-K_2(1+2a)x_1}x_1|\widetilde{x}_1|^2 e^{-\frac{s}{n}e^{-a(K_1|\widetilde{x}_1|+K_2x_1)}}|K_1\big]\big(\mathbb{E}\big[e^{-\frac{s}{n}e^{-a(K_1|\widetilde{x}_1|+K_2x_1)}}|K_1\big]\big)^{n-1}\mathrm{d}s\bigg]$$

$$+\frac{n-1}{n}\mathbb{E}\bigg[K_2\int_0^\infty s\mathbb{E}\big[e^{-K_1(|\widetilde{x}_1|+2a|\widetilde{x}_2|)-K_2(x_1+2ax_2)}|\widetilde{x}_1||\widetilde{x}_2|x_2$$

$$\cdot e^{-\frac{s}{n}e^{-a(K_1|\widetilde{x}_1|+K_2x_1)}-\frac{s}{n}e^{-a(K_1|\widetilde{x}_2|+K_2x_2)}}|K_1\big]\big(\mathbb{E}\big[e^{-\frac{s}{n}e^{-a(K_1|\widetilde{x}_1|+K_2x_1)}}|K_1\big]\big)^{n-2}\mathrm{d}s\bigg]$$

$$+\frac{n-1}{n}\mathbb{E}\bigg[K_2\int_0^\infty s\mathbb{E}\big[e^{-K_1(a|\widetilde{x}_1|+(a+1)|\widetilde{x}_2|)-K_2(ax_1+(a+1)x_2)}x_1|\widetilde{x}_2|^2$$

$$\cdot e^{-\frac{s}{n}e^{-a(K_1|\widetilde{x}_1|+K_2x_1)}-\frac{s}{n}e^{-a(K_1|\widetilde{x}_2|+K_2x_2)}}|K_1\big]\big(\mathbb{E}\big[e^{-\frac{s}{n}e^{-a(K_1|\widetilde{x}_1|+K_2x_1)}}|K_1\big]\big)^{n-2}\mathrm{d}s\bigg]$$

$$+\frac{n-1}{n}\mathbb{E}\bigg[K_2\int_0^\infty s\mathbb{E}\big[e^{-K_1(a|\widetilde{x}_1|+(a+1)|\widetilde{x}_2|)-K_2(ax_1+(a+1)x_2)}|\widetilde{x}_1||\widetilde{x}_2|x_2$$

$$\cdot e^{-\frac{s}{n}e^{-a(K_1|\widetilde{x}_1|+K_2x_1)}-\frac{s}{n}e^{-a(K_1|\widetilde{x}_2|+K_2x_2)}}|K_1\big]\big(\mathbb{E}\big[e^{-\frac{s}{n}e^{-a(K_1|\widetilde{x}_1|+K_2x_1)}}|K_1\big]\big)^{n-2}\mathrm{d}s\bigg]$$

$$+ n(n-1)(n-2)\mathbb{E}\bigg[K_2\int_0^\infty s\mathbb{E}\big[e^{-K_1(|\widetilde{x}_1|+a|\widetilde{x}_2|+a|\widetilde{x}_3|)-K_2(x_1+ax_2+ax_3)}|\widetilde{x}_1||\widetilde{x}_2|x_3$$

$$\cdot e^{-\frac{s}{n}e^{-a(K_1|\widetilde{x}_1|+K_2x_1)}-\frac{s}{n}e^{-a(K_1|\widetilde{x}_2|+K_2x_2)}-\frac{s}{n}e^{-a(K_1|\widetilde{x}_3|+K_2x_3)}}|K_1\big]\big(\mathbb{E}\big[e^{-\frac{s}{n}\frac{s}{n}e^{-a(K_1|\widetilde{x}_1|+K_2x_1)}}|K_1\big]\big)^{n-3}\mathrm{d}s\bigg]$$

$$=\frac{1}{n}\mathbb{E}\bigg[K_2\underbrace{\int_0^\infty sA_8(s/n,1+2a,a,K_1)[M(s/n,a,K_1)]^{n-1}\mathrm{d}s}_{(I)}\bigg]$$

$$+\frac{n-1}{n}\mathbb{E}\bigg[K_2\underbrace{\int_0^\infty sA_4(s/n,1,a,K_1)A_7(s/n,2a,a,K_1)[M(s/n,a,K_1)]^{n-2}\mathrm{d}s}_{(II)}\bigg]$$

$$+\frac{n-1}{n}\mathbb{E}\bigg[K_2\underbrace{\int_0^\infty sA_2(s/n,a,a,K_1)A_3(s/n,a+1,a,K_1)[M(s/n,a,K_1)]^{n-2}\mathrm{d}s}_{(III)}\bigg]$$

$$+\frac{n-1}{n}\mathbb{E}\bigg[K_2\underbrace{\int_0^\infty sA_4(s/n,a,a,K_1)A_7(s/n,a+1,a,K_1)[M(s/n,a,K_1)]^{n-2}\mathrm{d}s}_{(IV)}\bigg]$$

$$+\frac{(n-1)(n-2)}{n}\mathbb{E}\bigg[K_2\underbrace{\int_0^\infty sA_2(s/n,a,a,K_1)A_4(s/n,a,a,K_1)A_4(s/n,1,a,K_1)[M(s/n,a,K_1)]^{n-3}\mathrm{d}s}_{(V)}\bigg]. \quad (95)$$

Here, $A_2(\lambda,\alpha,a,K_1)$ and $M(\lambda,a,K_1)$ share the same definitions as in the proof in Lemma H.1, $A_3(\lambda,\alpha,a,K_1)$ and

$A_4(\lambda, \alpha, a, K_1)$ share the same definitions as in the proof in Lemma H.3, and $A_7(\lambda, \alpha, a, K_1)$ and $A_8(\lambda, \alpha, a, K_1)$ shares the same definition as in the proof in Lemma H.8. Then, through similar procedures in the proof of Lemma H.7 and H.8 (by utilizing the Lipschitz continuities of these functions), we can derive that:

$$\left| (I) + \frac{A_8(0, 2a+1, a, K_1)}{4e^{a^2\sigma^2}\Phi^2(-a\sigma K_1)} \right|$$

$$= \left| (I) + \frac{(1+2a)\sigma^4 K_2 e^{\frac{2a^2+4a+1}{2}\sigma^2} \left[ (1+(1+2a)^2 K_1^2 \sigma^2)\Phi(-(1+2a)K_1\sigma) - (1+2a)K_1\sigma\phi((1+2a)K_1\sigma) \right]}{2\Phi^2(-a\sigma K_1)} \right| \leq \frac{c_1(a, \sigma)}{n};$$

$$\left| (II) + \frac{A_4(0, 1, a, K_1)A_7(0, 2a, a, K_1)}{4e^{a^2\sigma^2}\Phi^2(-a\sigma K_1)} \right|$$

$$= \left| (II) + \frac{a\sigma^2 K_2 e^{a^2\sigma^2}\left(\sqrt{\frac{2}{\pi}}\sigma e^{\frac{\sigma^2 K_2^2}{2}} - 2K_1\sigma^2 e^{\frac{\sigma^2}{2}}\Phi(-K_1\sigma)\right)\left(\sigma\phi(2aK_1\sigma) - 2aK_1\sigma^2\Phi(-2aK_1\sigma)\right)}{\Phi^2(-a\sigma K_1)} \right| \leq \frac{c_2(a, \sigma)}{n};$$

$$\left| (III) + \frac{A_2(0, a, a, K_1)A_3(0, a+1, a, K_1)}{4e^{a^2\sigma^2}\Phi^2(-a\sigma K_1)} \right|$$

$$= \left| (III) + \frac{a\sigma^2 K_2 e^{\frac{(2a+1)\sigma^2}{2}}}{\Phi(-aK_1\sigma)}\left(\sigma^2(1+(a+1)^2 K_1^2\sigma^2)\Phi(-(a+1)K_1\sigma) - (a+1)K_1\sigma^3\phi((a+1)K_1\sigma)\right) \right| \leq \frac{c_3(a, \sigma)}{n};$$

$$\left| (IV) + \frac{A_4(0, a, a, K_1)A_7(0, a+1, a, K_1)}{4e^{a^2\sigma^2}\Phi^2(-a\sigma K_1)} \right|$$

$$= \left| (IV) + \frac{(a+1)\sigma^2 K_2 e^{\frac{(2a+1)\sigma^2}{2}}\left(\sqrt{\frac{2}{\pi}}\sigma e^{\frac{a^2\sigma^2 K_2^2}{2}} - 2aK_1\sigma^2 e^{\frac{a^2\sigma^2}{2}}\Phi(-aK_1\sigma)\right)}{2\Phi^2(-a\sigma K_1)} \right.$$

$$\left. \cdot \left(\sigma\phi((a+1)K_1\sigma) - (a+1)K_1\sigma^2\Phi(-(a+1)K_1\sigma)\right) \right| \leq \frac{c_4(a, \sigma)}{n}. \tag{96}$$

Specifically, for the term $(V)$, we have

$$\left| (V) + \frac{A_2(0, a, a, K_1)A_4(0, a, a, K_1)A_4(0, 1, a, K_1)}{4e^{a^2\sigma^2}\Phi^2(-a\sigma K_1)} \right|$$

$$= \left| (V) + \frac{a\sigma^2 K_2\left(\sqrt{\frac{2}{\pi}}\sigma e^{-\frac{a^2\sigma^2 K_1^2}{2}} - 2aK_1\sigma^2\Phi(-aK_1\sigma)\right)}{2\Phi(-a\sigma K_1)}\left(\sqrt{\frac{2}{\pi}}\sigma e^{\frac{\sigma^2 K_2^2}{2}} - 2K_1\sigma^2 e^{\frac{\sigma^2}{2}}\Phi(-K_1\sigma)\right) \right| \leq \frac{c_5(a, \sigma)}{n}. \tag{97}$$

Lastly, by utilizing Taylor's expansion regarding the function above w.r.t. $K_1$ at $K_1 = 0$, and the facts that $\mathbb{E}[K_1] = 0$ and $\mathbb{E}[K_1^2] = 1/d$, we can derive that

$$\left| \mathbb{E}[K_2(V)] + \frac{2}{\pi}a\sigma^4 e^{\frac{\sigma^2}{2}} \right| \leq \frac{c_5(a, \sigma)}{n} + \frac{c_6(a, \sigma)}{d}. \tag{98}$$

Substituting the results of (96), (97), and (98) into (95), we complete the proof. $\qquad\square$

**Lemma H.12.** *Let $x_1, x_2, \ldots, x_n \sim \mathcal{N}(0, \sigma^2)$ be $n$ i.i.d Gaussian random variables, and $\widetilde{x}_1, \widetilde{x}_2, \ldots, \widetilde{x}_n \sim \mathcal{N}(0, \sigma^2)$ be another $n$ i.i.d Gaussian random variables. In addition, $a$ is a positive scalar, and $\boldsymbol{\theta}_*, \boldsymbol{\theta}_0 \in \mathbb{R}^d$ are two independent random vectors following uniform $d-$dimensional sphere distribution, and we denote $K_1 = \langle \boldsymbol{\theta}_0, \boldsymbol{\theta}_* \rangle$, $K_2 = \|(\mathbf{I}_d - \boldsymbol{\theta}_*\boldsymbol{\theta}_*^\top)\boldsymbol{\theta}_0\|_2$. Then we have*

$$\left| \mathbb{E}\left[ \frac{K_2 \sum_{i_1=1}^n \sum_{i_2=1}^n e^{-K_1((1+a)|\widetilde{x}_{i_1}|+a|\widetilde{x}_{i_2}|) - K_2((1+a)x_{i_1}+ax_{i_2})}x_{i_2}}{\left(\sum_{i=1}^n e^{-aK_1|\widetilde{x}_i|-aK_2 x_i}\right)^2} \right] + a\sigma^2 e^{(2a+1)\sigma^2/2} \right| \leq \frac{f_{21}(a, \sigma)}{n} + \frac{f_{22}(a, \sigma)}{d}.$$

*Here, $f_{21}(a, \sigma)$ and $f_{22}(a, \sigma)$ are both analytic functions of $a$ and $\sigma$ while irrelevant with $n$ and $d$.*

*Proof of Lemma H.12.* Following a similar procedure in the proof of Lemma H.9, we can obtain that

$$\mathbb{E}\left[ \frac{K_2 \sum_{i_1=1}^n \sum_{i_2=1}^n e^{-K_1((1+a)|\widetilde{x}_{i_1}|+a|\widetilde{x}_{i_2}|) - K_2((1+a)x_{i_1}+ax_{i_2})}x_{i_2}}{\left(\sum_{i=1}^n e^{-aK_1|\widetilde{x}_i|-aK_2 x_i}\right)^2} \right]$$

$$= \int_0^\infty s' \mathbb{E}\left[ K_2 \sum_{i_1,i_2=1}^n e^{-K_1((1+a)|\widetilde{x}_{i_1}|+a|\widetilde{x}_{i_2}|)-K_2((1+a)x_{i_1}+ax_{i_2})} x_{i_2} \exp\left(-s' \sum_{i=1}^n e^{-aK_1|\widetilde{x}_i|-aK_2 x_i}\right)\right] ds'$$

$$= n\mathbb{E}\left[ K_2 \int_0^\infty s' \mathbb{E}\left[ e^{-K_1(1+2a)|\widetilde{x}_1|-K_2(1+2a)x_1} e^{-s'e^{-a(K_1|\widetilde{x}_1|+K_2 x_1)}} x_1 | K_1 \right] \left( \mathbb{E}\left[ e^{-s'e^{-a(K_1|\widetilde{x}_1|+K_2 x_1)}} | K_1 \right] \right)^{n-1} ds' \right]$$

$$+ n(n-1)\mathbb{E}\left[ K_2 \int_0^\infty s' \mathbb{E}\left[ e^{-K_1((1+a)|\widetilde{x}_1|+a|\widetilde{x}_2|)-K_2((1+a)x_1+ax_2)} x_2 e^{-s'e^{-a(K_1|\widetilde{x}_1|+K_2 x_1)}-s'e^{-a(K_1|\widetilde{x}_2|+K_2 x_2)}} | K_1 \right] \right.$$
$$\left. \cdot \left( \mathbb{E}\left[ e^{-s'e^{-a(K_1|\widetilde{x}_1|+K_2 x_1)}} | K_1 \right] \right)^{n-2} ds' \right]$$

$$= \frac{1}{n}\mathbb{E}\left[ K_2 \int_0^\infty s \mathbb{E}\left[ e^{-K_1(1+2a)|\widetilde{x}_1|-K_2(1+2a)x_1} e^{-\frac{s}{n}e^{-a(K_1|\widetilde{x}_1|+K_2 x_1)}} x_1 | K_1 \right] \left( \mathbb{E}\left[ e^{-\frac{s}{n}e^{-a(K_1|\widetilde{x}_1|+K_2 x_1)}} | K_1 \right] \right)^{n-1} ds \right]$$

$$+ \frac{n-1}{n}\mathbb{E}\left[ K_2 \int_0^\infty s \mathbb{E}\left[ e^{-K_1((1+a)|\widetilde{x}_1|+a|\widetilde{x}_2|)-K_2((1+a)x_1+ax_2)} x_2 e^{-\frac{s}{n}e^{-a(K_1|\widetilde{x}_1|+K_2 x_1)}-\frac{s}{n}e^{-a(K_1|\widetilde{x}_2|+K_2 x_2)}} | K_1 \right] \right.$$
$$\left. \cdot \left( \mathbb{E}\left[ e^{-\frac{s}{n}e^{-a(K_1|\widetilde{x}_1|+K_2 x_1)}} | K_1 \right] \right)^{n-2} ds \right]$$

$$= \frac{1}{n}\mathbb{E}\left[ K_2 \underbrace{\int_0^\infty s A_2(s/n, 1+2a, a, K_1)[M(s/n, a, K_1)]^{n-1} ds}_{(I)} \right]$$

$$+ \frac{n-1}{n}\mathbb{E}\left[ K_2 \underbrace{\int_0^\infty s A_5(s/n, 1+a, a, K_1) A_2(s/n, a, a, K_1)[M(s/n, a, K_1)]^{n-2} ds}_{(II)} \right], \tag{99}$$

Here, $A_2(\lambda, \alpha, a, K_1)$ and $M(\lambda, a, K_1)$ share the same definitions as in the proof in Lemma H.1, and $A_5(\lambda, \alpha, a, K_1)$ shares the same definitions as in the proof in Lemma H.5. Then, through similar procedures in the proof of previous lemmas (by utilizing the Lipschitz continuities of these functions), we can derive that:

$$\left| (I) + \frac{A_2(0, 2a+1, a, K_1)}{4e^{a^2\sigma^2}\Phi^2(-a\sigma K_1)} \right| = \left| (I) + \frac{(1+2a)\sigma^2 K_2 e^{\frac{(2a^2+4a+1)\sigma^2}{2}}\Phi(-(1+2a)K_1\sigma)}{2\Phi(-a\sigma K_1)} \right| \leq \frac{c_1(a,\sigma)}{n}, \tag{100}$$

and

$$\left| (II) + \frac{A_5(0, a+1, a, K_1) A_2(0, a, a, K_1)}{4e^{a^2\sigma^2}\Phi^2(-a\sigma K_1)} \right| = \left| (II) + \frac{a\sigma^2 K_2 e^{\frac{(2a+1)\sigma^2}{2}}\Phi(-(1+a)K_1\sigma)}{\Phi(-aK_1\sigma)} \right| \leq \frac{c_2(a,\sigma)}{n}. \tag{101}$$

Specifically, by considering Taylor's expansion at $K_1 = 0$ of the function above, we can obtain

$$\left| \mathbb{E}\left[ K_2(II) \right] + a\sigma^2 e^{(2a+1)\sigma^2/2} \right| \leq \frac{c_2(a,\sigma)}{n} + \frac{c_3(a,\sigma)}{d} \tag{102}$$

Substituting the results of (100), (101), and (102) into (99), we complete the proof. □

**Lemma H.13.** *Let $x_1, x_2, \ldots, x_n \sim \mathcal{N}(0, \sigma^2)$ be $n$ i.i.d Gaussian random variables, and $\widetilde{x}_1, \widetilde{x}_2, \ldots, \widetilde{x}_n \sim \mathcal{N}(0, \sigma^2)$ be another $n$ i.i.d Gaussian random variables. In addition, $a$ is a positive scalar, and $\boldsymbol{\theta}_*, \boldsymbol{\theta}_0 \in \mathbb{R}^d$ are two independent random vectors following uniform $d-$dimensional sphere distribution, and we denote $K_1 = \langle \boldsymbol{\theta}_0, \boldsymbol{\theta}_* \rangle$, $K_2 = \|(\mathbf{I}_d - \boldsymbol{\theta}_*\boldsymbol{\theta}_*^\top)\boldsymbol{\theta}_0\|_2$. Then we have*

$$\left| \mathbb{E}\left[ \frac{K_2 \sum_{i_1=1}^n \sum_{i_2=1}^n e^{-aK_1(|\widetilde{x}_{i_1}|+|\widetilde{x}_{i_2}|)-aK_2(x_{i_1}+x_{i_2})} x_{i_1} x_{i_2}^2}{\left( \sum_{i=1}^n e^{-aK_1|\widetilde{x}_i|-aK_2 x_i} \right)^2} \right] + a\sigma^4(1+a^2\sigma^2) \right| \leq \frac{f_{23}(a,\sigma)}{n} + \frac{f_{24}(a,\sigma)}{d}.$$

*Here, $f_{23}(a,\sigma)$, $f_{24}(a,\sigma)$ are both analytic functions of $a$ and $\sigma$ while irrelevant with $n$ and $d$.*

*Proof of Lemma H.13.* Following a similar technique utilized in the proof of previous lemmas, that $\frac{1}{S^2} = \int_0^\infty s e^{-sS} ds$, we can obtain that

$$\mathbb{E}\left[ \frac{K_2 \sum_{i_1=1}^n \sum_{i_2=1}^n e^{-aK_1(|\widetilde{x}_{i_1}|+|\widetilde{x}_{i_2}|)-aK_2(x_{i_1}+x_{i_2})} x_{i_1} x_{i_2}^2}{\left( \sum_{i=1}^n e^{-aK_1|\widetilde{x}_i|-aK_2 x_i} \right)^2} \right]$$

$$= \int_0^\infty s' \mathbb{E}\left[K_2 \sum_{i_1,i_2=1}^n e^{-aK_1(|\widetilde{x}_{i_1}|+|\widetilde{x}_{i_2}|)-aK_2(x_{i_1}+x_{i_2})} x_{i_1} x_{i_2}^2 \exp\left(-s' \sum_{i=1}^n e^{-aK_1|\widetilde{x}_i|-aK_2 x_i}\right)\right] \mathrm{d}s'$$

$$= n\mathbb{E}\left[K_2 \int_0^\infty s' \mathbb{E}\left[e^{-2aK_1|\widetilde{x}_1|-2aK_2 x_1} x_1^3 e^{-s' e^{-a(K_1|\widetilde{x}_1|+K_2 x_1)}} \Big| K_1\right] \left(\mathbb{E}\left[e^{-s' e^{-a(K_1|\widetilde{x}_1|+K_2 x_1)}} \Big| K_1\right]\right)^{n-1} \mathrm{d}s'\right]$$

$$+ n(n-1)\mathbb{E}\left[K_2 \int_0^\infty s' \mathbb{E}\left[e^{-aK_1(|\widetilde{x}_1|+|\widetilde{x}_2|)-aK_2(x_1+x_2)} x_1 x_2^2 e^{-s' e^{-a(K_1|\widetilde{x}_1|+K_2 x_1)} - s' e^{-a(K_1|\widetilde{x}_2|+K_2 x_2)}} \Big| K_1\right]\right.$$

$$\left. \cdot \left(\mathbb{E}\left[e^{-s' e^{-a(K_1|\widetilde{x}_1|+K_2 x_1)}} \Big| K_1\right]\right)^{n-2} \mathrm{d}s'\right]$$

$$= \frac{1}{n}\mathbb{E}\left[K_2 \int_0^\infty s \mathbb{E}\left[e^{-2aK_1|\widetilde{x}_1|-2aK_2 x_1} x_1^3 e^{-\frac{s}{n} e^{-a(K_1|\widetilde{x}_1|+K_2 x_1)}} \Big| K_1\right] \left(\mathbb{E}\left[e^{-\frac{s}{n} e^{-a(K_1|\widetilde{x}_1|+K_2 x_1)}} \Big| K_1\right]\right)^{n-1} \mathrm{d}s\right]$$

$$+ \frac{n-1}{n}\mathbb{E}\left[K_2 \int_0^\infty \mathbb{E}\left[e^{-aK_1(|\widetilde{x}_1|+|\widetilde{x}_2|)-aK_2(x_1+x_2)} x_1 x_2^2 e^{-\frac{s}{n} e^{-a(K_1|\widetilde{x}_1|+K_2 x_1)} - \frac{s}{n} e^{-a(K_1|\widetilde{x}_2|+K_2 x_2)}} \Big| K_1\right]\right.$$

$$\left. \cdot \left(\mathbb{E}\left[e^{-\frac{s}{n} e^{-a(K_1|\widetilde{x}_1|+K_2 x_1)}} \Big| K_1\right]\right)^{n-2} \mathrm{d}s\right]$$

$$= \frac{1}{n}\mathbb{E}\left[K_2 \underbrace{\int_0^\infty s A_6(s/n, 2a, a, K_1)[M(s/n, a, K_1)]^{n-1} \mathrm{d}s}_{(I)}\right]$$

$$+ \frac{n-1}{n}\mathbb{E}\left[K_2 \underbrace{\int_0^\infty s A_2(s/n, a, a, K_1) A_1(s/n, a, a, K_1)[M(s/n, a, K_1)]^{n-2} \mathrm{d}s}_{(II)}\right]. \tag{103}$$

Here, $A_1(\lambda, \alpha, a, K_1)$, $A_2(\lambda, \alpha, a, K_1)$ and $M(\lambda, a, K_1)$ share the same definitions as in the proof in Lemma H.1, and $A_6(\lambda, \alpha, a, K_1)$ shares the same definitions as in the proof in Lemma H.7. By utilizing the Lipschitz continuity of these terms established previously, we can obtain that

$$\left|(I) + \frac{A_6(0, 2a, a, K_1)}{4e^{a^2\sigma^2}\Phi^2(-a\sigma K_1)}\right| = \left|(I) + \frac{a\sigma^4 K_2 e^{a^2\sigma^2}(3 + 4a^2\sigma^2 K_2^2)\Phi(-2aK_1\sigma)}{\Phi^2(-a\sigma K_1)}\right| \leq \frac{c_1(a,\sigma)}{n}, \tag{104}$$

and

$$\left|(II) + \frac{A_2(0, a, a, K_1)A_1(0, a, a, K_1)}{4e^{a^2\sigma^2}\Phi^2(-a\sigma K_1)}\right| = \left|(II) + \sigma^4 a K_2(1 + a^2 K_2^2 \sigma^2)\right| \leq \frac{c_2(a,\sigma)}{n}. \tag{105}$$

Specifically, by considering Taylor's expansion of the function above at $K_1 = 0$, we can further derive that

$$\left|\mathbb{E}[K_2(II)] + a\sigma^4(1 + a^2\sigma^2)\right| \leq \frac{c_2(a,\sigma)}{n} + \frac{c_3(a,\sigma)}{d}. \tag{106}$$

Substituting the results of (104), (105), and (106) into (103), we complete the proof. $\qquad\square$

**Lemma H.14.** *Let $x_1, x_2, \ldots, x_n \sim \mathcal{N}(0, \sigma^2)$ be $n$ i.i.d Gaussian random variables, and $\widetilde{x}_1, \widetilde{x}_2, \ldots, \widetilde{x}_n \sim \mathcal{N}(0, \sigma^2)$ be another $n$ i.i.d Gaussian random variables. In addition, $a$ is a positive scalar, and $\boldsymbol{\theta}_*, \boldsymbol{\theta}_0 \in \mathbb{R}^d$ are two independent random vectors following uniform $d-$dimensional sphere distribution, and we denote $K_1 = \langle \boldsymbol{\theta}_0, \boldsymbol{\theta}_* \rangle$, $K_2 = \|(\mathbf{I}_d - \boldsymbol{\theta}_* \boldsymbol{\theta}_*^\top)\boldsymbol{\theta}_0\|_2$. Then we have*

$$\left|\mathbb{E}\left[\frac{K_2 \sum_{i_1=1}^n \sum_{i_2=1}^n e^{-aK_1(|\widetilde{x}_{i_1}|+|\widetilde{x}_{i_2}|)-aK_2(x_{i_1}+x_{i_2})} x_{i_2}|\widetilde{x}_{i_1}||\widetilde{x}_{i_2}|}{\left(\sum_{i=1}^n e^{-aK_1|\widetilde{x}_i|-aK_2 x_i}\right)^2}\right] + \frac{2a\sigma^4}{\pi}\right| \leq \frac{f_{25}(a,\sigma)}{n} + \frac{f_{26}(a,\sigma)}{d}.$$

*Here, $f_{25}(a,\sigma)$ and $f_{26}(a,\sigma)$ are both analytic functions of $a$ and $\sigma$ while irrelevant with $n$ and $d$.*

*Proof of Lemma H.14.* Following a similar procedure in the proof of Lemma H.13, we can obtain that

$$\mathbb{E}\left[\frac{K_2 \sum_{i_1=1}^n \sum_{i_2=1}^n e^{-aK_1(|\widetilde{x}_{i_1}|+|\widetilde{x}_{i_2}|)-aK_2(x_{i_1}+x_{i_2})} x_{i_2}|\widetilde{x}_{i_1}||\widetilde{x}_{i_2}|}{\left(\sum_{i=1}^n e^{-aK_1|\widetilde{x}_i|-aK_2 x_i}\right)^2}\right]$$

$$= \int_0^\infty s' \mathbb{E}\Bigg[ K_2 \sum_{i_1, i_2=1}^n e^{-aK_1(|\tilde{x}_{i_1}|+|\tilde{x}_{i_2}|)-aK_2(x_{i_1}+x_{i_2})} x_{i_1}|\tilde{x}_{i_1}||\tilde{x}_{i_2}| \exp\Big(-s' \sum_{i=1}^n e^{-aK_1|\tilde{x}_i|-aK_2 x_i}\Big)\Bigg] \mathrm{d}s'$$

$$= n\mathbb{E}\Bigg[ K_2 \int_0^\infty s' \mathbb{E}\big[ e^{-2aK_1|\tilde{x}_1|-2aK_2 x_1} x_1 \tilde{x}_1^2 e^{-s' e^{-a(K_1|\tilde{x}_1|+K_2 x_1)}} \big| K_1 \big] \Big( \mathbb{E}\big[ e^{-s' e^{-a(K_1|\tilde{x}_1|+K_2 x_1)}} \big| K_1 \big] \Big)^{n-1} \mathrm{d}s' \Bigg]$$

$$+ n(n-1)\mathbb{E}\Bigg[ K_2 \int_0^\infty s' \mathbb{E}\big[ e^{-aK_1(|\tilde{x}_1|+|\tilde{x}_2|)-aK_2(x_1+x_2)} x_1 |\tilde{x}_1||\tilde{x}_2| e^{-s' e^{-a(K_1|\tilde{x}_1|+K_2 x_1)}-s' e^{-a(K_1|\tilde{x}_2|+K_2 x_2)}} \big| K_1 \big]$$

$$\cdot \Big( \mathbb{E}\big[ e^{-s' e^{-a(K_1|\tilde{x}_1|+K_2 x_1)}} \big| K_1 \big] \Big)^{n-2} \mathrm{d}s' \Bigg]$$

$$= \frac{1}{n} \mathbb{E}\Bigg[ K_2 \int_0^\infty s \mathbb{E}\big[ e^{-2aK_1|\tilde{x}_1|-2aK_2 x_1} x_1 \tilde{x}_1^2 e^{-\frac{s}{n} e^{-a(K_1|\tilde{x}_1|+K_2 x_1)}} \big| K_1 \big] \Big( \mathbb{E}\big[ e^{-\frac{s}{n} e^{-a(K_1|\tilde{x}_1|+K_2 x_1)}} \big| K_1 \big] \Big)^{n-1} \mathrm{d}s \Bigg]$$

$$+ \frac{n-1}{n} \mathbb{E}\Bigg[ K_2 \int_0^\infty s \mathbb{E}\big[ e^{-aK_1(|\tilde{x}_1|+|\tilde{x}_2|)-aK_2(x_1+x_2)} x_1 |\tilde{x}_1||\tilde{x}_2| e^{-\frac{s}{n} e^{-a(K_1|\tilde{x}_1|+K_2 x_1)}-\frac{s}{n} e^{-a(K_1|\tilde{x}_2|+K_2 x_2)}} \big| K_1 \big]$$

$$\cdot \Big( \mathbb{E}\big[ e^{-\frac{s}{n} e^{-a(K_1|\tilde{x}_1|+K_2 x_1)}} \big| K_1 \big] \Big)^{n-2} \mathrm{d}s \Bigg]$$

$$= \frac{1}{n} \mathbb{E}\Bigg[ K_2 \underbrace{\int_0^\infty s A_8(s/n, 2a, a, K_1)[M(s/n, a, K_1)]^{n-1} \mathrm{d}s}_{(I)} \Bigg]$$

$$+ \frac{n-1}{n} \mathbb{E}\Bigg[ K_2 \underbrace{\int_0^\infty s A_7(s/n, a, a, K_1) A_4(s/n, a, a, K_1)[M(s/n, a, K_1)]^{n-2} \mathrm{d}s}_{(II)} \Bigg]. \tag{107}$$

Here, $M(\lambda, a, K_1)$ shares the same definitions as in the proof in Lemma H.1, $A_4(\lambda, \alpha, a, K_1)$ shares the same definitions as in the proof in Lemma H.3, and $A_7(\lambda, \alpha, a, K_1)$ and $A_8(\lambda, \alpha, a, K_1)$ shares the same definitions as in the proof in Lemma H.8. Through a similar technique to utilize the Lipschitz continuity of these terms, we can obtain that

$$\left| (I) + \frac{A_8(0, 2a, a, K_1)}{4e^{a^2\sigma^2}\Phi^2(-a\sigma K_1)} \right|$$
$$= \left| (I) + \frac{a\sigma^4 K_2 e^{a^2\sigma^2}\big( (1 + 4a^2 K_1^2 \sigma^2)\Phi(-2aK_1\sigma) - 2aK_1\sigma\phi(2aK_1\sigma) \big)}{\Phi^2(-a\sigma K_1)} \right| \le \frac{c_1(a, \sigma)}{n}, \tag{108}$$

and

$$\left| (II) + \frac{A_7(0, a, a, K_1) A_4(0, a, a, K_1)}{4e^{a^2\sigma^2}\Phi^2(-a\sigma K_1)} \right|$$
$$= \left| (II) + \frac{a\sigma^2 K_2[\sigma\phi(aK_1\sigma) - aK_1\sigma^2\Phi(-aK_1\sigma)]}{2\Phi^2(-a\sigma K_1)} \Big( \sqrt{\tfrac{2}{\pi}}\sigma e^{-\frac{a^2\sigma^2 K_1^2}{2}} - 2aK_1\sigma^2\Phi(-aK_1\sigma) \Big) \right| \le \frac{c_2(a, \sigma)}{n}. \tag{109}$$

Specifically, by considering Taylor's expansion of the function above at $K_1 = 0$, we can further derive that

$$\left| \mathbb{E}[K_2(II)] + \frac{2a\sigma^4}{\pi} \right| \le \frac{c_2(a, \sigma)}{n} + \frac{c_3(a, \sigma)}{d}. \tag{110}$$

Substituting the results of (108), (109), and (110) into (107), we complete the proof. $\qquad \square$

**Lemma H.15.** *Let* $x_1, x_2, \ldots, x_n \sim \mathcal{N}(0, \sigma^2)$ *be* $n$ *i.i.d Gaussian random variables, and* $\tilde{x}_1, \tilde{x}_2, \ldots, \tilde{x}_n \sim \mathcal{N}(0, \sigma^2)$ *be another* $n$ *i.i.d Gaussian random variables. In addition,* $a$ *is a positive scalar, and* $\boldsymbol{\theta}_*, \boldsymbol{\theta}_0 \in \mathbb{R}^d$ *are two independent random vectors following uniform* $d-$*dimensional sphere distribution, and we denote* $K_1 = \langle \boldsymbol{\theta}_0, \boldsymbol{\theta}_* \rangle$, $K_2 = \|(\mathbf{I}_d - \boldsymbol{\theta}_* \boldsymbol{\theta}_*^\top)\boldsymbol{\theta}_0\|_2$. *Then we have*

$$\left| \mathbb{E}\left[ \frac{K_2 \sum_{i=1}^n e^{-2aK_1|\tilde{x}_i|-2aK_2 x_i} x_i}{\big( \sum_{i=1}^n e^{-aK_1|\tilde{x}_i|-aK_2 x_i} \big)^2} \right] + \frac{2a\sigma^2 e^{a^2\sigma^2}}{n} \right| \le \frac{f_{27}(a, \sigma)}{n^2} + \frac{f_{28}(a, \sigma)}{nd}.$$

*Here,* $f_{27}(a, \sigma)$ *and* $f_{28}(a, \sigma)$ *are both analytic functions of* $a$ *and* $\sigma$ *while irrelevant with* $n$ *and* $d$.

*Proof of Lemma H.15.* Following a similar procedure in the proof of Lemma H.9, we can obtain that

$$
\mathbb{E}\left[\frac{K_2 \sum_{i=1}^n e^{-2aK_1|\widetilde{x}_i|-2aK_2 x_i} x_i}{\left(\sum_{i=1}^n e^{-aK_1|\widetilde{x}_i|-aK_2 x_i}\right)^2}\right]
$$

$$
= \int_0^\infty s' \mathbb{E}\left[K_2 \sum_{i=1}^n e^{-2aK_1|\widetilde{x}_i|-2aK_2 x_i} x_i \exp\left(-s'\sum_{i=1}^n e^{-aK_1|\widetilde{x}_i|-aK_2 x_i}\right)\right]\mathrm{d}s'
$$

$$
= n\mathbb{E}\left[K_2 \int_0^\infty s' \mathbb{E}\left[e^{-2aK_1|\widetilde{x}_1|-2aK_2 x_1} e^{-s' e^{-a(K_1|\widetilde{x}_1|+K_2 x_1)}} x_1 | K_1\right]\left(\mathbb{E}\left[e^{-s' e^{-a(K_1|\widetilde{x}_1|+K_2 x_1)}}|K_1\right]\right)^{n-1}\mathrm{d}s'\right]
$$

$$
= \frac{1}{n}\mathbb{E}\left[K_2 \int_0^\infty s \mathbb{E}\left[e^{-2aK_1|\widetilde{x}_1|-2aK_2 x_1} e^{-\frac{s}{n} e^{-a(K_1|\widetilde{x}_1|+K_2 x_1)}} x_1 | K_1\right]\left(\mathbb{E}\left[e^{-\frac{s}{n} e^{-a(K_1|\widetilde{x}_1|+K_2 x_1)}}|K_1\right]\right)^{n-1}\mathrm{d}s\right]
$$

$$
= \frac{1}{n}\mathbb{E}\left[K_2 \underbrace{\int_0^\infty s A_2(s/n, 2a, a, K_1)[M(s/n, a, K_1)]^{n-1}\mathrm{d}s}_{(I)}\right]. \tag{111}
$$

Here, $A_2(\lambda, \alpha, a, K_1)$ and $M(\lambda, a, K_1)$ share the same definitions as in the proof in Lemma H.1. By using the Lipschitz continuity of $A_2(\lambda, \alpha, a, K_1)$ established in the proof of Lemma H.1, we can obtain that

$$
\left|(I) + \frac{A_2(0, 2a, a, K_1)}{4e^{a^2\sigma^2}\Phi^2(-a\sigma K_1)}\right| = \left|(I) + \frac{a\sigma^2 K_2 e^{a^2\sigma^2}\Phi(-2aK_1\sigma)}{\Phi^2(-a\sigma K_1)}\right| \le \frac{c_1(a,\sigma)}{n}. \tag{112}
$$

Specifically, by considering Taylor's expansion of the function above at $K_1 = 0$, we can further derive that

$$
\left|\mathbb{E}[K_2(I)] + 2a\sigma^2 e^{a^2\sigma^2}\right| \le \frac{c_1(a,\sigma)}{n} + \frac{c_2(a,\sigma)}{d}. \tag{113}
$$

Substituting the results of (112), and (113) into (111), we complete the proof. $\square$

**Lemma H.16.** *Let* $x_1, x_2, \ldots, x_n \sim \mathcal{N}(0, \sigma^2)$ *be* $n$ *i.i.d Gaussian random variables, and* $\widetilde{x}_1, \widetilde{x}_2, \ldots, \widetilde{x}_n \sim \mathcal{N}(0, \sigma^2)$ *be another* $n$ *i.i.d Gaussian random variables. In addition,* $a$ *is a positive scalar, and* $\boldsymbol{\theta}_*, \boldsymbol{\theta}_0 \in \mathbb{R}^d$ *are two independent random vectors following uniform* $d-$*dimensional sphere distribution, and we denote* $K_1 = \langle\boldsymbol{\theta}_0, \boldsymbol{\theta}_*\rangle$, $K_2 = \|(\mathbf{I}_d - \boldsymbol{\theta}_*\boldsymbol{\theta}_*^\top)\boldsymbol{\theta}_0\|_2$. *Then we have*

$$
\left|\mathbb{E}\left[\frac{K_2 \sum_{i_1=1}^n \sum_{i_2=1}^n \sum_{i_3=1}^n e^{-aK_1(|\widetilde{x}_{i_1}|+|\widetilde{x}_{i_2}|+|\widetilde{x}_{i_3}|)-aK_2(x_{i_1}+x_{i_2}+x_{i_3})} x_{i_1} x_{i_2} x_{i_3}}{\left(\sum_{i=1}^n e^{-aK_1|\widetilde{x}_i|-aK_2 x_i}\right)^3}\right] + a^3\sigma^6\right| \le \frac{f_{29}(a,\sigma)}{n} + \frac{f_{30}(a,\sigma)}{d}.
$$

*Here,* $f_{29}(a,\sigma)$, $f_{30}(a,\sigma)$ *are both analytic functions of* $a$ *and* $\sigma$ *while irrelevant with* $n$ *and* $d$.

*Proof of Lemma H.16.* Following the identity that $\frac{1}{S^n} = \frac{1}{(n-1)!}\int_0^\infty s^{n-1} e^{-sS}\mathrm{d}s$, we can obtain that

$$
\mathbb{E}\left[\frac{K_2 \sum_{i_1=1}^n \sum_{i_2=1}^n \sum_{i_3=1}^n e^{-aK_1(|\widetilde{x}_{i_1}|+|\widetilde{x}_{i_2}|+|\widetilde{x}_{i_3}|)-aK_2(x_{i_1}+x_{i_2}+x_{i_3})} x_{i_1} x_{i_2} x_{i_3}}{\left(\sum_{i=1}^n e^{-aK_1|\widetilde{x}_i|-aK_2 x_i}\right)^3}\right]
$$

$$
= \frac{1}{2}\int_0^\infty (s')^2 \mathbb{E}\left[K_2 \sum_{i_1, i_2, i_3=1}^n e^{-aK_1(|\widetilde{x}_{i_1}|+|\widetilde{x}_{i_2}|+|\widetilde{x}_{i_3}|)-aK_2(x_{i_1}+x_{i_2}+x_{i_3})} x_{i_1} x_{i_2} x_{i_3} \exp\left(-s'\sum_{i=1}^n e^{-aK_1|\widetilde{x}_i|-aK_2 x_i}\right)\right]\mathrm{d}s'
$$

$$
= \frac{n}{2}\mathbb{E}\left[K_2 \int_0^\infty (s')^2 \mathbb{E}\left[e^{-3aK_1|\widetilde{x}_1|-3aK_2 x_1} x_1^3 e^{-s' e^{-a(K_1|\widetilde{x}_1|+K_2 x_1)}}|K_1\right]\left(\mathbb{E}\left[e^{-s' e^{-a(K_1|\widetilde{x}_1|+K_2 x_1)}}|K_1\right]\right)^{n-1}\mathrm{d}s'\right]
$$

$$
+ \frac{3n(n-1)}{2}\mathbb{E}\left[K_2 \int_0^\infty (s')^2 \mathbb{E}\left[e^{-aK_1(|\widetilde{x}_1|+2|\widetilde{x}_2|)-aK_2(x_1+2x_2)} x_1 x_2^2 e^{-s' e^{-a(K_1|\widetilde{x}_1|+K_2 x_1)} - s' e^{-a(K_1|\widetilde{x}_2|+K_2 x_2)}}|K_1\right]\right.
$$

$$
\left. \cdot \left(\mathbb{E}\left[e^{-s' e^{-a(K_1|\widetilde{x}_1|+K_2 x_1)}}|K_1\right]\right)^{n-2}\mathrm{d}s'\right]
$$

$$
+ \frac{n(n-1)(n-2)}{2}\mathbb{E}\left[K_2 \int_0^\infty (s')^2 \mathbb{E}\left[e^{-aK_1(|\widetilde{x}_1|+|\widetilde{x}_2|+|\widetilde{x}_3|)-aK_2(x_1+x_2+x_3)} x_1 x_2 x_3\right.\right.
$$

$$\cdot e^{-s'e^{-a(K_1|\widetilde{x}_1|+K_2x_1)}-s'e^{-a(K_1|\widetilde{x}_2|+K_2x_2)}-s'e^{-a(K_1|\widetilde{x}_3|+K_2x_3)}}|K_1]\left(\mathbb{E}\big[e^{-s'e^{-a(K_1|\widetilde{x}_1|+K_2x_1)}}|K_1\big]\right)^{n-3}ds'\Big]$$

$$=\frac{1}{2n^2}\mathbb{E}\Big[K_2\int_0^\infty s^2\mathbb{E}\big[e^{-3aK_1|\widetilde{x}_1|-3aK_2x_1}x_1^3e^{-\frac{s}{n}e^{-a(K_1|\widetilde{x}_1|+K_2x_1)}}|K_1\big]\left(\mathbb{E}\big[e^{-\frac{s}{n}e^{-a(K_1|\widetilde{x}_1|+K_2x_1)}}|K_1\big]\right)^{n-1}ds\Big]$$

$$+\frac{3(n-1)}{2n^2}\mathbb{E}\Big[K_2\int_0^\infty s^2\mathbb{E}\big[e^{-aK_1(|\widetilde{x}_1|+2|\widetilde{x}_2|)-aK_2(x_1+2x_2)}x_1x_2^2e^{-\frac{s}{n}e^{-a(K_1|\widetilde{x}_1|+K_2x_1)}-\frac{s}{n}e^{-a(K_1|\widetilde{x}_2|+K_2x_2)}}|K_1\big]$$

$$\cdot\left(\mathbb{E}\big[e^{-\frac{s}{n}e^{-a(K_1|\widetilde{x}_1|+K_2x_1)}}|K_1\big]\right)^{n-2}ds\Big]$$

$$+\frac{(n-1)(n-2)}{2n^2}\mathbb{E}\Big[K_2\int_0^\infty s\mathbb{E}\big[e^{-aK_1(|\widetilde{x}_1|+|\widetilde{x}_2|+|\widetilde{x}_3|)-aK_2(x_1+x_2+x_3)}x_1x_2x_3$$

$$\cdot e^{-\frac{s}{n}e^{-a(K_1|\widetilde{x}_1|+K_2x_1)}-\frac{s}{n}e^{-a(K_1|\widetilde{x}_2|+K_2x_2)}-\frac{s}{n}e^{-a(K_1|\widetilde{x}_3|+K_2x_3)}}|K_1\big]\left(\mathbb{E}\big[e^{-\frac{s}{n}e^{-a(K_1|\widetilde{x}_1|+K_2x_1)}}|K_1\big]\right)^{n-3}ds\Big]$$

$$=\frac{1}{2n^2}\mathbb{E}\Big[K_2\underbrace{\int_0^\infty s^2 A_6(s/n,3a,a,K_1)[M(s/n,a,K_1)]^{n-1}ds}_{(I)}\Big]$$

$$+\frac{3(n-1)}{n^2}\mathbb{E}\Big[K_2\underbrace{\int_0^\infty s^2 A_2(s/n,a,a,K_1)A_1(s/n,2a,a,K_1)[M(s/n,a,K_1)]^{n-2}ds}_{(II)}\Big]$$

$$+\frac{(n-1)(n-2)}{2n^2}\mathbb{E}\Big[K_2\underbrace{\int_0^\infty s^2 A_2^3(s/n,a,a,K_1)[M(s/n,a,K_1)]^{n-3}ds}_{(III)}\Big]. \tag{114}$$

Here, $A_1(\lambda,\alpha,a,K_1)$, $A_2(\lambda,\alpha,a,K_1)$ and $M(\lambda,a,K_1)$ share the same definitions as in the proof in Lemma H.1 and $A_6(\lambda,\alpha,a,K_1)$ shares the same definition as in the proof in Lemma H.7. Through similar procedures in the proof of Lemma H.10 (by utilizing the Lipschitz continuities of these functions), we can derive that:

$$\left|(I)+\frac{A_6(0,3a,a,K_1)}{4e^{3a^2\sigma^2/2}\Phi^3(-a\sigma K_1)}\right|=\left|(I)+\frac{3a\sigma^4 K_2 e^{3a^2\sigma^2}\left(3+9a^2\sigma^2 K_2^2\right)\Phi(-3aK_1\sigma)}{2\Phi^3(-a\sigma K_1)}\right|\leq\frac{c_1(a,\sigma)}{n};$$

$$\left|(II)+\frac{A_1(0,2a,a,K_1)A_2(0,a,a,K_1)}{4e^{3a^2\sigma^2/2}\Phi^3(-a\sigma K_1)}\right|=\left|(II)+\frac{a\sigma^4 K_2(1+4a^2K_2^2\sigma^2)e^{a^2\sigma^2}\Phi(-2aK_1\sigma)}{\Phi^2(-aK_1\sigma)}\right|\leq\frac{c_2(a,\sigma)}{n}. \tag{115}$$

Specifically, for the term $(IV)$, we have

$$\left|(III)+\frac{A_2^3(0,a,a,K_1)}{4e^{3a^2\sigma^2/2}\Phi^3(-a\sigma K_1)}\right|=\left|(IV)+2a^3\sigma^6 K_2^3\right|\leq\frac{c_3(a,\sigma)}{n}. \tag{116}$$

Lastly, by utilizing Taylor's expansion regarding the function $\mathbb{E}\big[2a^3\sigma^6 K_2^4\big]$ w.r.t. $K_1$ at $K_1=0$, and the facts that $\mathbb{E}[K_1]=0$ and $\mathbb{E}[K_1^2]=1/d$, we can derive that

$$\left|\mathbb{E}\big[K_2(III)\big]+2a^3\sigma^6\right|\leq\frac{c_3(a,\sigma)}{n}+\frac{c_4(a,\sigma)}{d}. \tag{117}$$

Substituting the results of (115), (116), and (117) into (114), we complete the proof. $\qquad\square$

**Lemma H.17.** *Let $x_1,x_2,\ldots,x_n\sim\mathcal{N}(0,\sigma^2)$ be $n$ i.i.d Gaussian random variables, and $\widetilde{x}_1,\widetilde{x}_2,\ldots,\widetilde{x}_n\sim\mathcal{N}(0,\sigma^2)$ be another $n$ i.i.d Gaussian random variables. In addition, $a$ is a positive scalar, and $\boldsymbol{\theta}_*,\boldsymbol{\theta}_0\in\mathbb{R}^d$ are two independent random vectors following uniform $d-$dimensional sphere distribution, and we denote $K_1=\langle\boldsymbol{\theta}_0,\boldsymbol{\theta}_*\rangle$, $K_2=\|(\mathbf{I}_d-\boldsymbol{\theta}_*\boldsymbol{\theta}_*^\top)\boldsymbol{\theta}_0\|_2$. Then we have*

$$\left|\mathbb{E}\left[\frac{K_2\sum_{i_1,i_2,i_3=1}^n e^{-aK_1(|\widetilde{x}_{i_1}|+|\widetilde{x}_{i_2}|+|\widetilde{x}_{i_3}|)-aK_2(x_{i_1}+x_{i_2}+x_{i_3})}|\widetilde{x}_{i_1}||\widetilde{x}_{i_2}|x_{i_3}}{(\sum_{i=1}^n e^{-aK_1|\widetilde{x}_i|-aK_2x_i})^3}\right]+\frac{2}{\pi}a\sigma^4\right|\leq\frac{f_{31}(a,\sigma)}{n}+\frac{f_{32}(a,\sigma)}{d}.$$

*Here, $f_{30}(a,\sigma)$, $f_{31}(a,\sigma)$ are both analytic functions of $a$ and $\sigma$ while irrelevant with $n$ and $d$.*

*Proof of Lemma H.17.* Following a similar procedure in the proof of Lemma H.16, we can obtain that

$$
\mathbb{E}\left[\frac{K_2 \sum_{i_1=1}^{n} \sum_{i_2=1}^{n} \sum_{i_3=1}^{n} e^{-aK_1(|\widetilde{x}_{i_1}|+|\widetilde{x}_{i_2}|+|\widetilde{x}_{i_3}|)-aK_2(x_{i_1}+x_{i_2}+x_{i_3})}|\widetilde{x}_{i_1}||\widetilde{x}_{i_2}|x_{i_3}}{(\sum_{i=1}^{n} e^{-aK_1|\widetilde{x}_i|-aK_2 x_i})^3}\right]
$$

$$
=\frac{1}{2}\int_0^{\infty}(s')^2 \mathbb{E}\left[K_2 \sum_{i_1,i_2,i_3=1}^{n} e^{-aK_1(|\widetilde{x}_{i_1}|+|\widetilde{x}_{i_2}|+|\widetilde{x}_{i_3}|)-aK_2(x_{i_1}+x_{i_2}+x_{i_3})}|\widetilde{x}_{i_1}||\widetilde{x}_{i_2}|x_{i_3}\exp\left(-s'\sum_{i=1}^{n} e^{-aK_1|\widetilde{x}_i|-aK_2 x_i}\right)\right]\mathrm{d}s'
$$

$$
=\frac{n}{2}\mathbb{E}\left[K_2 \int_0^{\infty}(s')^2 \mathbb{E}\left[e^{-3aK_1|\widetilde{x}_1|-3aK_2 x_1}x_1|\widetilde{x}_1|^2 e^{-s'e^{-a(K_1|\widetilde{x}_1|+K_2 x_1)}}\Big|K_1\right]\left(\mathbb{E}\left[e^{-s'e^{-a(K_1|\widetilde{x}_1|+K_2 x_1)}}\Big|K_1\right]\right)^{n-1}\mathrm{d}s'\right]
$$

$$
+n(n-1)\mathbb{E}\left[K_2 \int_0^{\infty}(s')^2 \mathbb{E}\left[e^{-aK_1(|\widetilde{x}_1|+2|\widetilde{x}_2|)-aK_2(x_1+2x_2)}|\widetilde{x}_1||\widetilde{x}_2|x_2\right.\right.
$$

$$
\left.\left.\cdot e^{-s'e^{-a(K_1|\widetilde{x}_1|+K_2 x_1)}-s'e^{-a(K_1|\widetilde{x}_2|+K_2 x_2)}}\Big|K_1\right]\left(\mathbb{E}\left[e^{-s'e^{-a(K_1|\widetilde{x}_1|+K_2 x_1)}}\Big|K_1\right]\right)^{n-2}\mathrm{d}s'\right]
$$

$$
+\frac{n(n-1)}{2}\mathbb{E}\left[K_2 \int_0^{\infty}(s')^2 \mathbb{E}\left[e^{-aK_1(|\widetilde{x}_1|+2|\widetilde{x}_2|)-aK_2(x_1+2x_2)}x_1|\widetilde{x}_2|^2\right.\right.
$$

$$
\left.\left.\cdot e^{-s'e^{-a(K_1|\widetilde{x}_1|+K_2 x_1)}-s'e^{-a(K_1|\widetilde{x}_2|+K_2 x_2)}}\Big|K_1\right]\left(\mathbb{E}\left[e^{-s'e^{-a(K_1|\widetilde{x}_1|+K_2 x_1)}}\Big|K_1\right]\right)^{n-2}\mathrm{d}s'\right]
$$

$$
+\frac{n(n-1)(n-2)}{2}\mathbb{E}\left[K_2 \int_0^{\infty}(s')^2 \mathbb{E}\left[e^{-aK_1(|\widetilde{x}_1|+|\widetilde{x}_2|+|\widetilde{x}_3|)-aK_2(x_1+x_2+x_3)}|\widetilde{x}_1||\widetilde{x}_2|x_3\right.\right.
$$

$$
\left.\left.\cdot e^{-s'e^{-a(K_1|\widetilde{x}_1|+K_2 x_1)}-s'e^{-a(K_1|\widetilde{x}_2|+K_2 x_2)}-s'e^{-a(K_1|\widetilde{x}_3|+K_2 x_3)}}\Big|K_1\right]\left(\mathbb{E}\left[e^{-s'e^{-a(K_1|\widetilde{x}_1|+K_2 x_1)}}\Big|K_1\right]\right)^{n-3}\mathrm{d}s'\right]
$$

$$
=\frac{1}{2n^2}\mathbb{E}\left[K_2 \int_0^{\infty}s^2 \mathbb{E}\left[e^{-3aK_1|\widetilde{x}_1|-3aK_2 x_1}x_1|\widetilde{x}_1|^2 e^{-\frac{s}{n}e^{-a(K_1|\widetilde{x}_1|+K_2 x_1)}}\Big|K_1\right]\left(\mathbb{E}\left[e^{-\frac{s}{n}e^{-a(K_1|\widetilde{x}_1|+K_2 x_1)}}\Big|K_1\right]\right)^{n-1}\mathrm{d}s\right]
$$

$$
+\frac{n-1}{n^2}\mathbb{E}\left[K_2 \int_0^{\infty}s^2 \mathbb{E}\left[e^{-aK_1(|\widetilde{x}_1|+2|\widetilde{x}_2|)-aK_2(x_1+2x_2)}|\widetilde{x}_1||\widetilde{x}_2|x_2\right.\right.
$$

$$
\left.\left.\cdot e^{-\frac{s}{n}e^{-a(K_1|\widetilde{x}_1|+K_2 x_1)}-\frac{s}{n}e^{-a(K_1|\widetilde{x}_2|+K_2 x_2)}}\Big|K_1\right]\left(\mathbb{E}\left[e^{-\frac{s}{n}e^{-a(K_1|\widetilde{x}_1|+K_2 x_1)}}\Big|K_1\right]\right)^{n-2}\mathrm{d}s\right]
$$

$$
+\frac{n-1}{2n^2}\mathbb{E}\left[K_2 \int_0^{\infty}s^2 \mathbb{E}\left[e^{-aK_1(|\widetilde{x}_1|+2|\widetilde{x}_2|)-aK_2(x_1+2x_2)}x_1|\widetilde{x}_2|^2\right.\right.
$$

$$
\left.\left.\cdot e^{-\frac{s}{n}e^{-a(K_1|\widetilde{x}_1|+K_2 x_1)}-\frac{s}{n}e^{-a(K_1|\widetilde{x}_2|+K_2 x_2)}}\Big|K_1\right]\left(\mathbb{E}\left[e^{-\frac{s}{n}e^{-a(K_1|\widetilde{x}_1|+K_2 x_1)}}\Big|K_1\right]\right)^{n-2}\mathrm{d}s\right]
$$

$$
+\frac{(n-1)(n-2)}{2n^2}\mathbb{E}\left[K_2 \int_0^{\infty}s^2 \mathbb{E}\left[e^{-aK_1(|\widetilde{x}_1|+|\widetilde{x}_2|+|\widetilde{x}_3|)-aK_2(x_1+x_2+x_3)}|\widetilde{x}_1||\widetilde{x}_2|x_3\right.\right.
$$

$$
\left.\left.\cdot e^{-\frac{s}{n}e^{-a(K_1|\widetilde{x}_1|+K_2 x_1)}-\frac{s}{n}e^{-a(K_1|\widetilde{x}_2|+K_2 x_2)}-\frac{s}{n}e^{-a(K_1|\widetilde{x}_3|+K_2 x_3)}}\Big|K_1\right]\left(\mathbb{E}\left[e^{-\frac{s}{n}\frac{s}{n}e^{-a(K_1|\widetilde{x}_1|+K_2 x_1)}}\Big|K_1\right]\right)^{n-3}\mathrm{d}s\right]
$$

$$
=\frac{1}{2n^2}\mathbb{E}\left[K_2 \underbrace{\int_0^{\infty}s^2 A_8(s/n,3a,a,K_1)[M(s/n,a,K_1)]^{n-1}\mathrm{d}s}_{(I)}\right]
$$

$$
+\frac{n-1}{n^2}\mathbb{E}\left[K_2 \underbrace{\int_0^{\infty}s^2 A_4(s/n,a,a,K_1)A_7(s/n,2a,a,K_1)[M(s/n,a,K_1)]^{n-2}\mathrm{d}s}_{(II)}\right]
$$

$$
+\frac{n-1}{2n^2}\mathbb{E}\left[K_2 \underbrace{\int_0^{\infty}s^2 A_2(s/n,a,a,K_1)A_3(s/n,2a,a,K_1)[M(s/n,a,K_1)]^{n-2}\mathrm{d}s}_{(III)}\right]
$$

$$
+\frac{(n-1)(n-2)}{2n^2}\mathbb{E}\left[K_2 \underbrace{\int_0^{\infty}s^2 A_2(s/n,a,a,K_1)A_4^2(s/n,a,a,K_1)[M(s/n,a,K_1)]^{n-3}\mathrm{d}s}_{(IV)}\right]. \tag{118}
$$

Here, $A_2(\lambda, \alpha, a, K_1)$ and $M(\lambda, a, K_1)$ share the same definitions as in the proof in Lemma H.1, $A_3(\lambda, \alpha, a, K_1)$ and $A_4(\lambda, \alpha, a, K_1)$ share the same definitions as in the proof in Lemma H.3, and $A_7(\lambda, \alpha, a, K_1)$ and $A_8(\lambda, \alpha, a, K_1)$ shares the same definition as in the proof in Lemma H.8. Then, through similar procedures in the proof of Lemma H.11 (by utilizing the Lipschitz continuities of these functions), we can derive that:

$$
\begin{aligned}
&\left|(I) + \frac{A_8(0, 3a, a, K_1)}{4e^{3a^2\sigma^2/2}\Phi^3(-a\sigma K_1)}\right| \\
&= \left|(I) + \frac{3a\sigma^4 K_2 e^{3a^2\sigma^2}\left((1+9a^2K_1^2\sigma^2)\Phi(-3aK_1\sigma) - 3aK_1\sigma\phi(3aK_1\sigma)\right)}{2\Phi^3(-a\sigma K_1)}\right| \leq \frac{c_1(a,\sigma)}{n}; \\
&\left|(II) + \frac{A_4(0, a, a, K_1)A_7(0, 2a, a, K_1)}{4e^{3a^2\sigma^2/2}\Phi^3(-a\sigma K_1)}\right| \\
&= \left|(II) + \frac{a\sigma^4 K_2 e^{a^2\sigma^2}}{\Phi^3(-aK_1\sigma)}\left(\phi(2aK_1\sigma) - 2aK_1\sigma\Phi(-2aK_1\sigma)\right)\left(\sqrt{\frac{2}{\pi}}e^{-\frac{a^2\sigma^2K_1^2}{2}} + 2aK_1\sigma\Phi(-aK_1\sigma)\right)\right| \leq \frac{c_2(a,\sigma)}{n}; \\
&\left|(III) + \frac{A_2(0, a, a, K_1)A_3(0, 2a, a, K_1)}{4e^{3a^2\sigma^2/2}\Phi^3(-a\sigma K_1)}\right| \\
&= \left|(III) + a\sigma^4 K_2 e^{a^2\sigma^2}\frac{(1+4a^2K_1^2\sigma^2)\Phi(-2aK_1\sigma) - 2aK_1\sigma\phi(2aK_1\sigma)}{\Phi^2(-aK_1\sigma)}\right| \leq \frac{c_3(a,\sigma)}{n}.
\end{aligned}
\tag{119}
$$

Specifically, for the term $(IV)$, we have

$$
\begin{aligned}
&\left|(IV) + \frac{A_2(0, a, a, K_1)A_4^2(0, a, a, K_1)}{4e^{3a^2\sigma^2/2}\Phi^3(-a\sigma K_1)}\right| \\
&= \left|(IV) + \frac{a\sigma^4 K_2}{2\Phi^2(-aK_1\sigma)}\left(\sqrt{\frac{2}{\pi}}e^{-\frac{a^2\sigma^2K_1^2}{2}} - 2aK_1\sigma\Phi(-aK_1\sigma)\right)^2\right| \leq \frac{c_4(a,\sigma)}{n}.
\end{aligned}
\tag{120}
$$

Lastly, by utilizing Taylor's expansion regarding the function above w.r.t. $K_1$ at $K_1 = 0$, and the facts that $\mathbb{E}[K_1] = 0$ and $\mathbb{E}[K_1^2] = 1/d$, we can derive that

$$
\left|\mathbb{E}\big[K_2(IV)\big] + \frac{4}{\pi}a\sigma^4\right| \leq \frac{c_4(a,\sigma)}{n} + \frac{c_5(a,\sigma)}{d}.
\tag{121}
$$

Substituting the results of (119), (120), and (121) into (118), we complete the proof. $\qquad\square$

**Lemma H.18.** *Let $x_1, x_2, \ldots, x_n \sim \mathcal{N}(0, \sigma^2)$ be $n$ i.i.d Gaussian random variables, and $\widetilde{x}_1, \widetilde{x}_2, \ldots, \widetilde{x}_n \sim \mathcal{N}(0, \sigma^2)$ be another $n$ i.i.d Gaussian random variables. In addition, $a$ is a positive scalar, and $\boldsymbol{\theta}_*, \boldsymbol{\theta}_0 \in \mathbb{R}^d$ are two independent random vectors following uniform $d-$dimensional sphere distribution, and we denote $K_1 = \langle\boldsymbol{\theta}_0, \boldsymbol{\theta}_*\rangle$, $K_2 = \|(\mathbf{I}_d - \boldsymbol{\theta}_*\boldsymbol{\theta}_*^\top)\boldsymbol{\theta}_0\|_2$. Then we have*

$$
\left|\mathbb{E}\left[\frac{K_2\sum_{i_1=1}^n\sum_{i_2=1}^n e^{-aK_1(2|\widetilde{x}_{i_1}|+|\widetilde{x}_{i_2}|)-aK_2(2x_{i_1}+x_{i_2})}x_{i_2}}{(\sum_{i=1}^n e^{-aK_1|\widetilde{x}_i|-aK_2x_i})^3}\right] + \frac{a\sigma^2 e^{a^2\sigma^2}}{n}\right| \leq \frac{f_{33}(a,\sigma)}{n^2} + \frac{f_{34}(a,\sigma)}{nd}.
$$

*Here, $f_{33}(a,\sigma)$ and $f_{34}(a,\sigma)$ are both analytic functions of $a$ and $\sigma$ while irrelevant with $n$ and $d$.*

*Proof of Lemma H.18.* Following a similar procedure in the proof of Lemma H.12, we can obtain that

$$
\begin{aligned}
&\mathbb{E}\left[\frac{K_2\sum_{i_1=1}^n\sum_{i_2=1}^n e^{-aK_1(2|\widetilde{x}_{i_1}|+|\widetilde{x}_{i_2}|)-aK_2(2x_{i_1}+x_{i_2})}x_{i_2}}{(\sum_{i=1}^n e^{-aK_1|\widetilde{x}_i|-aK_2x_i})^3}\right] \\
&= \frac{1}{2}\int_0^\infty (s')^2\mathbb{E}\left[K_2\sum_{i_1,i_2=1}^n e^{-aK_1(2|\widetilde{x}_{i_1}|+|\widetilde{x}_{i_2}|)-aK_2(2x_{i_1}+x_{i_2})}x_{i_2}\exp\left(-s'\sum_{i=1}^n e^{-aK_1|\widetilde{x}_i|-aK_2x_i}\right)\right]ds' \\
&= \frac{n}{2}\mathbb{E}\left[K_2\int_0^\infty (s')^2\mathbb{E}\left[e^{-3aK_1|\widetilde{x}_1|-3aK_2x_1}e^{-s'e^{-a(K_1|\widetilde{x}_1|+K_2x_1)}}x_1|K_1\right]\left(\mathbb{E}\left[e^{-s'e^{-a(K_1|\widetilde{x}_1|+K_2x_1)}}|K_1\right]\right)^{n-1}ds'\right]
\end{aligned}
$$

$$+ \frac{n(n-1)}{2}\mathbb{E}\bigg[K_2\int_0^\infty (s')^2\mathbb{E}\big[e^{-aK_1(2|\widetilde{x}_1|+|\widetilde{x}_2|)-aK_2(2x_1+x_2)}x_2 e^{-s'e^{-a(K_1|\widetilde{x}_1|+K_2x_1)}-s'e^{-a(K_1|\widetilde{x}_2|+K_2x_2)}}\big|K_1\big]$$

$$\cdot\Big(\mathbb{E}\big[e^{-s'e^{-a(K_1|\widetilde{x}_1|+K_2x_1)}}\big|K_1\big]\Big)^{n-2}\mathrm{d}s'\bigg]$$

$$=\frac{1}{2n^2}\mathbb{E}\bigg[K_2\int_0^\infty s^2\mathbb{E}\big[e^{-3aK_1|\widetilde{x}_1|-3aK_2x_1}e^{-\frac{s}{n}e^{-a(K_1|\widetilde{x}_1|+K_2x_1)}}x_1\big|K_1\big]\Big(\mathbb{E}\big[e^{-\frac{s}{n}e^{-a(K_1|\widetilde{x}_1|+K_2x_1)}}\big|K_1\big]\Big)^{n-1}\mathrm{d}s\bigg]$$

$$+\frac{n-1}{2n^2}\mathbb{E}\bigg[K_2\int_0^\infty s^2\mathbb{E}\big[e^{-aK_1(2|\widetilde{x}_1|+|\widetilde{x}_2|)-aK_2(2x_1+x_2)}x_2 e^{-\frac{s}{n}e^{-a(K_1|\widetilde{x}_1|+K_2x_1)}-\frac{s}{n}e^{-a(K_1|\widetilde{x}_2|+K_2x_2)}}\big|K_1\big]$$

$$\cdot\Big(\mathbb{E}\big[e^{-\frac{s}{n}e^{-a(K_1|\widetilde{x}_1|+K_2x_1)}}\big|K_1\big]\Big)^{n-2}\mathrm{d}s\bigg]$$

$$=\frac{1}{2n^2}\mathbb{E}\bigg[K_2\underbrace{\int_0^\infty s^2 A_2(s/n,3a,a,K_1)[M(s/n,a,K_1)]^{n-1}\mathrm{d}s}_{(I)}\bigg]$$

$$+\frac{n-1}{2n^2}\mathbb{E}\bigg[K_2\underbrace{\int_0^\infty s A_5(s/n,2a,a,K_1)A_2(s/n,a,a,K_1)[M(s/n,a,K_1)]^{n-2}\mathrm{d}s}_{(II)}\bigg], \tag{122}$$

Here, $A_2(\lambda,\alpha,a,K_1)$ and $M(\lambda,a,K_1)$ share the same definitions as in the proof in Lemma H.1, and $A_5(\lambda,\alpha,a,K_1)$ shares the same definitions as in the proof in Lemma H.5. Then, through similar procedures in the proof of previous lemmas (by utilizing the Lipschitz continuities of these functions), we can derive that:

$$\left|(I)+\frac{A_2(0,3a,a,K_1)}{4e^{3a^2\sigma^2/2}\Phi^3(-a\sigma K_1)}\right|=\left|(I)+\frac{3a\sigma^2 K_2 e^{3a^2\sigma^2}\Phi(-3aK_1\sigma)}{2\Phi^3(-a\sigma K_1)}\right|\le\frac{c_1(a,\sigma)}{n}, \tag{123}$$

and

$$\left|(II)+\frac{A_5(0,2a,a,K_1)A_2(0,a,a,K_1)}{4e^{3a^2\sigma^2/2}\Phi^3(-a\sigma K_1)}\right|=\left|(II)+\frac{a\sigma^2 K_2 e^{a^2\sigma^2}\Phi(-2aK_1\sigma)}{\Phi^2(-aK_1\sigma)}\right|\le\frac{c_2(a,\sigma)}{n}. \tag{124}$$

Specifically, by considering Taylor's expansion at $K_1=0$ of the function above, we can obtain

$$\left|\mathbb{E}\big[K_2(II)\big]+2a\sigma^2 e^{a^2\sigma^2}\right|\le\frac{c_2(a,\sigma)}{n}+\frac{c_3(a,\sigma)}{d} \tag{125}$$

Substituting the results of (123), (124), and (125) into (122), we complete the proof. $\qquad\square$

**Lemma H.19.** *Let* $x_1,x_2\ldots,x_n\sim\mathcal{N}(0,\sigma^2)$ *be* $n$ *Gaussian random variables and* $\widetilde{x}_1,\widetilde{x}_2\ldots,\widetilde{x}_n\sim\mathcal{N}(0,\sigma^2)$ *be another* $n$ *Gaussian random variables. In addition, let* $a,b_1,b_2,k$ *be any absolute constants satisfying that* $k$ *is an integer and* $b_1^2+b_2^2=1$. *Then for sufficiently large* $n$, *it holds that*

$$\frac{1}{n^k e^{\frac{ka^2\sigma^2}{2}}[2\Phi(-a\sigma b_1)]^k}\le\mathbb{E}\bigg[\Big(\sum_{i=1}^n e^{-a(b_1|\widetilde{x}_i|+b_2 x_i)}\Big)^{-k}\bigg]\le\frac{1}{n^k e^{\frac{ka^2\sigma^2}{2}}[2\Phi(-a\sigma b_1)]^k}+\frac{c_{a,\sigma,k}}{n^{k+1}},$$

*where* $c_{a,\sigma,k}=\frac{2\Phi(2|a|\sigma)}{n^{k+1}e^{(k-2)a^2\sigma^2/2}\Phi(-|a|\sigma)^{k+2}}+1$ *is a constant solely depending on* $a$, $\sigma$ *and* $k$, *and* $\Phi(\cdot)$ *denotes the c.d.f. of standard Gaussian random variable.*

*Proof of Lemma H.19.* We first denote that $y_i=e^{-a(b_1|\widetilde{x}_i|+b_2 x_i)}$, and $S_n=\sum_{i=1}^n y_i$. In addition, we can calculate that for any scalar $m$, $\mathbb{E}[y_i^m]=2e^{\frac{m^2a^2\sigma^2}{2}}\Phi(-ma\sigma b_1)$. For the lower bound, since the function $f(z)=z^{-k}$ is convex for any $z\in(0,\infty)$, we utilize the Jensen's inequality to directly derive that

$$\mathbb{E}\big[S_n^{-k}\big]\ge\big(\mathbb{E}[S_n]\big)^{-k}=\big(n\mathbb{E}[y_1]\big)^{-k}=\frac{1}{n^k e^{\frac{ka^2\sigma^2}{2}}[2\Phi(-a\sigma b_1)]^k}. \tag{126}$$

To provide the upper bound, we leverage the Chernoff bound to obtain that

$$\mathbb{P}\Big(S_n < ne^{a^2\sigma^2/2}\Phi(-a\sigma b_1)\Big) = \mathbb{P}\Big(e^{-\lambda S_n} \geq e^{-\lambda ne^{a^2\sigma^2/2}\Phi(-a\sigma b_1)}\Big) \leq \exp\big(n\varphi(\lambda)\big),$$

where $\lambda$ is any positive scalar and $\varphi(\lambda) = \lambda e^{a^2\sigma^2/2}\Phi(-a\sigma b_1) + \log \mathbb{E}\big[e^{-\lambda y_i}\big]$. By the fact that $\log(z) \leq z - 1$ and $e^{-z} \leq 1 - z + \frac{z^2}{2}$ for any $z \geq 0$, we can derive that

$$\log \mathbb{E}\big[e^{-\lambda y_i}\big] \leq \mathbb{E}\big[e^{-\lambda y_i}\big] - 1 \leq -\lambda \mathbb{E}[y_i] + \frac{\lambda^2}{2}\mathbb{E}[y_i^2] = -2\lambda e^{\frac{a^2\sigma^2}{2}}\Phi(-a\sigma b_1) + \lambda^2 e^{2a^2\sigma^2}\Phi(-2a\sigma b_1),$$

which implies that $\varphi(\lambda) \leq -\lambda e^{\frac{a^2\sigma^2}{2}}\Phi(-a\sigma b_1) + \lambda^2 e^{2a^2\sigma^2}\Phi(-2a\sigma b_1)$. By choosing $\lambda_0 = \frac{e^{-3a^2\sigma^2/2}\Phi(-a\sigma b_1)}{2\Phi(-2a\sigma b_1)}$, we have $\varphi(\lambda_0) \leq -\frac{\Phi(-a\sigma b_1)^2}{4\Phi(-2a\sigma b_1)}e^{-a^2\sigma^2} \leq -\frac{\Phi(-|a|\sigma)^2}{4\Phi(2|a|\sigma)}e^{-a^2\sigma^2} = -f(a,\sigma)$, and further result that

$$\mathbb{P}\Big(S_n < ne^{a^2\sigma^2/2}\Phi(-a\sigma b_1)\Big) \leq e^{-f(a,\sigma)n}. \tag{127}$$

Now, we can start to derive the upper bound based on the established concentration results above. With Taylor's expansion of $g(z) = z^{-k}$, we have

$$S_n^{-k} = \frac{1}{n^k e^{\frac{ka^2\sigma^2}{2}}[2\Phi(-a\sigma b_1)]^k} - \frac{k\big(S_n - 2ne^{\frac{a^2\sigma^2}{2}}\Phi(-a\sigma b_1)\big)}{n^{k+1}e^{(k+1)a^2\sigma^2/2}[2\Phi(-a\sigma b_1)]^{k+1}} + \frac{k(k+1)\big(S_n - 2ne^{\frac{a^2\sigma^2}{2}}\Phi(-a\sigma b_1)\big)^2}{\xi_n^{k+2}},$$

where $\xi_n$ is a random variable between $S_n$ and $2ne^{\frac{a^2\sigma^2}{2}}\Phi(-a\sigma b_1)$. We take the expectation on both sides and derive that

$$\mathbb{E}\big[S_n^{-k}\big] = \frac{1}{n^k e^{\frac{ka^2\sigma^2}{2}}[2\Phi(-a\sigma b_1)]^k} + k(k+1)\mathbb{E}\left[\frac{\big(S_n - 2ne^{\frac{a^2\sigma^2}{2}}\Phi(-a\sigma b_1)\big)^2}{\xi_n^{k+2}}\right].$$

In the next, we separate the term $\mathbb{E}\left[\frac{(S_n - 2ne^{\frac{a^2\sigma^2}{2}}\Phi(-a\sigma b_1))^2}{\xi_n^{k+2}}\right]$ with the event $\{S_n \geq ne^{\frac{a^2\sigma^2}{2}}\Phi(-a\sigma b_1)\}$ as

$$\mathbb{E}\left[\frac{\big(S_n - 2ne^{\frac{a^2\sigma^2}{2}}\Phi(-a\sigma b_1)\big)^2}{\xi_n^{k+2}}\right] = \mathbb{E}\left[\frac{\big(S_n - 2ne^{\frac{a^2\sigma^2}{2}}\Phi(-a\sigma b_1)\big)^2}{\xi_n^{k+2}}\mathbb{1}_{\{S_n \geq ne^{a^2\sigma^2/2}\Phi(-a\sigma b_1)\}}\right]$$

$$+ \mathbb{E}\left[\frac{\big(S_n - 2ne^{\frac{a^2\sigma^2}{2}}\Phi(-a\sigma b_1)\big)^2}{\xi_n^{k+2}}\mathbb{1}_{\{S_n < ne^{a^2\sigma^2/2}\Phi(-a\sigma b_1)\}}\right].$$

We also consider providing the upper bounds for these two components, respectively. For the first term, we have

$$\mathbb{E}\left[\frac{\big(S_n - 2ne^{\frac{a^2\sigma^2}{2}}\Phi(-a\sigma b_1)\big)^2}{\xi_n^{k+2}}\mathbb{1}_{\{S_n \geq ne^{a^2\sigma^2/2}\Phi(-a\sigma b_1)\}}\right] \leq \frac{\mathbb{E}\big[\big(S_n - 2ne^{\frac{a^2\sigma^2}{2}}\Phi(-a\sigma b_1)\big)^2\big]}{n^{k+2}e^{(k+2)a^2\sigma^2/2}\Phi(-|a|\sigma)^{k+2}}$$

$$\leq \frac{2\Phi(2|a|\sigma)}{n^{k+1}e^{(k-2)a^2\sigma^2/2}\Phi(-|a|\sigma)^{k+2}}, \tag{128}$$

where the first inequality holds if $\mathbb{1}_{\{S_n \geq ne^{a^2\sigma^2/2}\Phi(-a\sigma b_1)\}} = 1$, then $\xi_n$ is also larger than $ne^{\frac{a^2\sigma^2}{2}}\Phi(-a\sigma b_1)$, and the second inequality holds as $\mathbb{E}[(S_n - 2ne^{a^2\sigma^2/2}\Phi(-a\sigma b_1))^2] = n\mathrm{Var}(y_i) \leq n\mathbb{E}[y_1^2] \leq 2ne^{2a^2\sigma^2}\Phi(2|a|\sigma)$. For the second term, we can obtain that

$$\mathbb{E}\left[\frac{\big(S_n - 2ne^{\frac{a^2\sigma^2}{2}}\Phi(-a\sigma b_1)\big)^2}{\xi_n^{k+2}}\mathbb{1}_{\{S_n < ne^{a^2\sigma^2/2}\Phi(-a\sigma b_1)\}}\right]$$

$$\leq \mathbb{E}\left[\frac{\big(S_n - 2ne^{\frac{a^2\sigma^2}{2}}\Phi(-a\sigma b_1)\big)^2}{S_n^{k+2}}\mathbb{1}_{\{S_n < ne^{a^2\sigma^2/2}\Phi(-a\sigma b_1)\}}\right]$$

$$\leq \mathbb{E}\left[\frac{\big(S_n - 2ne^{\frac{a^2\sigma^2}{2}}\Phi(-a\sigma b_1)\big)^4}{S_n^{2k+4}}\right]^{1/2}\sqrt{\mathbb{P}\big(S_n < ne^{a^2\sigma^2/2}\Phi(-a\sigma b_1)\big)}$$

$$\leq \mathbb{E}[(S_n - 2ne^{a^2\sigma^2/2}\Phi(-a\sigma b_1))^8]^{1/4}\mathbb{E}[y_i^{-4k-8}]^{1/4}e^{-f(a,\sigma)n} \leq \frac{1}{k(k+1)n^{k+1}}. \tag{129}$$

Here, the first inequality holds as $\xi_n > S_n$ if $S_n < ne^{a^2\sigma^2/2}\Phi(-a\sigma b_1)$. The second and third inequalities are both derived by Cauchy-Schwarz's inequality, the facts that $S_n^{-k} \leq y_1^{-k}$, and $\mathbb{P}(S_n < ne^{a^2\sigma^2/2}\Phi(-a\sigma b_1)) \leq e^{-f(a,\sigma)n}$ derived in (127). The last inequality holds as the $\mathbb{E}[(S_n - 2ne^{a^2\sigma^2/2}\Phi(-a\sigma b_1))^8] \leq c'_{a,\sigma,k}n^4$ by Rosenthal's inequality (Rosenthal, 1970), and $\mathbb{E}[y_1^{-4k-8}]^{1/4} \leq 2e^{2(k+2)^2a^2\sigma^2}$. Combining the results of (128) and (129), we finish the proof for the upper bound. $\square$

## H.2. Properties of Gaussian random variables, uniform sphere random variables, log-concave random variables, and covering number on unit sphere

**Lemma H.20.** *Let $\mathbf{a}, \mathbf{b}$ be two independent random vectors, each distributed uniformly on the unit sphere $\mathbb{S}^{d-1} \subset \mathbb{R}^d$ (that is, each is a unit random vector with the rotation-invariant probability measure). Let $K = \langle \mathbf{a}, \mathbf{b} \rangle$. Then its probability density function is*

$$\mathbb{P}(K \leq k) := f_K(k) = \frac{\Gamma(\frac{d}{2})}{\sqrt{\pi}\,\Gamma(\frac{d-1}{2})}(1 - k^2)^{\frac{d-3}{2}}\mathbb{1}_{\{-1 \leq k \leq 1\}}.$$

*In addition, $K$ is independent with $\mathbf{a}$, and $\mathbf{b}$ respectively. Moreover $\mathbb{E}[K] = 0$, $\mathrm{Var}(K) = 1/d$, and $\mathbb{E}[\|(\mathbf{I}_d - \mathbf{a}\mathbf{a}^\top)\mathbf{b}\|_2] = \frac{\Gamma^2(\frac{d}{2})}{\Gamma(\frac{d-1}{2})\Gamma(\frac{d+1}{2})}$.*

*Proof of Lemma H.20.* Since $\mathbf{a}$ and $\mathbf{b}$ are rotation invariant, implying that for any orthogonal matrix $\mathbf{R}$, we have $\mathbf{R}\mathbf{a}$ and $\mathbf{R}\mathbf{b}$ still following the unit sphere distribution. Consequently, we can always find a specific $\mathbf{R}$ such that $\mathbf{R}\mathbf{a} = \mathbf{e}_1$. Then we have $K = \langle \mathbf{a}, \mathbf{b} \rangle = \langle \mathbf{R}\mathbf{a}, \mathbf{R}\mathbf{b} \rangle = (\mathbf{R}\mathbf{b})_1$, the first coordinate of a unit sphere random vector. And it is evident that this random variable would be independent with $\mathbf{a}$, and similarly also independent with $\mathbf{b}$. In the next we derive the p.d.f. of $K$. We have shown that $K$ has the same distribution as $\mathbf{a}_1$. In addition, the differential w.r.t. the polar coordinate system indicates that $\mathrm{d}S = \sin^{d-2}(\alpha_1)\sin^{d-3}(\alpha_2)\cdots\sin(\phi_{d-2})\mathrm{d}\alpha_1\cdots\mathrm{d}\alpha_{d-2}$, where $S$ is the area of unit sphere $\mathbb{S}^{d-1} \subset \mathbb{R}^d$, and $\alpha_1, \ldots, \alpha_{d-1}$ are the angles of the polar coordinate system. Since $\mathbf{a}_1 = \cos\alpha_1$, and the marginal density function of $\alpha_1$ is proportional to $\sin^{d-2}(\alpha_1)$, we have

$$f_K(k) \propto \sin^{d-2}(\alpha_1)\frac{1}{\sin(\alpha_1)} = (1 - k^2)^{\frac{d-3}{2}}\mathbb{1}_{\{-1 \leq k \leq 1\}}.$$

In addition, we have $\int_{-1}^{1}(1 - t^2)^{\frac{d-3}{2}}\mathrm{d}t = 2\int_0^1(1 - t^2)^{\frac{d-3}{2}}\mathrm{d}t = \mathrm{Beta}(\frac{1}{2}, \frac{d-1}{2})$, which proves the p.d.f. of $K$. By symmetry, $f_K$ is an even function, hence $\mathbb{E}[K] = 0$. To compute $\mathbb{E}[K^2]$, one can use either the density or a coordinate argument. Using coordinates, write $\mathbf{a} = (a_1, \ldots, a_d)$ and $\mathbf{b} = (b_1, \ldots, b_d)$. By independence and symmetry of the uniform spherical law,

$$\mathbb{E}[K^2] = \mathbb{E}[\langle \mathbf{a}, \mathbf{b} \rangle^2] = \mathbb{E}\Big[\sum_{i,j} a_ia_jb_ib_j\Big] = \sum_{i,j}\mathbb{E}[a_ia_j]\,\mathbb{E}[b_ib_j].$$

For a uniform unit vector on $\mathbb{S}^{d-1}$ we have $\mathbb{E}[a_ia_j] = \frac{1}{d}\delta_{ij}$. Hence

$$\mathbb{E}[K^2] = \sum_{i,j}\frac{1}{d}\delta_{ij} \cdot \frac{1}{d}\delta_{ij} = \sum_i \frac{1}{d^2} = \frac{1}{d}.$$

Thus $\mathrm{Var}(K) = \mathbb{E}[K^2] - (\mathbb{E}[K])^2 = 1/d$. In addition, we can calculate that

$$\|(\mathbf{I}_d - \mathbf{a}\mathbf{a}^\top)\mathbf{b}\|_2^2 = \|\mathbf{b}\|_2^2 - 2\langle \mathbf{a}, \mathbf{b} \rangle^2 + \langle \mathbf{a}, \mathbf{b} \rangle^2\|\mathbf{a}\|_2^2 = 1 - \langle \mathbf{a}, \mathbf{b} \rangle^2.$$

Then by the density function of $K = \langle \mathbf{a}, \mathbf{b} \rangle$ demonstrated previously, we can easily derive that

$$\mathbb{E}[\|(\mathbf{I}_d - \mathbf{a}\mathbf{a}^\top)\mathbf{b}\|_2] = \mathbb{E}[\sqrt{1 - \langle \mathbf{a}, \mathbf{b} \rangle^2}] = \frac{\Gamma^2(\frac{d}{2})}{\Gamma(\frac{d-1}{2})\Gamma(\frac{d+1}{2})}.$$

In addition, by Gautschi's inequality, we have

$$\frac{\Gamma^2(\frac{d}{2})}{\Gamma(\frac{d-1}{2})\Gamma(\frac{d+1}{2})} = \frac{\Gamma^2(\frac{d}{2})}{\frac{d-1}{2}\Gamma^2(\frac{d-1}{2})} \geq \frac{\frac{d-1}{2}}{d/2} = 1 - \frac{1}{d}.$$

This completes the proof of the lemma. □

**Lemma H.21.** *Let $\mathbf{a}$ be a random vector distributed uniformly on the unit sphere $\mathbb{S}^{d-1} \subset \mathbb{R}^d$, and $\mathbf{x} \in \mathbb{R}^d$ be a standard Gaussian random vector. Then it holds that $\langle \mathbf{a}, \mathbf{x} \rangle$ is independent with $\mathbf{a}$.*

*Proof of Lemma H.21.* It is evident that for any fixed $\mathbf{a} \in \mathbb{S}^{d-1}$, the conditional distribution $\langle \mathbf{a}, \mathbf{x} \rangle | \mathbf{a} \sim \mathcal{N}(0,1)$. Since this conditional distribution does not depend on the specific choice of $\mathbf{a}$, the random variable $\langle \mathbf{a}, \mathbf{x} \rangle$ is independent of the random vector $\mathbf{a}$. This completes the proof □

**Lemma H.22.** *Let $x_1, x_2$ be two Gaussian random variables with zero mean, and $y = \text{sign}(x_1)$. Then $y$ is independent with $y \cdot x_1$ and $y \cdot x_2$. Moreover, $y \cdot x_2$ also follows the normal distribution, which has zero mean and the same variance with $x_2$.*

*Proof of Lemma H.22.* W.L.O.G., we assume that $x_1, x_2$ are standard Gaussian random variables. We first prove that $y$ is independent with $y \cdot x_1$. It is clear that $y \cdot x_1 = |x_1|$, which is independent with $y$. Next, we prove that $y$ is independent with $y \cdot x_2$ and $y \cdot x_2$ follows the standard normal distribution. If $y = 1$ then $y \cdot x_2 = x_2 \sim \mathcal{N}(0,1)$. On the other hand when $y = -1$, $y \cdot x_2 = -x_2 \sim \mathcal{N}(0,1)$. Therefore, $y \cdot x_2 | y \sim \mathcal{N}(0,1)$. Since this conditional distribution does not depend on $y$, we have $y \cdot x_2$ is independent with $y$, and follows the standard Gaussian distribution. This completes the proof. □

**Lemma H.23** (Lemma 2 in Balcan & Long (2013))**.** *Suppose that $a$ is a one-dimensional isotropic log-concave random variable, with $f_a(x)$ denoting its probability density function, then $f_a(x) \leq 1$ for all $x \in \mathbb{R}$.*

**Lemma H.24** (Lemma 3 and Theorem 4 in Balcan & Long (2013))**.** *Suppose that $\mathbf{u}, \mathbf{v} \in \mathbb{S}^{d-1}$ are two $d$-dimensional unit vectors. In addition, let $\mathbf{x} \in \mathbb{R}^d$ be a random vector generated from an isotropic log-concave distribution, then there exist two absolute positive constants $c_- \leq c_+$ such that*

$$c_- \cdot \angle(\mathbf{u}, \mathbf{v}) \leq \mathbb{P}\big(\text{sign}(\langle \mathbf{u}, \mathbf{x} \rangle) \neq \text{sign}(\langle \mathbf{v}, \mathbf{x} \rangle)\big) \leq c_+ \cdot \angle(\mathbf{u}, \mathbf{v}),$$

*where $\angle(\mathbf{u}, \mathbf{v})$ denotes the angle between unit vectors $\mathbf{u}$ and $\mathbf{v}$.*

**Lemma H.25** (Paouris' inequality)**.** *Suppose that $\mathbf{x} \in \mathbb{R}^d$ follows $d$-dimensional isotropic log-concave distribution, then for any $s > 0$,*

$$\mathbb{P}\big(\|\mathbf{x}\|_2 \geq cs\sqrt{d}\big) \leq e^{-s\sqrt{d}},$$

*where $c$ is an absolute positive constant.*

**Lemma H.26** (Lemma 5.2 in Vershynin (2010))**.** *Let $N(\mathbb{S}^{d-1}, \epsilon)$ denotes the $\epsilon$-net on unit Euclidean sphere $\mathbb{S}^{d-1}$ equipped with the Euclidean metric. Then it holds that for any $\epsilon > 0$:*

$$\big|N(\mathbb{S}^{d-1}, \epsilon)\big| \leq \left(1 + \frac{2}{\epsilon}\right)^d.$$

The proofs of Lemmas H.23 and H.24 can be found in Balcan & Long (2013), the proof of Lemma H.25 can be found in Adamczak et al. (2012), and the proof of Lemma H.26 can be found in Vershynin (2010).

