# OpenReview forum: "Transformers Efficiently Perform In-Context Logistic Regression via Normalized Gradient Descent"
_ICML.cc/2026/Conference — ICML 2026 regular_

### Official Review · Reviewer_sfkP · 2026-03-10

**Soundness:** 3
**Presentation:** 3
**Significance:** 3
**Originality:** 3
**Overall Recommendation:** 3
**Confidence:** 3

**Summary:**

The theoretical processes upon which transformer models carry out in-context logistic regression are studied in this paper. The authors study the transformer architectures in terms of optimization and suggest that some transformer attention and value update functions implicitly use some form of normalized gradient descent, and which the model can approximate the behavior of logistic regression in in-context learning. It is based on previous studies, which view transformers as solving algorithmic tasks like gradient descent or gradient maximization inference.

The paper gives theoretical discussion of the implicit bias and convergence nature of these optimization dynamics and relates it to known findings in optimization theory and implicit bias of gradient descent. The authors propose formal notation, explain the dynamics of attention updates, and explain how transformers can be used to approximate the process of iterative optimization when learning in context. The findings are expected to present a theoretical description of the manner in which transformers can undertake statistical learning functions without updating parameters but via contextual inference.

**Compliance With Llm Reviewing Policy:**

Affirmed.

**Key Questions For Authors:**

The theoretical analysis presupposes certain architectural or training circumstances. To which extent do these assumptions correspond to the modern large-scale transformer models that are applied in practice?

Would the authors give empirical experiments that prove the existence of the predicted normalized gradient descent dynamics in in-context learning with transformer models?

What does the proposed analysis have to do with other newer theoretical accounts of in-context learning e.g. meta-learning or Bayesian inference frameworks?

What are some particular architectural modifications or training strategies this theory recommends that may enhance the performance of the transformer on the in-context learning tasks?

Are the authors able to explain how the theoretical findings can be generalized to more complicated learning tasks than logistic regression?

**Limitations:**

The article describes the extent of its theoretical study and admits that the findings are based on certain modeling conditions regarding the transformer architectures and the optimization dynamics. The authors also mention that the work mainly offers theoretical information about in-context learning instead of suggesting a method that can be directly applied. Nonetheless, one might have added the gap between the theoretical assumptions and practical large scale transformer systems and described the empirical validations of the hypothesis that might have happened.

**Strengths And Weaknesses:**

Strengths

Soundness

The article answers a significant theoretical question of the mechanism of in-context learning in transformers.

The discussion is based on the literature available regarding implicit bias of gradient descent and optimization theory.

Formal notation and citation of known optimization and statistical learning results are used to support the theoretical discussion.

Presentation

The paper is written in a typical theoretical structure where the sections on the background, notation, theoretical results, and related work are provided.

The work section is related because the research gives a concise account of the previous work explaining transformer attention and learning dynamics.

Significance

In-context learning of transformers is a valuable issue in the current theory of machine learning.

The article adds to a recent body of literature that has sought to give formal descriptions on transformer capabilities.

Originality

The article provides a theoretical viewpoint that relates the behavior of transformers to the dynamics of normalized gradient descent and logistic regression.

The contribution of the work to the theory is higher than a new model or algorithm.

Weaknesses

Soundness

There are some theoretical assumptions that can restrict the generalizability of the results to practical transformer architecture in larger language models.

The theoretical derivations are given much emphasis in the paper which only offers few empirical validation of the assertions.

Presentation

Some of them are mathematically packed and might not be easily understood by readers who do not have a solid background in optimization theory.

More explanatory diagrams or illustrations of the explanations can contribute to the accessibility.

Significance

Although the theoretical explanations are useful, the practical implications of the study on how the transformer architectures or training methodologies can be enhanced are not very well explained.

Originality

The contribution is based on a generalizing body of work of understanding transformers as optimization algorithms, and some conceptual insights are generalized by previous theoretical investigations.

---

> ### Author Rebuttal · Authors · 2026-03-31
>
> Thank you for your detailed comments. We address your questions as follows. Additional experiments and illustration diagrams are given in https://anonymous.4open.science/r/rebuttal-submission17005-AE72/Rebuttal_figures_Submission17005.pdf.
>
> **Q1:** Simplifications restrict generalizability.
>
> **A1:** Please refer to our **A2** to **reviewer g4DU**.
>
> **Q2:** Empirical validation is limited. Any experiments that prove the predicted NGD dynamics?
>
> **A2:** We first clarify that all theoretical findings have been rigorously validated by experiments. In the experimental section, Figures 1, 2, and 3 provide a comprehensive empirical validation of the convergence guarantee in Theorem 3.5, covering both loss convergence and parameter convergence. In addition, Figure 4 already validates the consistency between the transformer output and the NGD output. In these plots, the green curves show the difference between the ground truth and the transformer output, while the blue curves show the difference between the ground truth and the NGD output. We can observe that these two curves remain very close. Moreover, the red curves, which directly measure the difference between the transformer output and the NGD output, stay close to zero. We also conducted experiments on real image and language datasets, where the same conclusion still holds.
>
> **Q3:** Diagrams or illustrations to help explanations of the theory?
>
> **A3:** Thanks for your suggestion. Following your suggestion, we have plotted two diagrams to improve the accessibility of the theory, and both diagrams are provided in the supplementary material. The first diagram visualizes how the self-attention computation exactly implements one step of normalized gradient descent step-by-step. In particular, it highlights which blocks of the input and parameter matrices interact with each other, and how these interactions eventually produce exactly the NGD update. We have also added a diagram summarizing the logical dependence among the main assumptions and results in the paper, so that readers can more easily follow what the assumptions are required to establish theoretical results and how these different expressive-power, training, and generalization results connect to each other.
>
> **Q4:** Relation to other theoretical accounts of ICL like meta-learning or Bayesian inference?
>
> **A4:** Our work comprehensively demonstrates that transformers can implement in-context logistic regression by NGD, from both expressive power and training. Our work also reveals a non-trivial phenomenon: transformers are not merely algorithm imitators, but are also capable of discovering and executing an algorithm distinct from the teacher. We believe these findings and insights are novel, and to our knowledge, cannot be covered by those settings like Bayesian inference. We are happy to further discuss this point if the reviewer could indicate what specific connections they have in mind and point us to the most relevant references.
>
> **Q5:** Architectures/training strategies suggested by the theory? Generalization to more complicated tasks?
>
> **A5:** The goal of our work is to understand the in-context learning capability of transformers from a theoretical perspective and to provide provable guarantees from three complementary aspects: expressive power, training, and generalization. This paper does not propose, nor does it aim to develop, new architectures or training strategies, as such directions fall outside the scope of this work. In addition, analyses of training and convergence are inherently tied to the specific learning task under study, and are therefore typically highly nontrivial to extend to substantially different settings. To our knowledge, existing works on the theoretical understanding of ICL likewise do not propose or directly suggest architectural modifications or new training strategies, nor do they provide optimization or convergence analyses that directly extend to fundamentally different learning tasks.
>
> While our work does not directly propose practically impactful methods, we believe it can still provide valuable insights into the capabilities of transformers. In particular, as discussed in **A4**, our work rigorously demonstrates through concrete case studies that transformers are not merely algorithm imitators, but can also discover and execute algorithms distinct from those used by the teacher. We believe this novel and rigorously established finding may help motivate future transformer-based applications, such as algorithm discovery.

---

> > ### Author Rebuttal · Reviewer_sfkP · 2026-04-02
> >
> > Thanks so much in the comprehensive rebuttal. The extra explanations enhance the article especially on empirical confirmation and availability of the theoretical findings. The fact that the theoretical assertions (e.g., normalized gradient descent dynamics) are empirically supported, as well as the citation of the convergence behavior and the consistency of the transformer output and the NGD prediction, are helpful to mitigate the criticism of the absence of empirical support. The fact that more diagrams are included to elaborate the theoretical mechanisms is also an effective addition.
> >
> > Nevertheless, there are still certain issues that are not fully discussed. The relationship between the theoretical assumptions and pragmatic transformer frameworks applied in large-scale conditions remains not completely clear. Although the rebuttal discusses that the work is mostly theoretical, a more tangible conversation on how the assumptions compare with the real-life models would enhance the contribution.
> >
> > Also, empirical validation seems to be confined to fairly simple environments (e.g. logistic regression tasks), and it is unclear to what extent the theoretical insights apply to more complex tasks or non-linear environments. The location of the work in a larger context of interpretations of in-context learning (including meta-learning or the Bayesian viewpoint) is recognized but may be expanded more to place the contribution into better context.
> >
> > The follow-up questions I have are as follows:
> >
> > 1.Do the authors explain how the assumptions in the theoretical analysis are instantiated in modern transformer architectures in practice (e.g. attention scaling, normalization, training dynamics)?
> > 2. How far are the theoretical guarantees extended past logistic regression to more complicated or nonlinear activities?
> > 3.Are there any more ways through which the authors can explain the relevance of their results to the current views of in-context learning (e.g., meta-learning or Bayesian inference), in terms of similarities or major differences?
> >
> > All in all, the rebuttal makes it much more understandable and offers more supportive evidence, yet certain fundamental issues on the level of generalizability and practical applicability are still left.

---

> > > ### Author Response · Authors · 2026-04-04
> > >
> > > Thank you for your response. We address your further questions below.
> > >
> > > **Q1:** How are assumptions instantiated in modern models?
> > >
> > > **A1:** (Please refer to **A1** in our original rebuttal, **A2** to **reviewer g4DU**) We do not intend to capture every ingredient of modern transformer architectures. Rather, we build a rigorous theory based on slightly simplified attention-only transformers. Such a model already captures the core self-attention mechanism of transformers, and constitutes a standard setting in the theoretical literature [1-3]. Moreover, adding practical components such as MLPs, gated attention, positional encodings, and embedding layers does not affect our conclusion on transformers’ expressive power in performing NGD. This conclusion also applies to the attention scaling and layer normalization: By Remark 3.2, pre-softmax scaling factors equivalently rescales the context data. Layer normalization would encourage the model to perform a variant of NGD with normalization term $\\|\nabla \mathcal{L}(\theta)\\|_2$.
> > >
> > > **Q2:** Extensions to complicated nonlinear activities
> > >
> > > **A2:** We focus on the specific setting of training transformers to solve linear classification to give precise theoretical guarantees. Since our analysis relies on an accurate analysis of transformer training dynamics, extending it to substantially different tasks would require analyzing different dynamics and is therefore beyond the scope of this work. To our knowledge, existing theoretical works on ICL that study training guarantees are also tied to specific task settings.
> > >
> > > In addition, our work reveals that transformers are not merely algorithm imitators: they can also discover algorithms different from the teacher. We believe this finding offers insight for future transformer-based applications, such as algorithm discovery, and may extend to more general settings as well.
> > >
> > > **Q3:** Relation to meta-learning, Bayesian inference
> > >
> > > **A3:** Meta-learning is typically formulated as a bi-level optimization problem, where the outer model shares parameters across a family of tasks, while each task has its own task-specific inner model. In ICL, [1,4] interpret transformers as meta-learners: the transformers serve as the shared outer model, while the task-specific inner model is implicitly constructed from the context examples. Rather than running a separate inner-loop learning procedure for each task, transformers adapt implicitly through their forward pass.
> > >
> > > Our setting fits naturally into this framework. Def. 2.1 can be viewed as a meta-distribution over tasks: first, a ground-truth classifier $\theta^\*$ is sampled to define a task-specific distribution; then, a context dataset $D\_{n} = \\{(\mathbf{x}\_{i},y\_{ i})\\}\_{i=1}^n$ is sampled from this distribution. Under this interpretation, the shared transformer parameters constitute the outer model, while the classifier associated with a task serves as the task-specific inner model. Given the context dataset $D\_{n}$, the transformer implicitly identifies or approximates this task-specific classifier during inference. Therefore, we believe our work can contribute to the understanding of ICL from the view of meta-learning.
> > >
> > > Our setup is also related to Bayesian inference [5]. One may view the task-specific classifier $\theta^\*$ as a latent parameter drawn from a prior, and the context dataset $D\_{n}$ as observations generated conditionally on this latent parameter. However, there may not be a very clean interpretation of how the model performs weight prediction from the Bayesian inference view. For this reason, we prefer not to overstate the connection between our setting and Bayesian inference. At the same time, we believe our results are still of independent interest, even without a full Bayesian interpretation.
> > >
> > > Please note that, since no specific references were provided, it is possible that the settings discussed above do not fully align with those you had in mind. As this is our final opportunity to respond, we may not be able to give any further clarifications. If there are particular lines of work that you believe would be especially relevant, we would be happy to discuss them in the revision.
> > >
> > > We understand that you may be viewing this work primarily from an application-oriented perspective. However, we respectfully suggest that the paper should be assessed not only from that perspective, but also in the context of the existing literature on transformer training guarantees and in light of our theoretical contributions.
> > >
> > > [1] Transformers learn in-context by gradient descent. von Oswald et al.
> > >
> > > [2] Transformers learn to implement preconditioned gradient descent for in-context learning. Ahn et al.
> > >
> > > [3] Trained transformers learn linear models in-context. Zhang et al.
> > >
> > > [4] What learning algorithm is in-context learning? Investigations with linear models. Akyurek et al.
> > >
> > > [5] An explanation of in-context learning as implicit Bayesian inference. Xie et al.

---

### Official Review · Reviewer_rpPn · 2026-03-12

**Soundness:** 3
**Presentation:** 3
**Significance:** 3
**Originality:** 3
**Overall Recommendation:** 4
**Confidence:** 3

**Summary:**

The paper studies how multi-layer softmax transformers perform in-context learning on linear classification data. The theoretical contributions of the paper are: an exact result showing that each attention layer in the transformer executes one step of normalized gradient descent on the in-context exponential loss with a single head, a training convergence result showing that a GD teacher indeed induces NGD (not GD) on the learned model, and an O.O.D. generalization bound for the looped transformer that equals the PAC lower bound under a depth condition. Experiments on synthetic data verify all three parts.

**Compliance With Llm Reviewing Policy:**

Affirmed.

**Key Questions For Authors:**

*Q1*. The training convergence result (Theorem 3.5) as well as the generalization bound (Theorem 3.7) heavily rely on the noiseless label assumption. I am curious to know whether the results still hold even when a small level of noise is allowed in the labels (like Outliers or random noises in Anwar et al. [3]).

*Q2*. I am curious to know whether the exact equivalence result of Theorem 3.1 holds even when the architecture of the neural network is extended to incorporate gated attention mechanisms, MLPs, layer normalization, etc.

*Q3*. How broadly can the exact NGD condition of Theorem 3.1 be extended to other binary classification problems? Can we achieve a similar exact equivalence result for multi-class classification as well as for regression with non-quadratic loss functions?

*Q4*. All the experiments are conducted on synthetic Gaussian data with a linear decision boundary. I am curious to know whether the exact NGD construction of the transformer model still holds on real data for the problem of text classification.

**Limitations:**

The authors acknowledge the one-layer and orthogonal pattern restrictions, but I feel several important gaps are underacknowledged. The closest-to-query vulnerability fundamentally undermines the robustness narrative yet is not framed as a limitation, and the SST-2 experiment operates outside the theoretical regime without discussion. The appendix extensions to other SSMs and tasks are informal discussions rather then formal results, and no error bars are reported anywhere.

**Strengths And Weaknesses:**

*S1*. The precise relationship between one softmax attention head and one NGD step is nice, particularly when compared to Bai et al. (2024) who need O(epsilon^{-2}) ReLU heads for an approximate GD relationship. Well done, cleaner than I expected.

*S2*. The finding that a transformer supervised by a GD teacher actually learns NGD is interesting. I think this says something real about the inductive bias of softmax attention layer.


*W1*. The general idea that transformers implement optimization algorithms in context is by now well explored, and the framwork here is directly adopted from Huang et al. (2025). I might be missing something but the novelty seems primarily in the technical extension to classification and softmax. Relatedly, Dragutinovic et al. (2025) and Shen et al. (2024) study softmax attention for ICL classification as well.

*W2*. The entire analysis is based on noiseless labels, isotropic Gaussian features, and single-head attention without MLPs. It is not clear how much of this analysis would remain if any of these assumptions are violated, but this is not discussed in the paper.


References:
- Dragutinovic, Saxe, and Singh, 2025, "Softmax >= Linear: Transformers may learn to classify in-context by kernel gradient descent", arXiv:2510.10425
- Shen, Zhou, Yang, and Shen, 2024, "On the Training Convergence of Transformers for In-Context Classification of Gaussian Mixtures", arXiv:2410.11778
- Anwar, von Oswald, Kirsch, Krueger, and Frei, 2024, "Understanding In-Context Learning of Linear Models in Transformers Through an Adversarial Lens", arXiv:2411.05189

---

> ### Author Rebuttal · Authors · 2026-03-31
>
> Thanks for your insightful comments. We address your concerns as follows. Additional experiments are given in https://anonymous.4open.science/r/rebuttal-submission17005-AE72/Rebuttal_figures_Submission17005.pdf.
>
> **Q1:** The idea that transformers conduct algorithms is not new. Comparisons with prior works.
>
> **A1:** Besides the expressive power result that transformers can implement NGD, one of our key findings is a nontrivial phenomenon: even when guided by a GD teacher, transformers still learn a different algorithm, namely NGD, rather than merely imitating the teacher. This suggests that transformers are not only algorithm executors, but are also capable of discovering and executing an algorithm distinct from the teacher.  We believe this insight is new, and establishes the practical value of our work.
>
> It is true that we study the weight prediction task introduced in [1]. However, beyond this task setup, our setting, including softmax attention, miss-specified teacher algorithm, fully trainable parameters, etc., is fundamentally different from that of [1]. Moreover, our main conclusion that transformers learn NGD even when guided by GD is entirely unrelated to [1]. Therefore, in terms of both technical novelty and contribution, our work should not be viewed as an incremental extension of [1].
>
> Due to space limit, please refer to our response **A2** to **reviewer g4DU** for more detailed comparisons with [2, 4] and other related works.
>
> **Q2:** Analysis is based on simplifications. Does Thm 3.1 hold when incorporating more components? Does it hold on real data?
>
> **A2:** Please refer to our **A2** to **reviewer g4DU**.
>
> **Q3:** How do Thms 3.5  and 3.7 behave when noise is allowed in the labels?
>
> **A3:** We consider label flipping, common in classification tasks. Under this setting, the feature $\mathbf{x}_i$ and ground-truth label $y_i$ are still generated from Def 2.1. However, we only observe a noisy label $\tilde y_i=\zeta_iy_i$, where $\zeta_i=1$ with probability (w.p.) $1-p$ and $\zeta_i=-1$ w.p. $p$. The input vector is then $\mathbf z_i=\tilde y_i\mathbf x_i$. We next discuss the extensions of both Thms 3.5 and 3.7 to this setting.
>
> * For Thm 3.5, the conclusions remain unchanged. This is because transformer training is supervised by one-step GD, and the teacher itself is composed of input vectors $\mathbf z_i$ with flipped label $\tilde y_i$. Hence, label flipping affects both the transformer inputs and the teacher, and these effects can be cancelled. We conduct experiments with $p=0.05$, and the trajectories of $C_1^{(t)}$ and $C_2^{(t)}$ remain identical to the unflipped case, validating that Thm 3.5 still hold.
>
> * For Thm 3.7, the current proof and conclusion cannot be directly extended. This is because the previous proof relies on Corollary 3.3, which requires the in-context examples to be linearly separable. However, this condition no longer holds under label flipping, and a new proof is needed. Under $p\geq\Omega((d/n)^{1/3})$, which ensures that w.h.p. at least one label is flipped, we can obtain that w.p. $1-\delta$,
> $$\Bigg\\|\frac{\theta\_L}{\\|\theta\_L\\|\_2}-\theta^\*\Bigg\\|\_2\leq\frac{\\|\theta\_L-\hat{\theta}\_n\\|\_2}{\\|\theta\_L\\|\_2}+\Bigg\\|\frac{\hat{\theta}\_n}{\\|\theta\_L\\|\_2}-\theta^\*\Bigg\\|\_2\leq O\Bigg(\sqrt{\frac{d \log\rho}{L}}+\sqrt{\frac{d+\log\rho}{p^3n}}\Bigg),$$
> where $\hat{\theta}_n$ denotes the empirical minimizer of ICL loss, and $\rho =\max\\{n, d, 1/p, 1/\delta\\}$.
>
> **Q4:** Extensions to other cases, like multi-classifications, other loss functions and layer norms.
>
> **A4:** The current Thm 3.1 cannot be extended to these cases, as the self-attention(SA) layer cannot perfectly match the required functions.
>
> Take multi-classifications as an example. For $K$-classification, the exponential loss is $L(\Theta)=\frac{1}{n}\sum\_{i=1}^n\sum\_{k \neq y\_i}e^{\langle\theta\_k - \theta\_{y\_i}, \mathbf{x}\_i \rangle},$ a summation of $n(K-1)$ terms. Consequently, the NGD update for $k$-th classifier $\theta_k$ would be
> $$-\frac{\nabla_{\theta_k}L(\Theta)}{L(\Theta)}=\frac{\sum_{y_i=k}\sum_{j\neq k}e^{\langle\theta_j-\theta_k,\mathbf{x}\_i\rangle}\mathbf{x}\_i}{\sum\_{i=1}^n\sum\_{k\neq y\_i}e^{\langle\theta\_k - \theta\_{y\_i}, \mathbf{x}\_i \rangle}}
> -\frac{\sum_{y_i\neq k} e^{\langle \theta_k-\theta_{y_i},\mathbf{x}_i\rangle}\mathbf{x}_i}{\sum\_{i=1}^n\sum\_{k \neq y\_i}e^{\langle\theta\_k - \theta\_{y\_i}, \mathbf{x}\_i \rangle}},$$
> where denominators have $n(K-1)$  exponential terms. Since the sequence length of the input to transformers is $n+1$, unless a particular embedding is allowed to change the sequence length, the SA layer can only normalize over at most $n+1$ terms, and cannot easily realize a denominator with $n(K−1)$ terms. Similarly, the SA layer also cannot perfectly match other loss functions. In addition, when adding layer norms, the output of the SA layer is constrained with a fixed scale, which cannot match algorithms like GD or NGD.

---

> > ### Author Rebuttal · Reviewer_rpPn · 2026-04-03
> >
> > The rebuttal effectively addressed all the issues I raised. Since my initial assessment was already positive, I will retain my original score.

---

> > > ### Author Response · Authors · 2026-04-04
> > >
> > > We sincerely appreciate your insightful comments and efforts in reviewing our paper.

---

### Official Review · Reviewer_g4DU · 2026-03-12

**Soundness:** 4
**Presentation:** 3
**Significance:** 3
**Originality:** 2
**Overall Recommendation:** 5
**Confidence:** 4

**Summary:**

The paper studies in-context learning of logistic regression. It first construct transformer weight matrices to prove that a transformer can perform in-context logistic regression and further proves under a certain training procedure the transformer is indeed trained to the constructed weights. Experimental results also supports the theorems.

**Compliance With Llm Reviewing Policy:**

Affirmed.

**Final Justification:**

The paper considers not only the inference weight construction but also proves that the weight can be learned during pretraining. The experimental results strongly agrees with the theoretical claims, and the extension to softmax is also nontrivial in terms of theoretical contribution.

**Key Questions For Authors:**

1. Could you predict how the conclusions would change for a normal multi-layer transformer? What would be the main technical difficulty to extend Theorem 3.7 to a non-loop transformer?
2. Could you provide the inference time ICL performance plots and compare it directly with normalized GD?

**Limitations:**

yes

**Strengths And Weaknesses:**

Strengths:
1. The paper closes the gap between "a model can perform certain algorithmic update in ICL" and "whether a model is actually trained to do so" for logistic regression tasks.
2. Experimental results align well with the theoretical predictions, both in terms of convergence and weight construction.
3. The presentation and comprehensiveness of the paper is of high quality and meets the standard of related works accepted in top conferences.

Weaknesses:
1. The results on in-context logistic regression can't guide real world transformer development, e.g. how to improve ICL, and the concept that transformers perform GD in various ICL tasks is not new, neither surprising.
2. The logistic regression task is a somewhat toy example. These kinds of mathematical functions were initially considered to guide the understanding of ICL, but people don't use transformers or LLMs to in-context learn toy math examples in practice.
3. The simplifications of the transformer architecture (e.g. no positional encoding; no MLP etc.) and the real world input format (no embedding considered; ICL examples are formatted in the specific column-wise way) limits the general applicability of the theorems.

---

> ### Author Rebuttal · Authors · 2026-03-31
>
> Thanks for your positive evaluation! We address your concerns in the following. Additional experiments are provided in https://anonymous.4open.science/r/rebuttal-submission17005-AE72/Rebuttal_figures_Submission17005.pdf.
>
> **Q1:** Logistic regression is a toy example, can't guide real development. The idea that transformers perform GD in ICL is not new.
>
> **A1:** We first clarify that our finding is that transformers can perform NGD instead of GD. To our knowledge, this finding is novel.
>
> Moreover, while logistic regression is a simplified setting, this case study reveals a nontrivial phenomenon: even when guided by GD, transformers still learn a different algorithm, namely NGD, rather than merely imitating the teacher. This suggests that transformers are not only algorithm imitators, and are also capable of discovering new algorithms. We believe this insight is surprising, demonstrating the practical value of our work.
>
> **Q2:** The simplifications of architecture and input format limit the general applicability.
>
> **A2:** We address this concern from expressive power and training.
>
> * For expressive power, Thm 3.1 demonstrates that transformers can implement NGD. This theorem does not rely on any assumption on the input data distribution. Since the result concerns the expressive power of transformers, **showing that such a capability can already be achieved with a simpler architecture is a strength rather than a weakness.** Adding extra components, such as MLP layers, positional encodings, or gated attention, does not affect this expressive power conclusion. Intuitively, MLP layers and gated attention with certain parameters can easily behave as identity mappings and standard attention, and the self-attention parameters can be chosen to ignore concatenated positional encodings, implying the output would be identical to our proposed models. In addition, as detailed in our **A3** to **Reviewer nmrc**, standard input format $[\mathbf{x}_i^\top, y_i]$ can be transformed into the feature vector $y_i \cdot \mathbf{x}_i$ through an embedding model. We have conducted more experiments on real image and language data to validate Thm 3.1. These observations demonstrate the broad generality of Thm 3.1.
>
> * For training, we acknowledge that our current theory only covers a single self-attention(SA) layer. However, compared with most existing theoretical works on transformer training, our assumptions are already among the weakest. In particular, we study the SA layer with softmax attention, rather than linear attention, which is often adopted in prior works for analytical convenience [1, 2]. We also do not impose special training strategies or initialization schemes for simplification, allowing all parameters to be fully trainable. In comparison, [2] assumes that the value matrix $\mathbf{V}$ is frozen in a prescribed structure throughout training, and only $\mathbf{W}$ is trainable. [3] requires some blocks of $\mathbf{V}$ and $\mathbf{W}$ to remain identically zero, while other blocks must stay proportional to the identity matrix. Moreover, we develop a highly non-trivial convergence guarantee, while the training analysis is not covered in prior works [4, 5] about ICL on classification tasks. These comparisons highlight the technical strength of our work.
>
> **Q3:**  How would the conclusions (Thm 3.7) change for a normal multi-layer transformer?
>
> **A3:** Thm 3.7 can be directly extended to any transformers satisfying the parameterization form in Thm 3.1, and Thm 3.1 does not require a looped structure. Intuitively, this is because a non-looped model should even have stronger expressive power than a looped model. Moreover, as discussed in the previous response **A2**, this conclusion can even cover more complicated architectures incorporating other components like MLPs.
>
> **Q4:** Inference time ICL performance plots and compare with NGD?
>
> **A4:** Figure 4 already shows the results on three distributions. On these plots, the green curves present the difference between the ground truth and the output of transformers, while the blue curves present the difference between the ground truth and the output of NGD. We can observe that they stay close. In addition, the red curves directly indicate the difference between the output of transformers and NGD, which remains close to 0. As mentioned in **A2**, further experiments on real data are provided.
>
> [1]. Huang et al. Transformers learn to implement multi-step gradient descent with chain of thought.
>
> [2]. Shen et al. On the Training Convergence of Transformers for In-Context Classification of Gaussian Mixtures.
>
> [3]. Chen et al. In-Context Linear Regression Demystified: Training Dynamics and Mechanistic Interpretability of Multi-Head Softmax Attention.
>
> [4]. Dragutinovic et al. Softmax >= linear: transformers may learn to classify in-context by kernel gradient descent.
>
> [5]. Bai et al. Transformers as statisticians: provable in-context learning with in-context algorithm selection.

---

> > ### Author Rebuttal · Reviewer_g4DU · 2026-04-03
> >
> > I thank the authors for the rebuttal and apologize for missing figure4. I do appreciate the merit of strength 1 and the softmax architecture as well as the additional experiments on real world data. Thus I've increased my score to 5.

---

> > > ### Author Response · Authors · 2026-04-04
> > >
> > > Thank you for your careful follow-up and for revisiting the paper after reading the rebuttal. We sincerely appreciate your recognition of our technical contributions.

---

### Official Review · Reviewer_nmrc · 2026-03-17

**Soundness:** 3
**Presentation:** 3
**Significance:** 3
**Originality:** 3
**Overall Recommendation:** 5
**Confidence:** 4

**Summary:**

This paper studies attention only transformers on sequences corresponding to samples from a classification dataset (the parameter theta governing the classification rule is different for each sequence, hence the in-context). The paper show that there exists a transformer with L layer for which the forward pass corresponds to implementing L steps of gradient descent on an exponential classification loss (Th 3.1). This allows to obtain a bound on the optimality of the found parameter theta_L (Cor. 3.3). Then, the submission proves that such constructed model is actually learned by gradient descent on a looped model (i.e. the weights are the same at each layer, also known as tied-weights), and derive linear convergence rates (Th 3.5) for the optimization algorithm (the gradient descent on the model parameter, not to be confused by the meta gradient descent implemented by the transformer itself). Under additional assumptions on the data distribution D_x, OOD generalization guarantees are proved in Th 3.7. Numerical simulations validate the theoretical results.

**Compliance With Llm Reviewing Policy:**

Affirmed.

**Final Justification:**

The authors have adressed my concerns about the limitation of their work and will acknoledge these limitations in the paper. The main concern I had was regarding the use of an exponential loss whereas the paper claims it does logistic regression. I do believe the paper deserves to be published as it adresses rigourously a new theoretical question, backed up with numerical experiments. I therefore increased my score from 4 to 5.

**Key Questions For Authors:**

In this work, we adopt the exponential loss, i.e. ℓ(x) = e^−x to enable cleaner mathematical results -> isn't that actually fundamental for your analysis to work? It is precisely because you use a loss with this exponential form that you can implement your gradient descent using softmax attention right? This would not work for standard logistic loss l(x) = log(1 + e^-x)?

Corollary 3.3: can you comment on the log(n) dependency?

l.153: typo l is from R to R.

Why aren't the input sequences of the form (x_i, y_i , 0_d) rather than your  (y_i * x_i , 0_d)? Could your analysis extend to this setting? Why exactly did you choose this formatting for the input sequence?

For now, I am hesitating between weak accept and accept. Putting weak accept for now, but happy to revise my rating after discussion with the authors. edit: increasing to 5 after rebuttal

**Limitations:**

yes

**Strengths And Weaknesses:**

Soundness: Yes, the paper is sound. Claims are well supported, theorems are proved. Experiments are detailled.

Presentation: The submission is well written, I enjoyed reading it, things are rigourous and pedagogical.

Significance: Yes, the paper is significant because it extends previous studies done on linear regression to linear classification, and studies the general softmax attention case. However, The paper should be more honest in that it considers a very specific loss function (exponential) as an inner in-context objective.

Originality: Yes, the paper provides new insights on the theory of in-context learning

---

> ### Author Rebuttal · Authors · 2026-03-31
>
> Thanks for your insightful comments and recognition of our work! We address your concerns as follows.
>
> **Q1:** This paper considers a specific exponential loss function. The exact match form can not be extended to logistic loss.
>
> **A1:** Thanks for pointing this out. It is true that softmax attention exactly matches the gradient form of the exponential loss, which serves as the key insight behind our analysis. This exact correspondence does not directly extend to the logistic loss. More precisely, Thm 3.1 shows that transformers can exactly implement NGD for “in-context linear classification with the exponential loss”. In the implicit-bias literature, linear classification models with exponential-tailed loss functions are typically treated under the unified framework of logistic regression. Following this convention, we slightly abuse the term logistic regression to present the case of linear classification with the exponential loss for conciseness. To avoid potential overclaim or ambiguity, we will revise the manuscript to make this point explicit by stating in the introduction and in the discussion of Thm 3.1 that logistic regression is considered with the exponential loss. We will also delete the phrase “to enable cleaner mathematical results”, and modify “$L_{ICL}$” in Line 159 as “$L_{ICL}$ on exponential loss” to explicitly specify the scope of Thm 3.1.
>
> However, we would like to emphasize that the transformer here should be viewed as an algorithm executor. The in-context exponential loss is introduced solely to precisely characterize how transformers in-context learn a linear classifier. When evaluating whether transformers can solve a task, what ultimately matters is the output classifier. From this perspective, transformer models do solve in-context logistic regression, since they produce a linear classifier with the same limiting solution as logistic regression. This is because the implicit-bias literature has shown that linear classification with the logistic loss and with the exponential loss eventually converges to the same linear classifier.
>
> **Q2:** Explain the $\log(n)$ dependence in Corollary 3.3.
>
> Corollary 3.3 is obtained by directly applying Thm 4.4 in [1] . Hence, the $\log(n)$ term is also directly inherited from their result. Intuitively, if the $n$ data points $\mathbf z\_i$’s are identical, then the optimization procedure is equivalent to just using one data point, and such a $\log(n)$ term would disappear. Since $\mathbf  z\_i$’s are different and the SVM solution relies particularly on the “support vectors”, the discrepancy among $\mathbf z\_i$’s leads to such a $\log(n)$ term.
>
> **Q3:** Why the input sequence of the form $(y_i\cdot\mathbf{x}_i , \mathbf{0}_d)$ rather than $(\mathbf{x}_i, y_i ,\mathbf{0}_d)$? Could your analysis extend to this setting?
>
> A3: From the definition of in-context logistic regression that
> $$\mathcal{L}\_{\mathrm{ICL}}(\theta) = \frac{1}{n}\sum\_{i=1}^n \ell(\langle \theta, y\_i\cdot\mathbf{x}\_{i}\rangle),$$
> we can clearly observe that both the loss and its gradient depend on $y\_i$ and $\mathbf{x}\_i$ only through their product $y\_i\cdot \mathbf{x}\_i$. Consequently, we consider the embedding vector in the form of $\mathbf{z}\_i = y\_i\cdot \mathbf{x}\_i$ to enable the input of transformers to match that of logistic regression. Notably, treating the product $y\_i\cdot \mathbf{x}\_i$ as a new feature vector $\mathbf{z}\_i$ is widely considered in the implicit bias literature [1, 2], and our settings follow this convention.
>
> Moreover, the concatenated vector $[\mathbf{x}\_i^\top, y\_i]^\top\in \mathbb{R}^{d+1}$ can be converted to our embedding vector $\mathbf{z}\_i = y\_i\cdot \mathbf{x}\_i$ through a two-layer ReLU network. Define $f(\mathbf x; \mathbf W\_1,  \mathbf b\_1, \mathbf W\_2) =\mathbf W\_2\sigma(\mathbf W\_1 \mathbf x+ \mathbf b\_1)$, where $\mathbf x\in \mathbb{R}^{d+1}$ denotes the input, and $M$ is a sufficiently large constant such that $\\|\mathbf x\\|\_\infty\leq M$. In addition, the parameter matrices are set as
> $$\mathbf{W}\_1=
> \begin{bmatrix}
> \mathbf{I}\_d &M\mathbf{1}\_d\\\\
> -\mathbf{I}\_d &M\mathbf{1}\_d\\\\
> \mathbf{I}\_d &-M\mathbf{1}\_d\\\\
> -\mathbf{I}\_d &-M\mathbf{1}\_d
> \end{bmatrix}
> \in\mathbb{R}^{4d\times(d+1)},$$
> $\mathbf{b}\_1=-M\mathbf{1}\_{4d}\in\mathbb{R}^{4d},$
> and $\mathbf{W}\_2=
> \begin{bmatrix}
> \mathbf{I}\_d & -\mathbf{I}\_d & -\mathbf{I}\_d & \mathbf{I}\_d
> \end{bmatrix}
> \in\mathbb{R}^{d\times 4d}.$
>
> With this embedding layer, we can obtain the embedding vector $y\_i\cdot \mathbf{x}\_i$ through $y\_i\cdot \mathbf{x}\_i =f([\mathbf{x}\_i^\top, y\_i]^\top; \mathbf W\_1,  \mathbf b\_1, \mathbf W\_2)$. As an embedding layer is a common component of transformers, this demonstrates that our theory still applies to input vectors $[\mathbf{x}\_i^\top, y\_i]^\top$.
>
> [1] Ji and Telgarsky. Characterizing the implicit bias via a primal-dual analysis.
>
> [2] Soudry et al. The implicit bias of gradient descent on separable data.

---

> > ### Author Rebuttal · Reviewer_nmrc · 2026-04-03
> >
> > Dear authors
> >
> > Thank you for your rebuttal and addressing my concerns. I am increasing my score to 5. Please acknowledge in the paper the limitations of considering the exponential loss, and I also believe that a discussion on how the embeddings y_i x_i can be computed with the two linear ReLU network is necessary.
> >
> > One last advice I missed in my original review: the figures in the experimental part of the paper are really hard to read. I would suggest reducing the figsize in matplotlib, this will make the captions bigger. I think this will give more attraction to your paper.

---

> > > ### Author Response · Authors · 2026-04-04
> > >
> > > Thank you for your thoughtful suggestions and efforts in reviewing our paper. We sincerely appreciate your insightful comments, and we will revise the paper according to these discussions and adjust the size of the figures.

---

### Decision · Program_Chairs · 2026-04-30

**Decision:**

Accept (regular)

**Comment:**

This paper studies in-context learning for logistic regression. It focuses on multi-layer softmax attention and shows that it implements normalized gradient descent over in-context examples. The paper also provides theoretical results on convergence and optimization.

As noted by the reviewers, there is limited discussion on the connection between the theoretical setting and realistic transformer architectures and data distributions. However, the paper offers solid and rigorous theoretical analysis in a nontrivial setting involving softmax attention and multi-layer models, which extends prior work.

I encourage the authors to explicitly and clearly explain the novelty of their contributions compared to prior work, and to better discuss the implications of their results for practical transformer models.